# Provable Benefit of Cutout and CutMix for Feature Learning

**Junsoo Oh**
KAIST AI
junsoo.oh@kaist.ac.kr

**Chulhee Yun**
KAIST AI
chulhee.yun@kaist.ac.kr

## Abstract

Patch-level data augmentation techniques such as Cutout and CutMix have demonstrated significant efficacy in enhancing the performance of vision tasks. However, a comprehensive theoretical understanding of these methods remains elusive. In this paper, we study two-layer neural networks trained using three distinct methods: vanilla training without augmentation, Cutout training, and CutMix training. Our analysis focuses on a feature-noise data model, which consists of several label-dependent features of varying rarity and label-independent noises of differing strengths. Our theorems demonstrate that Cutout training can learn low-frequency features that vanilla training cannot, while CutMix training can learn even rarer features that Cutout cannot capture. From this, we establish that CutMix yields the highest test accuracy among the three. Our novel analysis reveals that CutMix training makes the network learn all features and noise vectors "evenly" regardless of the rarity and strength, which provides an interesting insight into understanding patch-level augmentation.

## 1 Introduction

Data augmentation is a crucial technique in deep learning, particularly in the image domain. It involves creating additional training examples by applying various transformations to the original data, thereby enhancing the generalization performance and robustness of deep learning models. Traditional data augmentation techniques typically focus on geometric transformations such as random rotations, horizontal and vertical flips, and cropping (Krizhevsky et al., 2012), or color-based adjustments such as color jittering (Simonyan and Zisserman, 2014).

In recent years, several new data augmentation techniques have appeared. Among them, patch-level data augmentation techniques like Cutout (DeVries and Taylor, 2017) and CutMix (Yun et al., 2019) have received considerable attention for their effectiveness in improving generalization. Cutout is a straightforward method where random rectangular regions of an image are removed during training. In comparison, CutMix adopts a more complex strategy by cutting and pasting sections from different images and using mixed labels, encouraging the model to learn from blended contexts. The success of Cutout and CutMix has triggered the development of numerous variants including Random Erasing (Zhong et al., 2020), GridMask (Chen et al., 2020a), CutBlur (Yoo et al., 2020), Puzzle Mix (Kim et al., 2020), and Co-Mixup (Kim et al., 2021). However, despite the empirical success of these patch-level data augmentation techniques in various image-related tasks, a lack of comprehensive theoretical understanding persists: *why and how do they work?*

In this paper, we aim to address this gap by offering a theoretical analysis of two important patch-level data augmentation techniques: Cutout and CutMix. Our theoretical framework draws inspiration from a study by Shen et al. (2022), which explores a data model comprising multiple label-dependent feature vectors and label-independent noises of varying frequencies and intensities. The key idea of this work is that learning features with low frequency can be challenging due to strong noises (i.e., low signal-to-noise ratio). We focus on how Cutout and CutMix can aid in learning such rare features.

38th Conference on Neural Information Processing Systems (NeurIPS 2024).

## 1.1 Our Contributions

In this paper, we consider a patch-wise data model consisting of features and noises, and use two-layer convolutional neural networks as learner networks. We focus on three different training methods: vanilla training without any augmentation, Cutout training, and CutMix training. We refer to these training methods in our problem setting as ERM, Cutout, and CutMix. We investigate how these methods affect the network's ability to learn features. We summarize our contributions below:

- We analyze ERM, Cutout, and CutMix, revealing that Cutout outperforms ERM since it enables the learning of rarer features compared to ERM (Theorem 3.1 and Theorem 3.2). Furthermore, CutMix demonstrates almost perfect performance (Theorem 3.3) by learning all features.

- Our main intuition behind the negative result for ERM is that ERM learns to classify training samples by memorizing noise vectors instead of learning meaningful features if the features do not appear frequently enough. Hence, ERM suffers low test accuracy because it cannot learn rare features. However, Cutout alleviates this challenge by removing some of the strong noise patches, allowing it to learn rare features to some extent.

- We prove the near-perfect performance of CutMix based on a novel technique that views the non-convex loss as a composition of a convex function and reparameterization. This enables us to characterize the global minimum of the loss and show that CutMix forces the model to activate almost uniformly across every patch of inputs, allowing it to learn all features.

## 1.2 Related Works

**Feature Learning Theory.** Our work aligns with a recent line of studies investigating how training methods and neural network architectures influence feature learning. These studies focus on a specific data distribution composed of two components: label-dependent features and label-independent noise. The key contribution of this body of work is the exploration of which training methods or neural networks are most effective at learning meaningful features and achieving good generalization performance. Allen-Zhu and Li (2020) demonstrate that an ensemble model can achieve near-perfect performance by learning diverse features, while a single model tends to learn only certain parts of the feature space, leading to lower test accuracy. In other works, Cao et al. (2022); Kou et al. (2023a) explore the phenomenon of benign overfitting when training a two-layer convolutional neural network. The authors identify the specific conditions under which benign overfitting occurs, providing valuable insights into how these networks behave during training. Several other studies seek to understand various aspects of deep learning through the lens of feature learning (Zou et al., 2021; Jelassi and Li, 2022; Chen et al., 2022, 2023; Li and Li, 2023; Huang et al., 2023a,b).

**Theoretical Analysis of Data Augmentation.** Several works aim to analyze traditional data augmentation from different perspectives, including kernel theory (Dao et al., 2019), margin-based approach (Rajput et al., 2019), regularization effects (Wu et al., 2020), group invariance (Chen et al., 2020b), and impact on optimization (Hanin and Sun, 2021). Moreover, many papers have explored various aspects of a recent technique called Mixup (Zhang et al., 2017). For example, studies have explored its regularization effects (Carratino et al., 2020; Zhang et al., 2020), its role in improving calibration (Zhang et al., 2022), its ability to find optimal decision boundaries (Oh and Yun, 2023) and its potential negative effects (Chidambaram et al., 2021; Chidambaram and Ge, 2024). Some works investigate the broader framework of Mixup, including CutMix, which aligns with the scope of our work. Park et al. (2022) study the regularization effect of mixed-sample data augmentation within a unified framework that contains both Mixup and CutMix. In Oh and Yun (2023), the authors analyze masking-based Mixup, which is a class of Mixup variants that also includes CutMix. In their context, they show that masking-based Mixup can deviate from the Bayes optimal classifier but require less training sample complexity. However, neither work provides a rigorous explanation for why CutMix has been successful. The studies most closely related to our work include Shen et al. (2022); Chidambaram et al. (2023); Zou et al. (2023). Shen et al. (2022) regard traditional data augmentation as a form of feature manipulation and investigate its advantages from a feature learning perspective. Both Chidambaram et al. (2023) and Zou et al. (2023) analyze Mixup within a feature learning framework. However, patch-level data augmentation such as Cutout and CutMix, which are the focus of our work, have not yet been explored within this context.

## 2 Problem Setting

In this section, we introduce the data distribution and neural network architecture, and formally describe the three training methods considered in this paper.

### 2.1 Data Distribution

We consider a binary classification problem on structured data, consisting of patches of label-dependent vectors (referred to as *features*) and label-independent vectors (referred to as *noise*).

**Definition 2.1** (Feature Noise Patch Data). We define a data distribution $\mathcal{D}$ on $\mathbb{R}^{d \times P} \times \{-1, 1\}$ such that $(\boldsymbol{X}, y) \sim \mathcal{D}$ where $\boldsymbol{X} = (\boldsymbol{x}^{(1)}, \dots, \boldsymbol{x}^{(P)}) \in \mathbb{R}^{d \times P}$ and $y \in \{\pm 1\}$ is constructed as follows.

1. Choose the *label* $y \in \{\pm 1\}$ uniformly at random.

2. Let $\{\boldsymbol{v}_{s,k}\}_{s \in \{\pm 1\}, k \in [K]} \subset \mathbb{R}^d$ be a set of orthonormal *feature vectors*. Choose the feature vector $\boldsymbol{v} \in \mathbb{R}^d$ for data point $\boldsymbol{X}$ as $\boldsymbol{v} = \boldsymbol{v}_{y,k}$ with probability $\rho_k$ from $\{\boldsymbol{v}_{y,k}\}_{k \in [K]} \subset \mathbb{R}^d$, where $\rho_1 + \cdots + \rho_K = 1$ and $\rho_1 \geq \cdots \geq \rho_K$. In our setting, there are three types of features with significantly different frequencies: *common features*, *rare features*, and *extremely rare features*, ordered from most to least frequent. The indices of these features partition $[K]$ into $(\mathcal{K}_C, \mathcal{K}_R, \mathcal{K}_E)$.

3. We construct $P$ patches of $\boldsymbol{X}$ as follows.

   - **Feature Patch**: Choose $p^*$ uniformly from $[P]$ and we set $\boldsymbol{x}^{(p^*)} = \boldsymbol{v}$.
   - **Dominant Noise Patch**: Choose $\tilde{p}$ uniformly from $[P] \setminus \{p^*\}$. We construct $\boldsymbol{x}^{(\tilde{p})} = \alpha \boldsymbol{u} + \xi^{(\tilde{p})}$ where $\alpha \boldsymbol{u}$ is *feature noise* drawn uniformly from $\{\alpha \boldsymbol{v}_{1,1}, \alpha \boldsymbol{v}_{-1,1}\}$ with $0 < \alpha < 1$ and $\xi^{(\tilde{p})}$ is Gaussian *dominant noise* drawn from $N(\boldsymbol{0}, \sigma_{\mathrm{d}}^2 \boldsymbol{\Lambda})$.
   - **Background Noise Patch**: The remaining patches $p \in [P] \setminus \{p^*, \tilde{p}\}$ consist of Gaussian *background noise*, i.e., we set $\boldsymbol{x}^{(p)} = \xi^{(p)}$ where $\xi^{(p)} \sim N(\boldsymbol{0}, \sigma_{\mathrm{b}}^2 \boldsymbol{\Lambda})$.

   Here, the noise covariance matrix is defined as $\boldsymbol{\Lambda} := \boldsymbol{I} - \sum_{s,k} \boldsymbol{v}_{s,k} \boldsymbol{v}_{s,k}^\top$ which ensures that Gaussian noises are orthogonal to all features. We assume that the dominant noise is stronger than the background noise, i.e., $\sigma_{\mathrm{b}} < \sigma_{\mathrm{d}}$.

Our data distribution captures characteristics of image data, where the input consists of several patches. Some patches contain information relevant to the image labels, such as cat faces for the label "cat," while other patches contain information irrelevant to the labels, such as the background. Intuitively, there are two ways to fit the given data: learning features or memorizing noise. If a model fits the data by learning features, it can correctly classify test data having the same features. However, if a model fits the data by memorizing noise, it cannot generalize to unseen data because noise patches are not relevant to labels. Thus, learning more features is crucial for achieving better generalization.

In real-world scenarios, different features may appear with varying frequencies. For instance, the occurrences of cat's faces and cat's tails in a dataset might differ significantly, although both are relevant to the "cat" label. Our data distribution reflects these characteristics by considering features with varying frequencies. To emphasize the distinctions between the three training methods we analyze, we categorize features into three groups: common, rare, and extremely rare. We refer to data points containing these features as *common data*, *rare data*, and *extremely rare data*, respectively. We emphasize that these terminologies are chosen merely to distinguish the three different levels of rarity, and even "extremely rare" features appear in a nontrivial fraction of the training data with high probability (see our assumptions in Section 2.4).

**Comparison to Previous Work.** Our data distribution is similar to those considered in Shen et al. (2022) and Zou et al. (2023), which investigate the benefits of standard data augmentation methods and Mixup by comparing them to vanilla training without any augmentation. These results consider two types of features—common and rare—with different levels of rarity, along with two types of noise: feature noise and Gaussian noise. However, we consider three types of features: common, rare, and extremely rare, and three types of noise: feature noise, dominant noise, and background noise. This distinction allows us to compare three distinct methods and demonstrate the differences between them, whereas Shen et al. (2022) and Zou et al. (2023) compared only two methods.

## 2.2 Neural Network Architecture

For the prediction model, we focus on the following two-layer convolutional neural network where the weights in the second layer are fixed at $1$ and $-1$, with only the first layer being trainable. Several works including Shen et al. (2022) and Zou et al. (2023) also focus on similar two-layer convolutional neural networks.

**Definition 2.2** (2-Layer CNN). We define 2-layer CNN $f_{\boldsymbol{W}} : \mathbb{R}^{d \times P} \to \mathbb{R}$ parameterized by $\boldsymbol{W} = \{\boldsymbol{w}_1, \boldsymbol{w}_{-1}\} \in \mathbb{R}^{d \times 2}$. For each input $\boldsymbol{X} = (\boldsymbol{x}^{(1)}, \dots, \boldsymbol{x}^{(P)}) \in \mathbb{R}^{d \times P}$, we define

$$f_{\boldsymbol{W}}(\boldsymbol{X}) := \sum_{p \in [P]} \phi\left(\left\langle \boldsymbol{w}_1, \boldsymbol{x}^{(p)} \right\rangle\right) - \sum_{p \in [P]} \phi\left(\left\langle \boldsymbol{w}_{-1}, \boldsymbol{x}^{(p)} \right\rangle\right),$$

where $\phi(\cdot)$ is a smoothed version of leaky ReLU activation, defined as follows.

$$\phi(z) := \begin{cases} z - \frac{(1-\beta)r}{2} & z \geq r \\ \frac{1-\beta}{2r} z^2 + \beta z & 0 \leq z \leq r \\ \beta z & z \leq 0 \end{cases},$$

where $0 < \beta \leq 1$ and $r > 0$.

Previous works on the theory of feature learning often consider neural networks with (smoothed) ReLU or polynomial activation functions. However, we adopt a smoothed leaky ReLU activation, which always has a positive slope, to exclude the possibility of neurons "dying" during the complex optimization trajectory. Using smoothed leaky ReLU to analyze the learning dynamics of neural networks is not entirely new; there is a body of work that studies phenomena such as benign overfitting (Frei et al., 2022a) and implicit bias (Frei et al., 2022b; Kou et al., 2023b) by analyzing neural networks with (smoothed) leaky ReLU activation.

A key difference between ReLU and leaky ReLU lies in the possibility of ReLU neurons "dying" in the negative region, where some negatively initialized neurons remain unchanged throughout training. As a result, using ReLU activation requires multiple neurons to ensure the survival of neurons at initialization, which becomes increasingly probable as the number of neurons increases. In contrast, the derivative of leaky ReLU is always positive, ensuring that a single neuron is often sufficient. Therefore, for mathematical simplicity, we consider the case where the network has a single neuron for each positive and negative output. We believe that our analysis can be extended to the multi-neuron case as we validate numerically in Appendix A.2.

## 2.3 Training Methods

Using a training set sampled from the distribution $\mathcal{D}$, we would like to train our network $f_{\boldsymbol{W}}$ to learn to correctly classify unseen data points from $\mathcal{D}$. We consider three learning methods: vanilla training without any augmentation, Cutout, and CutMix. We first introduce necessary notation for our data and parameters, and then formalize training methods within our framework.

**Training Data.** We consider a training set $\mathcal{Z} = \{(\boldsymbol{X}_i, y_i)\}_{i \in [n]}$ comprising $n$ data points, each independently drawn from $\mathcal{D}$. For each $i \in [n]$, we denote $\boldsymbol{X}_i = (\boldsymbol{x}_i^{(1)}, \dots, \boldsymbol{x}_i^{(P)})$.

**Initialization.** We initialize the model parameters in our neural network using random initialization. Specifically, we initialize the model parameter $\boldsymbol{W}^{(0)} = \{\boldsymbol{w}_1^{(0)}, \boldsymbol{w}_{-1}^{(0)}\}$, where $\boldsymbol{w}_1^{(0)}, \boldsymbol{w}_{-1}^{(0)} \overset{\text{i.i.d.}}{\sim} N(\boldsymbol{0}, \sigma_0^2 \boldsymbol{I}_d)$. Let us denote updated model parameters at iteration $t$ as $\boldsymbol{W}^{(t)} = \{\boldsymbol{w}_1^{(t)}, \boldsymbol{w}_{-1}^{(t)}\}$.

### 2.3.1 Vanilla Training

The vanilla approach to training a model $f_{\boldsymbol{W}}$ is solving the empirical risk minimization problem using gradient descent. We refer to this method as ERM. Then, ERM updates parameters $\boldsymbol{W}^{(t)}$ of a model using the following rule.

$$\boldsymbol{W}^{(t+1)} = \boldsymbol{W}^{(t)} - \eta \nabla_{\boldsymbol{W}} \mathcal{L}_{\text{ERM}}\left(\boldsymbol{W}^{(t)}\right),$$

where $\eta$ is a learning rate and $\mathcal{L}_{\mathrm{ERM}}(\cdot)$ is the ERM training loss defined as

$$\mathcal{L}_{\mathrm{ERM}}(\boldsymbol{W}) := \frac{1}{n} \sum_{i \in [n]} \ell(y_i f_{\boldsymbol{W}}(\boldsymbol{X}_i)), \tag{1}$$

where $\ell(\cdot)$ is the logistic loss $\ell(z) = \log(1 + e^{-z})$.

### 2.3.2 Cutout Training.

Cutout (DeVries and Taylor, 2017) is a data augmentation technique that randomly cuts out rectangular regions of image inputs. In our patch-wise data, we regard Cutout training as using inputs with masked patches from the original data. For each subset $\mathcal{C}$ of $[P]$ and $i \in [n]$, we define augmented data $\boldsymbol{X}_{i,\mathcal{C}} \in \mathbb{R}^{d \times P}$ as a data point generated by cutting the patches with indices in $\mathcal{C}$ out of $\boldsymbol{X}_i$. We can represent $\boldsymbol{X}_{i,\mathcal{C}}$ as

$$\boldsymbol{X}_{i,\mathcal{C}} = \left( \boldsymbol{x}_{i,\mathcal{C}}^{(1)}, \ldots, \boldsymbol{x}_{i,\mathcal{C}}^{(P)} \right), \text{ where } \boldsymbol{x}_{i,\mathcal{C}}^{(p)} = \begin{cases} \boldsymbol{x}_i^{(p)} & \text{if } p \notin \mathcal{C}, \\ \boldsymbol{0} & \text{otherwise.} \end{cases}$$

Note that the output of the model $f_{\boldsymbol{W}}(\cdot)$ on this augmented data point $\boldsymbol{X}_{i,\mathcal{C}}$ is

$$f_{\boldsymbol{W}}(\boldsymbol{X}_{i,\mathcal{C}}) = \sum_{p \notin \mathcal{C}} \phi\left( \left\langle \boldsymbol{w}_1, \boldsymbol{x}_i^{(p)} \right\rangle \right) - \sum_{p \notin \mathcal{C}} \phi\left( \left\langle \boldsymbol{w}_{-1}, \boldsymbol{x}_i^{(p)} \right\rangle \right).$$

Then, the objective function for Cutout training can be defined as

$$\mathcal{L}_{\mathrm{Cutout}}(\boldsymbol{W}) := \frac{1}{n} \sum_{i \in [n]} \mathbb{E}_{\mathcal{C} \sim \mathcal{D}_{\mathcal{C}}} [\ell(y_i f_{\boldsymbol{W}}(\boldsymbol{X}_{i,\mathcal{C}}))],$$

where $\mathcal{D}_{\mathcal{C}}$ is a uniform distribution on the collection of subsets of $[P]$ with cardinality $C$, where $C$ is a hyperparameter satisfying $1 \le C < \frac{P}{2}$.[1] We refer to the process of training our model using gradient descent on Cutout loss $\mathcal{L}_{\mathrm{Cutout}}(\boldsymbol{W})$ as Cutout, and its update rule is

$$\boldsymbol{W}^{(t+1)} = \boldsymbol{W}^{(t)} - \eta \nabla_{\boldsymbol{W}} \mathcal{L}_{\mathrm{Cutout}}\left( \boldsymbol{W}^{(t)} \right), \tag{2}$$

where $\eta$ is a learning rate.

### 2.3.3 CutMix Training.

CutMix (Yun et al., 2019) involves not only cutting parts of images, but also pasting them into different images as well as assigning them mixed labels. For each subset $\mathcal{S}$ of $[P]$ and $i, j \in [n]$, we define the augmented data point $\boldsymbol{X}_{i,j,S} \in \mathbb{R}^{d \times P}$ as the data obtained by cutting patches with indices in $\mathcal{S}$ from data $\boldsymbol{X}_i$ and pasting them into $\boldsymbol{X}_j$ at the same indices $\mathcal{S}$. We can write $\boldsymbol{X}_{i,j,S}$ as

$$\boldsymbol{X}_{i,j,S} = \left( \boldsymbol{x}_{i,j,S}^{(1)}, \ldots, \boldsymbol{x}_{i,j,S}^{(P)} \right), \text{ where } \boldsymbol{x}_{i,j,S}^{(p)} = \begin{cases} \boldsymbol{x}_i^{(p)} & \text{if } p \in \mathcal{S}, \\ \boldsymbol{x}_j^{(p)} & \text{otherwise.} \end{cases}$$

The one-hot encoding of the labels $y_i$ and $y_j$ are also mixed with proportions $\frac{|\mathcal{S}|}{P}$ and $1 - \frac{|\mathcal{S}|}{P}$, respectively. This mixed label results in the loss of the form

$$\frac{|\mathcal{S}|}{P} \ell(y_i f_{\boldsymbol{W}}(\boldsymbol{X}_{i,j,S})) + \left(1 - \frac{|\mathcal{S}|}{P}\right) \ell(y_j f_{\boldsymbol{W}}(\boldsymbol{X}_{i,j,S})).$$

From this, the CutMix training loss $\mathcal{L}_{\mathrm{CutMix}}(\boldsymbol{W})$ can be defined as

$$\mathcal{L}_{\mathrm{CutMix}}(\boldsymbol{W}) := \frac{1}{n^2} \sum_{i,j \in [n]} \mathbb{E}_{\mathcal{S} \sim \mathcal{D}_{\mathcal{S}}} \left[ \frac{|\mathcal{S}|}{P} \ell(y_i f_{\boldsymbol{W}}(\boldsymbol{X}_{i,j,S})) + \left(1 - \frac{|\mathcal{S}|}{P}\right) \ell(y_j f_{\boldsymbol{W}}(\boldsymbol{X}_{i,j,S})) \right],$$

where $\mathcal{D}_{\mathcal{S}}$ is a probability distribution on the set of subsets of $[P]$ which samples $\mathcal{S} \sim \mathcal{D}_{\mathcal{S}}$ as follows.[2]

---

[1]DeVries and Taylor (2017) also employ a moderate size of cutting, such as cutting $16 \times 16$ pixels on CIFAR-10 data, which originally has $32 \times 32$ pixels.

[2]Other types of distributions, such as those considered in Yun et al. (2019), make the same conclusion. We adopt this distribution to make presentation simpler.

1. Choose the cardinality $s$ of $\mathcal{S}$ uniformly at random from $\{0, 1, \dots, P\}$, and

2. Choose $\mathcal{S}$ uniformly at random from the collection of subsets of $[P]$ with cardinality $s$.

We refer to the process of training our network using gradient descent on CutMix loss $\mathcal{L}_{\text{CutMix}}(\boldsymbol{W})$ as CutMix, and its update rule is

$$\boldsymbol{W}^{(t+1)} = \boldsymbol{W}^{(t)} - \eta \nabla_{\boldsymbol{W}} \mathcal{L}_{\text{CutMix}} \left( \boldsymbol{W}^{(t)} \right), \tag{3}$$

where $\eta$ is a learning rate.

## 2.4 Assumptions on the Choice of Problem Parameters

To control the quantities that appear in the analysis of training dynamics, we make assumptions on several quantities in our problem setting. For simplicity, we use choices of problem parameters as a function of the dimension of patches $d$ and consider sufficiently large $d$.

We use the standard asymptotic notation $\mathcal{O}(\cdot), \Omega(\cdot), \Theta(\cdot), o(\cdot), \omega(\cdot)$ to express the dependency on $d$. We also use $\widetilde{\mathcal{O}}(\cdot), \widetilde{\Omega}(\cdot), \widetilde{\Theta}(\cdot)$ to hide logarithmic factors of $d$. Additionally, $\text{poly}(d)$ (or $\text{polylog}(d)$) represents quantities that increase faster than $d^{c_1}$ (or $(\log d)^{c_1}$) and slower than $d^{c_2}$ (or $(\log d)^{c_2}$) for some constant $0 < c_1 < c_2$. Similarly, $o(1/\text{poly}(d))$ (or $o(1/\text{polylog}(d))$) denotes some quantities that decrease faster than $1/d^c$ (or $1/(\log d)^c$) for any constant $c$. Finally, we use $f(d) = o(g(d)/\text{polylog}(d))$ when $f(d)/g(d) = o(1/\text{polylog}(d))$ for some function $f$ and $g$ of $d$.

**Assumptions.** We assume that $P = \Theta(1)$ and $P \geq 8$ for simplicity. Additionally, we consider a high-dimensional regime where the number of data points is much smaller than the dimension $d$, which is expressed as $n = o\left(\alpha \beta \sigma_{\text{d}}^{-1} \sigma_{\text{b}} d^{\frac{1}{2}} / \text{polylog}(d)\right)$. We also assume that $\rho_k n = \omega\left(n^{\frac{1}{2}} \log d\right)$ for all $k \in [K]$, which ensures the sufficiency of data points with each feature.

In addition, as we will describe in Section 4, the relative scales between the frequencies of features and the strengths of noises play crucial roles in our analysis, as they serve as a proxy for the "learning speed" in the initial phase. For common features $k \in \mathcal{K}_C$, we assume $\rho_k = \Theta(1)$ and the learning speed of common features is much faster than that of dominant noise, which translates into the assumption $\sigma_{\text{d}}^2 d = o(\beta n)$. For rare features $k \in \mathcal{K}_R$, we assume $\rho_k = \Theta(\rho_R)$ for some $\rho_R$, and we consider the case where the learning speed of rare features is much slower than that of dominant noise but faster than background noise, which is expressed as $\rho_R n = o\left(\alpha^2 \sigma_{\text{d}}^2 d / \text{polylog}(d)\right)$ and $\sigma_{\text{b}}^2 d = o(\beta \rho_R n)$. Finally, for extremely rare features $k \in \mathcal{K}_E$, we say $\rho_k = \Theta(\rho_E)$ for some $\rho_E$ and their learning is even slower than that of background noises, which can be expressed as $\rho_E n = o\left(\alpha^2 \sigma_{\text{b}}^2 d / \text{polylog}(d)\right)$.

Lastly, we assume the strength of feature noise satisfies $\alpha = o\left(n^{-1} \beta \sigma_{\text{d}}^2 d / \text{polylog}(d)\right)$, and $r, \sigma_0, \eta > 0$ are sufficiently small so that $\sigma_0, r = o(\alpha/\text{polylog}(d))$, $\eta = o\left(r \sigma_{\text{d}}^{-2} d^{-1} / \text{polylog}(d)\right)$.

We list our assumptions in Assumption B.1 and there are many choices of parameters satisfying the set of assumptions, including:

$$P = 8, C = 2, n = \Theta\left(d^{0.4}\right), \alpha = \Theta\left(d^{-0.02}\right), \beta = \frac{1}{\text{polylog}(d)}, \sigma_0 = \Theta(d^{-0.2}), r = \Theta(d^{-0.2}),$$

$$\sigma_{\text{d}} = \Theta\left(d^{-0.305}\right), \sigma_{\text{b}} = \Theta\left(d^{-0.375}\right), \rho_R = \Theta\left(d^{-0.1}\right), \rho_E = \Theta\left(d^{-0.195}\right), \eta = \Theta(d^{-1}).$$

## 3 Main Results

In this section, we provide a characterization of the high probability guarantees for the behavior of models trained using three distinct methods we have introduced. We denote by $T^*$ the maximum admissible training iterates and we assume $T^* = \frac{\text{poly}(d)}{\eta}$ with a sufficiently large polynomial in $d$. In all of our theorem statements, the randomness is over the sampling of training data and the initialization of models and all results hold under the condition that $d$ is sufficiently large.

The following theorem characterizes training accuracy and test accuracy achieved by ERM.

**Theorem 3.1.** *Let $\boldsymbol{W}^{(t)}$ be iterates of* ERM. *Then with probability at least $1 - o\left(\frac{1}{\mathrm{poly}(d)}\right)$, there exists $T_{\mathrm{ERM}}$ such that any $T \in [T_{\mathrm{ERM}}, T^*]$ satisfies the following:*

- *(Perfectly fits training set): For all $i \in [n]$, $y_i f_{\boldsymbol{W}^{(T)}}(\boldsymbol{X}_i) > 0$.*

- *(Random on (extremely) rare data): $\mathbb{P}_{(\boldsymbol{X},y) \sim \mathcal{D}}\left[y f_{\boldsymbol{W}^{(T)}}(\boldsymbol{X}) > 0\right] = 1 - \frac{1}{2} \sum\limits_{k \in \mathcal{K}_R \cup \mathcal{K}_E} \rho_k \pm o\left(\frac{1}{\mathrm{poly}(d)}\right).$*

The proof is provided in Appendix C.2. Theorem 3.1 demonstrates that ERM achieves perfect training accuracy; however, it performs almost like random guessing on unseen data points with rare and extremely rare features. This is because ERM can only learn common features and overfit rare or extremely rare data in the training set by memorizing noises to achieve perfect training accuracy.

In comparison, we show that Cutout can perfectly fit both augmented training data and original training data, and it can also learn rare features that ERM cannot. However, Cutout still makes random guesses on test data with extremely rare features. We state these in the following theorem with the proof provided in Appendix D.2:

**Theorem 3.2.** *Let $\boldsymbol{W}^{(t)}$ be iterates of* Cutout *training. Then with probability at least $1 - o\left(\frac{1}{\mathrm{poly}(d)}\right)$, there exists $T_{\mathrm{Cutout}}$ such that any $T \in [T_{\mathrm{Cutout}}, T^*]$ satisfies the following:*

- *(Perfectly fits augmented data): For all $i \in [n]$ and $\mathcal{C} \subset [P]$ with $|\mathcal{C}| = C$, $y_i f_{\boldsymbol{W}^{(T)}}(\boldsymbol{X}_{i,\mathcal{C}}) > 0$.*

- *(Perfectly fits original training data): For all $i \in [n]$, $y_i f_{\boldsymbol{W}^{(T)}}(\boldsymbol{X}_i) > 0$.*

- *(Random on extremely rare data): $\mathbb{P}_{(\boldsymbol{X},y) \sim \mathcal{D}}\left[y f_{\boldsymbol{W}^{(T)}}(\boldsymbol{X}) > 0\right] = 1 - \frac{1}{2} \sum\limits_{k \in \mathcal{K}_E} \rho_k \pm o\left(\frac{1}{\mathrm{poly}(d)}\right).$*

In the case of CutMix, it is challenging to discuss training accuracy directly because the augmented data have soft labels generated by mixing pairs of labels. Instead, we prove that CutMix achieves a sufficiently small gradient of the loss, and the training accuracy on the original training data is perfect. We also demonstrate that CutMix achieves almost perfect test accuracy, as it learns all types of features regardless of rarity.

**Theorem 3.3.** *Let $\boldsymbol{W}^{(t)}$ be iterates of* CutMix *training. Then with probability at least $1 - o\left(\frac{1}{\mathrm{poly}(d)}\right)$, there exists some $T_{\mathrm{CutMix}} \in [0, T^*]$ that satisfies the following:*

- *(Finds a near stationary point): $\left\|\nabla_{\boldsymbol{W}} \mathcal{L}_{\mathrm{CutMix}}\left(\boldsymbol{W}^{(T_{\mathrm{CutMix}})}\right)\right\| = \frac{1}{\mathrm{poly}(d)}$.*

- *(Perfectly fits original training data): For all $i \in [n]$, $y_i f_{\boldsymbol{W}^{(T_{\mathrm{CutMix}})}}(\boldsymbol{X}_i) > 0$.*

- *(Almost perfectly classifies test data): $\mathbb{P}_{(\boldsymbol{X},y) \sim \mathcal{D}}\left[y f_{\boldsymbol{W}^{(T_{\mathrm{CutMix}})}}(\boldsymbol{X}) > 0\right] = 1 - o\left(\frac{1}{\mathrm{poly}(d)}\right).$*

To prove Theorem 3.3, we characterize the global minimum of objective loss of CutMix. Surprisingly, at the global minimum, the model has the same outputs for all patches of the input data. In other words, the contributions of all feature vectors and noise vectors to the final outcome of the network are identical, regardless of their frequency and strength (see Section 4.2 for more details). Moreover, this uniform "contribution" is large enough, which allows the model to learn all types of features by reaching the global minimum. We provide the detailed proof in Appendix E.2.

Our three main theorems elucidate the benefits of Cutout and CutMix. Cutout enables a model to learn rarer features than ERM, while CutMix can even outperform Cutout. These advantages in learning rarer features lead to improvements in generalization performance.

## 4 Overview of Analysis

In this section, we discuss key proof ideas and the main challenges in our analysis. For ease of presentation, we consider the case $\alpha = 0$. Although our assumptions do not allow the choice $\alpha = 0$, the choice of nonzero $\alpha$ is to show guarantees on the test accuracy and does not significantly affect the feature learning aspect.

To provide the proof overview, let us introduce some additional notation. For each $i \in [n]$, recall that the corresponding input point can be written as $\boldsymbol{X}_i = (\boldsymbol{x}_i^{(1)}, \ldots, \boldsymbol{x}_i^{(P)})$. We use $p_i^*$ and $\tilde{p}_i$ to denote the indices of its feature patch and dominant noise patch, respectively. For each feature vector $\boldsymbol{v}_{s,k}$, where $s \in \{\pm 1\}$ and $k \in [K]$, let $\mathcal{V}_{s,k} \subset [n]$ represent the set of indices of data points having the feature vector $\boldsymbol{v}_{s,k}$, and $\mathcal{V}_s = \bigcup_{k=1}^{K} \mathcal{V}_{s,k}$ denotes the set of indices of data with label $s$. For each data point $i \in [n]$ and dominant or background noise patch $p \in [P] \setminus \{p_i^*\}$, we refer to the Gaussian noise inside $\boldsymbol{x}_i^{(p)}$ as $\xi_i^{(p)}$.

## 4.1 Vanilla Training and Cutout Training

We now explain why ERM fails to learn (extremely) rare features, while Cutout can learn rare features but not extremely rare features. Let us consider ERM. From (1), for $s, s' \in \{\pm 1\}, k \in [K], i \in [n]$ and $p \in [P] \setminus \{p_i^*\}$, the component of $\boldsymbol{w}_s$ in the feature vector $\boldsymbol{v}_{s',k}$'s direction is updated as

$$\left\langle \boldsymbol{w}_s^{(t+1)}, \boldsymbol{v}_{s',k} \right\rangle = \left\langle \boldsymbol{w}_s^{(t)}, \boldsymbol{v}_{s',k} \right\rangle - \frac{ss'\eta}{n} \sum_{j \in \mathcal{V}_{s',k}} \ell'(y_j f_{\boldsymbol{W}^{(t)}}(\boldsymbol{X}_j)) \phi'\left( \left\langle \boldsymbol{w}_s^{(t)}, \boldsymbol{v}_{s',k} \right\rangle \right), \quad (4)$$

and similarly, the "update" of inner product of $\boldsymbol{w}_s$ with a noise patch $\xi_i^{(p)}$ can be written as

$$\left\langle \boldsymbol{w}_s^{(t+1)}, \xi_i^{(p)} \right\rangle \approx \left\langle \boldsymbol{w}_s^{(t)}, \xi_i^{(p)} \right\rangle - \frac{sy_i\eta}{n} \ell'(y_i f_{\boldsymbol{W}^{(t)}}(\boldsymbol{X}_i)) \phi'\left( \left\langle \boldsymbol{w}_s^{(t)}, \xi_i^{(p)} \right\rangle \right) \left\| \xi_i^{(p)} \right\|^2, \quad (5)$$

where the approximation is due to the near-orthogonality of Gaussian random vectors in the high-dimensional regime. This approximation shows that $\langle \boldsymbol{w}_s^{(t+1)}, \boldsymbol{v}_{s',k} \rangle$'s and $\langle \boldsymbol{w}_s^{(t)}, \xi_i^{(p)} \rangle$'s are almost monotonically increasing or decreasing. We address the approximation errors using a variant of the technique introduced by Cao et al. (2022), as detailed in Appendix B.3.

From (4) and (5), we can observe that in the early phase of training satisfying $-\ell'(y_i f_{\boldsymbol{W}^{(t)}}(\boldsymbol{X}_i)) = \Theta(1)$, the main factor for the speed of learning features and noises are the number of feature occurrence $|\mathcal{V}_{s',k}|$ and the strength of noise $\|\xi_i^{(p)}\|^2$. From our assumptions introduced in Section 2.4, if we compare the learning speed of different components, we have

common features $\gg$ dominant noises $\gg$ rare features $\gg$ background noises $\gg$ extremely rare features,

in terms of "learning speed." Based on this observation, we conduct a three-phase analysis for ERM.

- **Phase 1**: Learning common features quickly.
- **Phase 2**: Fitting (extremely) rare data by memorizing dominant noises instead of learning features.
- **Phase 3**: A model cannot learn (extremely) rare features since gradients of all data are small.

The main intuition behind why ERM cannot learn (extremely) rare features is that the gradients of all data containing these features become small after quickly memorizing dominant noise patches. In contrast, since Cutout randomly cuts some patches out, there exist augmented data points that do not contain dominant noises and have only features and background noises. This allows Cutout to learn rare features, thanks to these augmented data. However, extremely rare features cannot be learned since the learning speed of background noise is much faster and there are too many background noise patches to cut them all out.

*Remark* 4.1. Shen et al. (2022) conduct analysis on vanilla training and training using standard data augmentation, sharing the same intuition in similar but different data models and neural networks. Also, we emphasize that we proved the model cannot learn (extremely) rare features even if we run $\frac{\text{poly}(d)}{\eta}$ iterations of GD, whereas Shen et al. (2022) only consider the first iteration that achieves perfect training accuracy.

**Practical Insights.**   In practice, images contain features and noise across several patches. A larger cutting size can be more effective in removing noise but may also remove important features that the model needs to learn. Thus, there is a trade-off in choosing the optimal cutting size, a trend also observed in DeVries and Taylor (2017). One limitation of Cutout is that it may not effectively remove dominant noise. Thus, dominant noise can persist in the augmented data, leading to potential noise memorization. We believe that developing strategies that can more precisely detect and remove these noise components from the image input could enhance the effectiveness of these methods.

## 4.2 CutMix Training

In learning dynamics of ERM and Cutout, inner products between weight and data patches evolve (approximately) monotonically, which makes the analysis much more feasible. However, analyzing the learning dynamics of CutMix involves non-monotone change of inner products, which is inevitable since CutMix uses mixed labels; this is also demonstrated in our experimental results (Section 5,especially the leftmost plot in Figure 1). Non-monotonicity and non-convexity of the problem necessitates novel proof strategies.

Let us define $\boldsymbol{Z} := \{z_{s,k}\}_{s\in\{\pm1\},k\in[K]} \cup \{z_i^{(p)}\}_{i\in[n],p\in[P]\setminus\{p_i^*\}}$ as a function of $\boldsymbol{W}$ as follows,

$$z_i^{(p)} := \phi\left(\left\langle \boldsymbol{w}_1, \xi_i^{(p)}\right\rangle\right) - \phi\left(\left\langle \boldsymbol{w}_{-1}, \xi_i^{(p)}\right\rangle\right), \quad z_{s,k} := \phi(\langle\boldsymbol{w}_1, \boldsymbol{v}_{s,k}\rangle) - \phi(\langle\boldsymbol{w}_{-1}, \boldsymbol{v}_{s,k}\rangle).$$

Then, $\boldsymbol{Z}$ represents the contribution of each noise patch and feature vector to the neural network output, and the nonconvex function $\mathcal{L}_{\text{CutMix}}(\boldsymbol{W})$ can be viewed as the composition of $\boldsymbol{Z}(\boldsymbol{W})$ and a convex function $h(\boldsymbol{Z})$. By using the convexity of $h(\boldsymbol{Z})$, we can characterize the global minimum of $\mathcal{L}_{\text{CutMix}}(\boldsymbol{W})$. Surprisingly, we show that any global minimizer $\boldsymbol{W}^* = \{\boldsymbol{w}_1^*, \boldsymbol{w}_{-1}^*\}$ satisfies

$$\phi\left(\left\langle \boldsymbol{w}_s^*, \boldsymbol{x}_i^{(p)}\right\rangle\right) - \phi\left(\left\langle \boldsymbol{w}_{-s}^*, \boldsymbol{x}_i^{(p)}\right\rangle\right) = C_s,$$

for all $s \in \{\pm1\}, i \in \mathcal{V}_s$, and $p \in [P]$, with some constants $C_1, C_{-1} = \Theta(1)$. In other words, at the global minimum, the output of model on each patch of the training data is uniform across the set of data with the same labels. We also prove that CutMix can achieve a point close to the global minimum within $\frac{\text{poly}(d)}{\eta}$ iterations. As a result, the model trained by CutMix can learn all features including extremely rare features. The complete proof of Theorem 3.3 appears in Appendix E.2.

*Remark* 4.2. Zou et al. (2023) investigate Mixup in a similar feature-noise model and show that Mixup can learn rarer features than vanilla training, with its benefits emerging from the early dynamics of training. However, our characterization of the global minimum of $\mathcal{L}_{\text{CutMix}}(\boldsymbol{W})$ and experimental results in our setting (Section 5, Figure 1) suggest that the benefits of CutMix, especially for learning extremely rare features, arise from the later stages of training. This suggests that Mixup and CutMix have different underlying mechanisms for promoting feature learning.

**Practical Insights.** The main underlying mechanism of CutMix is that it learns information almost uniformly from all patches in the training data. However, this approach also involves memorizing noise, which can potentially degrade performance in real-world scenarios. We believe that a more sophisticated strategy such as considering the positional information of patches as used in Puzzle Mix (Kim et al., 2020) or Co-Mixup (Kim et al., 2021) could improve the ability to learn more from patches containing features and reduce the impact of noise.

## 5 Experiments

We conduct experiments both in our setting and real-world data CIFAR-10 to support our theoretical findings and intuition. We defer CIFAR-10 experiment results to Appendix A.1.

For the numerical experiments on our setting, we set the number of patches $P = 3$, dimension $d = 2000$, number of data points $n = 300$, dominant noise strength $\sigma_{\text{d}} = 0.25$, background noise strength $\sigma_{\text{b}} = 0.15$, and feature noise strength $\alpha = 0.005$. The feature vectors are given as the standard basis $\boldsymbol{e}_1, \boldsymbol{e}_2, \boldsymbol{e}_3, \boldsymbol{e}_4, \boldsymbol{e}_5, \boldsymbol{e}_6 \in \mathbb{R}^d$, where $\boldsymbol{e}_1, \boldsymbol{e}_2, \boldsymbol{e}_3$ are features for the positive label $y = 1$ and $\boldsymbol{e}_4, \boldsymbol{e}_5, \boldsymbol{e}_6$ are features for the negative label $y = -1$. We categorize $\boldsymbol{e}_1$ and $\boldsymbol{e}_4$ as common features with a frequency of $0.8$, $\boldsymbol{e}_2$ and $\boldsymbol{e}_5$ as rare features with a frequency of $0.15$, and lastly, $\boldsymbol{e}_3$ and $\boldsymbol{e}_6$ as extremely rare features with a frequency of $0.05$. For the learner network, we set the slope of negative regime $\beta = 0.1$ and the length of the smoothed interval $r = 1$. We train models using three methods: ERM, Cutout, and CutMix with a learning rate $\eta = 1$. For Cutout, we cut a single patch of data ($C = 1$). We apply full-batch gradient descent for all methods; for Cutout and CutMix, we utilize all possible augmented data points.[3] We note that this choice of problem parameters does not exactly match the technical assumptions in Section 2.4. However, we empirically observe the same conclusions, which suggests that our analysis could be extended beyond our assumptions.

---

[3]For CutMix, this may induce different choices of $\mathcal{D}_{\mathcal{S}}$ from those assumed in our analysis, but we mention that other general choices of $\mathcal{D}_{\mathcal{S}}$ do not alter the conclusions in our analysis.

For each feature vector $\boldsymbol{v}$ of the positive label, we plot the output of the learned filters for the feature vector $\phi(\langle \boldsymbol{w}_1^{(t)}, \boldsymbol{v} \rangle) - \phi(\langle \boldsymbol{w}_{-1}^{(t)}, \boldsymbol{v} \rangle)$ throughout training in Figure 1. Our numerical findings confirm that ERM can only learn common features, Cutout can learn common and rare features but cannot learn extremely rare features, and CutMix can learn all types of features. Especially, CutMix learn common features, rare features, and extremely rare features almost evenly. Also, we observed non-monotone behavior of the output in the case of CutMix, which motivated our novel proof technique. The same trends are observed with different architectures, such as a smoothed (leaky) ReLU network with multiple neurons, as detailed in Appendix A.2.

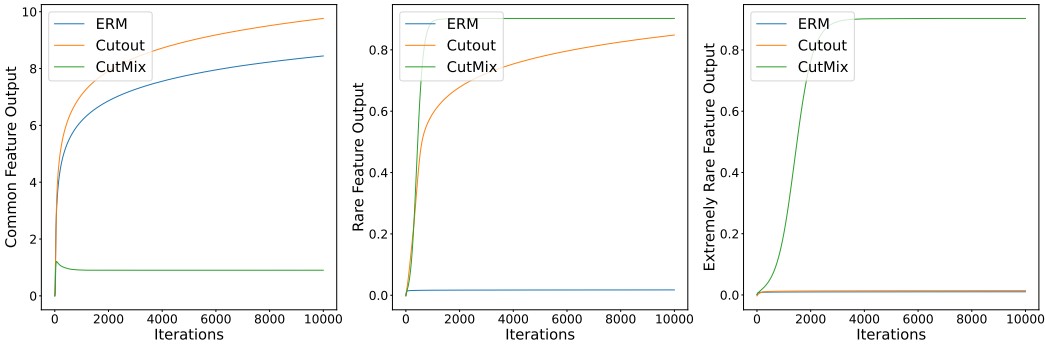

Figure 1: Numerical results on our problem setting. We validate our findings on the trends of ERM, Cutout, and CutMix in learning common feature (Left), rare feature (Center), and extremely rare feature (Right). The output of the common feature trained by CutMix shows non-monotone behavior.

# 6    Conclusion

We studied how Cutout and CutMix influence the ability to learn features in a patch-wise feature-noise data model learning with two-layer convolutional neural networks by comparing them with vanilla training. We showed that Cutout enables the learning of rare features that cannot be learned through vanilla training by mitigating the problem of memorizing label-independent noises instead of learning label-dependent features. Surprisingly, we further proved that CutMix can learn extremely rare features that Cutout cannot learn. We also present our theoretical insights on the underlying mechanism of these methods and provide experimental support.

**Limitation and Future Work.**  Our work has some limitations related to the neural network architecture, specifically, the use of a 2-layer two-neuron smoothed leaky ReLU network. Extending our results to neural networks with deeper, wider, and more general activation functions is a direction for future work. Another future direction is to develop patch-level data augmentation based on our theoretical findings. Also, it would be interesting to perform theoretical analysis on state-of-the-art patch-level data augmentation such as Puzzle Mix (Kim et al., 2020) or Co-Mixup (Kim et al., 2021). These methods utilize patch location information, thus it may require the development of a theoretical framework capturing more complex characteristics of image data.

# Acknowledgement

This work was supported by three Institute of Information & communications Technology Planning & Evaluation (IITP) grants (No. RS-2019-II190075, Artificial Intelligence Graduate School Program (KAIST); No. RS-2022-II220184, Development and Study of AI Technologies to Inexpensively Conform to Evolving Policy on Ethics; No. RS-2024-00457882, AI Research Hub Project) funded by the Korean government (MSIT), and a National Research Foundation of Korea (NRF) grant (No. RS-2019-NR040050) funded by the Korean government (MSIT). CY acknowledges support from a grant funded by Samsung Electronics Co., Ltd.

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

# Contents

# A  Additional Experimental Results

For all experiments described in this section and in Section 5, we use NVIDIA RTX A6000 GPUs.

## A.1  Experiments on CIFAR-10 Dataset

### A.1.1  Experimental Support for Our Intuition

We compare three methods, ERM training, Cutout training, and CutMix training on CIFAR-10 classification. For ERM training, we apply only random cropping and random horizontal flipping on train dataset. In comparison, for Cutout training and CutMix training, we additionally apply Cutout and CutMix, respectively, on training data. For Cutout training, we randomly cut $16 \times 16$ pixels of input images, and for CutMix training, we sample the mixing ratio from a beta distribution $\text{Beta}(0.5, 0.5)$. We train ResNet-18 (He et al., 2016) for 200 epochs with a batch size of 128 using SGD with a learning rate 0.1, momentum 0.9, and weight decay $5 \times 10^{-4}$. Trained models using ERM, Cutout, and CutMix achieve test accuracy 95.16%, 96.05%, and 96.29%, respectively.

We randomly generate augmented data using CutMix from pairs of cat images and dog images in CIFAR-10 with varying mixing ratios $\lambda = 1, 0.8, 0.6$ (Dog:Cat $= \lambda : 1 - \lambda$). We randomly make $5,000$ (cat, dot)-pairs in CIFAR-10 training set and apply CutMix randomly 10 times. By repeating this procedure 10 times, we generate total $5,000 \times 10 \times 10 = 500,000$ augmented samples for each mixing ratio $\lambda$. We plot a histogram of dog prediction output subtracted by cat prediction output (before applying the softmax function), evaluated on $500,000$ augmented data in Figure 2.

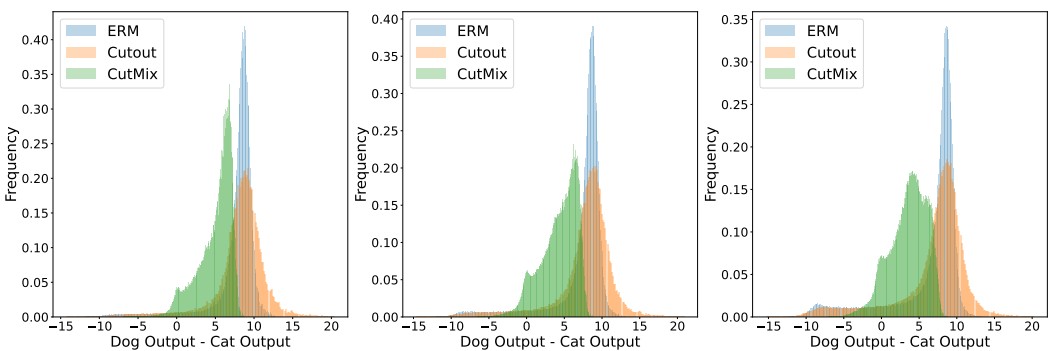

Figure 2: Histogram of dog prediction output subtracted by cat prediction output evaluated on data points augmented by CutMix data using cat data and dog data with varying mixing ratio $\lambda$ (Dog : Cat $= \lambda : 1 - \lambda$) (Left) $\lambda = 1$ , (Center) $\lambda = 0.8$, (Right) $\lambda = 0.6$

The leftmost plot represents the evaluation results for original dog images, as it uses a mixing ratio of $\lambda = 1$. We can observe that the output of the model trained using Cutout is skewed toward higher values compared to the output of the model trained using other methods. We believe this aligns with the theoretical intuition that Cutout learns more information from the original image using augmented data.

The remaining two plots show the output for randomly augmented data using CutMix. We observe that the models trained with CutMix exhibit a shorter tail, supporting our intuition from the CutMix analysis that the models learn uniformly across all patches.

### A.1.2  Experimental Support for Our Findings

We train ResNet-18 using ERM training, Cutout training, and CutMix training following the same experimental details described in Appendix A.1.1, except using only 10% of the training set. This data-hungry setting is intended to highlight the benefits of Cutout and CutMix. We then evaluated the trained models on the remaining 90% of the CIFAR-10 training dataset. The reason for evaluating the remaining training dataset is to analyze the misclassified data using C-score (Jiang et al., 2021), which is publicly available only for the training dataset.

C-score measures the structural regularity of data, with lower values indicating examples that are more difficult to classify correctly. In our framework, data with harder-to-learn features (corresponding to rarer features) would likely have lower C-scores. Since directly extracting and quantitatively evaluating features learned by the models is challenging, we use the C-score as a proxy to evaluate the misclassified data across models trained by ERM, Cutout, and CutMix.

Table 1 illustrates that Cutout tends to misclassify data with lower C-scores compared to ERM, indicating that Cutout learns more hard-to-learn features than vanilla training. Furthermore, the data misclassified by CutMix has even lower C-scores than those misclassified by Cutout, suggesting that CutMix is effective at learning features that are the most challenging to classify. This observation aligns with our theoretical findings, demonstrating that CutMix captures even more difficult features compared to both ERM and Cutout.

Table 1: Mean and quantiles of the C-score on misclassified data across models trained with ERM, Cutout, and CutMix. The results indicate that Cutout tends to misclassify data with lower C-scores compared to ERM, while CutMix exhibits even lower C-scores.

| Method | Mean | Q1 | Q2 | Q3 |
|--------|------|------|------|------|
| **ERM** | 0.687 | 0.615 | 0.782 | 0.841 |
| **Cutout** | 0.679 | 0.599 | 0.775 | 0.837 |
| **CutMix** | 0.670 | 0.575 | 0.767 | 0.835 |

Since directly visualizing features learned by a model is challenging, we present data that were misclassified by the model trained with ERM but correctly classified by the model trained with Cutout instead. In Figure 3, we show 7 samples per class with the lowest C-scores, which are considered to have rare features. Similarly, we also visualize data misclassified by the model trained with Cutout but correctly classified by the model trained with CutMix to represent extremely rare data in Figure 4. This approach allows us to interpret some (extremely) rare features in CIFAR-10, such as frogs with unusual colors.

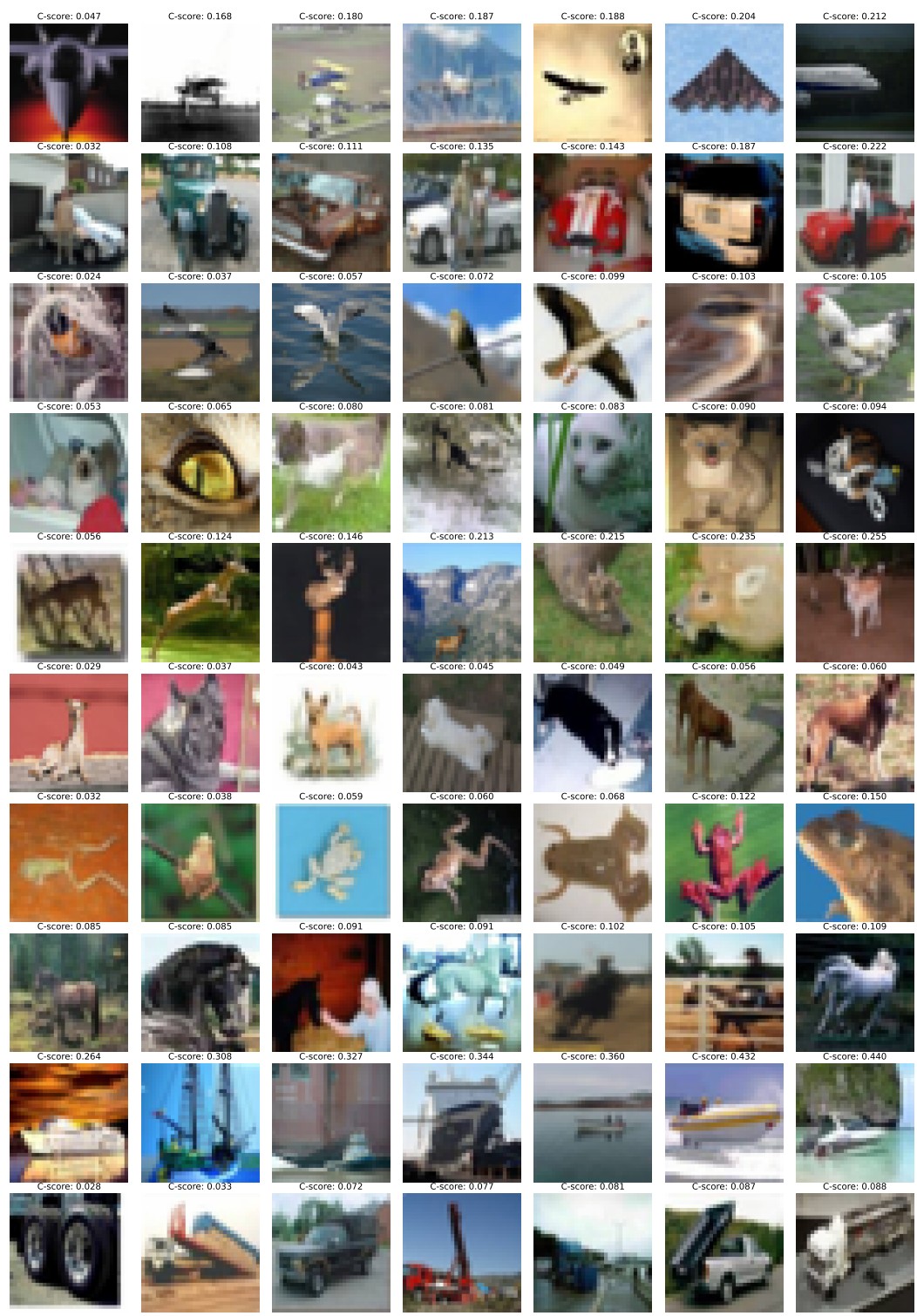

Figure 3: Examples of rare data in CIFAR-10

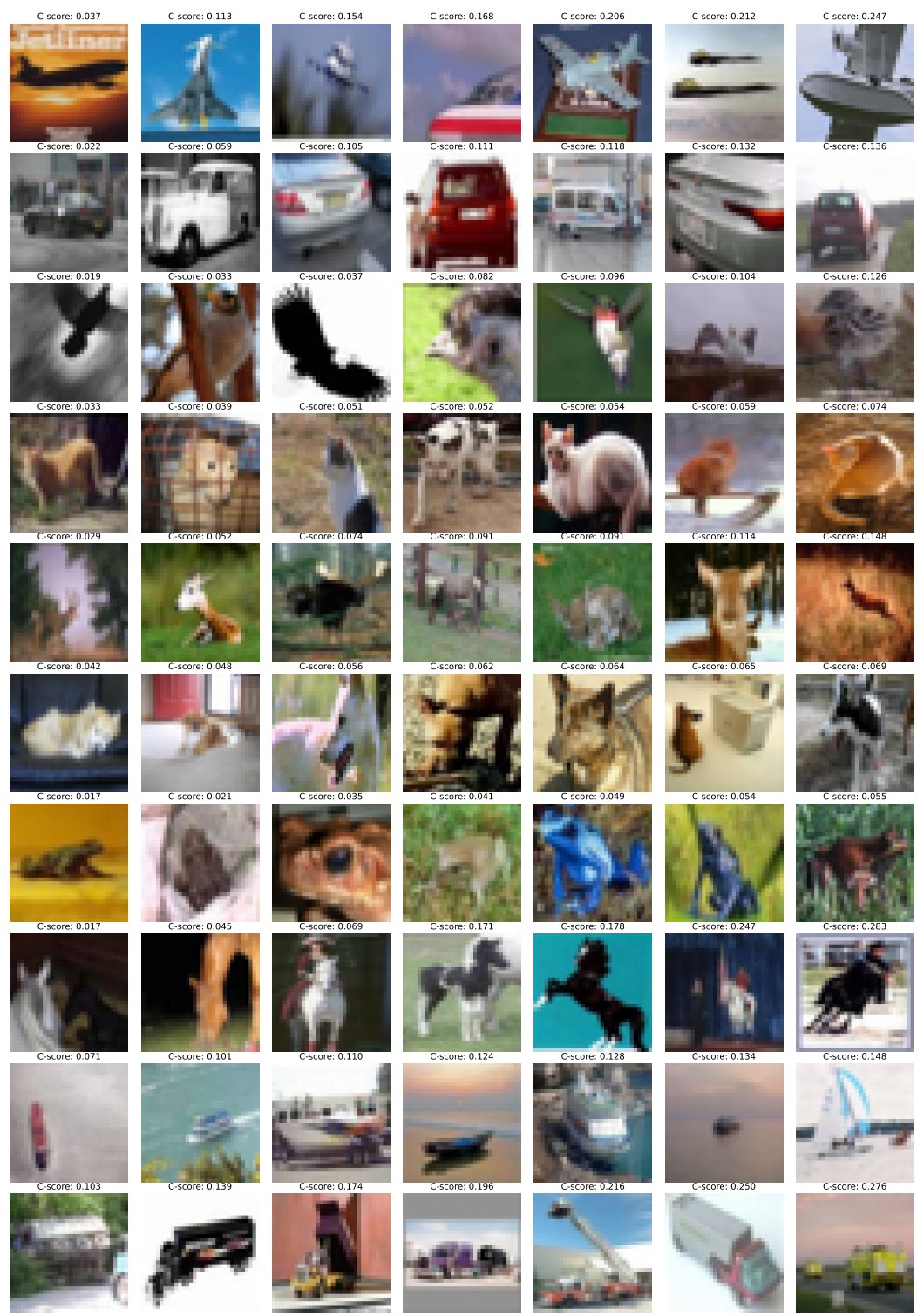

Figure 4: Examples of extreme data in CIFAR-10

## A.2 Additional Experimental Results on Our Data Distribution

In addition to the results described in Section 5, we further conducted numerical experiments on our data distribution by applying two variations to our architecture: increasing the number of neurons, and increasing the number of neurons with a smoothed ReLU activation (instead of smoothed leaky ReLU). We observed the same trends as predicted by our theoretical findings and shown in Figure 1.

Let us describe the setting of our experiments in detail. In both cases, We set the number of patches $P = 3$, dimension $d = 2000$, and the number of data $n = 300$. The feature vectors are given by the standard basis $e_1, e_2, e_3, e_4, e_5, e_6 \in \mathbb{R}^d$, where $e_1, e_2, e_3$ are features for the positive label $y = 1$ and $e_4, e_5, e_6$ are features for the negative label $y = -1$. We categorize $e_1$ and $e_4$ as common features, $e_2$ and $e_5$ as rare features, and lastly, $e_3$ and $e_6$ as extremely rare features. We apply full-batch gradient descent with learning rate $\eta = 1$ and for Cutout and CutMix, we utilize all possible augmented data.

For the multi-neuron with smoothed Leaky ReLU case (Figure 5), we use 10 neurons for each positive/negative output with the slope of negative regime $\beta = 0.1$ and the length of polynomial regime $r = 1$. We set the strength of dominant noise $\sigma_{\mathrm{d}} = 0.25$, the strength of background noise $\sigma_{\mathrm{b}} = 0.12$, and the strength of feature noise $\alpha = 0.05$. In addition, frequencies of common features, rare features, and extremely rare features are set to $0.72$, $0.15$, and $0.03$, respectively.

For the multi-neuron with smoothed ReLU case i.e., $\beta = 0$ (Figure 6), we set the length of the polynomial regime as $r = 1$, and we use 10 neurons for each positive/negative output. We set the remaining problem parameters as follows: the strength of dominant noise $\sigma_{\mathrm{d}} = 0.25$, the strength of background noise $\sigma_{\mathrm{b}} = 0.12$, and the strength of feature noise $\alpha = 0.05$. In addition, frequencies of common features, rare features, and extremely rare features are set to $0.75$, $0.2$, and $0.05$, respectively.

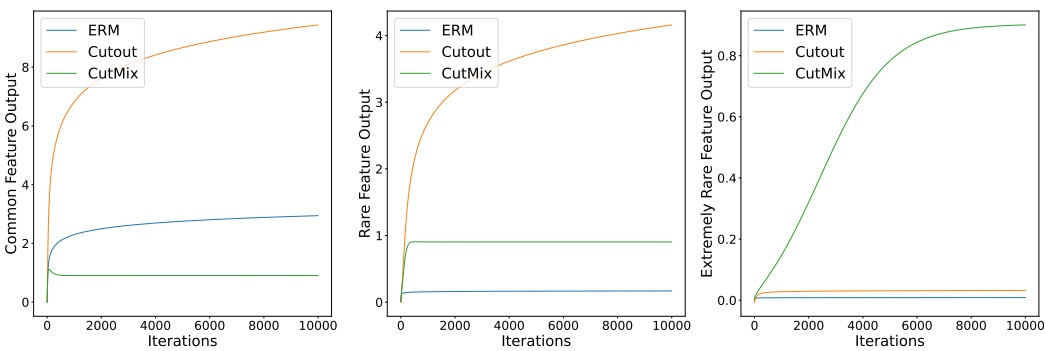

Figure 5: Multi-neuron with a smoothed leaky ReLU actiation

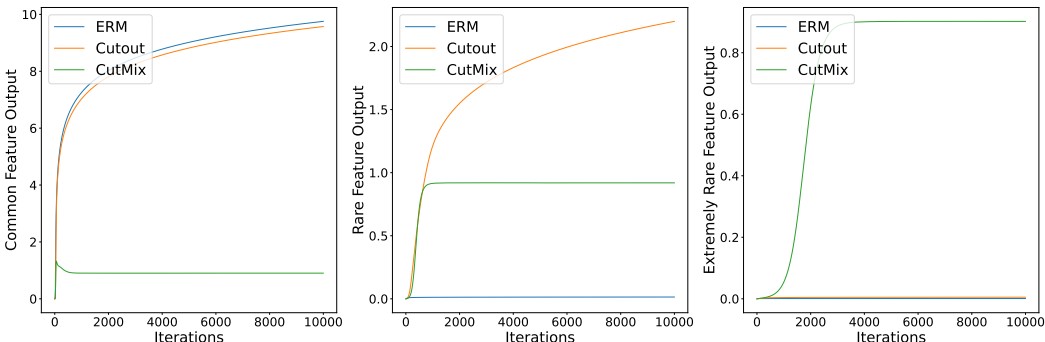

Figure 6: Multi-neuron with a smoothed ReLU

# B  Proof Preliminaries

## B.1  Properties of the Choice of Problem Parameters

In our analysis, we consider the choice of problem parameters as a function of the dimension of patches $d$ and consider sufficiently large $d$. Let us summarize the assumptions on the parameters for the problem setting and assume they hold.

**Assumption B.1.** The following conditions hold.

A1. (The number of patches) $P = \Theta(1)$ and $P \geq 8$.

A2. (Overparameterized regime): $n = o\left(\alpha\beta\sigma_{\mathrm{d}}^{-1}\sigma_{\mathrm{b}}d^{\frac{1}{2}}/\mathrm{polylog}(d)\right)$.

A3. (Sufficient feature data): For all $k \in [K]$, $\rho_k n = \omega\left(n^{\frac{1}{2}}\log d\right)$.

A4. (Common feature vs dominant noise): For all $k \in \mathcal{K}_C$, $\rho_k = \Theta(1)$ and $\sigma_{\mathrm{d}}^2 d = o(\beta n)$.

A5. (Rare feature vs noise): For all $k \in \mathcal{K}_R$, $\rho_k = \Theta(\rho_R)$ with $\rho_R n = o\left(\alpha^2\sigma_{\mathrm{d}}^2 d/\mathrm{polylog}(d)\right)$ and $\sigma_{\mathrm{b}}^2 d = o(\beta\rho_R n)$.

A6. (Extremely rare feature vs background noise) For all $k \in \mathcal{K}_E$, $\rho_k = \Theta(\rho_E)$ with $\rho_E n = o\left(\alpha^2\sigma_{\mathrm{b}}^2 d/\mathrm{polylog}(d)\right)$.

A7. (Strength of feature noise) $\alpha = o\left(n^{-1}\beta\sigma_{\mathrm{d}}^2 d/\mathrm{polylog}(d)\right)$.

A8. $\sigma_0\sigma_{\mathrm{d}}^2 d, r = o\left(\alpha/\mathrm{polylog}(d)\right), \eta = o\left(r\sigma_{\mathrm{d}}^{-2}d^{-1}/\mathrm{polylog}(d)\right)$

We now present some properties derived from Assumption B.1, which are frequently used throughout our proof.

From (A3), for all $k \in [K]$, we have the following inequality:

$$n \geq \rho_1 n \geq \rho_k^2 n = \omega\left(\log^2 d\right) \tag{6}$$

From (A1) and (A2), and given that $\beta < 1, \sigma_{\mathrm{b}} < \sigma_{\mathrm{d}}$, we have

$$d > (\beta\sigma_{\mathrm{d}}^{-1}\sigma_{\mathrm{b}}d^{\frac{1}{2}})^2 > n^2 P > nP. \tag{7}$$

From (A2), (A3), and (A6), and given that $\alpha, \beta < 1$, we have

$$\sigma_{\mathrm{d}}^2 d > \sigma_{\mathrm{b}}^2 d = \omega(\rho_E n) = \omega(1). \tag{8}$$

From (A1), (A2) and the fact that $0 < \alpha < 1$, we have

$$nP\beta^{-1}\sigma_{\mathrm{d}}\sigma_{\mathrm{b}}^{-1}d^{-\frac{1}{2}} = o\left(\frac{\alpha}{\mathrm{polylog}(d)}\right) = o\left(\frac{1}{\mathrm{polylog}(d)}\right) \tag{9}$$

From (A7) and (A4), we have

$$\alpha\beta^{-1} < \alpha\beta^{-2} = o\left(\frac{n^{-1}\beta^{-1}\sigma_{\mathrm{d}}^2 d}{\mathrm{polylog}(d)}\right) = o\left(\frac{1}{\mathrm{polylog}(d)}\right) \tag{10}$$

From (8) and (A8), $\eta = o(1)$ and then we have

$$\eta \leq \frac{\log(\eta T^*)}{2}. \tag{11}$$

From (A2), (A3), (A4), and (A5) we have

$$\alpha^{-2} = o\left(\frac{\sigma_{\mathrm{d}}^2 d}{\rho_R n}\right) = o\left(\rho_R^{-1}\right) = o\left(n^{\frac{1}{2}}\right) = o\left(d^{\frac{1}{4}}\right). \tag{12}$$

## B.2 Quantities at the Beginning

We characterize some quantities at the beginning of training.

**Lemma B.2.** *Let $E_{\mathrm{init}}$ the event such that all the following holds:*

- $\frac{25}{52}n \le |\mathcal{V}_1|, |\mathcal{V}_{-1}| \le \frac{27}{52}n$

- *For each $s \in \{\pm 1\}$ and $k \in [K]$, $\frac{\rho_k n}{4} \le |\mathcal{V}_{s,k}| \le \frac{3\rho_k n}{4}$*

- $\cup_{i \in \mathcal{V}_{1,1}} \{p_i^*\} = [P]$

- *For any $s, s' \in \{\pm 1\}$ and $k \in [K]$, $\left| \left\langle \boldsymbol{w}_s^{(0)}, \boldsymbol{v}_{s',k} \right\rangle \right| \le \sigma_0 \log d.$*

- *For any $s \in \{\pm 1\}$ and $i \in [n]$, $\left| \left\langle \boldsymbol{w}_s^{(0)}, \xi_i^{(\tilde{p}_i)} \right\rangle \right| \le \sigma_0 \sigma_{\mathrm{d}} d^{\frac{1}{2}} \log d.$*

- *For any $s \in \{\pm 1\}, i \in [n]$ and $p \in [P] \setminus \{p_i^*, \tilde{p}_i\}$, $\left| \left\langle \boldsymbol{w}_s^{(0)}, \xi_i^{(p)} \right\rangle \right| \le \sigma_0 \sigma_{\mathrm{b}} d^{\frac{1}{2}} \log d.$*

- *For any $i, j \in [n]$ with $i \ne j$, $\frac{1}{2} \sigma_{\mathrm{d}}^2 d \le \left\| \xi_i^{(\tilde{p}_i)} \right\|^2 \le \frac{3}{2} \sigma_{\mathrm{d}}^2 d$ and $\left| \left\langle \xi_i^{(\tilde{p}_i)}, \xi_j^{(\tilde{p}_j)} \right\rangle \right| \le \sigma_{\mathrm{d}}^2 d^{\frac{1}{2}} \log d.$*

- *For any $i, j \in [n]$ and $p \in [P] \setminus \{p_j^*, \tilde{p}_j\}$, $\left| \left\langle \xi_i^{(\tilde{p}_i)}, \xi_j^{(p)} \right\rangle \right| \le \sigma_{\mathrm{d}} \sigma_{\mathrm{b}} d^{\frac{1}{2}} \log d.$*

- *For any $i, j \in [n]$ and $p \in [P] \setminus \{p_i^*, \tilde{p}_i\}, q \in [P] \setminus \{p_j^*, \tilde{p}_j\}$ with $(i, p) \ne (j, q)$,*
  $\frac{1}{2} \sigma_{\mathrm{b}}^2 d \le \left\| \xi_i^{(p)} \right\|^2 \le \frac{3}{2} \sigma_{\mathrm{b}}^2 d$ *and* $\left| \left\langle \xi_i^{(p)}, \xi_j^{(q)} \right\rangle \right| \le \sigma_{\mathrm{b}}^2 d^{\frac{1}{2}} \log d.$

- $\{\boldsymbol{v}_{s,k}\}_{s \in \{\pm 1\}, k \in [K]} \cup \{\boldsymbol{x}_i^{(p)}\}_{i \in [n], p \in [P] \setminus \{p_i^*\}}$ *is linearly independent.*

*Then, the event $E_{\mathrm{init}}$ occurs with probability at least $1 - o\left( \frac{1}{\mathrm{poly}(d)} \right)$. Also, if $\xi \sim N(\boldsymbol{0}, \sigma^2 \Lambda)$ is independent of $\boldsymbol{w}_1^{(0)}, \boldsymbol{w}_{-1}^{(0)}$ and $\{(\boldsymbol{X}_i, y_i)\}_{i \in [n]}$, we have*

$$\left| \left\langle \boldsymbol{w}_1^{(0)}, \xi \right\rangle \right|, \left| \left\langle \boldsymbol{w}_{-1}^{(0)}, \xi \right\rangle \right| \le \sigma_0 \sigma d^{\frac{1}{2}} \log d, \text{ and } \left| \left\langle \xi, \xi_i^{(p)} \right\rangle \right| \le \sigma \sigma_{\mathrm{d}} d^{\frac{1}{2}} \log d,$$

*for all $i \in [n]$ and $p \in [P] \setminus \{p_i^*\}$, with probability at least $1 - o\left( \frac{1}{\mathrm{poly}(d)} \right)$.*

*Proof of Lemma B.2.* Let us prove the first three points hold with probability at least $1 - o\left( \frac{1}{\mathrm{poly}(d)} \right)$. By Höeffding's inequality,

$$\mathbb{P}\left[ \left| |\mathcal{V}_1| - \frac{n}{2} \right| > \frac{n}{52} \right] = \mathbb{P}\left[ \left| \sum_{i \in [n]} \left( \mathbb{1}_{y_i = 1} - \mathbb{E}[\mathbb{1}_{y_i = 1}] \right) \right| > \frac{n}{52} \right]$$
$$\le 2 \exp\left( -\frac{2}{52^2} n \right) = o\left( \frac{1}{\mathrm{poly}(d)} \right),$$

where the last equality is due to (6). In addition, for each $s \in \{\pm 1\}, k \in [K]$, by Höeffding's inequality

$$\mathbb{P}\left[ \left| |\mathcal{V}_{s,k}| - \frac{\rho_k}{2} n \right| > \frac{\rho_k}{4} n \right] = \mathbb{P}\left[ \left| \sum_{i \in [n]} \left( \mathbb{1}_{i \in \mathcal{V}_{s,k}} - \mathbb{E}[\mathbb{1}_{i \in \mathcal{V}_{s,k}}] \right) \right| > \frac{\rho_k}{4} n \right]$$
$$\le 2 \exp\left( -\frac{\rho_k^2}{8} n \right) = o\left( \frac{1}{\mathrm{poly}(d)} \right),$$

where the last equality is due to (6). Also, for each $i \in [n]$ and $p \in [P]$,

$$\mathbb{P}[\{i \in \mathcal{V}_{1,1}\} \cap \{p_i^* = p\}] = \frac{\rho_1}{P}.$$

Hence,

$$\mathbb{P}\left[\cup_{i\in\mathcal{V}_{1,1}}\{p_i^*\} \neq [P]\right] \leq \sum_{p\in[P]} \mathbb{P}\left[\cap_{i\in[n]}\left(\left(\{i\in\mathcal{V}_{1,1}\}\cap\{p_i^*=p\}\right)^{\complement}\right)\right]$$

$$= P\left(1-\frac{\rho_1}{P}\right)^n \leq P\exp\left(-\frac{\rho_1}{P}n\right)$$

$$= o\left(\frac{1}{\mathrm{poly}(d)}\right).$$

Next, we will prove the remaining. Let us refer to the standard deviation of the Gaussian noise vector in $p$-th patch of $i$-th data as $\sigma_{i,p}$. In other words, for each $i\in[n]$ and $p\in[P]\setminus\{p_i^*\}$,

$$\sigma_{i,p} = \begin{cases} \sigma_{\mathrm{d}} & \text{if } p=\tilde{p}_i, \\ \sigma_{\mathrm{b}} & \text{otherwise.} \end{cases}$$

For each $s,s'\in\{\pm1\}$ and $k\in[K]$, $\left\langle\boldsymbol{w}_s^{(0)},\boldsymbol{v}_{s',k}\right\rangle \sim N(0,\sigma_0)$. Hence, by Höeffding's inequality, we have

$$\mathbb{P}\left[\left|\left\langle\boldsymbol{w}_s^{(0)},\boldsymbol{v}_{s',k}\right\rangle\right| > \sigma_0\log d\right] \leq 2\exp\left(-\frac{(\sigma_0\log d)^2}{2\sigma_0^2}\right) = o\left(\frac{1}{\mathrm{poly}(d)}\right).$$

Let $\{\boldsymbol{u}_l\}_{l\in[d-2K]}$ be an orthonormal basis of the orthogonal complement of $\mathrm{Span}(\{\boldsymbol{v}_{s,k}\}_{s\in\{\pm1\},k\in[K]})$. Note that for each $s\in\{\pm1\}, i\in[n]$ and $p\in[P]\setminus\{p_i^*\}$, we can write $\xi_i^{(p)}$ and $\xi$ as

$$\boldsymbol{w}_s(0) = \sigma_0\sum_{l\in[d-2K]}\mathbf{z}_{s,l}\boldsymbol{u}_l, \quad \xi_i^{(p)} = \sigma_{i,p}\sum_{l\in[d-2K]}\mathbf{z}_{i,l}^{(p)}\boldsymbol{u}_l, \quad \xi = \sigma\sum_{l\in[d-2K]}\mathbf{z}_l\boldsymbol{u}_l$$

where $\mathbf{z}_{s,l},\mathbf{z}_{i,l}^{(p)},\mathbf{z}_l \overset{i.i.d.}{\sim} N(0,1)$. The sub-gaussian norm of standard normal distribution $N(0,1)$ is $\sqrt{\frac{8}{3}}$. Then $\left(\mathbf{z}_{i,l}^{(p)}\right)^2 - 1$'s are mean zero sub-exponential with sub-exponential norm $\frac{8}{3}$ (Lemma 2.7.6 in Vershynin (2018)). In addition, $\mathbf{z}_{s,l}\mathbf{z}_{i,l}^{(p)}$'s, $\mathbf{z}_{i,l}^{(p)}\mathbf{z}_{j,l}^{(q)}$'s and $\mathbf{z}_{i,l}^{(p)}\mathbf{z}_l$'s are mean zero sub-exponential with sub-exponential norm less than or equal to $\frac{8}{3}$ (Lemma 2.7.7 in Vershynin (2018)). We use Bernstein's inequality (Theorem 2.8.1 in Vershynin (2018)), with $c$ being the absolute constant stated therein. We then have the following:

$$1 - \mathbb{P}\left[\frac{1}{2}\sigma_{i,p}^2 d \leq \left\|\xi_i^{(p)}\right\|^2 \leq \frac{3}{2}\sigma_{i,p}^2 d\right] \leq \mathbb{P}\left[\left|\left\|\xi_i^{(p)}\right\|^2 - \sigma_{i,p}^2(d-2K)\right| \geq \sigma_{i,p}^2 d^{\frac{1}{2}}\log d\right]$$

$$= \mathbb{P}\left[\left|\sum_{l\in[d-2K]}\left(\left(\mathbf{z}_{i,l}^{(p)}\right)^2 - 1\right)\right| \geq d^{\frac{1}{2}}\log d\right]$$

$$\leq 2\exp\left(-\frac{9cd\log^2 d}{64(d-2K)}\right)$$

$$\leq 2\exp\left(-\frac{9c\log^2 d}{64}\right) = o\left(\frac{1}{\mathrm{poly}(d)}\right),$$

in addition,

$$\mathbb{P}\left[\left|\left\langle\xi_i^{(p)},\xi_j^{(q)}\right\rangle\right| \geq \sigma_{i,p}\sigma_{j,q}d^{\frac{1}{2}}\log d\right] = \mathbb{P}\left[\left|\sum_{l\in[d-2K]}\mathbf{z}_{i,l}^{(p)}\mathbf{z}_{j,l}^{(q)}\right| \geq d^{\frac{1}{2}}\log d\right]$$

$$\leq 2\exp\left(-\frac{9cd\log^2 d}{64(d-2K)}\right)$$

$$\leq 2\exp\left(-\frac{9c\log^2 d}{64}\right) = o\left(\frac{1}{\mathrm{poly}(d)}\right).$$

Similarly, we have

$$\mathbb{P}\left[\left|\left\langle \boldsymbol{w}_s^{(0)}, \xi_i^{(p)}\right\rangle\right| \geq \sigma_0 \sigma_{i,p} d^{\frac{1}{2}} \log d\right] \leq 2\exp\left(-\frac{9c\log^2 d}{64}\right) = o\left(\frac{1}{\mathrm{poly}(d)}\right).$$

Lastly, the last result holds almost surely due to (7). Applying the union bound to all events, each of which is at most $\mathrm{poly}(d)$ due to (7), leads us to our first conclusion.

In addition, for each $s \in \{\pm 1\}, i \in [n]$ and $p \in [P] \setminus \{p_i^*\}$,

$$\mathbb{P}\left[\left|\left\langle \boldsymbol{w}_s^{(0)}, \xi\right\rangle\right| \geq \sigma_0 \sigma d^{\frac{1}{2}} \log d\right] \leq 2\exp\left(-\frac{9c\log^2 d}{64}\right) = o\left(\frac{1}{\mathrm{poly}(d)}\right),$$

and

$$\mathbb{P}\left[\left|\left\langle \xi_i^{(p)}, \xi\right\rangle\right| \geq \sigma_{i,p} \sigma d^{\frac{1}{2}} \log d\right] \leq 2\exp\left(-\frac{9c\log^2 d}{64}\right) = o\left(\frac{1}{\mathrm{poly}(d)}\right).$$

Applying the union bound to all events, each of which is at most $\mathrm{poly}(d)$ due to (7), leads us to our second conclusion. $\qquad\square$

### B.3 Feature Noise Decomposition

In our analysis, we use a technique that analyzes the coefficients of linear combinations of feature and noise vectors. A similar technique in a different data and network setting is introduced by Cao et al. (2022).

**Lemma B.3.** *If we run one of* ERM, Cutout, *and* CutMix *training to update parameters $\boldsymbol{W}^{(t)}$ of a model $f_{\boldsymbol{W}^{(t)}}$, then there exist coefficients (corresponding to each method) $\gamma_s^{(t)}(s', k)$'s and $\rho_s^{(t)}(i, p)$'s so that we can write $\boldsymbol{W}^{(t)} = \{\boldsymbol{w}_1^{(t)}, \boldsymbol{w}_{-1}^{(t)}\}$ as*

$$\boldsymbol{w}_s^{(t)} = \boldsymbol{w}_s^{(0)} + \sum_{k\in[K]} \gamma_s^{(t)}(s,k)\boldsymbol{v}_{s,k} - \sum_{k\in[K]} \gamma_s^{(t)}(-s,k)\boldsymbol{v}_{-s,k}$$

$$+ \sum_{i\in\mathcal{V}_s, p\in[P]\setminus\{p_i^*\}} \rho_s^{(t)}(i,p)\frac{\xi_i^{(p)}}{\left\|\xi_i^{(p)}\right\|^2} - \sum_{i\in\mathcal{V}_{-s}, p\in[P]\setminus\{p_i^*\}} \rho_s^{(t)}(i,p)\frac{\xi_i^{(p)}}{\left\|\xi_i^{(p)}\right\|^2}$$

$$+ \alpha\left(\sum_{i\in\mathcal{F}_s} sy_i\rho_s^{(t)}(i,\tilde{p}_i)\frac{\boldsymbol{v}_{s,1}}{\left\|\xi_i^{(\tilde{p}_i)}\right\|^2} + \sum_{i\in\mathcal{F}_{-s}} sy_i\rho_s^{(t)}(i,\tilde{p}_i)\frac{\boldsymbol{v}_{-s,1}}{\left\|\xi_i^{(\tilde{p}_i)}\right\|^2}\right)$$

*where $\mathcal{F}_s$ denotes the set of indices of data with feature noise $\boldsymbol{v}_{s,1}$. Furthermore, if we run one of* ERM *and* Cutout, *the coefficients $\gamma_s^{(t)}(s', k)$'s and $\rho_s^{(t)}(i, p)$'s are monotone increasing.*

We provide proof of Lemma B.3 for ERM in Appendix C.1, for Cutout in Appendix D.1 and for CutMix in Appendix E.1.

Since Gaussian vectors in a high-dimensional regime are nearly orthogonal, we can use the coefficients to approximate the inner products or outputs of neurons. The following lemma quantifies the approximation error.

**Lemma B.4.** *Suppose the event $E_{\mathrm{init}}$ occurs and $0 \leq \gamma_s^{(t)}(s', k), \rho_s^{(t)}(i, p) \leq \widetilde{\mathcal{O}}(\beta^{-1})$ for all $s, s' \in \{\pm 1\}, k \in [K], i \in [n]$ and $p \in [P] \setminus \{p_i^*\}$ at iteration $t$. Then, for each $s \in \{\pm 1\}, k \in [K], i \in [n]$, and $p \in [P] \setminus \{p_i^*\}$, the following holds:*

- $\left|\left\langle \boldsymbol{w}_s^{(t)}, \boldsymbol{v}_{s,k}\right\rangle - \gamma_s^{(t)}(s,k)\right|, \left|\phi\left(\left\langle \boldsymbol{w}_s^{(t)}, \boldsymbol{v}_{s,k}\right\rangle\right) - \gamma_s^{(t)}(s,k)\right| = o\left(\frac{1}{\mathrm{polylog}(d)}\right)$

- $\left|\left\langle \boldsymbol{w}_s^{(t)}, \boldsymbol{v}_{-s,k}\right\rangle + \gamma_s^{(t)}(-s,k)\right|, \left|\phi\left(\left\langle \boldsymbol{w}_s^{(t)}, \boldsymbol{v}_{-s,k}\right\rangle\right) + \beta\gamma_s^{(t)}(-s,k)\right| = o\left(\frac{1}{\mathrm{polylog}(d)}\right)$

- $\left|\left\langle \boldsymbol{w}_{y_i}^{(t)}, \xi_i^{(p)}\right\rangle - \rho_{y_i}^{(t)}(i,p)\right|, \left|\phi\left(\left\langle \boldsymbol{w}_{y_i}^{(t)}, \xi_i^{(p)}\right\rangle\right) - \rho_{y_i}^{(t)}(i,p)\right| = o\left(\frac{1}{\mathrm{polylog}(d)}\right)$

- $\left|\left\langle \boldsymbol{w}_{-y_i}^{(t)}, \xi_i^{(p)}\right\rangle + \rho_{-y_i}^{(t)}(i,p)\right|, \left|\phi\left(\left\langle \boldsymbol{w}_{-y_i}^{(t)}, \xi_i^{(p)}\right\rangle\right) + \beta\rho_{-y_i}^{(t)}(i,p)\right| = o\left(\frac{1}{\mathrm{polylog}(d)}\right)$

- $\left|\phi\left(\left\langle \boldsymbol{w}_{y_i}^{(t)}, \boldsymbol{x}_i^{(\tilde{p}_i)}\right\rangle\right) - \rho_{y_i}^{(t)}(i, \tilde{p}_i)\right|, \left|\phi\left(\left\langle \boldsymbol{w}_{-y_i}^{(t)}, \boldsymbol{x}_i^{(\tilde{p}_i)}\right\rangle\right) + \beta\rho_{-y_i}^{(t)}(i, \tilde{p}_i)\right| = o\left(\frac{1}{\mathrm{polylog}(d)}\right)$

*Proof of Lemma B.4.* For each $s \in \{\pm 1\}, k \in [K] \setminus \{1\}$, by (A8) and (8), we have

$$\left|\left\langle \boldsymbol{w}_s^{(t)}, \boldsymbol{v}_{s,k}\right\rangle - \gamma_s^{(t)}(s, k)\right| = \left|\left\langle \boldsymbol{w}_s^{(0)}, \boldsymbol{v}_{s,k}\right\rangle\right| = \widetilde{\mathcal{O}}(\sigma_0) = o\left(\frac{1}{\mathrm{polylog}(d)}\right).$$

Similarly, by (A8) and (8),

$$\left|\left\langle \boldsymbol{w}_s^{(t)}, \boldsymbol{v}_{-s,k}\right\rangle + \gamma_s^{(t)}(-s, k)\right| = \left|\left\langle \boldsymbol{w}_s^{(0)}, \boldsymbol{v}_{-s,k}\right\rangle\right| = \widetilde{\mathcal{O}}(\sigma_0) = o\left(\frac{1}{\mathrm{polylog}(d)}\right),$$

Next, we will consider the case of $\boldsymbol{v}_{1,1}$ and $\boldsymbol{v}_{-1,1}$. For each $s \in \{\pm 1\}$, we have

$$\left|\left\langle \boldsymbol{w}_s^{(t)}, \boldsymbol{v}_{s,1}\right\rangle - \gamma_s^{(t)}(s, 1)\right|$$
$$\leq \left|\left\langle \boldsymbol{w}_s^{(0)}, \boldsymbol{v}_{s,1}\right\rangle\right| + \alpha \sum_{i \in [n]} \rho_s^{(t)}(i, \tilde{p}_i)\left\|\xi_i^{(\tilde{p}_i)}\right\|^{-2}$$
$$\leq \widetilde{\mathcal{O}}(\sigma_0) + \widetilde{\mathcal{O}}\left(\alpha n \beta^{-1} \sigma_{\mathrm{d}}^{-2} d^{-1}\right)$$
$$= o\left(\frac{1}{\mathrm{polylog}(d)}\right),$$

where the last equality is due to (8) and (A7). Similarly, we have

$$\left|\left\langle \boldsymbol{w}_s^{(t)}, \boldsymbol{v}_{-s,1}\right\rangle + \gamma_s^{(t)}(-s, 1)\right|$$
$$\leq \left|\left\langle \boldsymbol{w}_s^{(0)}, \boldsymbol{v}_{-s,1}\right\rangle\right| + \alpha \sum_{i \in [n]} \rho_s^{(t)}(i, \tilde{p}_i)\left\|\xi_i^{(\tilde{p}_i)}\right\|^{-2}$$
$$\leq \widetilde{\mathcal{O}}(\sigma_0) + \widetilde{\mathcal{O}}\left(\alpha n \beta^{-1} \sigma_{\mathrm{d}}^{-2} d^{-1}\right)$$
$$= o\left(\frac{1}{\mathrm{polylog}(d)}\right).$$

Hence, from (A8) and the fact that $|\phi(z) - z| \leq \frac{(1-\beta)r}{2}$ for any $z \geq 0$, we have

$$\left|\phi\left(\left\langle \boldsymbol{w}_s^{(t)}, \boldsymbol{v}_{s,k}\right\rangle\right) - \gamma_s^{(t)}(s, k)\right|$$
$$\leq \left|\phi\left(\left\langle \boldsymbol{w}_s^{(t)}, \boldsymbol{v}_{s,k}\right\rangle\right) - \phi\left(\gamma_s^{(t)}(s, k)\right)\right| + \left|\phi\left(\gamma_s^{(t)}(s, k)\right) - \gamma_s^{(t)}(s, k)\right|$$
$$\leq \left|\left\langle \boldsymbol{w}_s^{(t)}, \boldsymbol{v}_{s,k}\right\rangle - \gamma_s^{(t)}(s, k)\right| + \frac{(1-\beta)r}{2}$$
$$= o\left(\frac{1}{\mathrm{polylog}(d)}\right).$$

and

$$\left|\phi\left(\left\langle \boldsymbol{w}_s^{(t)}, \boldsymbol{v}_{-s,k}\right\rangle\right) + \beta\gamma_s^{(t)}(-s, k)\right| = \left|\phi\left(\left\langle \boldsymbol{w}_s^{(t)}, \boldsymbol{v}_{-s,k}\right\rangle\right) - \phi\left(-\gamma_s^{(t)}(-s, k)\right)\right|$$
$$\leq \left|\left\langle \boldsymbol{w}_s^{(t)}, \boldsymbol{v}_{-s,k}\right\rangle + \gamma_s^{(t)}(-s, k)\right|$$
$$= o\left(\frac{1}{\mathrm{polylog}(d)}\right).$$

For each $i \in [n]$, and $p \in [P] \setminus \{p_i^*\}$, we have

$$\left|\left\langle \boldsymbol{w}_{y_i}^{(t)}, \xi_i^{(p)}\right\rangle - \rho_{y_i}^{(t)}(i, p)\right| \leq \left|\left\langle \boldsymbol{w}_{y_i}^{(0)}, \xi_i^{(p)}\right\rangle\right| + \sum_{\substack{j \in [n], q \in [P] \setminus \{p_i^*\} \\ (j,q) \neq (i,p)}} \rho_{y_i}^{(t)}(j, q) \frac{\left|\left\langle \xi_i^{(p)}, \xi_j^{(q)}\right\rangle\right|}{\left\|\xi_j^{(q)}\right\|^2}$$

$$\le \widetilde{\mathcal{O}}\left(\sigma_0 \sigma_{\mathrm{d}} d^{\frac{1}{2}}\right) + \widetilde{\mathcal{O}}\left(nP\beta^{-1}\sigma_{\mathrm{d}}\sigma_{\mathrm{b}}^{-1} d^{-\frac{1}{2}}\right)$$

$$= o\left(\frac{1}{\mathrm{polylog}(d)}\right),$$

where the last equality is due to (A8) and (9). By triangular inequality, (A8), and the fact that $\phi' \le 1$ and $|\phi(z) - z| \le \frac{(1-\beta)r}{2}$ for any $z \ge 0$, we have

$$\left|\phi\left(\left\langle \boldsymbol{w}_{y_i}^{(t)}, \xi_i^{(p)}\right\rangle\right) - \rho_{y_i}^{(t)}(i,p)\right|$$

$$\le \left|\phi\left(\left\langle \boldsymbol{w}_{y_i}^{(t)}, \xi_i^{(p)}\right\rangle\right) - \phi\left(\rho_{y_i}^{(t)}(i,p)\right)\right| + \left|\phi\left(\rho_{y_i}^{(t)}(i,p)\right) - \rho_{y_i}^{(t)}(i,p)\right|$$

$$\le \left|\left\langle \boldsymbol{w}_{y_i}^{(t)}, \xi_i^{(p)}\right\rangle - \rho_{y_i}^{(t)}(i,p)\right| + \frac{(1-\beta)r}{2}$$

$$= o\left(\frac{1}{\mathrm{polylog}(d)}\right).$$

Also, if $i \in \mathcal{F}_s$ for some $s \in \{\pm 1\}$,

$$\left|\phi\left(\left\langle \boldsymbol{w}_{y_i}^{(t)}, \boldsymbol{x}_i^{(\tilde{p}_i)}\right\rangle\right) - \rho_{y_i}^{(t)}(i,\tilde{p}_i)\right|$$

$$\le \left|\phi\left(\left\langle \boldsymbol{w}_{y_i}^{(t)}, \xi_i^{(\tilde{p}_i)}\right\rangle\right) - \rho_{y_i}^{(t)}(i,\tilde{p}_i)\right| + \left|\phi\left(\left\langle \boldsymbol{w}_{y_i}^{(t)}, \boldsymbol{x}_i^{(\tilde{p}_i)}\right\rangle\right) - \phi\left(\left\langle \boldsymbol{w}_{y_i}^{(t)}, \xi_i^{(\tilde{p}_i)}\right\rangle\right)\right|$$

$$\le \left|\phi\left(\left\langle \boldsymbol{w}_{y_i}^{(t)}, \xi_i^{(\tilde{p}_i)}\right\rangle\right) - \rho_{y_i}^{(t)}(i,\tilde{p}_i)\right| + \alpha\left|\left\langle \boldsymbol{w}_{y_i}^{(t)}, \boldsymbol{v}_{s,1}\right\rangle\right|$$

$$\le \left|\phi\left(\left\langle \boldsymbol{w}_{y_i}^{(t)}, \xi_i^{(\tilde{p}_i)}\right\rangle\right) - \rho_{y_i}^{(t)}(i,\tilde{p}_i)\right| + \alpha\gamma_{y_i}^{(t)}(s,1) + \alpha \cdot o\left(\frac{1}{\mathrm{polylog}(d)}\right)$$

$$\le \widetilde{\mathcal{O}}\left(\alpha\beta^{-1}\right) + o\left(\frac{1}{\mathrm{polylog}(d)}\right)$$

$$= o\left(\frac{1}{\mathrm{polylog}(d)}\right),$$

where we apply the triangular inequality, the fact that $\phi' \le 1$, the triangular inequality again, $\rho_{y_i}^{(t)}(s,1) = \widetilde{\mathcal{O}}(\beta^{-1})$ and (10) sequentially.

Similarly,

$$\left|\left\langle \boldsymbol{w}_{-y_i}^{(t)}, \xi_i^{(p)}\right\rangle + \rho_{-y_i}^{(t)}(i,p)\right| \le \left|\left\langle \boldsymbol{w}_{-y_i}^{(0)}, \xi_i^{(p)}\right\rangle\right| + \sum_{\substack{j\in[n], q\in[P]\setminus\{p_i^*\} \\ (j,q)\neq(i,p)}} \rho_{-y_i}^{(t)}(j,q) \frac{\left|\left\langle \xi_i^{(p)}, \xi_j^{(q)}\right\rangle\right|}{\left\|\xi_j^{(q)}\right\|^2}$$

$$\le \widetilde{\mathcal{O}}(\sigma_0 \sigma_{\mathrm{d}} d^{\frac{1}{2}}) + \widetilde{\mathcal{O}}\left(nP\beta^{-1}\sigma_{\mathrm{d}}\sigma_{\mathrm{b}}^{-1} d^{-\frac{1}{2}}\right)$$

$$= o\left(\frac{1}{\mathrm{polylog}(d)}\right),$$

and

$$\left|\phi\left(\left\langle \boldsymbol{w}_{-y_i}^{(t)}, \xi_i^{(p)}\right\rangle\right) + \beta\rho_{-y_i}^{(t)}(i,p)\right| = \left|\phi\left(\left\langle \boldsymbol{w}_{-y_i}^{(t)}, \xi_i^{(p)}\right\rangle\right) - \phi\left(-\rho_{-y_i}^{(t)}(i,p)\right)\right|$$

$$\le \left|\left\langle \boldsymbol{w}_{-y_i}^{(t)}, \xi_i^{(p)}\right\rangle + \rho_{-y_i}^{(t)}(i,p)\right|$$

$$= o\left(\frac{1}{\mathrm{polylog}(d)}\right),$$

Also, if $i \in \mathcal{F}_s$ for some $s \in \{\pm 1\}$,

$$\left|\phi\left(\left\langle \boldsymbol{w}_{-y_i}^{(t)}, \boldsymbol{x}_i^{(\tilde{p}_i)}\right\rangle\right) + \beta\rho_{-y_i}^{(t)}(i,\tilde{p}_i)\right|$$

$$= \left|\phi\left(\left\langle \boldsymbol{w}_{-y_i}^{(t)}, \xi_i^{(\tilde{p}_i)}\right\rangle\right) + \beta\rho_{-y_i}^{(t)}(i,\tilde{p}_i)\right| + \left|\phi\left(\left\langle \boldsymbol{w}_{-y_i}^{(t)}, \boldsymbol{x}_i^{(\tilde{p}_i)}\right\rangle\right) - \phi\left(\left\langle \boldsymbol{w}_{-y_i}^{(t)}, \xi_i^{(\tilde{p}_i)}\right\rangle\right)\right|$$

$$\leq \left| \phi\left( \left\langle \boldsymbol{w}_{-y_i}^{(t)}, \xi_i^{(\tilde{p}_i)} \right\rangle \right) + \beta \rho_{-y_i}^{(t)}(i, \tilde{p}_i) \right| + \alpha \left| \left\langle \boldsymbol{w}_{-y_i}^{(t)}, \boldsymbol{v}_{s,1} \right\rangle \right|$$

$$\leq \left| \phi\left( \left\langle \boldsymbol{w}_{-y_i}^{(t)}, \xi_i^{(\tilde{p}_i)} \right\rangle \right) + \beta \rho_{-y_i}^{(t)}(i, \tilde{p}_i) \right| + \alpha \gamma_{-y_i}^{(t)}(s, 1) + \alpha \cdot o\left( \frac{1}{\mathrm{polylog}(d)} \right)$$

$$\leq \tilde{\mathcal{O}}\left( \alpha \beta^{-1} \right) + o\left( \frac{1}{\mathrm{polylog}(d)} \right)$$

$$= o\left( \frac{1}{\mathrm{polylog}(d)} \right).$$

$\square$

We define the set $\mathcal{W}$ as the collection of $\boldsymbol{W} = \{\boldsymbol{w}_1, \boldsymbol{w}_{-1}\}$, where $\boldsymbol{w}_1 - \boldsymbol{w}_1^{(0)}, \boldsymbol{w}_{-1} - \boldsymbol{w}_{-1}^{(0)}$ are elements of the subspace spanned by $\{\boldsymbol{v}_{s,k}\}_{s \in \{\pm 1\}, k \in [K]} \cup \left\{ \boldsymbol{x}_i^{(p)} \right\}_{i \in [n], p \in [P] \setminus \{p_i^*\}}$. The following lemma guarantees the unique expression of any $\boldsymbol{W} \in \mathcal{W}$ in the form of the feature noise decomposition.

**Lemma B.5.** *Suppose the event* $E_{\mathrm{init}}$ *occurs. For each element* $\boldsymbol{W} = \{\boldsymbol{w}_1, \boldsymbol{w}_{-1}\} \in \mathcal{W}$, *there exist unique coefficients* $\gamma_s(s', k)$'s *and* $\rho_s(i, p)$'s *such that*

$$\boldsymbol{w}_s = \boldsymbol{w}_s^{(0)} + \sum_{k \in [K]} \gamma_s(s, k) \boldsymbol{v}_{s,k} - \sum_{k \in [K]} \gamma_s(-s, k) \boldsymbol{v}_{-s,k}$$

$$+ \sum_{\substack{i \in \mathcal{V}_s \\ p \in [P] \setminus \{p_i^*\}}} \rho_s(i, p) \frac{\xi_i^{(p)}}{\left\| \xi_i^{(p)} \right\|^2} - \sum_{\substack{i \in \mathcal{V}_{-s} \\ p \in [P] \setminus \{p_i^*\}}} \rho_s(i, p) \frac{\xi_i^{(p)}}{\left\| \xi_i^{(p)} \right\|^2}$$

$$+ \alpha \left( \sum_{i \in \mathcal{F}_s} s y_i \rho_s(i, \tilde{p}_i) \frac{\boldsymbol{v}_{s,1}}{\left\| \xi_i^{(\tilde{p}_i)} \right\|^2} + \sum_{i \in \mathcal{F}_{-s}} s y_i \rho_s(i, \tilde{p}_i) \frac{\boldsymbol{v}_{-s,1}}{\left\| \xi_i^{(\tilde{p}_i)} \right\|^2} \right)$$

*for each* $s \in \{\pm 1\}$. *Using this fact, for each* $s^* \in \{\pm 1\}$ *and* $k^* \in [K]$, *we can introduce a function* $Q^{(s^*, k^*)} : \mathcal{W} \to \mathbb{R}^{d \times 2}$ *such that for each* $\boldsymbol{W} = \{\boldsymbol{w}_1, \boldsymbol{w}_{-1}\} \in \mathcal{W}$,

$$Q^{(s^*, k^*)}(\boldsymbol{W}) = \left\{ Q_1^{(s^*, k^*)}(\boldsymbol{w}_1), Q_{-1}^{(s^*, k^*)}(\boldsymbol{w}_{-1}) \right\}$$

*is given by:*

$$Q_s^{(s^*, k^*)}(\boldsymbol{w}_s) = s s^* \gamma_s(s^*, k^*) \boldsymbol{v}_{s^*, k^*} + s s^* \sum_{i \in \mathcal{V}_{s^*, k^*}, p \in [P] \setminus \{p_i^*\}} \rho_s(i, p) \frac{\xi_i^{(p)}}{\left\| \xi_i^{(p)} \right\|^2}$$

$$+ \alpha \left( \sum_{i \in \mathcal{F}_s \cap \mathcal{V}_{s^*, k^*}} s s^* \rho_s(i, \tilde{p}_i) \frac{\boldsymbol{v}_{s,1}}{\left\| \xi_i^{(\tilde{p}_i)} \right\|^2} + \sum_{i \in \mathcal{F}_{-s} \cap \mathcal{V}_{s^*, k^*}} s s^* \rho_s(i, \tilde{p}_i) \frac{\boldsymbol{v}_{-s,1}}{\left\| \xi_i^{(\tilde{p}_i)} \right\|^2} \right).$$

The function $Q^{(s^*, k^*)}$ plays a crucial role in Section C.2.4 and Section D.2.4. The key intuition behind our definition of $Q^{(s^*, k^*)}$ is that $Q^{(s^*, k^*)}(\boldsymbol{W}^{(t)})$ represents the term updated by the data having the feature vector $\boldsymbol{v}_{s^*, k^*}$, where $\boldsymbol{W}^{(t)}$ are the iterates of either ERM or Cutout. As expected from this intuition, if we sum all $Q_1^{(s^*, k^*)}(\boldsymbol{w}_1)$ and $Q_{-1}^{(s^*, k^*)}(\boldsymbol{w}_{-1})$ over all $s^* \in \{\pm 1\}$ and $k^* \in [K]$, the result will be equal to $\boldsymbol{w}_1 - \boldsymbol{w}_1^{(0)}$ and $\boldsymbol{w}_{-1} - \boldsymbol{w}_{-1}^{(0)}$, respectively.

*Proof.* From linear independency of $\{\boldsymbol{v}_{s,k}\}_{s \in \{\pm 1\}, k \in [K]} \cup \left\{ \boldsymbol{x}_i^{(p)} \right\}_{i \in [n], p \in [P] \setminus \{p_i^*\}}$, we can express any element $\boldsymbol{W} = \{\boldsymbol{w}_1, \boldsymbol{w}_{-1}\} \in \mathcal{W}$ as

$$\boldsymbol{w}_s = \boldsymbol{w}_s^{(0)} + \sum_{k \in [K]} \tilde{\gamma}_s(s, k) \boldsymbol{v}_{s,k} - \sum_{k \in [K]} \tilde{\gamma}_s(-s, k) \boldsymbol{v}_{-s,k}$$

$$+ \sum_{\substack{i \in \mathcal{V}_s, \\ p \in [P] \setminus \{p_i^*\}}} \rho_s(i,p) \frac{\xi_i^{(p)}}{\left\| \xi_i^{(p)} \right\|} - \sum_{\substack{i \in \mathcal{V}_{-s}, \\ p \in [P] \setminus \{p_i^*\}}} \rho_s(i,p) \frac{\xi_i^{(p)}}{\left\| \xi_i^{(p)} \right\|} \tag{13}$$

with unique $\{\tilde{\gamma}_s(s,k), \tilde{\gamma}_s(-s,k)\}_{s \in \{\pm 1\}, k \in [K]}$ and $\{\rho_s(i,p)\}_{s \in \{\pm 1\}, i \in [n], p \in [P] \setminus \{i^*\}}$. If we define $\gamma_s(s,k)$ and $\gamma_s(-s,k)$ as $\gamma_s(s,k) = \tilde{\gamma}_s(s,k), \gamma_s(-s,k) = \tilde{\gamma}_s(-s,k)$ for $k \neq 1$, and

$$\gamma_s(s,1) = \tilde{\gamma}_s(s,1) - \alpha \sum_{i \in \mathcal{F}_s} s y_i \rho_s(i,\tilde{p}_i) \left\| \xi_i^{(\tilde{p}_i)} \right\|^{-2},$$

$$\gamma_s(-s,1) = \tilde{\gamma}_s(-s,1) + \alpha \sum_{i \in \mathcal{F}_{-s}} s y_i \rho_s(i,\tilde{p}_i) \left\| \xi_i^{(\tilde{p}_i)} \right\|^{-2},$$

then we have

$$\boldsymbol{w}_s = \boldsymbol{w}_s^{(0)} + \sum_{k \in [K]} \gamma_s(s,k) \boldsymbol{v}_{s,k} - \sum_{k \in [K]} \gamma_s(-s,k) \boldsymbol{v}_{-s,k}$$

$$+ \sum_{\substack{i \in \mathcal{V}_s \\ p \in [P] \setminus \{p_i^*\}}} \rho_s(i,p) \frac{\xi_i^{(p)}}{\left\| \xi_i^{(p)} \right\|^2} - \sum_{\substack{i \in \mathcal{V}_{-s} \\ p \in [P] \setminus \{p_i^*\}}} \rho_s(i,p) \frac{\xi_i^{(p)}}{\left\| \xi_i^{(p)} \right\|^2}$$

$$+ \alpha \left( \sum_{i \in \mathcal{F}_s} s y_i \rho_s(i,\tilde{p}_i) \frac{\boldsymbol{v}_{s,1}}{\left\| \xi_i^{(\tilde{p}_i)} \right\|^2} + \sum_{i \in \mathcal{F}_{-s}} s y_i \rho_s(i,\tilde{p}_i) \frac{\boldsymbol{v}_{-s,1}}{\left\| \xi_i^{(\tilde{p}_i)} \right\|^2} \right).$$

Next, we want to show the uniqueness part. Suppose $\{\hat{\gamma}_s(s,k), \hat{\gamma}_s(-s,k)\}_{s \in \{\pm 1\}, k \in [K]}$ and $\{\hat{\rho}_s(i,p)\}_{s \in \{\pm 1\}, i \in [n], p \in [P] \setminus \{i^*\}}$ satisfies

$$\boldsymbol{w}_s = \boldsymbol{w}_s^{(0)} + \sum_{k \in [K]} \hat{\gamma}_s(s,k) \boldsymbol{v}_{s,k} - \sum_{k \in [K]} \hat{\gamma}_s(-s,k) \boldsymbol{v}_{-s,k}$$

$$+ \sum_{\substack{i \in \mathcal{V}_s \\ p \in [P] \setminus \{p_i^*\}}} \hat{\rho}_s(i,p) \frac{\xi_i^{(p)}}{\left\| \xi_i^{(p)} \right\|^2} - \sum_{\substack{i \in \mathcal{V}_{-s} \\ p \in [P] \setminus \{p_i^*\}}} \hat{\rho}_s(i,p) \frac{\xi_i^{(p)}}{\left\| \xi_i^{(p)} \right\|^2}$$

$$+ \alpha \left( \sum_{i \in \mathcal{F}_s} s y_i \hat{\rho}_s(i,\tilde{p}_i) \frac{\boldsymbol{v}_{s,1}}{\left\| \xi_i^{(\tilde{p}_i)} \right\|^2} + \sum_{i \in \mathcal{F}_{-s}} s y_i \hat{\rho}_s(i,\tilde{p}_i) \frac{\boldsymbol{v}_{-s,1}}{\left\| \xi_i^{(\tilde{p}_i)} \right\|^2} \right).$$

We have

$$\boldsymbol{w}_s = \boldsymbol{w}_s^{(0)} + \sum_{k \in [K] \setminus \{1\}} \hat{\gamma}_s(s,k) \boldsymbol{v}_{s,k} - \sum_{k \in [K] \setminus \{1\}} \hat{\gamma}_s(-s,k) \boldsymbol{v}_{-s,k}$$

$$+ \left( \hat{\gamma}_s(s,1) + \alpha \sum_{i \in \mathcal{F}_s} s y_i \hat{\rho}_s(i,\tilde{p}_i) \left\| \xi_i^{(\tilde{p}_i)} \right\|^{-2} \right) \boldsymbol{v}_{s,1}$$

$$- \left( \hat{\gamma}_s(-s,1) - \alpha \sum_{i \in \mathcal{F}_{-s}} s y_i \hat{\rho}_s(i,\tilde{p}_i) \left\| \xi_i^{(\tilde{p}_i)} \right\|^{-2} \right) \boldsymbol{v}_{-s,1}$$

$$+ \sum_{\substack{i \in \mathcal{V}_s \\ p \in [P] \setminus \{p_i^*\}}} \hat{\rho}_s(i,p) \frac{\xi_i^{(p)}}{\left\| \xi_i^{(p)} \right\|^2} - \sum_{\substack{i \in \mathcal{V}_{-s} \\ p \in [P] \setminus \{p_i^*\}}} \hat{\rho}_s(i,p) \frac{\xi_i^{(p)}}{\left\| \xi_i^{(p)} \right\|^2}.$$

From the uniqueness of (13), we have

$$\hat{\gamma}_s(s,k) = \tilde{\gamma}_s(s,k) = \gamma_s(s,k), \quad \hat{\gamma}_s(-s,k) = \tilde{\gamma}_s(-s,k) = \gamma_s(-s,k),$$

for each $s \in \{\pm 1\}, k \in [K] \setminus \{1\}$, and $\hat{\rho}_s(i,p) = \rho_s(i,p)$ for each $i \in [n], p \in [P] \setminus \{p_i^*\}$. Furthermore,

$$\hat{\gamma}_s(s,1) + \alpha \sum_{i \in \mathcal{F}_s} s y_i \hat{\rho}_s(i,\tilde{p}_i) \left\| \xi_i^{(\tilde{p}_i)} \right\|^{-2} = \tilde{\gamma}_s(s,1) = \gamma_s(s,1) + \alpha \sum_{i \in \mathcal{F}_s} s y_i \rho_s(i,\tilde{p}_i) \left\| \xi_i^{(\tilde{p}_i)} \right\|^{-2},$$

and

$$\hat{\gamma}_s(-s,1) - \alpha \sum_{i \in \mathcal{F}_{-s}} sy_i \hat{\rho}_s(i,\tilde{p}_i) \left\| \xi_i^{(\tilde{p}_i)} \right\|^{-2} = \tilde{\gamma}_s(-s,1) = \gamma_s(-s,1) - \alpha \sum_{i \in \mathcal{F}_{-s}} sy_i \rho_s(i,\tilde{p}_i) \left\| \xi_i^{(\tilde{p}_i)} \right\|^{-2}.$$

Hence, we obtain the uniqueness of the expression and $Q^{(s^*,k^*)}$ is well defined for each $s^* \in \{\pm 1\}$ and $k^* \in [K]$. $\qquad\square$

## C Proof for ERM

In this section, we use $g_i^{(t)} := \frac{1}{1+\exp\left(y_i f_{\boldsymbol{W}^{(t)}}(\boldsymbol{X}_i)\right)}$ for each data $i$ and iteration $t$, for simplicity.

### C.1 Proof of Lemma B.3 for ERM

For $s \in \{\pm 1\}$ and iterate $t$,

$$
\begin{aligned}
&\boldsymbol{w}_s^{(t+1)} - \boldsymbol{w}_s^{(t)} \\
&= -\eta \nabla_{\boldsymbol{w}_s} \mathcal{L}_{\mathrm{ERM}}\left(\boldsymbol{W}^{(t)}\right) \\
&= \frac{\eta}{n} \sum_{i\in[n]} s y_i g_i^{(t)} \sum_{p\in[P]} \phi'\left(\left\langle \boldsymbol{w}_s^{(t)}, \boldsymbol{x}_i^{(p)}\right\rangle\right) \boldsymbol{x}_i^{(p)} \\
&= \frac{\eta}{n} \left( \sum_{i\in\mathcal{V}_s} g_i^{(t)} \sum_{p\in[P]} \phi'\left(\left\langle \boldsymbol{w}_s^{(t)}, \boldsymbol{x}_i^{(p)}\right\rangle\right) \boldsymbol{x}_i^{(p)} - \sum_{i\in\mathcal{V}_{-s}} g_i^{(t)} \sum_{p\in[P]} \phi'\left(\left\langle \boldsymbol{w}_s^{(t)}, \boldsymbol{x}_i^{(p)}\right\rangle\right) \boldsymbol{x}_i^{(p)} \right),
\end{aligned}
$$

and we have

$$
\begin{aligned}
&\sum_{i\in\mathcal{V}_s} g_i^{(t)} \sum_{p\in[P]} \phi'\left(\left\langle \boldsymbol{w}_s^{(t)}, \boldsymbol{x}_i^{(p)}\right\rangle\right) \boldsymbol{x}_i^{(p)} \\
&= \sum_{k\in[K]} \sum_{i\in\mathcal{V}_{s,k}} g_i^{(t)} \phi'\left(\left\langle \boldsymbol{w}_s^{(t)}, \boldsymbol{v}_{s,k}\right\rangle\right) \boldsymbol{v}_{s,k} + \sum_{i\in\mathcal{V}_s} g_i^{(t)} \sum_{p\in[P]\setminus\{p_i^*,\tilde{p}_i\}} \phi'\left(\left\langle \boldsymbol{w}_s^{(t)}, \xi_i^{(p)}\right\rangle\right) \xi_i^{(p)} \\
&\quad + \sum_{i\in\mathcal{V}_s\cap\mathcal{F}_s} g_i^{(t)} \phi'\left(\left\langle \boldsymbol{w}_s^{(t)}, \alpha\boldsymbol{v}_{s,1} + \xi_i^{(\tilde{p}_i)}\right\rangle\right) \left(\alpha\boldsymbol{v}_{s,1} + \xi_i^{(\tilde{p}_i)}\right) \\
&\quad + \sum_{i\in\mathcal{V}_s\cap\mathcal{F}_{-s}} g_i^{(t)} \phi'\left(\left\langle \boldsymbol{w}_s^{(t)}, \alpha\boldsymbol{v}_{-s,1} + \xi_i^{(\tilde{p}_i)}\right\rangle\right) \left(\alpha\boldsymbol{v}_{-s,1} + \xi_i^{(\tilde{p}_i)}\right),
\end{aligned}
$$

and

$$
\begin{aligned}
&\sum_{i\in\mathcal{V}_{-s}} g_i^{(t)} \sum_{p\in[P]} \phi'\left(\left\langle \boldsymbol{w}_s^{(t)}, \boldsymbol{x}_i^{(p)}\right\rangle\right) \boldsymbol{x}_i^{(p)} \\
&= \sum_{k\in[K]} \sum_{i\in\mathcal{V}_{-s,k}} g_i^{(t)} \phi'\left(\left\langle \boldsymbol{w}_s^{(t)}, \boldsymbol{v}_{-s,k}\right\rangle\right) \boldsymbol{v}_{-s,k} + \sum_{i\in\mathcal{V}_{-s}} g_i^{(t)} \sum_{p\in[P]\setminus\{p_i^*,\tilde{p}_i\}} \phi'\left(\left\langle \boldsymbol{w}_s^{(t)}, \xi_i^{(p)}\right\rangle\right) \xi_i^{(p)} \\
&\quad + \sum_{i\in\mathcal{V}_{-s}\cap\mathcal{F}_s} g_i^{(t)} \phi'\left(\left\langle \boldsymbol{w}_s^{(t)}, \alpha\boldsymbol{v}_{s,1} + \xi_i^{(\tilde{p}_i)}\right\rangle\right) \left(\alpha\boldsymbol{v}_{s,1} + \xi_i^{(\tilde{p}_i)}\right) \\
&\quad + \sum_{i\in\mathcal{V}_{-s}\cap\mathcal{F}_{-s}} g_i^{(t)} \phi'\left(\left\langle \boldsymbol{w}_s^{(t)}, \alpha\boldsymbol{v}_{-s,1} + \xi_i^{(\tilde{p}_i)}\right\rangle\right) \left(\alpha\boldsymbol{v}_{-s,1} + \xi_i^{(\tilde{p}_i)}\right).
\end{aligned}
$$

Hence, if we define $\gamma_s^{(t)}(s',k)$'s and $\rho_s^{(t)}(i,p)$'s recursively by using the rule

$$
\gamma_s^{(t+1)}(s',k) = \gamma_s^{(t)}(s',k) + \frac{\eta}{n} \sum_{i\in\mathcal{V}_{s',k}} g_i^{(t)} \phi'\left(\left\langle \boldsymbol{w}_s^{(t)}, \boldsymbol{v}_{s',k}\right\rangle\right), \tag{14}
$$

$$
\rho_s^{(t+1)}(i,p) = \rho_s^{(t)}(i,p) + \frac{\eta}{n} g_i^{(t)} \phi'\left(\left\langle \boldsymbol{w}_s^{(t)}, \boldsymbol{x}_i^{(p)}\right\rangle\right) \left\|\xi_i^{(p)}\right\|^2, \tag{15}
$$

starting from $\gamma_s^{(0)}(s',k) = \rho_s^{(0)}(i,p) = 0$ for each $s,s' \in \{\pm 1\}, k \in [K], i \in [n]$ and $p \in [P]\setminus\{p_i^*\}$, then we have

$$
\begin{aligned}
\boldsymbol{w}_s^{(t)} = \boldsymbol{w}_s^{(0)} + &\sum_{k\in[K]} \gamma_s^{(t)}(s,k)\boldsymbol{v}_{s,k} - \sum_{k\in[K]} \gamma_s^{(t)}(-s,k)\boldsymbol{v}_{-s,k} \\
&+ \sum_{i\in\mathcal{V}_s, p\in[P]\setminus\{p_i^*\}} \rho_s^{(t)}(i,p)\frac{\xi_i^{(p)}}{\left\|\xi_i^{(p)}\right\|^2} - \sum_{i\in\mathcal{V}_{-s}, p\in[P]\setminus\{p_i^*\}} \rho_s^{(t)}(i,p)\frac{\xi_i^{(p)}}{\left\|\xi_i^{(p)}\right\|^2}
\end{aligned}
$$

$$+ \alpha \left( \sum_{i \in \mathcal{F}_s} s y_i \rho_s^{(t)}(i, \tilde{p}_i) \frac{\boldsymbol{v}_{s,1}}{\left\| \xi_i^{(\tilde{p}_i)} \right\|^2} + \sum_{i \in \mathcal{F}_{-s}} s y_i \rho_s^{(t)}(i, \tilde{p}_i) \frac{\boldsymbol{v}_{-s,1}}{\left\| \xi_i^{(\tilde{p}_i)} \right\|^2} \right),$$

for each $s \in \{\pm 1\}$. Furthermore, $\gamma_s^{(t)}(s', k)$'s and $\rho_s^{(t)}(i, p)$'s are monotone increasing. $\qquad \square$

## C.2 Proof of Theorem 3.1

To show Theorem 3.1, we present a structured proof comprising the following five steps:

1. Establish upper bounds on $\gamma_s^{(t)}(s', k)$'s and $\rho_s^{(t)}(i, p)$'s to apply Lemma B.4 (Section C.2.1).
2. Demonstrate that the model learns common features quickly (Section C.2.2).
3. Show that the model overfits dominant noise in (extremely) rare data instead of learning its feature (Section C.2.3).
4. Confirm the persistence of this tendency until $T^*$ iterates (Section C.2.4).
5. Characterize train accuracy and test accuracy (Section C.2.5).

### C.2.1 Bounds on the Coefficients in Feature Noise Decomposition

The following lemma provides upper bounds on Lemma B.3 during $T^*$ iterations.

**Lemma C.1.** *Suppose the event $E_{\mathrm{init}}$ occurs. For any $t \in [0, T^*]$, we have*

$$0 \le \gamma_s^{(t)}(s, k) + \beta \gamma_{-s}^{(t)}(s, k) \le 4 \log(\eta T^*), \quad 0 \le \rho_{y_i}^{(t)}(i, p) + \beta \rho_{-y_i}^{(t)}(i, p) \le 4 \log(\eta T^*),$$

*for all $s \in \{\pm 1\}, k \in [K], i \in [n]$ and $p \in [P] \setminus \{p_i^*\}$. Consequently, $\gamma_s^{(t)}(s', k), \rho_s^{(t)}(i, p) = \widetilde{\mathcal{O}}(\beta^{-1})$ for all $s, s' \in \{\pm 1\}, k \in [K], i \in [n]$ and $p \in [P] \setminus \{p_i^*\}$.*

*Proof of Lemma C.1.* The first argument implies the second argument since $\log(\eta T^*) = \mathrm{polylog}(d)$ and

$$\gamma_s^{(t)}(s', k) \le \beta^{-1} \left( \gamma_{s'}^{(t)}(s', k) + \beta \gamma_{s'}^{(t)}(s', k) \right), \quad \rho_s^{(t)}(i, p) \le \beta^{-1} \left( \rho_{y_i}^{(t)}(i, p) + \beta \rho_{-y_i}^{(t)}(i, p) \right),$$

for all $s, s' \in \{\pm 1\}, k \in [K], i \in [n]$ and $p \in [P] \setminus \{p_i^*\}$.

We will prove this by using induction on $t$. The initial case $t = 0$ is trivial. Suppose the given statement holds at $t = T$ and consider the case $t = T + 1$.

Let $\tilde{T}_{s,k} \le T$ denote the smallest iteration where $\gamma_s^{(\tilde{T}_{s,k}+1)}(s, k) + \beta \gamma_{-s}^{(\tilde{T}_{s,k}+1)}(s, k) > 2 \log(\eta T^*)$. We assume the existence of $\tilde{T}_{s,k}$, as its absence would directly lead to our desired conclusion; to see why, note that the following holds, due to (14) and (11):

$$\gamma_s^{(T+1)}(s, k) + \beta \gamma_{-s}^{(T+1)}(s, k)$$

$$= \gamma_s^{(T)}(s, k) + \beta \gamma_{-s}^{(T)}(s, k) + \frac{\eta}{n} \sum_{i \in \mathcal{V}_{s,k}} g_i^{(T)} \left( \phi' \left( \left\langle \boldsymbol{w}_s^{(T)}, \boldsymbol{v}_{s,k} \right\rangle \right) + \beta \phi' \left( \left\langle \boldsymbol{w}_{-s}^{(T)}, \boldsymbol{v}_{s,k} \right\rangle \right) \right)$$

$$\le 2 \log(\eta T^*) + 2\eta \le 4 \log(\eta T^*)$$

Now suppose there exists such $\tilde{T}_{s,k} \le T$. By (14), we have

$$\gamma_s^{(T+1)}(s, k) + \beta \gamma_{-s}^{(T+1)}(s, k)$$

$$= \gamma_s^{(\tilde{T}_{s,k})}(s, k) + \beta \gamma_{-s}^{(\tilde{T}_{s,k})}(s, k)$$

$$+ \sum_{t=\tilde{T}_{s,k}}^{T} \left( \gamma_s^{(t+1)}(s, k) + \beta \gamma_{-s}^{(t+1)}(s, k) - \gamma_s^{(t)}(s, k) - \beta \gamma_{-s}^{(t)}(s, k) \right)$$

$$\le 2 \log(\eta T^*) + \log(\eta T^*) + \frac{\eta}{n} \sum_{t=\tilde{T}_{s,k}+1}^{T} \sum_{i \in \mathcal{V}_{s,k}} g_i^{(t)} \left( \phi' \left( \left\langle \boldsymbol{w}_s^{(t)}, \boldsymbol{v}_{s,k} \right\rangle \right) + \beta \phi' \left( \left\langle \boldsymbol{w}_{-s}^{(t)}, \boldsymbol{v}_{s,k} \right\rangle \right) \right).$$

The inequality is due to $\gamma_s^{(\tilde{T}_{s,k})}(s,k) + \beta\gamma_{-s}^{(\tilde{T}_{s,k})}(s,k) \leq 2\log(\eta T^*)$ from our choice of $\tilde{T}_{s,k}$ and

$$\frac{\eta}{n}\sum_{i\in\mathcal{V}_{s,k}} g_i^{(\tilde{T}_{s,k})}\left(\phi'\left(\left\langle\boldsymbol{w}_s^{(\tilde{T}_{s,k})},\boldsymbol{v}_{s,k}\right\rangle\right) + \beta\phi'\left(\left\langle\boldsymbol{w}_{-s}^{(\tilde{T}_{s,k})},\boldsymbol{v}_{s,k}\right\rangle\right)\right) \leq 2\eta \leq \log(\eta T^*),$$

from (11).

For each $t = \tilde{T}_{s,k}+1,\ldots T$, and $i\in\mathcal{V}_{s,k}$, we have

$$y_i f_{\boldsymbol{W}^{(t)}}(\boldsymbol{X}_i)$$
$$= \phi\left(\left\langle\boldsymbol{w}_s^{(t)},\boldsymbol{v}_{s,k}\right\rangle\right) - \phi\left(\left\langle\boldsymbol{w}_{-s}^{(t)},\boldsymbol{v}_{s,k}\right\rangle\right) + \sum_{p\in[P]\backslash\{p_i^*\}}\left(\phi\left(\left\langle\boldsymbol{w}_s^{(t)},\boldsymbol{x}_i^{(p)}\right\rangle\right) - \phi\left(\left\langle\boldsymbol{w}_{-s}^{(t)},\boldsymbol{x}_i^{(p)}\right\rangle\right)\right)$$
$$\geq \gamma_s^{(t)}(s,k) + \beta\gamma_{-s}^{(t)}(s,k) + \sum_{p\in[P]\backslash\{p_i^*\}}\left(\rho_s^{(t)}(i,p) + \beta\rho_{-s}^{(t)}(i,p)\right) - 2P\cdot o\left(\frac{1}{\text{polylog}(d)}\right)$$
$$\geq \frac{3}{2}\log(\eta T^*)$$

The first inequality is due to Lemma B.4 and the second inequality holds due to (A7), (8), and our choice of $t$, $\gamma_s^{(t)}(s,k) + \beta\gamma_{-s}^{(t)}(s,k) \geq 2\log(\eta T^*)$.

Hence, we obtain

$$\frac{\eta}{n}\sum_{t=\tilde{T}_{s,k}}^{T}\sum_{i\in\mathcal{V}_{s,k}} g_i^{(t)}\left(\phi'\left(\left\langle\boldsymbol{w}_s^{(t)},\boldsymbol{v}_{s,k}\right\rangle\right) + \beta\phi'\left(\left\langle\boldsymbol{w}_{-s}^{(t)},\boldsymbol{v}_{s,k}\right\rangle\right)\right)$$
$$\leq \frac{2\eta}{n}\sum_{t=\tilde{T}_{s,k}}^{T}\sum_{i\in\mathcal{V}_{s,k}}\exp\left(-y_i f_{\boldsymbol{W}^{(t)}}(\boldsymbol{X}_i)\right)$$
$$\leq \frac{2|\mathcal{V}_{s,k}|}{n}(\eta T^*)\exp\left(-\frac{3}{2}\log(\eta T^*)\right)$$
$$\leq \frac{2}{\sqrt{\eta T^*}} \leq \log(\eta T^*),$$

where the last inequality holds for any reasonably large $T^*$. Merging all inequalities together, we have $\gamma_s^{(T+1)}(s,k) + \beta\gamma_{-s}^{(T+1)}(s,k) \leq 4\log(\eta T^*)$.

Next, we will follow similar arguments to show that

$$\rho_{y_i}^{(T+1)}(i,p) + \beta\rho_{-y_i}^{(T+1)}(i,p) \leq 4\log(\eta T^*)$$

for each $i\in[n]$ and $p\in[P]\backslash\{p_i^*\}$.

Let $\tilde{T}_i^{(p)} \leq T$ be the smallest iteration such that $\rho_{y_i}^{(\tilde{T}_i^{(p)}+1)}(i,p) + \beta\rho_{-y_i}^{(\tilde{T}_i^{(p)}+1)}(i,p) > 2\log(\eta T^*)$. We assume the existence of $\tilde{T}_i^{(p)}$, as its absence would directly lead to our desired conclusion; to see why, note that the following holds, due to (15) and (11):

$$\rho_{y_i}^{(T+1)}(i,p) + \beta\rho_{-y_i}^{(T+1)}(i,p)$$
$$= \rho_{y_i}^{(T)}(i,p) + \beta\rho_{-y_i}^{(T)}(i,p) + \frac{\eta}{n}g_i^{(T)}\left(\phi'\left(\left\langle\boldsymbol{w}_s^{(t)},\boldsymbol{x}_i^{(p)}\right\rangle\right) + \beta\phi'\left(\left\langle\boldsymbol{w}_s^{(t)},\boldsymbol{x}_i^{(p)}\right\rangle\right)\right)\left\|\xi_i^{(p)}\right\|^2$$
$$\leq 2\log(\eta T^*) + 2\eta \leq 4\log(\eta T^*),$$

where the first inequality is due to $\left\|\xi_i^{(p)}\right\| \leq \frac{3}{2}\sigma_{\text{d}}^2 d$ and (A4), and the last inequality is due to (11).

Now suppose there exists such $\tilde{T}_i^{(p)} \leq T$. By (15), we have

$$\rho_{y_i}^{(T+1)}(i,p) + \beta\rho_{-y_i}^{(T+1)}(i,p)$$

$$= \rho_{y_i}^{(\tilde{T}_i^{(p)})}(i,p) + \beta \rho_{-y_i}^{(\tilde{T}_i^{(p)})}(i,p)$$

$$+ \sum_{t=\tilde{T}_i^{(p)}}^{T} \left( \rho_{y_i}^{(t+1)}(i,p) + \beta \rho_{-y_i}^{(t+1)}(i,p) - \rho_{y_i}^{(t)}(i,p) - \beta \rho_{-y_i}^{(t)}(i,p) \right)$$

$$\leq 2\log(\eta T^*) + \log(\eta T^*)$$

$$+ \frac{\eta}{n} \sum_{t=\tilde{T}_i^{(p)}+1}^{T} g_i^{(t)} \left( \phi' \left( \left\langle \boldsymbol{w}_s^{(t)}, \boldsymbol{x}_i^{(p)} \right\rangle \right) + \beta \phi' \left( \left\langle \boldsymbol{w}_{-s}^{(t)}, \boldsymbol{x}_i^{(p)} \right\rangle \right) \right) \left\| \xi_i^{(p)} \right\|^2$$

The inequality is due to $\rho_{y_i}^{(\tilde{T}_i^{(p)})}(i,p) + \beta \rho_{-y_i}^{(\tilde{T}_i^{(p)})}(i,p) \leq 2\log(\eta T^*)$ from our choice of $\tilde{T}_i^{(p)}$ and

$$\frac{\eta}{n} g_i^{(\tilde{T}_i^{(p)})} \left[ \phi' \left( \left\langle \boldsymbol{w}_s^{(\tilde{T}_i^{(p)})}, \boldsymbol{x}_i^{(p)} \right\rangle \right) + \beta \phi' \left( \left\langle \boldsymbol{w}_{-s}^{(\tilde{T}_i^{(p)})}, \boldsymbol{x}_i^{(p)} \right\rangle \right) \right] \left\| \xi_i^{(p)} \right\|^2 \leq 2\eta \leq \log(\eta T^*),$$

from $\left\| \xi_i^{(p)} \right\|^2 \leq \frac{3}{2}\sigma_d^2 d$, (A4), and (11).

For each $t = \tilde{T}_i^{(p)} + 1, \ldots, T$, if $i \in \mathcal{V}_{s,k}$, then we have

$$y_i f_{\boldsymbol{W}^{(t)}}(\boldsymbol{X}_i)$$

$$= \phi \left( \left\langle \boldsymbol{w}_{y_i}^{(t)}, \boldsymbol{x}_i^{(p)} \right\rangle \right) - \phi \left( \left\langle \boldsymbol{w}_{-y_i}^{(t)}, \boldsymbol{x}_i^{(p)} \right\rangle \right) + \sum_{q \in [P] \setminus \{p\}} \left( \phi \left( \left\langle \boldsymbol{w}_{y_i}^{(t)}, \boldsymbol{x}_i^{(p)} \right\rangle \right) - \phi \left( \left\langle \boldsymbol{w}_{-y_i}^{(t)}, \boldsymbol{x}_i^{(p)} \right\rangle \right) \right)$$

$$\geq \rho_{y_i}^{(t)}(i,p) + \beta \rho_{-y_i}^{(t)}(i,p) + \gamma_{y_i}^{(t)}(s,k) + \beta \gamma_{-y_i}^{(t)}(s,k)$$

$$+ \sum_{q \in [P] \setminus \{p, p_i^*\}} \left( \rho_{y_i}^{(t)}(i,q) + \beta \rho_{-y_i}^{(t)}(i,q) \right) - 2P \cdot o\left( \frac{1}{\text{polylog}(d)} \right)$$

$$\geq \frac{3}{2} \log(\eta T^*).$$

The first inequality is due to Lemma B.4 and the second inequality holds because from our choice of $t$, $\rho_{y_i}^{(t)}(i,p) + \beta \rho_{-y_i}^{(t)}(i,p) \geq 2\log(\eta T^*)$.

Therefore, we have

$$\frac{\eta}{n} \sum_{t=\tilde{T}_i^{(p)}+1}^{T} g_i^{(t)} \left( \phi' \left( \left\langle \boldsymbol{w}_{y_i}^{(t)}, \boldsymbol{x}_i^{(p)} \right\rangle \right) + \beta \phi' \left( \left\langle \boldsymbol{w}_{-y_i}^{(t)}, \boldsymbol{x}_i^{(p)} \right\rangle \right) \right) \left\| \xi_i^{(p)} \right\|^2$$

$$\leq \eta \sum_{t=\tilde{T}_i^{(p)}+1}^{T} \exp\left( -y_i f_{\boldsymbol{W}^{(t)}}(\boldsymbol{X}_i) \right) \leq (\eta T^*) \exp\left( -\frac{3}{2} \log(\eta T^*) \right)$$

$$\leq \frac{1}{\sqrt{\eta T^*}} \leq \log(\eta T^*),$$

where the first inequality is due to $\left\| \xi_i^{(p)} \right\|^2 \leq \frac{3}{2}\sigma_d^2$, (A4) and the last inequality holds for any reasonably large $T^*$. Merging all inequalities together, we conclude $\rho_{y_i}^{(T+1)}(i,p) + \beta \rho_{-y_i}^{(T+1)}(i,p) \leq 4\log(\eta T^*)$. □

### C.2.2 Learning Common Features

In the initial stages of training, the model quickly learns common features while exhibiting minimal overfitting to Gaussian noise.

First, we establish lower bounds on the number of iterations ensuring that noise coefficients $\rho_s^{(t)}(i,p)$ remain small, up to the order of $\frac{1}{P}$.

**Lemma C.2.** *Suppose the event $E_{\text{init}}$ occurs. There exists $\tilde{T} > \frac{n}{6\eta P \sigma_d^2 d}$ such that $\rho_s^{(t)}(i,p) \leq \frac{1}{4P}$ for all $0 \leq t < \tilde{T}, s \in \{\pm 1\}, i \in [n]$ and $p \in [P] \setminus \{p_i^*\}$.*

*Proof of Lemma C.2.* Let $\tilde{T}$ be the smallest iteration such that $\rho_s^{(\tilde{T})}(i,p) \geq \frac{1}{4P}$ for some $s \in \{\pm 1\}, i \in [n]$ and $p \in [P] \setminus \{p_i^*\}$. We assume the existence of $\tilde{T}$, as its absence would directly lead to our conclusion. Then, for any $0 \leq t < \tilde{T}$, we have

$$\rho_s^{(t+1)}(i,p) = \rho_s^{(t)}(i,p) + \frac{\eta}{n} g_i^{(t)} \phi'\left(\left\langle \boldsymbol{w}_s^{(t)}, \boldsymbol{x}_i^{(p)} \right\rangle\right) \left\|\xi_i^{(p)}\right\|^2 \leq \rho_s^{(t)}(i,p) + \frac{3\eta\sigma_{\mathrm{d}}^2 d}{2n},$$

where the inequality is due to $g_i^{(t)} < 1$, $\phi' \leq 1$, and $\left\|\xi_i^{(p)}\right\|^2 \leq \frac{3}{2}\sigma_{\mathrm{d}}^2 d$. Hence, we have

$$\frac{1}{4P} \leq \rho_s^{(\tilde{T})}(i,p) = \sum_{t=0}^{\tilde{T}-1} \left(\rho_s^{(t+1)}(i,p) - \rho_s^{(t)}(i,p)\right) < \frac{3\eta\sigma_{\mathrm{d}}^2 d}{2n}\tilde{T},$$

and we conclude $\tilde{T} > \frac{n}{6\eta P \sigma_{\mathrm{d}}^2 d}$ which is the desired result. $\qquad\square$

Next, we will show that the model learns common features in at least constant order within $\tilde{T}$ iterates.

**Lemma C.3.** *Suppose the event $E_{\mathrm{init}}$ occurs and $\rho_k = \omega\left(\frac{\sigma_{\mathrm{d}}^2 d}{\beta n}\right)$ for some $k \in [K]$. Then, for each $s \in \{\pm 1\}$, there exists $T_{s,k} \leq \frac{9n}{\eta\beta|\mathcal{V}_{s,k}|}$ such that $\gamma_s^{(t)}(s,k) + \beta\gamma_{-s}^{(t)}(s,k) \geq 1$ for any $t > T_{s,k}$.*

*Proof of Lemma C.3.* Suppose $\gamma_s^{(t)}(s,k) + \beta\gamma_{-s}^{(t)}(s,k) < 1$ for all $0 \leq t \leq \frac{n}{6\eta P \sigma_{\mathrm{d}}^2 d}$. For each $i \in \mathcal{V}_{s,k}$, we have

$$y_i f_{\boldsymbol{W}^{(t)}}(\boldsymbol{X}_i)$$

$$= \phi\left(\left\langle \boldsymbol{w}_s^{(t)}, \boldsymbol{v}_{s,k} \right\rangle\right) - \phi\left(\left\langle \boldsymbol{w}_{-s}^{(t)}, \boldsymbol{v}_{s,k} \right\rangle\right) + \sum_{p \in [P]\setminus\{p_i^*\}} \left(\phi\left(\left\langle \boldsymbol{w}_s^{(t)}, \boldsymbol{x}_i^{(p)} \right\rangle\right) - \phi\left(\left\langle \boldsymbol{w}_{-s}^{(t)}, \boldsymbol{x}_i^{(p)} \right\rangle\right)\right)$$

$$\leq \gamma_s^{(t)}(s,k) + \beta\gamma_{-s}^{(t)}(s,k) + \sum_{p \in [P]\setminus\{p_i^*\}} \left(\rho_s^{(t)}(i,p) + \beta\rho_{-s}^{(t)}(i,p)\right) + 2P \cdot o\left(\frac{1}{\mathrm{polylog}(d)}\right)$$

$$\leq 1 + 2P \cdot \frac{1}{4P} + 2P \cdot o\left(\frac{1}{\mathrm{polylog}(d)}\right)$$

$$\leq 2.$$

The first inequality is due to Lemma B.4, the second inequality holds since we can apply Lemma C.2, and the last inequality is due to (A1). Thus, $g_i^{(t)} = \frac{1}{1+\exp\left(y_i f_{\boldsymbol{W}^{(t)}}(\boldsymbol{X}_i)\right)} > \frac{1}{9}$ and we have

$$\gamma_s^{(t+1)}(s,k) + \beta\gamma_{-s}^{(t+1)}(s,k)$$

$$= \gamma_s^{(t)}(s,k) + \beta\gamma_{-s}^{(t)}(s,k) + \frac{\eta}{n}\sum_{i \in \mathcal{V}_{s,k}} g_i^{(t)}\left(\phi'\left(\left\langle \boldsymbol{w}_s^{(t)}, \boldsymbol{v}_{s,k} \right\rangle\right) + \beta\phi'\left(\left\langle \boldsymbol{w}_{-s}^{(t)}, \boldsymbol{v}_{s,k} \right\rangle\right)\right)$$

$$\geq \gamma_s^{(t)}(s,k) + \beta\gamma_{-s}^{(t)}(s,k) + \frac{\eta\beta|\mathcal{V}_{s,k}|}{9n}.$$

Notice that $|\mathcal{V}_{s,k}| = \rho_k n$. From the condition in the lemma statement, we have $\frac{9n}{\eta\beta|\mathcal{V}_{s,k}|} = o\left(\frac{n}{6\eta P \sigma_{\mathrm{d}}^2 d}\right)$. If we choose $t_0 \in \left[\frac{9n}{\eta\beta|\mathcal{V}_{s,k}|}, \frac{n}{6\eta P \sigma_{\mathrm{d}}^2 d}\right]$, then

$$1 > \gamma_s^{(t_0)}(s,k) + \beta\gamma_{-s}^{(t_0)}(s,k) \geq \frac{\eta\beta|\mathcal{V}_{s,k}|}{9n}t_0 \geq 1,$$

and this is contradictory; therefore, it cannot hold that $\gamma_s^{(t)}(s,k) + \beta\gamma_{-s}^{(t)}(s,k) < 1$ for all $0 \leq t \leq \frac{n}{6\eta P \sigma_{\mathrm{d}}^2 d}$. Hence, there exists $0 \leq T_{s,k} < \frac{n}{6\eta P \sigma_{\mathrm{d}}^2 d}$ such that $\gamma_s^{(T_{s,k}+1)}(s,k) + \beta\gamma_{-s}^{(T_{s,k}+1)}(s,k) \geq 1$ and choose the smallest one. Then we obtain

$$1 > \gamma_s^{(T_{s,k})}(s,k) + \beta\gamma_{-s}^{(T_{s,k})}(s,k) \geq \frac{\eta\beta|\mathcal{V}_{s,k}|}{9n}T_{s,k}.$$

Therefore, $T_{s,k} < \frac{9n}{\eta\beta|\mathcal{V}_{s,k}|}$ and this is what we desired. $\qquad\square$

**What We Have So Far.** For any common feature $\boldsymbol{v}_{s,k}$ with $s \in \{\pm 1\}$ and $k \in \mathcal{K}_C$, it satisfies $\rho_k = w\left(\frac{\sigma_{\mathrm{d}}^2 d}{\beta n}\right)$ due to (A4). By Lemma C.3, at any iterate $t \in \left[\bar{T}_1, T^*\right]$ with $\bar{T}_1 := \max_{s \in \{\pm 1\}, k \in \mathcal{K}_C} T_{s,k}$, the following properties hold if the event $E_{\mathrm{init}}$ occurs:

- (Learn common features): For any $s \in \{\pm 1\}$ and $k \in \mathcal{K}_C$,

$$\gamma_s^{(t)}(s,k) + \beta \gamma_{-s}^{(t)}(s,k) = \Omega(1).$$

- For any $s \in \{\pm 1\}, i \in [n]$, and $p \in [P] \setminus \{p_i^*\}$, $\rho_s^{(t)}(i,p) = \widetilde{\mathcal{O}}\left(\beta^{-1}\right)$.

### C.2.3 Overfitting (extremely) Rare Data

In the previous step, we have shown that common data can be well-classified by learning common features. In this step, we will show that the model correctly classifies (extremely) rare data by overfitting dominant noise instead of learning its features.

We first introduce lower bounds on the number of iterates such that feature coefficients $\gamma_s^{(t)}(s',k)$ remain small, up to the order of $\alpha^2 \beta^{-1}$. This lemma holds for any kind of features, but we will focus on (extremely) rare features. This does not contradict the results from Section C.2.2 for common features since the upper bound on the number of iterations in Lemma C.3 is larger than the lower bound on the number of iterations in this lemma.

**Lemma C.4.** *Suppose the event $E_{\mathrm{init}}$ occurs. For each $s \in \{\pm 1\}$ and $k \in [K]$, there exists $\tilde{T}_{s,k} > \frac{n\alpha^2}{\eta\beta|\mathcal{V}_{s,k}|}$ such that $\gamma_{s'}^{(t)}(s,k) \leq \alpha^2 \beta^{-1}$ for any $0 \leq t < \tilde{T}_{s,k}$ and $s' \in \{\pm 1\}$.*

*Proof of Lemma C.4.* Let $\tilde{T}_{s,k}$ be the smallest iterate such that $\gamma_{s'}^{(\tilde{T}_{s,k})}(s,k) > \alpha^2 \beta^{-1}$ for some $s' \in \{\pm 1\}$. We assume the existence of $\tilde{T}_{s,k}$, as its absence would directly lead to our conclusion.

For any $0 \leq t < \tilde{T}_{s,k}$,

$$\gamma_{s'}^{(t+1)}(s,k) = \gamma_{s'}^{(t)}(s,k) + \frac{\eta}{n}\sum_{i \in \mathcal{V}_{s,k}} g_i^{(t)} \phi'\left(\left\langle \boldsymbol{w}_{s'}^{(t)}, \boldsymbol{v}_{s,k}\right\rangle\right) \leq \gamma_{s'}^{(t)}(s,k) + \frac{\eta|\mathcal{V}_{s,k}|}{n},$$

and we have

$$\alpha^2\beta^{-1} < \gamma_{s'}^{(\tilde{T}_{s,k})}(s,k) = \sum_{t=0}^{\tilde{T}_{s,k}-1}\left(\gamma_{s'}^{(t+1)}(s,k) - \gamma_{s'}^{(t)}(s,k)\right) \leq \frac{\eta|\mathcal{V}_{s,k}|}{n}\tilde{T}_{s,k}.$$

We conclude $\tilde{T}_{s,k} > \frac{n\alpha^2}{\eta\beta|\mathcal{V}_{s,k}|}$ which is the desired result. $\qquad\square$

Next, we will show that the model overfits (extremely) rare data by memorizing dominant noise patches in at least constant order within $\tilde{T}_{s,k}$ iterates.

**Lemma C.5.** *Suppose the event $E_{\mathrm{init}}$ occurs and $\rho_k = o\left(\frac{\alpha^2\sigma_{\mathrm{d}}^2 d}{n}\right)$. Then, for each $i \in \mathcal{V}_{s,k}$, there exists $T_i \in \left[\bar{T}_1, \frac{18n}{\eta\beta\sigma_{\mathrm{d}}^2 d}\right]$ such that*

$$\sum_{p \in [P]\setminus\{p_i^*\}}\left(\rho_s^{(t)}(i,p) + \beta\rho_{-s}^{(t)}(i,p)\right) \geq 1,$$

*for any $t > T_i$.*

*Proof of Lemma C.5.* Suppose $\sum_{p \in [P]\setminus\{p_i^*\}}\left(\rho_s^{(t)}(i,p) + \beta\rho_{-s}^{(t)}(i,p)\right) < 1$ for $0 \leq t \leq \frac{n\alpha^2}{\eta\beta|\mathcal{V}_{s,k}|}$.

From Lemma B.4 and Lemma C.4, we have

$$y_i f_{\boldsymbol{W}^{(t)}}(\boldsymbol{X}_i)$$

$$= \phi\left(\left\langle \boldsymbol{w}_s^{(t)}, \boldsymbol{v}_{s,k} \right\rangle\right) - \phi\left(\left\langle \boldsymbol{w}_{-s}^{(t)}, \boldsymbol{v}_{s,k} \right\rangle\right) + \sum_{p \in [P] \setminus \{p_i^*\}} \left( \phi\left(\left\langle \boldsymbol{w}_s^{(t)}, \boldsymbol{x}_i^{(p)} \right\rangle\right) - \phi\left(\left\langle \boldsymbol{w}_{-s}^{(t)}, \boldsymbol{x}_i^{(p)} \right\rangle\right) \right)$$

$$\leq \gamma_s^{(t)}(s,k) + \beta\gamma_{-s}^{(t)}(s,k) + \sum_{p \in [P] \setminus \{p_i^*\}} \left( \rho_s^{(t)}(i,p) + \beta\rho_{-s}^{(t)}(i,p) \right) + 2P \cdot o\left(\frac{1}{\text{polylog}(d)}\right)$$

$$\leq (1+\beta)\alpha^2\beta^{-1} + 1 + 2P \cdot o\left(\frac{1}{\text{polylog}(d)}\right)$$

$$\leq 2,$$

where the last inequality is due to (10). Thus, we have $g_i^{(t)} = \frac{1}{1+\exp\left(y_i f_{\boldsymbol{W}^{(t)}}(\boldsymbol{X}_i)\right)} \geq \frac{1}{9}$. Also,

$$\rho_s^{(t+1)}(i,\tilde{p}_i) + \beta\rho_{-s}^{(t+1)}(i,\tilde{p}_i)$$

$$= \rho_s^{(t)}(i,\tilde{p}_i) + \beta\rho_{-s}^{(t)}(i,\tilde{p}_i) + \frac{\eta}{n}g_i^{(t)}\left(\phi'\left(\left\langle \boldsymbol{w}_s^{(t)}, \boldsymbol{x}_i^{(\tilde{p}_i)} \right\rangle\right) + \beta\phi'\left(\left\langle \boldsymbol{w}_{-s}^{(t)}, \boldsymbol{x}_i^{(\tilde{p}_i)} \right\rangle\right)\right)\left\|\xi_i^{(\tilde{p}_i)}\right\|^2$$

$$\geq \rho_s^{(t)}(i,\tilde{p}_i) + \beta\rho_{-s}^{(t)}(i,\tilde{p}_i) + \frac{\eta\beta\sigma_d^2 d}{18n},$$

where the last inequality is due to $\left\|\xi_i^{(\tilde{p}_i)}\right\|^2 \geq \frac{1}{2}\sigma_d^2 d$ and $\phi' \geq \beta$.

Notice that $|\mathcal{V}_{s,k}| = \rho_k n$. From the given condition in the lemma statement, we have $\frac{18n}{\eta\beta\sigma_d^2 d} = o\left(\frac{n\alpha^2}{\eta\beta|\mathcal{V}_{s,k}|}\right)$. If we choose $t_0 \in \left[\frac{18n}{\eta\beta\sigma_d^2 d}, \frac{n\alpha^2}{\eta\beta|\mathcal{V}_{s,k}|}\right]$, then we have

$$1 > \sum_{p \in [P] \setminus \{p_i^*\}} \left( \rho_s^{(t_0)}(i,p) + \beta\rho_{-s}^{(t_0)}(i,p) \right) \geq \rho_s^{(t_0)}(i,\tilde{p}_i) + \beta\rho_{-s}^{(t_0)}(i,\tilde{p}_i) \geq \frac{\eta\beta\sigma_d^2 d}{18n}t_0 \geq 1.$$

This is a contradiction; therefore it cannot hold that $\sum_{p \in [P] \setminus \{p_i^*\}} \left( \rho_s^{(t)}(i,p) + \beta\rho_{-s}^{(t)}(i,p) \right) < 1$ for all $0 \leq t \leq \frac{n\alpha^2}{\eta\beta|\mathcal{V}_{s,k}|}$. Hence, we can choose the smallest $0 \leq T_i < \frac{n\alpha^2}{\eta\beta|\mathcal{V}_{s,k}|}$ such that $\sum_{p \in [P] \setminus \{p_i^*\}} \left( \rho_s^{(T_i+1)}(i,p) + \beta\rho_{-s}^{(T_i+1)}(i,p) \right) \geq 1$.

For any $0 \leq t < T_i$,

$$1 \geq \sum_{p \in [P] \setminus \{p_i^*\}} \left( \rho_s^{(T_i)}(i,p) + \beta\rho_{-s}^{(T_i)}(i,p) \right) \geq \rho_s^{(T_i)}(i,\tilde{p}_i) + \beta\rho_{-s}^{(T_i)}(i,\tilde{p}_i) \geq \frac{\eta\beta\sigma_d^2 d}{18n}T_i,$$

and we conclude that $T_i \leq \frac{18n}{\eta\beta\sigma_d^2 d}$.

Lastly, we move on to prove $T_i > \bar{T}_1$. Combining Lemma C.2 and Lemma C.3 leads to

$$\sum_{p \in [P] \setminus \{p_i^*\}} \left( \rho_s^{(\bar{T}_1)}(i,p) + \beta\rho_{-s}^{(\bar{T}_1)}(i,p) \right) \leq \frac{1}{2}.$$

Thus, we have $T_i > \bar{T}_1$ and this is what we desired. $\qquad\square$

**What We Have So Far.** For any $k \in \mathcal{K}_R \cup \mathcal{K}_E$, it satisfies $\rho_k = o\left(\frac{\alpha^2\sigma_d^2 d}{n}\right)$ due to (A5). By Lemma C.5 at iterate $t \in [T_{\text{ERM}}, T^*]$ with

$$T_{\text{ERM}} := \max_{\substack{s \in \{\pm 1\} \\ k \in \mathcal{K}_R \cup \mathcal{K}_E}} \max_{i \in \mathcal{V}_{s,k}} T_i \quad \in \left[\bar{T}_1, T^*\right]$$

the following properties hold if the event $E_{\text{init}}$ occurs:

- (Learn common features): For $s \in \{\pm 1\}$ and $k \in \mathcal{K}_C$,

$$\gamma_s^{(t)}(s,k) + \beta\gamma_{-s}^{(t)}(s,k) = \Omega(1),$$

- (Overfit (extremely) rare data): For any $s \in \{\pm 1\}$, $k \in \mathcal{K}_R \cup \mathcal{K}_E$, and $i \in \mathcal{V}_{s,k}$,

$$\sum_{p \in [P] \setminus \{p_i^*\}} \left( \rho_s^{(t)}(i,p) + \beta \rho_{-s}^{(t)}(i,p) \right) = \Omega(1),$$

- (Do not learn (extremely) rare features at $T_{\mathrm{ERM}}$): For any $s, s' \in \{\pm 1\}$ and $k \in \mathcal{K}_R \cup \mathcal{K}_E$, $\gamma_{s'}^{(T_{\mathrm{ERM}})}(s,k) \leq \alpha^2 \beta^{-1}$.

- For any $s \in \{\pm 1\}$, $i \in [n]$, and $p \in [P] \setminus \{p_i^*\}$, $\rho_s^{(t)}(i,p) = \widetilde{\mathcal{O}}\left( \beta^{-1} \right)$.

### C.2.4 ERM cannot Learn (extremely) Rare Features Within Polynomial Times

In this step, we will show that ERM cannot learn (extremely) rare features within the maximum admissible iterations $T^* = \frac{\mathrm{poly}(d)}{\eta}$.

From now on, we fix any $s^* \in \{\pm 1\}$ and $k^* \in \mathcal{K}_R \cup \mathcal{K}_E$. Recall that we defined the set $\mathcal{W}$ and the function $Q^{(s^*, k^*)} : \mathcal{W} \to \mathbb{R}^{d \times 2}$ in Lemma B.5. Let us omit superscripts for simplicity. For each iteration $t$, $Q(\boldsymbol{W}^{(t)})$ represents the cumulative updates contributed by data points with feature vector $\boldsymbol{v}_{s^*, k^*}$ until $t$-th iteration. We will sequentially introduce several technical lemmas and by combining these lemmas, quantify update by data with feature vector $\boldsymbol{v}_{s^*, k^*}$ after $T_{\mathrm{ERM}}$ and derive our conclusion.

Let us define $\boldsymbol{W}^* = \{\boldsymbol{w}_1^*, \boldsymbol{w}_{-1}^*\}$, where

$$\boldsymbol{w}_s^* = \boldsymbol{w}_s^{(T_{\mathrm{ERM}})} + M \sum_{i \in \mathcal{V}_{s^*, k^*}} \frac{\xi_i^{(\tilde{p}_i)}}{\left\| \xi_i^{(\tilde{p}_i)} \right\|^2},$$

for each $s \in \{\pm 1\}$ with $M = 4\beta^{-1} \log \left( \frac{2\eta \beta^2 T^*}{\alpha^2} \right)$. Note that (12), $\beta < 1$, and $T^* = \frac{\mathrm{poly}(d)}{\eta}$ together imply $M = \widetilde{\mathcal{O}}\left( \beta^{-1} \right)$. Note that $\boldsymbol{W}^{(t)}, \boldsymbol{W}^* \in \mathcal{W}$ for any $t \geq 0$.

**Lemma C.6.** *Suppose the event $E_{\mathrm{init}}$ occurs. Then,*

$$\left\| Q\left( \boldsymbol{W}^{(T_{\mathrm{ERM}})} \right) - Q(\boldsymbol{W}^*) \right\|^2 \leq 12 M^2 |\mathcal{V}_{s^*, k^*}| \sigma_{\mathrm{d}}^{-2} d^{-1},$$

*where $\|\cdot\|$ denotes the Frobenius norm.*

*Proof of Lemma C.6.* For each $s \in \{\pm 1\}$,

$$ss^* \left( Q_s\left( \boldsymbol{w}_s^* \right) - Q_s\left( \boldsymbol{w}_s^{(T_{\mathrm{ERM}})} \right) \right)$$

$$= Q_s \left( ss^* M \sum_{i \in \mathcal{V}_{s^*, k^*}} \frac{\xi_i^{(\tilde{p}_i)}}{\left\| \xi_i^{(\tilde{p}_i)} \right\|} \right)$$

$$= M \sum_{i \in \mathcal{V}_{s^*, k^*}} \frac{\xi_i^{(\tilde{p}_i)}}{\left\| \xi_i^{(\tilde{p}_i)} \right\|^2} + \alpha M \left( \sum_{i \in \mathcal{F}_s \cap \mathcal{V}_{s^*, k^*}} \frac{\boldsymbol{v}_{s,1}}{\left\| \xi_i^{(\tilde{p}_i)} \right\|^2} + \sum_{i \in \mathcal{F}_{-s} \cap \mathcal{V}_{s^*, k^*}} \frac{\boldsymbol{v}_{-s,1}}{\left\| \xi_i^{(\tilde{p}_i)} \right\|^2} \right),$$

and we have

$$\left\| Q\left( \boldsymbol{W}^{(T_{\mathrm{ERM}})} \right) - Q(\boldsymbol{W}^*) \right\|^2$$

$$= \left\| Q_1(\boldsymbol{w}_1^*) - Q_1\left( \boldsymbol{w}_1^{(T_{\mathrm{ERM}})} \right) \right\|^2 + \left\| Q_{-1}(\boldsymbol{w}_{-1}^*) - Q_{-1}\left( \boldsymbol{w}_{-1}^{(T_{\mathrm{ERM}})} \right) \right\|^2$$

$$\leq 2M^2 \left( \sum_{i \in \mathcal{V}_{s^*, k^*}} \left\| \xi_i^{(\tilde{p}_i)} \right\|^{-2} + \sum_{i,j \in \mathcal{V}_{s^*, k^*}, i \neq j} \frac{\left| \left\langle \xi_i^{(\tilde{p}_i)}, \xi_j^{(\tilde{p}_j)} \right\rangle \right|}{\left\| \xi_i^{(\tilde{p}_i)} \right\|^2 \left\| \xi_j^{(\tilde{p}_j)} \right\|^2} \right)$$

$$+ 2M^2 \left( \alpha^2 \left( \sum_{i \in \mathcal{F}_s \cap \mathcal{V}_{s^*, k^*}} \left\| \xi_i^{(\tilde{p}_i)} \right\|^{-2} \right)^2 + \alpha^2 \left( \sum_{i \in \mathcal{F}_{-s} \cap \mathcal{V}_{s^*, k^*}} \left\| \xi_i^{(\tilde{p}_i)} \right\|^{-2} \right)^2 \right).$$

From the event $E_{\mathrm{init}}$ defined in Lemma B.2 and (A2), we have

$$\sum_{i, j \in \mathcal{V}_{s^*, k^*}, i \neq j} \frac{\left| \left\langle \xi_i^{(\tilde{p}_i)}, \xi_j^{(\tilde{p}_j)} \right\rangle \right|}{\left\| \xi_i^{(\tilde{p}_i)} \right\|^2 \left\| \xi_j^{(\tilde{p}_j)} \right\|^2} \leq \sum_{i \in \mathcal{V}_{s^*, k^*}} \sum_{j \in \mathcal{V}_{s^*, k^*}} \left\| \xi_i^{(\tilde{p}_i)} \right\|^{-2} \widetilde{\mathcal{O}} \left( d^{-\frac{1}{2}} \right)$$

$$\leq \sum_{i \in \mathcal{V}_{s^*, k^*}} \left\| \xi_i^{(\tilde{p}_i)} \right\|^{-2} \widetilde{\mathcal{O}} \left( n d^{-\frac{1}{2}} \right)$$

$$\leq \sum_{i \in \mathcal{V}_{s^*, k^*}} \left\| \xi_i^{(\tilde{p}_i)} \right\|^{-2}$$

In addition, we have

$$\alpha^2 \left( \sum_{i \in \mathcal{F}_s \cap \mathcal{V}_{s^*, k^*}} \left\| \xi_i^{(\tilde{p}_i)} \right\|^{-2} \right)^2 + \alpha^2 \left( \sum_{i \in \mathcal{F}_{-s} \cap \mathcal{V}_{s^*, k^*}} \left\| \xi_i^{(\tilde{p}_i)} \right\|^{-2} \right)^2$$

$$\leq \left( \sum_{i \in \mathcal{F}_s \cap \mathcal{V}_{s^*, k^*}} \left\| \xi_i^{(\tilde{p}_i)} \right\|^{-2} \right)^2 + \left( \sum_{i \in \mathcal{F}_{-s} \cap \mathcal{V}_{s^*, k^*}} \left\| \xi_i^{(\tilde{p}_i)} \right\|^{-2} \right)^2$$

$$\leq \sum_{i \in \mathcal{F}_s \cap \mathcal{V}_{s^*, k^*}} \left\| \xi_i^{(\tilde{p}_i)} \right\|^{-2} + \sum_{i \in \mathcal{F}_{-s} \cap \mathcal{V}_{s^*, k^*}} \left\| \xi_i^{(\tilde{p}_i)} \right\|^{-2}$$

$$= \sum_{i \in \mathcal{V}_{s^*, k^*}} \left\| \xi_i^{(\tilde{p}_i)} \right\|^{-2},$$

where the first inequality is due to $\alpha < 1$ and the second inequality is due to $\sum_{i \in \mathcal{V}_{s^*, k^*}} \left\| \xi_i^{(\tilde{p}_i)} \right\|^{-2} \leq 2 |\mathcal{V}_{s^*, k^*}| \sigma_{\mathrm{d}}^{-2} d^{-1} < 1$ from (A5). Hence, from $E_{\mathrm{init}}$, we obtain

$$\left\| Q \left( \boldsymbol{W}^{(T_{\mathrm{ERM}})} \right) - Q(\boldsymbol{W}^*) \right\|^2 \leq 6 M^2 \sum_{i \in \mathcal{V}_{s^*, k^*}} \left\| \xi_i^{(\tilde{p}_i)} \right\|^{-2} \leq 12 M^2 |\mathcal{V}_{s^*, k^*}| \sigma_{\mathrm{d}}^{-2} d^{-1}.$$

$\square$

**Lemma C.7.** *Suppose the $E_{\mathrm{init}}$ occurs. For any $t \geq T_{\mathrm{ERM}}$ and $i \in \mathcal{V}_{s^*, k^*}$, it holds that*

$$\langle y_i \nabla_{\boldsymbol{W}} f_{\boldsymbol{W}^{(t)}}(\boldsymbol{X}_i), Q(\boldsymbol{W}^*) \rangle \geq \frac{M \beta}{2}.$$

*Proof of Lemma C.7.* We have

$$\langle y_i \nabla_{\boldsymbol{W}} f_{\boldsymbol{W}^{(t)}}(\boldsymbol{X}_i), Q(\boldsymbol{W}^*) \rangle$$

$$= \sum_{p \in [P]} \left( \phi' \left( \left\langle \boldsymbol{w}_{s^*}^{(t)}, \boldsymbol{x}_i^{(p)} \right\rangle \right) \left\langle Q_{s^*}(\boldsymbol{w}_{s^*}^*), \boldsymbol{x}_i^{(p)} \right\rangle - \phi' \left( \left\langle \boldsymbol{w}_{-s^*}^{(t)}, \boldsymbol{x}_i^{(p)} \right\rangle \right) \left\langle Q_{-s^*}(\boldsymbol{w}_{-s^*}^*), \boldsymbol{x}_i^{(p)} \right\rangle \right).$$

For any $s \in \{\pm 1\}$ and $p \in [P] \setminus \{p_i^*, \tilde{p}_i\}$,

$$s s^* \left\langle Q_s(\boldsymbol{w}_s^*), \xi_i^{(p)} \right\rangle$$

$$= \rho_s^{(T_{\mathrm{ERM}})}(i, p) + \sum_{\substack{j \in \mathcal{V}_{s^*, k^*}, q \in [P] \setminus \{p_j^*\} \\ (j, q) \neq (i, p)}} \rho_s^{(T_{\mathrm{ERM}})}(j, q) \frac{\left\langle \xi_i^{(p)}, \xi_j^{(q)} \right\rangle}{\left\| \xi_j^{(q)} \right\|^2} + \sum_{j \in \mathcal{V}_{s^*, k^*}} M \frac{\left\langle \xi_i^{(p)}, \xi_j^{(\tilde{p}_j)} \right\rangle}{\left\| \xi_j^{(\tilde{p}_j)} \right\|^2}$$

$$\geq -\widetilde{\mathcal{O}}\left(nP\beta^{-1}\sigma_{\mathrm{d}}\sigma_{\mathrm{b}}^{-1}d^{-\frac{1}{2}}\right) - \widetilde{\mathcal{O}}\left(nM\sigma_{\mathrm{b}}\sigma_{\mathrm{d}}^{-1}d^{-\frac{1}{2}}\right)$$

$$= -o\left(\frac{1}{\mathrm{polylog}(d)}\right), \tag{16}$$

where the last equality is due to (9) and $M = \widetilde{\mathcal{O}}\left(\beta^{-1}\right)$. Also, for any $s \in \{\pm 1\}$, $ss^*\langle Q_s(\boldsymbol{w}_s^*), \boldsymbol{v}_{s^*,k^*}\rangle = \gamma_s^{(T_{\mathrm{ERM}})}(s^*,k^*) \geq 0$. In addition,

$$ss^*\left\langle Q_s(\boldsymbol{w}_s^*), \boldsymbol{x}_i^{(\tilde{p}_i)}\right\rangle$$

$$= ss^*\left\langle Q_s(\boldsymbol{w}_s^*), \xi_i^{(\tilde{p}_i)}\right\rangle + ss^*\left\langle Q_s(\boldsymbol{w}_s^*), \boldsymbol{x}_i^{(\tilde{p}_i)} - \xi_i^{(\tilde{p}_i)}\right\rangle$$

$$\geq ss^*\left\langle Q_s(\boldsymbol{w}_s^*), \xi_i^{(\tilde{p}_i)}\right\rangle - \widetilde{\mathcal{O}}\left(\alpha^2\beta^{-1}\rho_{k^*}n\sigma_{\mathrm{d}}^{-2}d^{-1}\right)$$

$$= M + \rho_s^{(T_{\mathrm{ERM}})}(i,\tilde{p}_i) + \sum_{\substack{j\in\mathcal{V}_{s^*,k^*},q\in[P]\setminus\{p_i^*\} \\ (j,q)\neq(i,\tilde{p}_i)}} \rho_s^{(T_{\mathrm{ERM}})}(j,q)\frac{\left\langle \xi_i^{(\tilde{p}_i)},\xi_j^{(q)}\right\rangle}{\left\|\xi_j^{(q)}\right\|^2}$$

$$+ \sum_{j\in\mathcal{V}_{s^*,k^*}\setminus\{i\}} M\frac{\left\langle \xi_i^{(\tilde{p}_i)},\xi_j^{(\tilde{p}_j)}\right\rangle}{\left\|\xi_j^{(\tilde{p}_j)}\right\|^2} - \widetilde{\mathcal{O}}\left(\alpha^2\beta^{-1}\rho_{k^*}n\sigma_{\mathrm{d}}^{-2}d^{-1}\right)$$

$$\geq M - \widetilde{\mathcal{O}}\left(nP\beta^{-1}\sigma_{\mathrm{d}}\sigma_{\mathrm{b}}^{-1}d^{-\frac{1}{2}}\right) - \widetilde{\mathcal{O}}\left(\alpha^2\beta^{-1}\rho_{k^*}n\sigma_{\mathrm{d}}^{-2}d^{-1}\right)$$

$$= M - o\left(\frac{1}{\mathrm{polylog}(d)}\right)$$

$$\geq \frac{M}{2}, \tag{17}$$

where the first inequality is due to the definition of $Q$ and the second-to-last line is due to (9) and (A7).

Hence, applying (16) and (17) for $s = s^*, -s^*$ and combining with $\phi' \geq \beta$, we have

$$\langle y_i \nabla_{\boldsymbol{W}} f_{\boldsymbol{W}^{(t)}}(\boldsymbol{X}_i), Q(\boldsymbol{W}^*)\rangle \geq M\beta - o\left(\frac{1}{\mathrm{polylog}(d)}\right) \geq \frac{M\beta}{2}.$$

$\square$

By combining Lemma C.6 and Lemma C.7, we can obtain the following result.

**Lemma C.8.** *Suppose the event $E_{\mathrm{init}}$ occurs.*

$$\frac{\eta}{n}\sum_{t=T_{\mathrm{ERM}}}^{T^*}\sum_{i\in\mathcal{V}_{s^*,k^*}}\ell\left(y_i f_{\boldsymbol{W}^{(t)}}(\boldsymbol{X}_i)\right) \leq \left\|Q\left(\boldsymbol{W}^{(T_{\mathrm{ERM}})}\right) - Q(\boldsymbol{W}^*)\right\|^2 + 2\eta T^* e^{-\frac{M\beta}{4}},$$

*where $\|\cdot\|$ denotes the Frobenius norm.*

*Proof of Lemma C.8.* Note that for any $T_{\mathrm{ERM}} \leq t < T^*$,

$$Q\left(\boldsymbol{W}^{(t+1)}\right) = Q\left(\boldsymbol{W}^{(t)}\right) - \frac{\eta}{n}\nabla_{\boldsymbol{W}}\sum_{i\in\mathcal{V}_{s^*,k^*}}\ell\left(y_i f_{\boldsymbol{W}^{(t)}}(\boldsymbol{X}_i)\right),$$

and thus

$$\left\|Q\left(\boldsymbol{W}^{(t)}\right) - Q(\boldsymbol{W}^*)\right\|^2 - \left\|Q\left(\boldsymbol{W}^{(t+1)}\right) - Q(\boldsymbol{W}^*)\right\|^2$$

$$= \frac{2\eta}{n}\left\langle \nabla_{\boldsymbol{W}}\sum_{i\in\mathcal{V}_{s^*,k^*}}\ell\left(y_i f_{\boldsymbol{W}^{(t)}}(\boldsymbol{X}_i)\right), Q\left(\boldsymbol{W}^{(t)}\right) - Q(\boldsymbol{W}^*)\right\rangle - \frac{\eta^2}{n^2}\left\|\nabla_{\boldsymbol{W}}\sum_{i\in\mathcal{V}_{s^*,k^*}}\ell\left(y_i f_{\boldsymbol{W}^{(t)}}(\boldsymbol{X}_i)\right)\right\|^2$$

$$= \frac{2\eta}{n} \left\langle \nabla_{\boldsymbol{W}} \sum_{i \in \mathcal{V}_{s^*,k^*}} \ell(y_i f_{\boldsymbol{W}^{(t)}}(\boldsymbol{X}_i)), Q\left(\boldsymbol{W}^{(t)}\right) \right\rangle$$

$$- \frac{2\eta}{n} \sum_{i \in \mathcal{V}_{s^*,k^*}} \ell'(y_i f_{\boldsymbol{W}^{(t)}}(\boldsymbol{X}_i)) \left\langle \nabla_{\boldsymbol{W}} y_i f_{\boldsymbol{W}^{(t)}}(\boldsymbol{X}_i), Q\left(\boldsymbol{W}^*\right) \right\rangle - \frac{\eta^2}{n^2} \left\| \nabla_{\boldsymbol{W}} \sum_{i \in \mathcal{V}_{s^*,k^*}} \ell\left(y_i f_{\boldsymbol{W}^{(t)}}(\boldsymbol{X}_i)\right) \right\|^2$$

$$\geq \frac{2\eta}{n} \left\langle \nabla_{\boldsymbol{W}} \sum_{i \in \mathcal{V}_{s^*,k^*}} \ell(y_i f_{\boldsymbol{W}^{(t)}}(\boldsymbol{X}_i)), Q\left(\boldsymbol{W}^{(t)}\right) \right\rangle$$

$$- \frac{M\beta\eta}{n} \sum_{i \in \mathcal{V}_{s^*,k^*}} \ell'(y_i f_{\boldsymbol{W}^{(t)}}(\boldsymbol{X}_i)) - \frac{\eta^2}{n^2} \left\| \nabla_{\boldsymbol{W}} \sum_{i \in \mathcal{V}_{s^*,k^*}} \ell\left(y_i f_{\boldsymbol{W}^{(t)}}(\boldsymbol{X}_i)\right) \right\|^2,$$

where the last inequality is due to Lemma C.7. By the chain rule, we have

$$\left\langle \nabla_{\boldsymbol{W}} \sum_{i \in \mathcal{V}_{s^*,k^*}} \ell(y_i f_{\boldsymbol{W}^{(t)}}(\boldsymbol{X}_i)), Q\left(\boldsymbol{W}^{(t)}\right) \right\rangle$$

$$= \sum_{i \in \mathcal{V}_{s^*,k^*}} \left[ \ell'(y_i f_{\boldsymbol{W}^{(t)}}(\boldsymbol{X}_i)) \right.$$

$$\left. \times \sum_{p \in [P]} \left( \phi'\left(\left\langle \boldsymbol{w}_{s^*}^{(t)}, \boldsymbol{x}_i^{(p)} \right\rangle\right) \left\langle Q_{s^*}\left(\boldsymbol{w}_{s^*}^{(t)}\right), \boldsymbol{x}_i^{(p)} \right\rangle - \phi'\left(\left\langle \boldsymbol{w}_{-s^*}^{(t)}, \boldsymbol{x}_i^{(p)} \right\rangle\right) \left\langle Q_{-s^*}\left(\boldsymbol{w}_{-s^*}^{(t)}\right), \boldsymbol{x}_i^{(p)} \right\rangle \right) \right].$$

For each $s \in \{\pm 1\}$, $i \in \mathcal{V}_{s^*,k^*}$, and $p \in [P]$,

$$\left| \left\langle \boldsymbol{w}_s^{(t)}, \boldsymbol{x}_i^{(p)} \right\rangle - \left\langle Q_s\left(\boldsymbol{w}_s^{(t)}\right), \boldsymbol{x}_i^{(p)} \right\rangle \right|$$

$$= \left| \left\langle \boldsymbol{w}_s^{(t)} - Q_s\left(\boldsymbol{w}_s^{(t)}\right), \boldsymbol{x}_i^{(p)} \right\rangle \right|$$

$$\leq \sum_{j \in [n] \setminus \mathcal{V}_{s^*,k^*}, q \in [P] \setminus \{p_i^*\}} \left| \left\langle \rho_s^{(t)}(j,q) \frac{\xi_j^{(q)}}{\left\| \xi_j^{(q)} \right\|^2}, \boldsymbol{x}_i^{(p)} \right\rangle \right|$$

$$+ \alpha \sum_{j \in \mathcal{F}_1 \setminus \mathcal{V}_{s^*,k^*}} \rho_s^{(t)}(j,\tilde{p}_j) \left\| \xi_j^{(\tilde{p}_j)} \right\|^{-2} \left| \left\langle \boldsymbol{v}_{1,1}, \boldsymbol{x}_i^{(p)} \right\rangle \right|$$

$$+ \alpha \sum_{j \in \mathcal{F}_{-1} \setminus \mathcal{V}_{s^*,k^*}} \rho_s^{(t)}(j,\tilde{p}_j) \left\| \xi_j^{(\tilde{p}_j)} \right\|^{-2} \left| \left\langle \boldsymbol{v}_{-1,1}, \boldsymbol{x}_i^{(p)} \right\rangle \right|$$

$$\leq \widetilde{\mathcal{O}}\left( n P \beta^{-1} \sigma_{\mathrm{d}} \sigma_{\mathrm{b}}^{-1} d^{-\frac{1}{2}} \right) + \widetilde{\mathcal{O}}\left( \alpha^2 \beta^{-1} n \sigma_{\mathrm{d}}^{-2} d^{-1} \right)$$

$$= o\left( \frac{1}{\mathrm{polylog}(d)} \right),$$

where the last inequality is due to Lemma C.1 and the event $E_{\mathrm{init}}$. By Lemma F.1,

$$\sum_{p \in [P]} \left( \phi'\left(\left\langle \boldsymbol{w}_{s^*}^{(t)}, \boldsymbol{x}_i^{(p)} \right\rangle\right) \left\langle Q_{s^*}\left(\boldsymbol{w}_{s^*}^{(t)}\right), \boldsymbol{x}_i^{(p)} \right\rangle - \phi'\left(\left\langle \boldsymbol{w}_{-s^*}^{(t)}, \boldsymbol{x}_i^{(p)} \right\rangle\right) \left\langle Q_{-s^*}\left(\boldsymbol{w}_{s^*}^{(t)}\right), \boldsymbol{x}_i^{(p)} \right\rangle \right)$$

$$\leq \sum_{p \in [P]} \left( \phi\left(\left\langle \boldsymbol{w}_{s^*}^{(t)}, \boldsymbol{x}_i^{(p)} \right\rangle\right) - \phi\left(\left\langle \boldsymbol{w}_{-s^*}^{(t)}, \boldsymbol{x}_i^{(p)} \right\rangle\right) \right) + rP + o\left( \frac{1}{\mathrm{polylog}(d)} \right)$$

$$= y_i f_{\boldsymbol{W}^{(t)}}(\boldsymbol{X}_i) + o\left( \frac{1}{\mathrm{polylog}(d)} \right)$$

where the last equality is due to $r = o\left( \frac{1}{\mathrm{polylog}(d)} \right)$. Therefore, we have

$$\left\| Q\left(\boldsymbol{W}^{(t)}\right) - Q\left(\boldsymbol{W}^*\right) \right\|^2 - \left\| Q\left(\boldsymbol{W}^{(t+1)}\right) - Q(\boldsymbol{W}^*) \right\|^2$$

$$\geq \frac{2\eta}{n} \sum_{i \in \mathcal{V}_{s^*,k^*}} \ell'(y_i f_{\boldsymbol{W}^{(t)}}(\boldsymbol{X}_i)) \left(y_i f_{\boldsymbol{W}^{(t)}}(\boldsymbol{X}_i) + o\left(\frac{1}{\text{polylog}(d)}\right) - \frac{M\beta}{2}\right)$$

$$- \frac{\eta^2}{n^2} \left\|\nabla_{\boldsymbol{W}} \sum_{i \in \mathcal{V}_{s^*,k^*}} \ell(y_i f_{\boldsymbol{W}^{(t)}}(\boldsymbol{X}_i))\right\|^2$$

$$\geq \frac{2\eta}{n} \sum_{i \in \mathcal{V}_{s^*,k^*}} \ell'(y_i f_{\boldsymbol{W}^{(t)}}(\boldsymbol{X}_i)) \left(y_i f_{\boldsymbol{W}^{(t)}}(\boldsymbol{X}_i) - \frac{M\beta}{4}\right)$$

$$- \frac{\eta^2}{n^2} \left\|\nabla_{\boldsymbol{W}} \sum_{i \in \mathcal{V}_{s^*,k^*}} \ell(y_i f_{\boldsymbol{W}^{(t)}}(\boldsymbol{X}_i))\right\|^2.$$

From the convexity of $\ell(\cdot)$,

$$\sum_{i \in \mathcal{V}_{s^*,k^*}} \ell'(y_i f_{\boldsymbol{W}^{(t)}}(\boldsymbol{X}_i)) \left(y_i f_{\boldsymbol{W}^{(t)}}(\boldsymbol{X}_i) - \frac{M\beta}{4}\right) \geq \sum_{i \in \mathcal{V}_{s^*,k^*}} \left(\ell(y_i f_{\boldsymbol{W}^{(t)}}(\boldsymbol{X}_i)) - \ell\left(\frac{M\beta}{4}\right)\right)$$

$$\geq \sum_{i \in \mathcal{V}_{s^*,k^*}} \ell(y_i f_{\boldsymbol{W}^{(t)}}(\boldsymbol{X}_i)) - ne^{-\frac{M\beta}{4}}.$$

In addition, by Lemma F.2,

$$\frac{\eta^2}{n^2} \left\|\nabla \sum_{i \in \mathcal{V}_{s^*,k^*}} \ell(y_i f_{\boldsymbol{W}^{(t)}}(\boldsymbol{X}_i))\right\|^2 \leq \frac{8\eta^2 P^2 \sigma_{\mathrm{d}}^2 d |\mathcal{V}_{s^*,k^*}|}{n^2} \sum_{i \in \mathcal{V}_{s^*,k^*}} \ell(y_i f_{\boldsymbol{W}^{(t)}}(\boldsymbol{X}_i))$$

$$\leq \frac{\eta}{n} \sum_{i \in \mathcal{V}_{s^*,k^*}} \ell(y_i f_{\boldsymbol{W}^{(t)}}(\boldsymbol{X}_i)),$$

where the last inequality is due to (A8), and we have

$$\left\|Q\left(\boldsymbol{W}^{(t)}\right) - Q(\boldsymbol{W}^*)\right\|^2 - \left\|Q\left(\boldsymbol{W}^{(t+1)}\right) - Q(\boldsymbol{W}^*)\right\|^2$$

$$\geq \frac{\eta}{n} \sum_{i \in \mathcal{V}_{s^*,k^*}} \ell(y_i f_{\boldsymbol{W}^{(t)}}(\boldsymbol{X}_i)) - 2\eta e^{-\frac{M\beta}{4}}.$$

From telescoping summation, we have

$$\frac{\eta}{n} \sum_{t=T_{\text{ERM}}}^{T^*} \sum_{i \in \mathcal{V}_{s^*,k^*}} \ell(y_i f_{\boldsymbol{W}^{(t)}}(\boldsymbol{X}_i)) \leq \left\|Q\left(\boldsymbol{W}^{(T_{\text{ERM}})}\right) - Q\left(\boldsymbol{W}^*\right)\right\|^2 + 2\eta T^* e^{-\frac{M\beta}{4}}.$$

$$\square$$

Finally, we can prove that the model cannot learn (extremely) rare features within $T^*$ iterations.

**Lemma C.9.** *Suppose the event $E_{\text{init}}$ occurs. For any $T \in [T_{\text{ERM}}, T^*]$, we have $\gamma_s^{(T)}(s^*, k^*) = \widetilde{\mathcal{O}}\left(\alpha^2 \beta^{-2}\right)$ for each $s \in \{\pm 1\}$.*

*Proof of Lemma C.9.* For any $T \in [T_{\text{ERM}}, T^*]$, we have

$$\gamma_s^{(T)}(s, k) = \gamma_s^{(T_{\text{ERM}})}(s^*, k^*) + \frac{\eta}{n} \sum_{t=T_{\text{ERM}}}^{T-1} \sum_{i \in \mathcal{V}_{s^*,k^*}} g_i^{(t)} \phi'\left(\left\langle \boldsymbol{w}_s^{(t)}, \boldsymbol{v}_{s^*,k^*}\right\rangle\right)$$

$$\leq \gamma_s^{(T_{\text{ERM}})}(s^*, k^*) + \frac{\eta}{n} \sum_{t=T_{\text{ERM}}}^{T-1} \sum_{i \in \mathcal{V}_{s^*,k^*}} g_i^{(t)}$$

$$\leq \gamma_s^{(T_{\text{ERM}})}(s^*, k^*) + \frac{\eta}{n} \sum_{t=T_{\text{ERM}}}^{T-1} \sum_{i \in \mathcal{V}_{s^*, k^*}} \ell\left(y_i f_{\boldsymbol{W}^{(t)}}(\boldsymbol{X}_i)\right),$$

where the first inequality is due to $\phi' \leq 1$ and the second inequality is due to $-\ell' \leq \ell$. From the result of Section C.2.3 we know $\gamma_s^{(T_{\text{ERM}})}(s^*, k^*) \leq \alpha^2 \beta^{-1}$. Additionally, by Lemma C.8 and Lemma C.6, we have

$$\frac{\eta}{n} \sum_{t=T_{\text{ERM}}}^{(T-1)} \sum_{i \in \mathcal{V}_{s^*, k^*}} \ell\left(y_i f_{\boldsymbol{W}^{(t)}}(\boldsymbol{X}_i)\right) \leq \frac{\eta}{n} \sum_{t=T_{\text{ERM}}}^{(T^*)} \sum_{i \in \mathcal{V}_{s^*, k^*}} \ell\left(y_i f_{\boldsymbol{W}^{(t)}}(\boldsymbol{X}_i)\right)$$

$$\leq \left\| Q\left(\boldsymbol{W}^{(T_{\text{ERM}})}\right) - Q(\boldsymbol{W}^*)\right\|^2 + 2\eta T^* e^{-\frac{M\beta}{4}}$$

$$\leq 12M^2 |\mathcal{V}_{s^*, k^*}| \sigma_{\text{d}}^{-2} d^{-1} + 2\eta T^* e^{-\frac{M\beta}{4}}$$

$$= \widetilde{\mathcal{O}}\left(\alpha^2 \beta^{-2}\right).$$

The last line is due to (A5) and our choice $M = 4\beta^{-1} \log\left(\frac{2\eta\beta^2 T^*}{\alpha^2}\right)$. Thus, we have our conclusion. $\square$

**What We Have So Far.** Suppose the event $E_{\text{init}}$ occurs. For any $t \in [T_{\text{ERM}}, T^*]$, we have

- (Learn common features): For each $s \in \{\pm 1\}$ and $k \in \mathcal{K}_C$,

$$\gamma_s^{(t)}(s, k) + \beta \gamma_{-s}^{(t)}(s, k) = \Omega(1).$$

- (Overfit (extremely) rare data): For each $s \in \{\pm 1\}, k \in \mathcal{K}_R \cup \mathcal{K}_E$ and $i \in \mathcal{V}_{s,k}$,

$$\sum_{p \in [P] \setminus \{p_i^*\}} \left(\rho_s^{(t)}(i, p) + \beta \rho_{-s}^{(t)}(i, p)\right) = \Omega(1).$$

- (Cannot learn (extremely) rare features): For each $s \in \{\pm 1\}$ and $k \in \mathcal{K}_R \cup \mathcal{K}_E$,

$$\gamma_s^{(t)}(s, k), \gamma_{-s}^{(t)}(s, k) = \mathcal{O}\left(\alpha^2 \beta^{-2}\right).$$

- For any $s \in \{\pm 1\}, i \in [n]$, and $p \in [P] \setminus \{p_i^*\}$, $\rho_s^{(t)}(i, p) = \widetilde{\mathcal{O}}\left(\beta^{-1}\right)$,

### C.2.5 Train and Test Accuracy

In this step, we will prove that the model trained by ERM has perfect training accuracy but has near-random guesses on (extremely) rare data.

For any $i \in \mathcal{V}_{s,k}$ with $s \in \{\pm 1\}$ and $k \in \mathcal{K}_C$, by Lemma B.4, we have

$$y_i f_{\boldsymbol{W}^{(t)}}(\boldsymbol{X}_i)$$
$$= \sum_{p \in [P]} \left(\phi\left(\left\langle \boldsymbol{w}_s^{(t)}, \boldsymbol{x}_i^{(p)}\right\rangle\right) - \phi\left(\left\langle \boldsymbol{w}_{-s}^{(t)}, \boldsymbol{x}_i^{(p)}\right\rangle\right)\right)$$
$$\geq \gamma_s^{(t)}(s, k) + \beta \gamma_{-s}^{(t)}(s, k) + \sum_{p \in [P] \setminus \{p_i^*\}} \left(\rho_s^{(t)}(i, p) + \beta \rho_{-s}^{(t)}(i, p)\right) - 2P \cdot o\left(\frac{1}{\text{polylog}(d)}\right)$$
$$\geq \gamma_s^{(t)}(s, k) + \beta \gamma_{-s}^{(t)}(s, k) - o\left(\frac{1}{\text{polylog}(d)}\right)$$
$$= \Omega(1) - o\left(\frac{1}{\text{polylog}(d)}\right)$$
$$> 0,$$

for any $t \in [T_{\text{ERM}}, T^*]$. In addition, for any $i \in \mathcal{V}_{s,k}$ with $s \in \{\pm 1\}$ and $k \in \mathcal{K}_R \cup \mathcal{K}_E$, we have

$$y_i f_{\boldsymbol{W}^{(t)}}(\boldsymbol{X}_i)$$

$$= \sum_{p \in [P]} \left( \phi\left( \left\langle \boldsymbol{w}_s^{(t)}, \boldsymbol{x}_i^{(p)} \right\rangle \right) - \phi\left( \left\langle \boldsymbol{w}_{-s}^{(t)}, \boldsymbol{x}_i^{(p)} \right\rangle \right) \right)$$

$$= \gamma_s^{(t)}(s,k) + \beta\gamma_{-s}^{(t)}(s,k) + \sum_{p \in [P] \setminus \{p_i^*\}} \left( \rho_s^{(t)}(i,p) + \beta\rho_{-s}^{(t)}(i,p) \right) - 2P \cdot o\left( \frac{1}{\mathrm{polylog}(d)} \right)$$

$$\geq \sum_{p \in [P] \setminus \{p_i^*\}} \left( \rho_s^{(t)}(i,p) + \beta\rho_{-s}^{(t)}(i,p) \right) - o\left( \frac{1}{\mathrm{polylog}(d)} \right)$$

$$= \Omega(1) - o\left( \frac{1}{\mathrm{polylog}(d)} \right)$$

$$> 0,$$

for any $t \in [T_{\mathrm{ERM}}, T^*]$. We can conclude that ERM with $t \in [T_{\mathrm{ERM}}, T^*]$ iterates achieve perfect training accuracy.

Next, let us move on to the test accuracy part. Let $(\boldsymbol{X}, y) \sim \mathcal{D}$ be a test data with $\boldsymbol{X} = \left( \boldsymbol{x}^{(1)}, \ldots, \boldsymbol{x}^{(P)} \right) \in \mathbb{R}^{d \times P}$ having feature patch index $p^*$, dominant noise patch index $\tilde{p}$, and feature vector $\boldsymbol{v}_{y,k}$. We have $\boldsymbol{x}^{(p)} \sim N(\boldsymbol{0}, \sigma_{\mathrm{b}}^2 \boldsymbol{\Lambda})$ for each $p \in [P] \setminus \{p^*, \tilde{p}\}$ and $\boldsymbol{x}^{(\tilde{p})} - \alpha\boldsymbol{v}_{s,1} \sim N(\boldsymbol{0}, \sigma_{\mathrm{d}}^2 \boldsymbol{\Lambda})$ for some $s \in \{\pm 1\}$. Therefore, for all $t \in [T_{\mathrm{ERM}}, T^*]$ and $p \in [P] \setminus \{p^*, \tilde{p}\}$,

$$\left| \phi\left( \left\langle \boldsymbol{w}_1^{(t)}, \boldsymbol{x}^{(p)} \right\rangle \right) - \phi\left( \left\langle \boldsymbol{w}_{-1}^{(t)}, \boldsymbol{x}^{(p)} \right\rangle \right) \right|$$

$$\leq \left| \left\langle \boldsymbol{w}_1^{(t)} - \boldsymbol{w}_{-1}^{(t)}, \boldsymbol{x}^{(p)} \right\rangle \right|$$

$$\leq \left| \left\langle \boldsymbol{w}_1^{(0)} - \boldsymbol{w}_{-1}^{(0)}, \boldsymbol{x}^{(p)} \right\rangle \right| + \sum_{i \in [n], q \in [P] \setminus \{p_i^*\}} \left| \rho_1^{(t)}(i,q) - \rho_{-1}^{(t)}(i,q) \right| \frac{\left| \left\langle \xi_i^{(q)}, \boldsymbol{x}^{(p)} \right\rangle \right|}{\left\| \xi_i^{(q)} \right\|^2}$$

$$\leq \tilde{\mathcal{O}}\left( \sigma_0 \sigma_{\mathrm{b}} d^{\frac{1}{2}} \right) + \tilde{\mathcal{O}}\left( nP\beta^{-1} \sigma_{\mathrm{d}} \sigma_{\mathrm{b}}^{-1} d^{-\frac{1}{2}} \right)$$

$$= o\left( \frac{\alpha}{\mathrm{polylog}(d)} \right), \tag{18}$$

with probability at least $1 - o\left( \frac{1}{\mathrm{poly}(d)} \right)$ due to Lemma B.2, (A8), (8), and (9). In addition, for any $s' \in \{\pm 1\}$, we have

$$\left| \left\langle \boldsymbol{w}_{s'}^{(t)}, \boldsymbol{x}^{(\tilde{p})} - \alpha\boldsymbol{v}_{s,1} \right\rangle \right|$$

$$\leq \left| \left\langle \boldsymbol{w}_{s'}^{(0)}, \boldsymbol{x}^{(\tilde{p})} - \alpha\boldsymbol{v}_{s,1} \right\rangle \right| + \sum_{i \in [n], q \in [P] \setminus \{p_i^*\}} \rho_{s'}^{(t)}(i,q) \frac{\left| \left\langle \xi_i^{(q)}, \boldsymbol{x}^{(\tilde{p})} - \alpha\boldsymbol{v}_{s,1} \right\rangle \right|}{\left\| \xi_i^{(q)} \right\|^2}$$

$$= \tilde{\mathcal{O}}\left( \sigma_0 \sigma_{\mathrm{d}} d^{\frac{1}{2}} \right) + \tilde{\mathcal{O}}\left( nP\beta^{-1} \sigma_{\mathrm{d}} \sigma_{\mathrm{b}}^{-1} d^{-\frac{1}{2}} \right)$$

$$= o\left( \frac{\alpha}{\mathrm{polylog}(d)} \right), \tag{19}$$

with probability at least $1 - o\left( \frac{1}{\mathrm{poly}(d)} \right)$ due to Lemma B.2, (A8), (8), and (9).

**Case 1:** $k \in \mathcal{K}_C$

By Lemma B.2, (A8), and (10),

$$\left| \phi\left( \left\langle \boldsymbol{w}_1^{(t)}, \boldsymbol{x}^{(\tilde{p})} \right\rangle \right) - \phi\left( \left\langle \boldsymbol{w}_{-1}^{(t)}, \boldsymbol{w}^{(\tilde{p})} \right\rangle \right) \right|$$

$$\leq \left| \left\langle \boldsymbol{w}_1^{(t)} - \boldsymbol{w}_{-1}^{(t)}, \boldsymbol{x}^{(\tilde{p})} \right\rangle \right|$$

$$\leq \alpha \left| \left\langle \boldsymbol{w}_1^{(t)} - \boldsymbol{w}_{-1}^{(t)}, \boldsymbol{v}_{s,1} \right\rangle \right| + \left| \left\langle \boldsymbol{w}_1^{(t)} - \boldsymbol{w}_{-1}^{(t)}, \boldsymbol{x}^{(p)} - \alpha\boldsymbol{v}_{s,1} \right\rangle \right|$$

$$\leq \alpha \left( \gamma_1^{(t)}(s, 1) + \gamma_{-1}^{(t)}(s, 1) \right) + \alpha \left| \left\langle \boldsymbol{w}_1^{(0)}, \boldsymbol{v}_{s,1} \right\rangle \right| + \alpha \left| \left\langle \boldsymbol{w}_{-1}^{(0)}, \boldsymbol{v}_{s,1} \right\rangle \right| + o \left( \frac{1}{\mathrm{polylog}(d)} \right)$$

$$\leq \tilde{\mathcal{O}} \left( \alpha \beta^{-1} \right) + \tilde{\mathcal{O}} \left( \alpha \sigma_0 \right) + o \left( \frac{1}{\mathrm{polylog}(d)} \right)$$

$$= o \left( \frac{1}{\mathrm{polylog}(d)} \right), \tag{20}$$

with probability at least $1 - o \left( \frac{1}{\mathrm{poly}(d)} \right)$. Suppose (18) and (20) holds. By Lemma B.4, we have

$$y f_{\boldsymbol{W}^{(t)}}(\boldsymbol{X})$$
$$= \left( \phi \left( \left\langle \boldsymbol{w}_y^{(t)}, \boldsymbol{v}_{y,k} \right\rangle \right) - \phi \left( \left\langle \boldsymbol{w}_{-y}^{(t)}, \boldsymbol{v}_{y,k} \right\rangle \right) \right)$$
$$+ \sum_{p \in [P] \setminus \{p^*\}} \left( \phi \left( \left\langle \boldsymbol{w}_y^{(t)}, \boldsymbol{x}^{(p)} \right\rangle \right) - \phi \left( \left\langle \boldsymbol{w}_{-y}^{(t)}, \boldsymbol{x}^{(p)} \right\rangle \right) \right)$$
$$= \gamma_y^{(t)}(y, k) + \beta \gamma_{-y}^{(t)}(y, k) - o \left( \frac{1}{\mathrm{polylog}(d)} \right)$$
$$= \Omega(1) - o \left( \frac{1}{\mathrm{polylog}(d)} \right)$$
$$> 0.$$

Therefore, we have

$$\mathbb{P}_{(\boldsymbol{X}, y) \sim \mathcal{D}} \left[ y f_{\boldsymbol{W}^{(t)}}(\boldsymbol{X}) > 0 \mid \boldsymbol{x}^{(p^*)} = \boldsymbol{v}_{y,k}, k \in \mathcal{K}_C \right] \geq 1 - o \left( \frac{1}{\mathrm{poly}(d)} \right). \tag{21}$$

**Case 2:** $k \in \mathcal{K}_R \cup \mathcal{K}_E$     By triangular inequality and $\phi' \leq 1$, we have

$$\phi \left( \left\langle \boldsymbol{w}_s^{(t)}, \boldsymbol{x}^{(\tilde{p})} \right\rangle \right) - \phi \left( \left\langle \boldsymbol{w}_{-s}^{(t)}, \boldsymbol{x}^{(\tilde{p})} \right\rangle \right)$$
$$= \phi \left( \left\langle \boldsymbol{w}_s^{(t)}, \alpha \boldsymbol{v}_{s,1} \right\rangle \right) - \phi \left( \left\langle \boldsymbol{w}_{-s}^{(t)}, \alpha \boldsymbol{v}_{s,1} \right\rangle \right)$$
$$+ \left( \phi \left( \left\langle \boldsymbol{w}_s^{(t)}, \boldsymbol{x}^{(\tilde{p})} \right\rangle \right) - \phi \left( \left\langle \boldsymbol{w}_s^{(t)}, \alpha \boldsymbol{v}_{s,1} \right\rangle \right) \right) - \left( \phi \left( \left\langle \boldsymbol{w}_{-s}^{(t)}, \boldsymbol{x}^{(\tilde{p})} \right\rangle \right) - \phi \left( \left\langle \boldsymbol{w}_{-s}^{(t)}, \alpha \boldsymbol{v}_{s,1} \right\rangle \right) \right)$$
$$\geq \phi \left( \left\langle \boldsymbol{w}_s^{(t)}, \alpha \boldsymbol{v}_{s,1} \right\rangle \right) - \phi \left( \left\langle \boldsymbol{w}_{-s}^{(t)}, \alpha \boldsymbol{v}_{s,1} \right\rangle \right)$$
$$- \left| \left\langle \boldsymbol{w}_s^{(t)}, \boldsymbol{x}^{(\tilde{p})} - \alpha \boldsymbol{v}_{s,1} \right\rangle \right| - \left| \left\langle \boldsymbol{w}_{-s}^{(t)}, \boldsymbol{x}^{(\tilde{p})} - \alpha \boldsymbol{v}_{s,1} \right\rangle \right|.$$

In addition,

$$\phi \left( \left\langle \boldsymbol{w}_s^{(t)}, \alpha \boldsymbol{v}_{s,1} \right\rangle \right) - \phi \left( \left\langle \boldsymbol{w}_{-s}^{(t)}, \alpha \boldsymbol{v}_{s,1} \right\rangle \right)$$
$$= \left( \phi \left( \alpha \gamma_s^{(t)}(s, 1) \right) - \phi \left( -\alpha \gamma_{-s}^{(t)}(s, 1) \right) \right)$$
$$+ \left( \phi \left( \left\langle \boldsymbol{w}_s^{(t)}, \alpha \boldsymbol{v}_{s,1} \right\rangle \right) - \phi \left( \alpha \gamma_s^{(t)}(s, 1) \right) \right)$$
$$- \left( \phi \left( \left\langle \boldsymbol{w}_{-s}^{(t)}, \alpha \boldsymbol{v}_{s,1} \right\rangle \right) - \phi \left( -\alpha \gamma_{-s}^{(t)}(s, 1) \right) \right)$$
$$\geq \left( \phi \left( \alpha \gamma_s^{(t)}(s, 1) \right) - \phi \left( -\alpha \gamma_{-s}^{(t)}(s, 1) \right) \right)$$
$$- \alpha \left| \left\langle \boldsymbol{w}_s^{(t)}, \boldsymbol{v}_{s,1} \right\rangle - \gamma_s^{(t)}(s, 1) \right| - \alpha \left| \left\langle \boldsymbol{w}_{-s}^{(t)}, \boldsymbol{v}_{s,1} \right\rangle + \gamma_{-s}^{(t)}(s, 1) \right|$$
$$= \alpha \left( \gamma_s^{(t)}(s, 1) + \beta \gamma_{-s}^{(t)}(s, 1) \right) - \alpha \cdot o \left( \frac{1}{\mathrm{polylog}(d)} \right)$$

$$= \Omega(\alpha),$$

where the second equality is due to Lemma B.4 and (A8). If (19) holds, we have

$$\phi\left(\left\langle \boldsymbol{w}_s^{(t)}, \boldsymbol{x}^{(\tilde{p})}\right\rangle\right) - \phi\left(\left\langle \boldsymbol{w}_{-s}^{(t)}, \boldsymbol{x}^{(\tilde{p})}\right\rangle\right) = \Omega(\alpha) - o\left(\frac{\alpha}{\mathrm{polylog}(d)}\right) = \Omega(\alpha). \qquad (22)$$

Note that

$$y f_{\boldsymbol{W}^{(t)}}(\boldsymbol{X})$$
$$= \phi\left(\left\langle \boldsymbol{w}_y^{(t)}, \boldsymbol{v}_{y,k}\right\rangle\right) - \phi\left(\left\langle \boldsymbol{w}_{-y}^{(t)}, \boldsymbol{v}_{y,k}\right\rangle\right) + \phi\left(\left\langle \boldsymbol{w}_y^{(t)}, \boldsymbol{x}^{(\tilde{p})}\right\rangle\right) - \phi\left(\left\langle \boldsymbol{w}_{-y}^{(t)}, \boldsymbol{x}^{(\tilde{p})}\right\rangle\right)$$
$$+ \sum_{p \in [P] \setminus \{p^*, \tilde{p}\}} \left(\phi\left(\left\langle \boldsymbol{w}_y^{(t)}, \boldsymbol{x}^{(p)}\right\rangle\right) - \phi\left(\left\langle \boldsymbol{w}_{-y}^{(t)}, \boldsymbol{x}^{(p)}\right\rangle\right)\right),$$

and

$$\left|\phi\left(\left\langle \boldsymbol{w}_y^{(t)}, \boldsymbol{v}_{y,k}\right\rangle\right) - \phi\left(\left\langle \boldsymbol{w}_{-y}^{(t)}, \boldsymbol{v}_{y,k}\right\rangle\right)\right|$$
$$+ \left|\sum_{p \in [P] \setminus \{p^*, \tilde{p}\}} \left(\phi\left(\left\langle \boldsymbol{w}_y^{(t)}, \boldsymbol{x}^{(p)}\right\rangle\right) - \phi\left(\left\langle \boldsymbol{w}_{-y}^{(t)}, \boldsymbol{x}^{(p)}\right\rangle\right)\right)\right|$$
$$\leq \left|\left\langle \boldsymbol{w}_y^{(t)} - \boldsymbol{w}_{-y}^{(t)}, \boldsymbol{v}_{y,k}\right\rangle\right| + o\left(\frac{\alpha}{\mathrm{polylog}(d)}\right)$$
$$\leq \gamma_1^{(t)}(y,k) + \gamma_{-1}^{(t)}(y,k) + \left|\left\langle \boldsymbol{w}_y^{(0)} - \boldsymbol{w}_{-y}^{(0)}, \boldsymbol{v}_{y,k}\right\rangle\right| + o\left(\frac{\alpha}{\mathrm{polylog}(d)}\right)$$
$$\leq \mathcal{O}(\alpha^2 \beta^{-2}) + \widetilde{\mathcal{O}}(\sigma_0) + o\left(\frac{\alpha}{\mathrm{polylog}(d)}\right)$$
$$= o\left(\frac{\alpha}{\mathrm{polylog}(d)}\right)$$
$$< \phi\left(\left\langle \boldsymbol{w}_s^{(t)}, \boldsymbol{x}^{(\tilde{p})}\right\rangle\right) - \phi\left(\left\langle \boldsymbol{w}_{-s}^{(t)}, \boldsymbol{x}^{(\tilde{p})}\right\rangle\right),$$

where the first inequality is due to (18), second-to-last line is due to (A8), (8) and (10), and the last inequality is due to (22). Therefore, we have $y f_{\boldsymbol{W}^{(t)}}(\boldsymbol{X}) > 0$ if $y = s$. Otherwise, $y f_{\boldsymbol{W}^{(t)}}(\boldsymbol{X}) < 0$. Therefore, we have

$$\mathbb{P}_{(\boldsymbol{X},y) \sim \mathcal{D}}\left[y f_{\boldsymbol{W}^{(t)}}(\boldsymbol{X}) > 0 \mid \boldsymbol{x}^{(p^*)} = \boldsymbol{v}_{y,k}, k \in \mathcal{K}_R \cup \mathcal{K}_E\right] = \frac{1}{2} \pm o\left(\frac{1}{\mathrm{poly}(d)}\right). \qquad (23)$$

Hence, combining (21) and (23) implies

$$\mathbb{P}_{(\boldsymbol{X},y) \sim \mathcal{D}}\left[y f_{\boldsymbol{W}^{(t)}}(\boldsymbol{X}) > 0\right] = \sum_{k \in \mathcal{K}_C} \rho_k + \frac{1}{2}\left(1 - \sum_{k \in \mathcal{K}_C} \rho_k\right) \pm o\left(\frac{1}{\mathrm{poly}(d)}\right)$$
$$= 1 - \frac{1}{2}\sum_{k \in \mathcal{K}_R \cup \mathcal{K}_E} \rho_k \pm o\left(\frac{1}{\mathrm{poly}(d)}\right).$$

$\square$

# D Proof for Cutout

In this section, we use $g_{i,\mathcal{C}}^{(t)} := \frac{1}{1+\exp\left(y_i f_{\boldsymbol{W}^{(t)}}(\boldsymbol{X}_{i,\mathcal{C}})\right)}$ for each data $i$, $\mathcal{C} \subset [P]$ with $|\mathcal{C}| = C$ and iteration $t$, for simplicity.

## D.1 Proof of Lemma B.3 for Cutout

For $s \in \{\pm 1\}$ and iterate $t$,

$$\boldsymbol{w}_s^{(t+1)} - \boldsymbol{w}_s^{(t)}$$

$$= -\eta \nabla_{\boldsymbol{w}_s} \mathcal{L}_{\text{Cutout}}\left(\boldsymbol{W}^{(t)}\right)$$

$$= \frac{\eta}{n} \sum_{i \in [n]} sy_i \mathbb{E}_{\mathcal{C} \sim \mathcal{D}_C} \left[ g_{i,\mathcal{C}}^{(t)} \sum_{p \notin \mathcal{C}} \phi'\left(\left\langle \boldsymbol{w}_s^{(t)}, \boldsymbol{x}_i^{(p)} \right\rangle\right) \boldsymbol{x}_i^{(p)} \right]$$

$$= \frac{\eta}{n} \left( \sum_{i \in \mathcal{V}_s} \mathbb{E}_{\mathcal{C} \sim \mathcal{D}_C} \left[ g_{i,\mathcal{C}}^{(t)} \sum_{p \notin \mathcal{C}} \phi'\left(\left\langle \boldsymbol{w}_s^{(t)}, \boldsymbol{x}_i^{(p)} \right\rangle\right) \boldsymbol{x}_i^{(p)} \right] \right.$$

$$\left. - \sum_{i \in \mathcal{V}_{-s}} \mathbb{E}_{\mathcal{C} \sim \mathcal{D}_C} \left[ g_{i,\mathcal{C}}^{(t)} \sum_{p \notin \mathcal{C}} \phi'\left(\left\langle \boldsymbol{w}_s^{(t)}, \boldsymbol{x}_i^{(p)} \right\rangle\right) \boldsymbol{x}_i^{(p)} \right] \right),$$

and we have

$$\sum_{i \in \mathcal{V}_s} \mathbb{E}_{\mathcal{C} \sim \mathcal{D}_C} \left[ g_{i,\mathcal{C}}^{(t)} \sum_{p \notin \mathcal{C}} \phi'\left(\left\langle \boldsymbol{w}_s^{(t)}, \boldsymbol{x}_i^{(p)} \right\rangle\right) \boldsymbol{x}_i^{(p)} \right]$$

$$= \sum_{k \in [K]} \sum_{i \in \mathcal{V}_{s,k}} \mathbb{E}_{\mathcal{C} \sim \mathcal{D}_C} \left[ g_{i,\mathcal{C}}^{(t)} \phi'\left(\left\langle \boldsymbol{w}_s^{(t)}, \boldsymbol{v}_{s,k} \right\rangle\right) \cdot \mathbb{1}_{p_i^* \notin \mathcal{C}} \right] \boldsymbol{v}_{s,k}$$

$$+ \sum_{i \in \mathcal{V}_s} \sum_{p \in [P] \setminus \{p_i^*, \tilde{p}_i\}} \mathbb{E}_{\mathcal{C} \sim \mathcal{D}_C} \left[ g_{i,\mathcal{C}}^{(t)} \phi'\left(\left\langle \boldsymbol{w}_s^{(t)}, \xi_i^{(p)} \right\rangle\right) \cdot \mathbb{1}_{p \notin \mathcal{C}} \right] \xi_i^{(p)}$$

$$+ \sum_{i \in \mathcal{V}_s \cap \mathcal{F}_s} \mathbb{E}_{\mathcal{C} \sim \mathcal{D}_C} \left[ g_{i,\mathcal{C}}^{(t)} \phi'\left(\left\langle \boldsymbol{w}_s^{(t)}, \alpha \boldsymbol{v}_{s,1} + \xi_i^{(\tilde{p}_i)} \right\rangle\right) \cdot \mathbb{1}_{\tilde{p}_i \notin \mathcal{C}} \right] \left(\alpha \boldsymbol{v}_{s,1} + \xi_i^{(\tilde{p}_i)}\right)$$

$$+ \sum_{i \in \mathcal{V}_s \cap \mathcal{F}_{-s}} \mathbb{E}_{\mathcal{C} \sim \mathcal{D}_C} \left[ g_{i,\mathcal{C}}^{(t)} \phi'\left(\left\langle \boldsymbol{w}_s^{(t)}, \alpha \boldsymbol{v}_{-s,1} + \xi_i^{(\tilde{p}_i)} \right\rangle\right) \cdot \mathbb{1}_{\tilde{p}_i \notin \mathcal{C}} \right] \left(\alpha \boldsymbol{v}_{-s,1} + \xi_i^{(\tilde{p}_i)}\right),$$

and

$$\sum_{i \in \mathcal{V}_{-s}} \mathbb{E}_{\mathcal{C} \sim \mathcal{D}_C} \left[ g_{i,\mathcal{C}}^{(t)} \sum_{p \notin \mathcal{C}} \phi'\left(\left\langle \boldsymbol{w}_s^{(t)}, \boldsymbol{x}_i^{(p)} \right\rangle\right) \boldsymbol{x}_i^{(p)} \right]$$

$$= \sum_{k \in [K]} \sum_{i \in \mathcal{V}_{-s,k}} \mathbb{E}_{\mathcal{C} \sim \mathcal{D}_C} \left[ g_{i,\mathcal{C}}^{(t)} \phi'\left(\left\langle \boldsymbol{w}_s^{(t)}, \boldsymbol{v}_{-s,k} \right\rangle\right) \cdot \mathbb{1}_{p_i^* \notin \mathcal{C}} \right] \boldsymbol{v}_{-s,k}$$

$$+ \sum_{i \in \mathcal{V}_{-s}} \sum_{p \in [P] \setminus \{p_i^*, \tilde{p}_i\}} \mathbb{E}_{\mathcal{C} \sim \mathcal{D}_C} \left[ g_{i,\mathcal{C}}^{(t)} \phi'\left(\left\langle \boldsymbol{w}_s^{(t)}, \xi_i^{(p)} \right\rangle\right) \cdot \mathbb{1}_{p \notin \mathcal{C}} \right] \xi_i^{(p)}$$

$$+ \sum_{i \in \mathcal{V}_{-s} \cap \mathcal{F}_s} \mathbb{E}_{\mathcal{C} \sim \mathcal{D}_C} \left[ g_{i,\mathcal{C}}^{(t)} \phi'\left(\left\langle \boldsymbol{w}_s^{(t)}, \alpha \boldsymbol{v}_{s,1} + \xi_i^{(\tilde{p}_i)} \right\rangle\right) \cdot \mathbb{1}_{\tilde{p}_i \notin \mathcal{C}} \right] \left(\alpha \boldsymbol{v}_{s,1} + \xi_i^{(\tilde{p}_i)}\right)$$

$$+ \sum_{i \in \mathcal{V}_{-s} \cap \mathcal{F}_{-s}} \mathbb{E}_{\mathcal{C} \sim \mathcal{D}_C} \left[ g_{i,\mathcal{C}}^{(t)} \phi'\left(\left\langle \boldsymbol{w}_s^{(t)}, \alpha \boldsymbol{v}_{-s,1} + \xi_i^{(\tilde{p}_i)} \right\rangle\right) \cdot \mathbb{1}_{\tilde{p}_i \notin \mathcal{C}} \right] \left(\alpha \boldsymbol{v}_{-s,1} + \xi_i^{(\tilde{p}_i)}\right).$$

Hence, if we define $\gamma_s^{(t)}(s', k)$'s and $\rho_s^{(t)}(i, p)$'s recursively by using the rule

$$\gamma_s^{(t+1)}(s', k) = \gamma_s^{(t)}(s', k) + \frac{\eta}{n} \sum_{i \in \mathcal{V}_{s',k}} \mathbb{E}_{\mathcal{C} \sim \mathcal{D}_C} \left[ g_{i,\mathcal{C}}^{(t)} \phi'\left(\left\langle \boldsymbol{w}_s^{(t)}, \boldsymbol{v}_{s',k} \right\rangle\right) \cdot \mathbb{1}_{p_i^* \notin \mathcal{C}} \right], \qquad (24)$$

$$\rho_s^{(t+1)}(i,p) = \rho_s^{(t)}(i,p) + \frac{\eta}{n}\mathbb{E}_{\mathcal{C}\sim\mathcal{D}_\mathcal{C}}\left[g_{i,\mathcal{C}}^{(t)}\phi'\left(\left\langle \boldsymbol{w}_s^{(t)}, \boldsymbol{x}_i^{(p)}\right\rangle\right)\cdot\mathbb{1}_{p\notin\mathcal{C}}\right]\left\|\xi_i^{(p)}\right\|^2, \qquad (25)$$

starting from $\gamma_s^{(0)}(s',k) = \rho_s^{(0)}(i,p) = 0$ for each $s,s'\in\{\pm1\}, k\in[K], i\in[n]$ and $p\in[P]\setminus\{p_i^*\}$, then we have

$$\boldsymbol{w}_s^{(t)} = \boldsymbol{w}_s^{(0)} + \sum_{k\in[K]}\gamma_s^{(t)}(s,k)\boldsymbol{v}_{s,k} - \sum_{k\in[K]}\gamma_s^{(t)}(-s,k)\boldsymbol{v}_{-s,k}$$

$$+ \sum_{i\in\mathcal{V}_s, p\in[P]\setminus\{p_i^*\}}\rho_s^{(t)}(i,p)\frac{\xi_i^{(p)}}{\left\|\xi_i^{(p)}\right\|^2} - \sum_{i\in\mathcal{V}_{-s}, p\in[P]\setminus\{p_i^*\}}\rho_s^{(t)}(i,p)\frac{\xi_i^{(p)}}{\left\|\xi_i^{(p)}\right\|^2}$$

$$+ \alpha\left(\sum_{i\in\mathcal{F}_s}sy_i\rho_s^{(t)}(i,\tilde{p}_i)\frac{\boldsymbol{v}_{s,1}}{\left\|\xi_i^{(\tilde{p}_i)}\right\|^2} + \sum_{i\in\mathcal{F}_{-s}}sy_i\rho_s^{(t)}(i,\tilde{p}_i)\frac{\boldsymbol{v}_{-s,1}}{\left\|\xi_i^{(\tilde{p}_i)}\right\|^2}\right),$$

for each $s\in\{\pm1\}$. Furthermore, $\gamma_s^{(t)}(s',k)$'s and $\rho_s^{(t)}(i,p)$'s are monotone increasing. $\qquad\square$

## D.2 Proof of Theorem 3.2

To show Theorem 3.2, we present a structured proof comprising the following five steps:

1. Establish upper bounds on $\gamma_s^{(t)}(s',k)$'s and $\rho_s^{(t)}(i,p)$'s to apply Lemma B.4 (Section D.2.1).
2. Demonstrate that the model quickly learns common and rare features (Section D.2.2).
3. Show that the model overfits augmented data if it does not contain common or rare features (Section D.2.3).
4. Confirm the persistence of this tendency until $T^*$ iterates (Section D.2.4).
5. Characterize train accuracy and test accuracy (Section D.2.5).

### D.2.1 Bounds on the Coefficients in Feature Noise Decomposition

The following lemma provides upper bounds on Lemma B.3 during $T^*$ iterations.

**Lemma D.1.** *Suppose the event $E_{\text{init}}$ occurs. For any $0\le t\le T^*$, we have*

$$0\le\gamma_s^{(t)}(s,k) + \beta\gamma_{-s}^{(t)}(s,k)\le 4\log(\eta T^*), \quad 0\le\rho_{y_i}^{(t)}(i,p) + \beta\rho_{-y_i}^{(t)}(i,p)\le 4\log(\eta T^*),$$

*for all $s\in\{\pm1\}, k\in[K], i\in[n]$ and $p\in[P]\setminus\{p_i^*\}$. Consequently, $\gamma_s^{(t)}(s',k), \rho_s^{(t)}(i,p) = \widetilde{\mathcal{O}}(\beta^{-1})$ for all $s,s'\in\{\pm1\}, k\in[K], i\in[n]$ and $p\in[P]\setminus\{p_i^*\}$.*

*Proof of Lemma D.1.* The first argument implies the second argument since $\log(\eta T^*) = \text{polylog}(d)$ and

$$\gamma_s^{(t)}(s',k)\le\beta^{-1}\left(\gamma_{s'}^{(t)}(s',k) + \beta\gamma_{s'}^{(t)}(s',k)\right), \quad \rho_s^{(t)}(i,p)\le\beta^{-1}\left(\rho_{y_i}^{(t)}(i,p) + \beta\rho_{-y_i}^{(t)}(i,p)\right),$$

for all $s,s'\in\{\pm1\}, k\in[K], i\in[n]$ and $p\in[P]\setminus\{p_i^*\}$.

We will prove the first argument by using induction on $t$. The initial case $t=0$ is trivial. Suppose the statement holds at $t=T$ and consider the case $t=T+1$.

Let $\tilde{T}_{s,k}\le T$ denote the smallest iteration where $\gamma_s^{(\tilde{T}_{s,k}+1)}(s,k) + \beta\gamma_{-s}^{(\tilde{T}_{s,k}+1)}(s,k) > 2\log(\eta T^*)$. We assume the existence of $\tilde{T}_{s,k}$, as its absence would directly lead to our desired conclusion; to see why, note that the following holds, due to (24) and (11):

$$\gamma_s^{(T+1)}(s,k) + \beta\gamma_{-s}^{(T+1)}(s,k)$$
$$= \gamma_s^{(T)}(s,k) + \beta\gamma_{-s}^{(T)}(s,k)$$
$$+ \frac{\eta}{n}\sum_{i\in\mathcal{V}_{s,k}}\mathbb{E}_{\mathcal{C}\sim\mathcal{D}_\mathcal{C}}\left[g_{i,\mathcal{C}}^{(T)}\cdot\mathbb{1}_{p_i^*\notin\mathcal{C}}\right]\left(\phi'\left(\left\langle\boldsymbol{w}_s^{(T)},\boldsymbol{v}_{s,k}\right\rangle\right) + \beta\phi'\left(\left\langle\boldsymbol{w}_{-s}^{(T)},\boldsymbol{v}_{s,k}\right\rangle\right)\right)$$

$$\leq 2\log(\eta T^*) + 2\eta \leq 4\log(\eta T^*)$$

By (24), we have

$$\gamma_s^{(T+1)}(s,k) + \beta\gamma_{-s}^{(T+1)}(s,k)$$

$$= \gamma_s^{(\tilde{T}_{s,k})}(s,k) + \beta\gamma_{-s}^{(\tilde{T}_{s,k})}(s,k)$$

$$+ \sum_{t=\tilde{T}_{s,k}}^{T} \left( \gamma_s^{(t+1)}(s,k) + \beta\gamma_{-s}^{(t+1)}(s,k) - \gamma_s^{(t)}(s,k) - \beta\gamma_{-s}^{(t)}(s,k) \right)$$

$$\leq 2\log(\eta T^*) + \log(\eta T^*)$$

$$+ \frac{\eta}{n} \sum_{t=\tilde{T}_{s,k}+1}^{T} \sum_{i\in\mathcal{V}_{s,k}} \mathbb{E}_{\mathcal{C}\sim\mathcal{D}_\mathcal{C}} \left[ g_{i,\mathcal{C}}^{(t)} \left( \phi'\left( \left\langle \boldsymbol{w}_s^{(t)}, \boldsymbol{v}_{s,k} \right\rangle \right) + \beta\phi'\left( \left\langle \boldsymbol{w}_{-s}^{(t)}, \boldsymbol{v}_{s,k} \right\rangle \right) \right) \cdot \mathbb{1}_{p_i^*\notin\mathcal{C}} \right].$$

The inequality is due to $\gamma_s^{(\tilde{T}_{s,k})}(s,k) + \beta\gamma_{-s}^{(\tilde{T}_{s,k})}(s,k) \leq 2\log(\eta T^*)$ and

$$\frac{\eta}{n} \sum_{i\in\mathcal{V}_{s,k}} \mathbb{E}_{\mathcal{C}\sim\mathcal{D}_\mathcal{C}} \left[ g_{i,\mathcal{C}}^{(\tilde{T}_{s,k})} \left( \phi'\left( \left\langle \boldsymbol{w}_s^{(\tilde{T}_{s,k})}, \boldsymbol{v}_{s,k} \right\rangle \right) + \beta\phi'\left( \left\langle \boldsymbol{w}_{-s}^{(\tilde{T}_{s,k})}, \boldsymbol{v}_{s,k} \right\rangle \right) \right) \cdot \mathbb{1}_{p_i^*\notin\mathcal{C}} \right]$$

$$\leq 2\eta \leq \log(\eta T^*),$$

from our choice of $\tilde{T}_{s,k}$ and $\eta$.

For each $t = \tilde{T}_{s,k} + 1, \ldots T$, $i \in \mathcal{V}_{s,k}$, and $\mathcal{C} \subset [P]$ such that $|\mathcal{C}| = C$ and $p_i^* \notin \mathcal{C}$, we have

$$y_i f_{\boldsymbol{W}^{(t)}}(\boldsymbol{X}_{i,\mathcal{C}})$$

$$= \phi\left( \left\langle \boldsymbol{w}_s^{(t)}, \boldsymbol{v}_{s,k} \right\rangle \right) - \phi\left( \left\langle \boldsymbol{w}_{-s}^{(t)}, \boldsymbol{v}_{s,k} \right\rangle \right) + \sum_{p\notin\mathcal{C}\cup\{p_i^*\}} \left( \phi\left( \left\langle \boldsymbol{w}_s^{(t)}, \boldsymbol{x}_i^{(p)} \right\rangle \right) - \phi\left( \left\langle \boldsymbol{w}_{-s}^{(t)}, \boldsymbol{x}_i^{(p)} \right\rangle \right) \right)$$

$$\geq \gamma_s^{(t)}(s,k) + \beta\gamma_{-s}^{(t)}(s,k) + \sum_{p\notin\mathcal{C}\cup\{p_i^*\}} \left( \rho_s^{(t)}(i,p) + \beta\rho_{-s}^{(t)}(i,p) \right) - 2P \cdot o\left( \frac{1}{\text{polylog}(d)} \right)$$

$$\geq \frac{3}{2}\log(\eta T^*)$$

The first inequality is due to Lemma B.4 and the second inequality holds due to (A7), (8), and our choice of $t$, $\gamma_s^{(t)}(s,k) + \beta\gamma_{-s}^{(t)}(s,k) \geq 2\log(\eta T^*)$.

Hence, we obtain

$$\frac{\eta}{n} \sum_{t=\tilde{T}_{s,k}}^{T} \sum_{i\in\mathcal{V}_{s,k}} \mathbb{E}_{\mathcal{C}\sim\mathcal{D}_\mathcal{C}} \left[ g_{i,\mathcal{C}}^{(t)} \left( \phi'\left( \left\langle \boldsymbol{w}_s^{(t)}, \boldsymbol{v}_{s,k} \right\rangle \right) + \beta\phi'\left( \left\langle \boldsymbol{w}_{-s}^{(t)}, \boldsymbol{v}_{s,k} \right\rangle \right) \right) \cdot \mathbb{1}_{p_i^*\notin\mathcal{C}} \right]$$

$$\leq \frac{2\eta}{n} \sum_{t=\tilde{T}_{s,k}}^{T} \sum_{i\in\mathcal{V}_{s,k}} \mathbb{E}_{\mathcal{C}\sim\mathcal{D}_\mathcal{C}} \left[ \exp\left( -y_i f_{\boldsymbol{W}^{(t)}}(\boldsymbol{X}_{i,\mathcal{C}}) \right) \cdot \mathbb{1}_{p_i^*\notin\mathcal{C}} \right]$$

$$\leq \frac{2|\mathcal{V}_{s,k}|}{n}(\eta T^*)\exp\left( -\frac{3}{2}\log(\eta T^*) \right)$$

$$\leq \frac{2}{\sqrt{\eta T^*}} \leq \log(\eta T^*),$$

where the last inequality holds for any reasonably large $T^*$. Merging all inequalities together, we have $\gamma_s^{(T+1)}(s,k) + \beta\gamma_{-s}^{(T+1)}(s,k) \leq 4\log(\eta T^*)$.

Next, we will follow similar arguments to show that

$$\rho_{y_i}^{(T+1)}(i,p) + \beta\rho_{-y_i}^{(T+1)}(i,p) \leq 4\log(\eta T^*)$$

for each $i \in [n]$ and $p \in [P] \setminus \{p_i^*\}$.

Let $\tilde{T}_i^{(p)} \leq T$ be the smallest iteration such that $\rho_{y_i}^{(\tilde{T}_i^{(p)}+1)}(i,p) + \beta\rho_{-y_i}^{(\tilde{T}_i^{(p)}+1)}(i,p) > 2\log(\eta T^*)$. We assume the existence of $\tilde{T}_i^{(p)}$, , as its absence would directly lead to our desired conclusion; to see why, note that the following holds, due to (25) and (11):

$$
\rho_{y_i}^{(T+1)}(i,p) + \beta\rho_{-y_i}^{(T+1)}(i,p)
$$
$$
= \rho_{y_i}^{(T)}(i,p) + \beta\rho_{-y_i}^{(T)}(i,p)
$$
$$
+ \frac{\eta}{n}\mathbb{E}_{\mathcal{C}\sim\mathcal{D}_\mathcal{C}}\left[g_{i,\mathcal{C}}^{(T)} \cdot \mathbb{1}_{p\notin\mathcal{C}}\right]\left(\phi'\left(\left\langle \boldsymbol{w}_s^{(t)}, \boldsymbol{x}_i^{(p)}\right\rangle\right) + \beta\phi'\left(\left\langle \boldsymbol{w}_s^{(t)}, \boldsymbol{x}_i^{(p)}\right\rangle\right)\right)\left\|\xi_i^{(p)}\right\|^2
$$
$$
\leq 2\log(\eta T^*) + 2\eta \leq 4\log(\eta T^*),
$$

where the first inequality is due to $\left\|\xi_i^{(p)}\right\| \leq \frac{3}{2}\sigma_{\mathrm{d}}^2 d$ and (A4), and the last inequality is due to (11).

Now we suppose there exists such $\tilde{T}_i \leq T$. By (25), we have

$$
\rho_{y_i}^{(T+1)}(i,p) + \beta\rho_{-y_i}^{(T+1)}(i,p)
$$
$$
= \rho_{y_i}^{(\tilde{T}_i^{(p)})}(i,p) + \beta\rho_{-y_i}^{(\tilde{T}_i^{(p)})}(i,p)
$$
$$
+ \sum_{t=\tilde{T}_i^{(p)}}^{T}\left(\rho_{y_i}^{(t+1)}(i,p) + \beta\rho_{-y_i}^{(t+1)}(i,p) - \rho_{y_i}^{(t)}(i,p) - \beta\rho_{-y_i}^{(t)}(i,p)\right)
$$
$$
\leq 2\log(\eta T^*) + \log(\eta T^*)
$$
$$
+ \frac{\eta}{n}\sum_{t=\tilde{T}_i^{(p)}+1}^{T}\mathbb{E}_{\mathcal{C}\sim\mathcal{D}_\mathcal{C}}\left[g_{i,\mathcal{C}}^{(t)} \cdot \mathbb{1}_{p\notin\mathcal{C}}\right]\left(\phi'\left(\left\langle \boldsymbol{w}_s^{(t)}, \boldsymbol{x}_i^{(p)}\right\rangle\right) + \beta\phi'\left(\left\langle \boldsymbol{w}_{-s}^{(t)}, \boldsymbol{x}_i^{(p)}\right\rangle\right)\right)\left\|\xi_i^{(p)}\right\|^2
$$

The inequality is due to $\rho_{y_i}^{(t)}(i,p) + \beta\rho_{-y_i}^{(t)}(i,p) \leq 2\log(\eta T^*)$ our choice of $\tilde{T}_i^{(p)}$ and

$$
\frac{\eta}{n}\mathbb{E}_{\mathcal{C}\sim\mathcal{D}_\mathcal{C}}\left[g_{i,\mathcal{C}}^{(\tilde{T}_i^{(p)})} \cdot \mathbb{1}_{p\notin\mathcal{C}}\right]\left(\phi'\left(\left\langle \boldsymbol{w}_s^{(\tilde{T}_i^{(p)})}, \boldsymbol{x}_i^{(p)}\right\rangle\right) + \beta\phi'\left(\left\langle \boldsymbol{w}_{-s}^{(\tilde{T}_i^{(p)})}, \boldsymbol{x}_i^{(p)}\right\rangle\right)\right)\left\|\xi_i^{(p)}\right\|^2
$$
$$
\leq 2\eta \leq \log(\eta T^*),
$$

from $\left\|\xi_i^{(p)}\right\|^2 \leq \frac{3}{2}\sigma_{\mathrm{d}}^2 d$, (A4), and (11).

For each $t = \tilde{T}_i^{(p)}+1, \ldots, T$, and $\mathcal{C}\subset[P]$ such that $|\mathcal{C}| = C$ and $p\notin\mathcal{C}$, we have

$$
y_i f_{\boldsymbol{W}^{(t)}}(\boldsymbol{X}_{i,\mathcal{C}})
$$
$$
= \phi\left(\left\langle \boldsymbol{w}_{y_i}^{(t)}, \boldsymbol{x}_i^{(p)}\right\rangle\right) - \phi\left(\left\langle \boldsymbol{w}_{-y_i}^{(t)}, \boldsymbol{x}_i^{(p)}\right\rangle\right) + \sum_{q\notin\mathcal{C}\cup\{p\}}\left(\phi\left(\left\langle \boldsymbol{w}_{y_i}^{(t)}, \boldsymbol{x}_i^{(q)}\right\rangle\right) - \phi\left(\left\langle \boldsymbol{w}_{-y_i}^{(t)}, \boldsymbol{x}_i^{(q)}\right\rangle\right)\right)
$$
$$
\geq \rho_{y_i}^{(t)}(i,p) + \beta\rho_{-y_i}^{(t)}(i,p) - 2P\cdot o\left(\frac{1}{\mathrm{polylog}(d)}\right)
$$
$$
\geq \frac{3}{2}\log(\eta T^*).
$$

The first inequality is due to Lemma B.4 and the second inequality holds since from our choice of $t$, $\rho_{y_i}^{(t)}(i,p) + \beta\rho_{-y_i}^{(t)}(i,p) \geq 2\log(\eta T^*)$.

Therefore, we have

$$
\frac{\eta}{n}\sum_{t=\tilde{T}_i^{(p)}+1}^{T}\mathbb{E}_{\mathcal{C}\sim\mathcal{D}_\mathcal{C}}\left[g_{i,\mathcal{C}}^{(t)} \cdot \mathbb{1}_{p\notin\mathcal{C}}\right]\left(\phi'\left(\left\langle \boldsymbol{w}_{y_i}^{(t)}, \boldsymbol{x}_i^{(p)}\right\rangle\right) + \beta\phi'\left(\left\langle \boldsymbol{w}_{-y_i}^{(t)}, \boldsymbol{x}_i^{(p)}\right\rangle\right)\right)\left\|\xi_i^{(p)}\right\|^2
$$
$$
\leq \eta\sum_{t=\tilde{T}_i^{(p)}+1}^{T}\mathbb{E}_{\mathcal{C}\sim\mathcal{D}_\mathcal{C}}\left[\exp\left(-y_i f_{\boldsymbol{W}^{(t)}}(\boldsymbol{X}_{i,\mathcal{C}})\right)\mathbb{1}_{p\notin\mathcal{C}}\right] \leq (\eta T^*)\exp\left(-\frac{3}{2}\log(\eta T^*)\right)
$$

$$\leq \frac{1}{\sqrt{\eta T^*}} \leq \log(\eta T^*),$$

where the first inequality is due to $\left\| \xi_i^{(p)} \right\|^2 \leq \frac{3}{2}\sigma_{\mathrm{d}}^2 d$ and (A4). Hence, we conclude $\rho_{y_i}^{(T+1)}(i,p) + \beta\rho_{-y_i}^{(T+1)}(i,p) \leq 4\log(\eta T^*)$. $\qquad\square$

### D.2.2 Learning Common Features and Rare Features

In the initial stages of training, the model quickly learns common features while exhibiting minimal overfitting to Gaussian noise.

First, we establish lower bounds on the number of iterations, ensuring that background noise coefficients $\rho_s^{(t)}(i,p)$ for $p \neq p_i^*, \tilde{p}_i$ remain small, up to the order of $\frac{1}{P}$.

**Lemma D.2.** *Suppose the event $E_{\mathrm{init}}$ occurs. There exists $\tilde{T} > \frac{n}{6\eta P\sigma_{\mathrm{b}}^2 d}$ such that $\rho_s^{(t)}(i,p) \leq \frac{1}{4P}$ for all $0 \leq t < \tilde{T}, s \in \{\pm 1\}, i \in [n]$ and $p \in [P] \setminus \{p_i^*, \tilde{p}_i\}$.*

*Proof of Lemma D.2.* Let $\tilde{T}$ be the smallest iteration such that $\rho_s^{(\tilde{T})}(i,p) \geq \frac{1}{4P}$ for some $s \in \{\pm 1\}, i \in [n]$ and $p \in [P] \setminus \{p_i^*\}$. We assume the existence of $\tilde{T}$, as its absence would directly lead to our conclusion. Then, for any $0 \leq t < \tilde{T}$, we have

$$\rho_s^{(t+1)}(i,p) = \rho_s^{(t)}(i,p) + \frac{\eta}{n}\mathbb{E}_{\mathcal{C}\sim\mathcal{D}_{\mathcal{C}}}\left[ g_{i,\mathcal{C}}^{(t)}\phi'\left(\left\langle \boldsymbol{w}_s^{(t)}, \boldsymbol{x}_i^{(p)} \right\rangle\right) \cdot \mathbb{1}_{p\notin\mathcal{C}}\right]\left\| \xi_i^{(p)} \right\|^2 < \rho_s^{(t)}(i,p) + \frac{3\eta\sigma_{\mathrm{b}}^2 d}{2n},$$

where the inequality is due to $g_{i,\mathcal{C}}^{(t)} < 1$, $\phi' \leq 1$, and $\left\| \xi_i^{(p)} \right\|^2 \leq \frac{3}{2}\sigma_{\mathrm{b}}^2 d$. Hence, we have

$$\frac{1}{4P} \leq \rho_s^{(\tilde{T})}(i,p) = \sum_{t=0}^{\tilde{T}-1}\left(\rho_s^{(t+1)}(i,p) - \rho_s^{(t)}(i,p)\right) < \frac{3\eta\sigma_{\mathrm{b}}^2 d}{2n}\tilde{T},$$

and we conclude $\tilde{T} > \frac{n}{6\eta P\sigma_{\mathrm{b}}^2 d}$ which is the desired result. $\qquad\square$

Next, we will show that the model learns common features in at least constant order within $\tilde{T}$ iterates.

**Lemma D.3.** *Suppose the event $E_{\mathrm{init}}$ occurs and $\rho_k = \omega\left(\frac{\sigma_{\mathrm{b}}^2 d}{\beta n}\right)$ for some $k \in [K]$. Then, for each $s \in \{\pm 1\}$. there exists $T_{s,k} \leq \frac{9nP}{\eta\beta|\mathcal{V}_{s,k}|}$ such that $\gamma_s^{(t)}(s,k) + \beta\gamma_{-s}^{(t)}(s,k) \geq 1$ for any $t > T_{s,k}$.*

*Proof of Lemma D.3.* Suppose $\gamma_s^{(t)}(s,k) + \beta\gamma_{-s}^{(t)}(s,k) < 1$ for all $0 \leq t \leq \frac{n}{6\eta P\sigma_{\mathrm{b}}^2 d}$. For each $i \in \mathcal{V}_{s,k}$ and $\mathcal{C} \subset [P]$ with $|\mathcal{C}| = C$ such that $p_i^* \notin \mathcal{C}$ and $\tilde{p}_i \in \mathcal{C}$, we have

$$y_i f_{\boldsymbol{W}^{(t)}}(\boldsymbol{X}_{i,\mathcal{C}})$$
$$= \phi\left(\left\langle \boldsymbol{w}_s^{(t)}, \boldsymbol{v}_{s,k} \right\rangle\right) - \phi\left(\left\langle \boldsymbol{w}_{-s}^{(t)}, \boldsymbol{v}_{s,k} \right\rangle\right) + \sum_{p\notin\mathcal{C}\cup\{p_i^*\}}\left(\phi\left(\left\langle \boldsymbol{w}_s^{(t)}, \boldsymbol{x}_i^{(p)} \right\rangle\right) - \phi\left(\left\langle \boldsymbol{w}_{-s}^{(t)}, \boldsymbol{x}_i^{(p)} \right\rangle\right)\right)$$
$$\leq \gamma_s^{(t)}(s,k) + \beta\gamma_{-s}^{(t)}(s,k) + \sum_{p\notin\mathcal{C}\cup\{p_i^*\}}\left(\rho_s^{(t)}(i,p) + \beta\rho_{-s}^{(t)}(i,p)\right) + 2P \cdot o\left(\frac{1}{\mathrm{polylog}(d)}\right)$$
$$\leq 1 + 2P \cdot \frac{1}{4P} + 2P \cdot o\left(\frac{1}{\mathrm{polylog}(d)}\right)$$
$$\leq 2.$$

The first inequality is due to Lemma B.4, the second inequality holds since we can apply Lemma D.2, and the last inequality is due to (A1). Thus, $g_{i,\mathcal{C}}^{(t)} = \frac{1}{1+\exp\left(y_i f_{\boldsymbol{W}^{(t)}}(\boldsymbol{X}_{i,\mathcal{C}})\right)} > \frac{1}{9}$ and we have

$$\gamma_s^{(t+1)}(s,k) + \beta\gamma_{-s}^{(t+1)}(s,k)$$

$$
\begin{aligned}
&= \gamma_s^{(t)}(s,k) + \beta\gamma_{-s}^{(t)}(s,k) \\
&\quad + \frac{\eta}{n} \sum_{i \in \mathcal{V}_{s,k}} \mathbb{E}_{\mathcal{C} \sim \mathcal{D}_{\mathcal{C}}} \left[ g_{i,\mathcal{C}}^{(t)} \left( \phi'\left( \left\langle \boldsymbol{w}_s^{(t)}, \boldsymbol{v}_{s,k} \right\rangle \right) + \beta\phi'\left( \left\langle \boldsymbol{w}_{-s}^{(t)}, \boldsymbol{v}_{s,k} \right\rangle \right) \right) \cdot \mathbb{1}_{p_i^* \notin \mathcal{C}} \right] \\
&\geq \gamma_s^{(t)}(s,k) + \beta\gamma_{-s}^{(t)}(s,k) + \frac{\eta\beta}{9n} \sum_{i \in \mathcal{V}_{s,k}} \mathbb{E}_{\mathcal{C} \sim \mathcal{D}_{\mathcal{C}}}[\mathbb{1}_{p_i^* \notin \mathcal{C} \wedge \tilde{p}_i \in \mathcal{C}}] \\
&= \gamma_s^{(t)}(s,k) + \beta\gamma_{-s}^{(t)}(s,k) + \frac{\eta\beta|\mathcal{V}_{s,k}|C(P-C)}{9nP(P-1)} \\
&\geq \gamma_s^{(t)}(s,k) + \beta\gamma_{-s}^{(t)}(s,k) + \frac{\eta\beta|\mathcal{V}_{s,k}|}{9nP}.
\end{aligned}
$$

From the given condition in the lemma statement, we have $\frac{9nP}{\eta\beta|\mathcal{V}_{s,k}|} = o\left(\frac{n}{6\eta P\sigma_{\mathrm{b}}^2 d}\right)$. If we choose $t_0 \in \left[\frac{9nP}{\eta\beta|\mathcal{V}_{s,k}|}, \frac{n}{6\eta P\sigma_{\mathrm{b}}^2 d}\right]$, then

$$
1 > \gamma_s^{(t_0)}(s,k) + \beta\gamma_{-s}^{(t_0)}(s,k) \geq \frac{\eta\beta|\mathcal{V}_{s,k}|}{9nP}t_0 \geq 1,
$$

and this is contradictory; therefore, it cannot hold that $\gamma_s^{(t)}(s,k) + \beta\gamma_{-s}^{(t)}(s,k) < 1$ for all $0 \leq t \leq \frac{n}{6\eta P\sigma_{\mathrm{b}}^2 d}$. Hence, there exists $0 \leq T_{s,k} < \frac{n}{6\eta P\sigma_{\mathrm{b}}^2 d}$ such that $\gamma_s^{(T_{s,k}+1)}(s,k) + \beta\gamma_{-s}^{(T_{s,k}+1)}(s,k) \geq 1$ and choose the smallest one. Then we obtain

$$
1 \geq \gamma_s^{(t)}(s,k) + \beta\gamma_{-s}^{(t)}(s,k) \geq \frac{\eta\beta|\mathcal{V}_{s,k}|}{9nP}T_{s,k}.
$$

Therefore, $T_{s,k} \leq \frac{9nP}{\eta\beta|\mathcal{V}_{s,k}|}$ and this is what we desired. $\qquad\square$

**What We Have So Far.** For any common feature or rare feature $\boldsymbol{v}_{s,k}$ with $s \in \{\pm 1\}$ and $k \in \mathcal{K}_C \cup \mathcal{K}_R$, it satisfies $\rho_k = \omega\left(\frac{\sigma_{\mathrm{b}}^2 d}{\beta n}\right)$ due to (A5). By Lemma D.3, at any iterate $t \in \left[\bar{T}_1, T^*\right]$ with $\bar{T}_1 := \max_{s \in \{\pm 1\}, k \in \mathcal{C}} T_{s,k}$, the following properties hold if the event $E_{\mathrm{init}}$ occurs:

- (Learn common/rare features): For $s \in \{\pm 1\}$ and $k \in \mathcal{K}_C \cup \mathcal{K}_R$, $\gamma_s^{(t)}(s,k) + \beta\gamma_{-s}^{(t)}(s,k) = \Omega(1)$,

- For any $s \in \{\pm 1\}, i \in [n]$, and $p \in [P] \setminus \{p_i^*\}$, $\rho_s^{(t)}(i,p) = \widetilde{\mathcal{O}}\left(\beta^{-1}\right)$.

### D.2.3 Overfitting Augmented Data

In the previous step, we have shown that data containing common or rare features can be well-classified by learning common and rare features. In this step, we will show that the model correctly classifies the remaining training data by overfitting background noise instead of learning its features.

We first introduce lower bounds on the number of iterates such that feature coefficients $\gamma_s^{(t)}(s',k)$ remain small, up to the order of $\alpha^2\beta^{-1}$. This lemma holds to any kind of features, but we will focus on extremely rare features. This does not contradict the results from Section D.2.2 for common features and rare features since the upper bound on the number of iterations in Lemma D.3 is larger than the lower bound on the number of iterations in this lemma.

**Lemma D.4.** *Suppose the event $E_{\mathrm{init}}$ occurs. For each $s \in \{\pm 1\}$ and $k \in [K]$, there exists $\tilde{T}_{s,k} \geq \frac{n\alpha^2}{\eta\beta|\mathcal{V}_{s,k}|}$ such that $\gamma_{s'}^{(t)}(s,k) \leq \alpha^2\beta^{-1}$ for any $0 \leq t < \tilde{T}_{s,k}$ and $s' \in \{\pm 1\}$.*

*Proo of Lemma D.4.* Let $\tilde{T}_{s,k}$ be the smallest iterate such that $\gamma_{s'}^{(t)}(s,k) > \alpha^2\beta^{-1}$ for some $s' \in \{\pm 1\}$. We assume the existence of $\tilde{T}_{s,k}$, as its absence would directly lead to our conclusion.

For any $0 \leq t < \tilde{T}_{s,k}$,

$$
\gamma_{s'}^{(t+1)}(s,k) = \gamma_{s'}^{(t)}(s,k) + \frac{\eta}{n} \sum_{i \in \mathcal{V}_{s,k}} \mathbb{E}_{\mathcal{C} \sim \mathcal{D}_{\mathcal{C}}} \left[ g_{i,\mathcal{C}}^{(t)} \phi'\left( \left\langle \boldsymbol{w}_{s'}^{(t)}, \boldsymbol{v}_{s,k} \right\rangle \right) \cdot \mathbb{1}_{p_i^* \notin \mathcal{C}} \right] \leq \gamma_{s'}^{(t)}(s,k) + \frac{\eta|\mathcal{V}_{s,k}|}{n},
$$

and we have

$$\alpha^2 \beta^{-1} \le \gamma_{s'}^{(\tilde{T}_{s,k})}(s,k) = \sum_{t=0}^{\tilde{T}_{s,k}-1} \left( \gamma_{s'}^{(t+1)}(s,k) - \gamma_{s'}^{(t)}(s,k) \right) \le \frac{\eta |\mathcal{V}_{s,k}|}{n} \tilde{T}_{s,k}.$$

We conclude $\tilde{T}_{s,k} \ge \frac{n\alpha^2}{\eta\beta|\mathcal{V}_{s,k}|}$ which is the desired result. $\qquad\square$

Next, we will show that the model overfits data augmented not containing common or rare features in at least constant order within $\tilde{T}_{s,k}$ iterates.

**Lemma D.5.** *Suppose the event $E_{\mathrm{init}}$ occurs and $\rho_k = o\left(\frac{\alpha^2 \sigma_b^2 d}{n}\right)$. For each $i \in [n]$ and $\mathcal{C} \subset [P]$ with $|\mathcal{C}| = C$, if (1) $i \in \mathcal{V}_{y_i,k}$ and $p_i^* \notin \mathcal{C}$ or (2) $i \in [n]$ and $p_i^* \in \mathcal{C}$, then there exists $T_{i,\mathcal{C}} \in \left[\bar{T}_1, \frac{18n\binom{P}{C}}{\eta\beta\sigma_b^2 d}\right]$ such that*

$$\sum_{p \notin \mathcal{C} \cup \{p_i^*\}} \left( \rho_{y_i}^{(t)}(i,p) + \beta\rho_{-y_i}^{(t)}(i,p) \right) \ge 1,$$

*for any $t > T_{i,\mathcal{C}}$.*

*Proof of Lemma D.5.* We can address both cases in the statement simultaneously. Suppose $\sum_{p \notin \mathcal{C} \cup \{p_i^*\}} \left( \rho_{y_i}^{(t)}(i,p) + \beta\rho_{-y_i}^{(t)}(i,p) \right) < 1$ for all $0 \le t \le \frac{n\alpha^2}{\eta\beta|\mathcal{V}_{y_i,k}|}$.

From Lemma B.4 and Lemma D.4, we have

$$y_i f_{\boldsymbol{W}^{(t)}}(\boldsymbol{X}_{i,\mathcal{C}})$$

$$= \sum_{p \notin \mathcal{C}} \left( \phi\left(\left\langle \boldsymbol{w}_{y_i}^{(t)}, \boldsymbol{x}_i^{(p)} \right\rangle\right) - \phi\left(\left\langle \boldsymbol{w}_{-y_i}^{(t)}, \boldsymbol{x}_i^{(p)} \right\rangle\right) \right)$$

$$\le \gamma_{y_i}^{(t)}(y_i, k) + \beta\gamma_{-y_i}^{(t)}(y_i, k) + \sum_{p \notin \mathcal{C} \cup \{p_i^*\}} \left( \rho_{y_i}^{(t)}(i,p) + \beta\rho_{-y_i}^{(t)}(i,p) \right) + 2P \cdot o\left(\frac{1}{\mathrm{polylog}(d)}\right)$$

$$\le (1+\beta)\alpha^2\beta^{-1} + 1 + 2P \cdot o\left(\frac{1}{\mathrm{polylog}(d)}\right)$$

$$\le 2,$$

and $g_{i,\mathcal{C}}^{(t)} = \frac{1}{1+\exp\left(y_i f_{\boldsymbol{W}^{(t)}}(\boldsymbol{X}_{i,\mathcal{C}})\right)} \ge \frac{1}{9}$. Also, for each $p \notin \mathcal{C} \cup \{p_i^*\}$, we have

$$\rho_s^{(t+1)}(i,p) + \beta\rho_{-s}^{(t+1)}(i,p)$$

$$\ge \rho_s^{(t)}(i,p) + \beta\rho_{-s}^{(t)}(i,p)$$

$$\quad + \frac{\eta}{n}\mathbb{P}_{\mathcal{C}'\sim\mathcal{D}_{\mathcal{C}}}[\mathcal{C}' = \mathcal{C}]g_{i,\mathcal{C}}^{(t)} \left( \phi'\left(\left\langle \boldsymbol{w}_s^{(t)}, \boldsymbol{x}_i^{(p)} \right\rangle\right) + \beta\phi'\left(\left\langle \boldsymbol{w}_{-s}^{(t)}, \boldsymbol{x}_i^{(p)} \right\rangle\right) \right) \left\| \xi_i^{(p)} \right\|^2$$

$$\ge \rho_s^{(t)}(i,p) + \beta\rho_{-s}^{(t)}(i,p) + \frac{\eta\beta\sigma_b^2 d}{18n\binom{P}{C}},$$

where the last inequality is due to $\left\| \xi_i^{(p)} \right\|^2 \ge \frac{1}{2}\sigma_b^2 d$ and $\phi' \ge \beta$. We also have

$$\sum_{p \notin \mathcal{C} \cup \{p_i^*\}} \left( \rho_s^{(t+1)}(i,p) + \beta\rho_{-s}^{(t+1)}(i,p) \right) \ge \sum_{p \notin \mathcal{C} \cup \{p_i^*\}} \left( \rho_s^{(t)}(i,p) + \beta\rho_{-s}^{(t)}(i,p) \right) + \frac{\eta\beta\sigma_b^2 d}{18n\binom{P}{C}}$$

From the given condition in the lemma statement, we have $\frac{18n\binom{P}{C}}{\eta\beta\sigma_b^2 d} = o\left(\frac{n\alpha^2}{\eta\beta|\mathcal{V}_{s,k}|}\right)$. If we choose $t_0 \in \left[\frac{18n\binom{P}{C}}{\eta\beta\sigma_b^2 d}, \frac{n\alpha^2}{\eta\beta|\mathcal{V}_{s,k}|}\right]$, then we have

$$1 > \sum_{p \notin \mathcal{C} \cup \{p_i^*\}} \left( \rho_s^{(t_0)}(i,p) + \beta\rho_{-s}^{(t_0)}(i,p) \right) \ge \frac{\eta\beta\sigma_b^2 d}{18n\binom{P}{C}} t_0 \ge 1,$$

and this is a contradiction; therefore, it cannot hold that $\sum_{p \notin \mathcal{C} \cup \{p_i^*\}} \left( \rho_{y_i}^{(t)}(i,p) + \beta \rho_{-y_i}^{(t)}(i,p) \right) < 1$ for all $0 \leq t \leq \frac{n\alpha^2}{\eta\beta|\mathcal{V}_{y_i,k}|}$. Thus, there exists $0 \leq T_{i,\mathcal{C}} < \frac{n\alpha^2}{\eta\beta|\mathcal{V}_{s,k}|}$ satisfying $\sum_{p \notin \mathcal{C} \cup \{p_i^*\}} \left( \rho_s^{(T_{i,\mathcal{C}}+1)}(i,p) + \beta \rho_{-s}^{(T_{i,\mathcal{C}}+1)}(i,p) \right) \geq 1$ and let us choose the smallest one.

For any $0 \leq t < T_{i,\mathcal{C}}$, we have

$$1 \geq \sum_{p \notin \mathcal{C} \cup \{p_i^*\}} \left( \rho_s^{(T_{i,\mathcal{C}})}(i,p) + \beta \rho_{-s}^{(T_{i,\mathcal{C}})}(i,p) \right) \geq \frac{\eta \sigma_{\mathrm{b}}^2 d}{18n\binom{P}{C}} T_i,$$

and we conclude that $T_{i,\mathcal{C}} \leq \frac{18n\binom{P}{C}}{\eta\beta\sigma_{\mathrm{b}}^2 d}$.

Lastly, we move on to prove $T_{i,\mathcal{C}} > \bar{T}_1$. Combining Lemma D.2 and Lemma D.3 leads to

$$\sum_{p \notin \mathcal{C} \cup \{p_i^*\} \setminus \{p_i^*\}} \left( \rho_s^{(\bar{T}_1)}(i,p) + \beta \rho_{-s}^{(\bar{T}_1)}(i,p) \right) \leq \frac{1}{2}.$$

Thus, we have $T_{i,\mathcal{C}} > \bar{T}_1$ and this is what we desired. □

**What We Have So Far.** For any $k \in \mathcal{K}_E$, it satisfies $\rho_k = o\left( \frac{\alpha^2 n}{\sigma_{\mathrm{b}}^2 d} \right)$ due to (A6). By Lemma D.5 at iterate $t \in [T_{\mathrm{Cutout}}, T^*]$ with

$$T_{\mathrm{Cutout}} := \max \left\{ \max_{k \in \mathcal{K}_E, i \in \mathcal{V}_{y_i,k}, p_i^* \notin \mathcal{C}} T_{i,\mathcal{C}}, \max_{i \in [n], p_i^* \in \mathcal{C}} T_{i,\mathcal{C}} \right\} \in [\bar{T}_1, T^*]$$

the following properties hold if the event $E_{\mathrm{init}}$ occurs:

- (Learn common/rare features): For any $s \in \{\pm 1\}$ and $k \in \mathcal{K}_C \cup \mathcal{K}_R$,
$$\gamma_s^{(t)}(s,k) + \beta \gamma_{-s}^{(t)}(s,k) = \Omega(1),$$

- (Overfit augmented data with extremely rare features or no feature): For each $i \in [n], k \in \mathcal{K}_E$, $\mathcal{C} \subset [P]$ with $|\mathcal{C}| = C$ such that (1) $i \in \mathcal{V}_{y_i,k}$ and $p_i^* \notin \mathcal{C}$ or (2) $i \in [n]$ and $p_i^* \in \mathcal{C}$
$$\sum_{p \notin \mathcal{C} \cup \{p_i^*\}} \left( \rho_{y_i}^{(t)}(i,p) + \beta \rho_{-y_i}^{(t)}(i,p) \right) = \Omega(1).$$

- (Do not learn extremely rare features at $T_{\mathrm{Cutout}}$): For any $s, s' \in \{\pm 1\}$ and $k \in \mathcal{K}_E$,
$$\gamma_{s'}^{(T_{\mathrm{Cutout}})}(s,k) \leq \alpha^2 \beta^{-1}.$$

- For any $s \in \{\pm 1\}, i \in [n]$, and $p \in [P] \setminus \{p_i^*\}$, $\rho_s^{(t)}(i,p) = \widetilde{\mathcal{O}}\left( \beta^{-1} \right)$.

### D.2.4 Cutout cannot Learn Extremely Rare Features Within Polynomial Times

In this step, We will show that Cutout cannot learn extremely rare features within the maximum admissible iterate $T^* = \frac{\mathrm{poly}(d)}{\eta}$.

we fix any $s^* \in \{\pm 1\}$ and $k^* \in \mathcal{K}_E$. Recall the function $Q^{(s^*,k^*)} : \mathcal{W} \to \mathbb{R}^{d \times 2}$, defined in Lemma B.5 and omit superscripts for simplicity. For each iteration $t$, $Q(\boldsymbol{W}^{(t)})$ represents quantities updates by data with feature vector $\boldsymbol{v}_{s^*,k^*}$ until $t$-th iteration. We will sequentially introduce several technical lemmas and by combining these lemmas, quantify update by data with feature vector $\boldsymbol{v}_{s^*,k^*}$ after $T_{\mathrm{Cutout}}$ and derive our conclusion.

Let us define $\boldsymbol{W}^* = \{\boldsymbol{w}_1^*, \boldsymbol{w}_{-1}^*\}$, where

$$\boldsymbol{w}_s^* = \boldsymbol{w}_s^{(T_{\mathrm{Cutout}})} + M \sum_{i \in \mathcal{V}_{s^*,k^*}} \sum_{p \in [P] \setminus \{p_i^*, \tilde{p}_i\}} \frac{\xi_i^{(p)}}{\left\| \xi_i^{(p)} \right\|^2},$$

where $M = 4\beta^{-1} \log \left( \frac{2\eta\beta^2 T^*}{\alpha^2} \right)$. Note that (12), $\beta < 1$, and $T^* = \frac{\mathrm{poly}(d)}{\eta}$ together imply $M = \widetilde{\mathcal{O}}\left( \beta^{-1} \right)$. Note that $\boldsymbol{W}^{(t)}, \boldsymbol{W}^* \in \mathcal{W}$ for any $t \geq 0$.

**Lemma D.6.** *Suppose the event $E_{\mathrm{init}}$ occurs. Then,*

$$\left\| Q\left(\boldsymbol{W}^{(T_{\mathrm{Cutout}})}\right) - Q(\boldsymbol{W}^*)\right\|^2 \leq 8M^2 P |\mathcal{V}_{s^*,k^*}|\sigma_{\mathrm{b}}^{-2}d^{-1}.$$

*where $\|\cdot\|$ denotes the Frobenius norm.*

*Proof of Lemma D.6.* For each $s \in \{\pm 1\}$,

$$ss^*\left(Q_s\left(\boldsymbol{w}_s^*\right) - Q_s\left(\boldsymbol{w}_s^{(T_{\mathrm{Cutout}})}\right)\right)$$

$$= Q_s\left(ss^*M \sum_{i\in\mathcal{V}_{s^*,k^*}}\sum_{p\in[P]\setminus\{p_i^*,\tilde{p}_i\}}\frac{\xi_i^{(p)}}{\left\|\xi_i^{(p)}\right\|}\right)$$

$$= M \sum_{i\in\mathcal{V}_{s^*,k^*}}\sum_{p\in[P]\setminus\{p_i^*,\tilde{p}_i\}}\frac{\xi_i^{(p)}}{\left\|\xi_i^{(p)}\right\|^2},$$

and we have

$$\left\| Q\left(\boldsymbol{W}^{(T_{\mathrm{Cutout}})}\right) - Q(\boldsymbol{W}^*)\right\|^2$$

$$= \left\| Q_1(\boldsymbol{w}_1^*) - Q_1\left(\boldsymbol{w}_1^{(T_{\mathrm{Cutout}})}\right)\right\|^2 + \left\| Q_{-1}(\boldsymbol{w}_{-1}^*) - Q_{-1}\left(\boldsymbol{w}_{-1}^{(T_{\mathrm{Cutout}})}\right)\right\|^2$$

$$\leq 2M^2 \left(\sum_{i\in\mathcal{V}_{s^*,k^*},p\in[P]\setminus\{p_i^*,\tilde{p}_i\}}\left\|\xi_i^{(p)}\right\|^{-2} + \sum_{\substack{i,j\in\mathcal{V}_{s^*,k^*}\\ p\in[P]\setminus\{p_i^*,\tilde{p}_i\},q\in[P]\setminus\{p_j^*,\tilde{p}_j\}\\ (i,p)\neq(j,q)}}\frac{\left|\left\langle\xi_i^{(p)},\xi_j^{(q)}\right\rangle\right|}{\left\|\xi_i^{(p)}\right\|^2\left\|\xi_j^{(q)}\right\|^2}\right).$$

From $E_{\mathrm{init}}$ and (A2), we have

$$\sum_{\substack{i,j\in\mathcal{V}_{s^*,k^*}\\ p\in[P]\setminus\{p_i^*,\tilde{p}_i\},q\in[P]\setminus\{p_j^*,\tilde{p}_j\}\\ (i,p)\neq(j,q)}}\frac{\left|\left\langle\xi_i^{(p)},\xi_j^{(q)}\right\rangle\right|}{\left\|\xi_i^{(p)}\right\|^2\left\|\xi_j^{(q)}\right\|^2} \leq \sum_{\substack{i\in\mathcal{V}_{s^*,k^*}\\ p\in[P]\setminus\{p_i^*,\tilde{p}_i\}}}\sum_{\substack{j\in\mathcal{V}_{s^*,k^*}\\ p\in[P]\setminus\{p_j^*,\tilde{p}_j\}}}\left\|\xi_i^{(\tilde{p})}\right\|^{-2}\widetilde{\mathcal{O}}\left(d^{-\frac{1}{2}}\right)$$

$$\leq \sum_{\substack{i\in\mathcal{V}_{s^*,k^*}\\ p\in[P]\setminus\{p_i^*,\tilde{p}_i\}}}\left\|\xi_i^{(\tilde{p})}\right\|^{-2}\widetilde{\mathcal{O}}\left(nPd^{-\frac{1}{2}}\right)$$

$$\leq \sum_{\substack{i\in\mathcal{V}_{s^*,k^*}\\ p\in[P]\setminus\{p_i^*,\tilde{p}_i\}}}\left\|\xi_i^{(p)}\right\|^{-2}$$

From the event $E_{\mathrm{init}}$ defined in Lemma B.2, we have

$$\sum_{\substack{i\in\mathcal{V}_{s^*,k^*}\\ p\in[P]\setminus\{p_i^*,\tilde{p}_i\}}}\left\|\xi_i^{(p)}\right\|^{-2} \leq 2P|\mathcal{V}_{s^*,k^*}|\sigma_{\mathrm{d}}^{-2}d^{-1},$$

and we obtain

$$\left\| Q\left(\boldsymbol{W}^{(T_{\mathrm{Cutout}})}\right) - Q(\boldsymbol{W}^*)\right\|^2 \leq 4M^2 \sum_{i\in\mathcal{V}_{s^*,k^*},p\in[P]\setminus\{p_i^*\}}\left\|\xi_i^{(p)}\right\|^{-2} \leq 8M^2 P|\mathcal{V}_{s^*,k^*}|\sigma_{\mathrm{b}}^{-2}d^{-1}.$$

$\square$

**Lemma D.7.** *Suppose the $E_{\mathrm{init}}$ occurs. For any $t \geq T_{\mathrm{Cutout}}$, $i \in \mathcal{V}_{s^*,k^*}$ and any $\mathcal{C} \subset [P]$ with $|\mathcal{C}| = C$, it holds that*

$$\langle y_i\nabla_{\boldsymbol{W}}f_{\boldsymbol{W}^{(t)}}(\boldsymbol{X}_{i,\mathcal{C}}), Q(\boldsymbol{W}^*)\rangle \geq \frac{M\beta}{2}.$$

*Proof of Lemma D.7.* We have

$$\langle y_i \nabla_{\boldsymbol{W}} f_{\boldsymbol{W}^{(t)}}(\boldsymbol{X}_{i,\mathcal{C}}), Q(\boldsymbol{W}^*)\rangle$$
$$= \sum_{p \notin \mathcal{C}} \left( \phi'\left(\left\langle \boldsymbol{w}_{s^*}^{(t)}, \boldsymbol{x}_i^{(p)}\right\rangle\right) \left\langle Q_{s^*}(\boldsymbol{w}_{s^*}^*), \boldsymbol{x}_i^{(p)}\right\rangle - \phi'\left(\left\langle \boldsymbol{w}_{-s^*}^{(t)}, \boldsymbol{x}_i^{(p)}\right\rangle\right) \left\langle Q_{-s^*}(\boldsymbol{w}_{-s^*}^*), \boldsymbol{x}_i^{(p)}\right\rangle \right).$$

For any $s \in \{\pm 1\}$ and $p \in [P] \setminus \{p_i^*, \tilde{p}_i\}$,

$$ss^* \left\langle Q_s(\boldsymbol{w}_s^*), \xi_i^{(p)}\right\rangle$$

$$\geq M + \rho_s^{(T_{\text{Cutout}})}(i, p) - \sum_{\substack{j \in [n], q \in [P] \setminus \{p_j^*\} \\ (j,q) \neq (i,p)}} \rho_s^{(T_{\text{Cutout}})}(j, q) \frac{\left|\left\langle \xi_i^{(p)}, \xi_j^{(q)}\right\rangle\right|}{\left\|\xi_j^{(q)}\right\|^2}$$

$$- M \sum_{\substack{j \in \mathcal{V}_{s^*,k^*}, q \in [P] \setminus \{p_j^*, \tilde{p}_j\} \\ (j,q) \neq (i,p)}} \frac{\left|\left\langle \xi_i^{(p)}, \xi_j^{(q)}\right\rangle\right|}{\left\|\xi_j^{(q)}\right\|^2}$$

$$\geq M - \widetilde{\mathcal{O}}\left(nP\beta^{-1}\sigma_{\text{d}}\sigma_{\text{b}}^{-1}d^{-\frac{1}{2}}\right) = M - o\left(\frac{1}{\text{polylog}(d)}\right)$$

$$\geq \frac{M}{2}, \tag{26}$$

where the last equality is due to (9). Also, for any $s \in \{\pm 1\}$, $ss^* \langle Q_s(\boldsymbol{w}_s^*), \boldsymbol{v}_{s^*,k^*}\rangle = \gamma_s^{(T_{\text{Cutout}})}(s^*, k^*) \geq 0$. In addition,

$$ss^* \left\langle Q_s(\boldsymbol{w}_s^*), \boldsymbol{x}_i^{(\tilde{p}_i)}\right\rangle$$

$$= ss^* \left\langle Q_s(\boldsymbol{w}_s^*), \xi_i^{(\tilde{p}_i)}\right\rangle + ss^* \left\langle Q_s(\boldsymbol{w}_s^*), \boldsymbol{x}_i^{(\tilde{p}_i)} - \xi_i^{(\tilde{p}_i)}\right\rangle$$

$$= ss^* \left\langle Q_s(\boldsymbol{w}_s^*), \xi_i^{(\tilde{p}_i)}\right\rangle - \widetilde{\mathcal{O}}\left(\alpha^2 \beta^{-1} \rho_{k^*} n \sigma_{\text{d}}^{-2} d^{-1}\right)$$

$$= \rho_s^{(T_{\text{Cutout}})}(i, \tilde{p}_i) + \sum_{\substack{j \in [n], q \in [P] \setminus \{p_i^*\} \\ (j,q) \neq (i,\tilde{p}_i)}} \rho_s^{(T_{\text{Cutout}})}(j, q) \frac{\left\langle \xi_i^{(\tilde{p}_i)}, \xi_j^{(q)}\right\rangle}{\left\|\xi_j^{(q)}\right\|^2} - \widetilde{\mathcal{O}}\left(\alpha^2 \beta^{-1} \rho_{k^*} n \sigma_{\text{d}}^{-2} d^{-1}\right)$$

$$\geq -\widetilde{\mathcal{O}}\left(nP\beta^{-1}\sigma_{\text{d}}\sigma_{\text{b}}^{-1}d^{-\frac{1}{2}}\right) - \widetilde{\mathcal{O}}\left(\alpha^2 \beta^{-1} \rho_{k^*} n \sigma_{\text{d}}^{-2} d^{-1}\right)$$

$$= -o\left(\frac{1}{\text{polylog}(d)}\right), \tag{27}$$

where the last equality is due to (9) and (A7).

For any $\mathcal{C} \subset [P]$ with $|\mathcal{C}| = C$, there exists $p \in [P] \setminus \{p_i^*, \tilde{p}_i\}$ such that $p \neq \mathcal{C}$ since $C < \frac{P}{2}$. By applying (26) and (27) for $s = s^*, -s^*$ and combining with $\phi' \geq \beta$, we have

$$\langle y_i \nabla_{\boldsymbol{W}} f_{\boldsymbol{W}^{(t)}}(\boldsymbol{X}_{i,\mathcal{C}}), Q(\boldsymbol{W}^*)\rangle$$

$$\geq \left( \phi'\left(\left\langle \boldsymbol{w}_{s^*}^{(t)}, \boldsymbol{x}_i^{(p)}\right\rangle\right) \left\langle Q_{s^*}(\boldsymbol{w}_{s^*}^*), \boldsymbol{x}_i^{(p)}\right\rangle - \phi'\left(\left\langle \boldsymbol{w}_{-s^*}^{(t)}, \boldsymbol{x}_i^{(p)}\right\rangle\right) \left\langle Q_{-s^*}(\boldsymbol{w}_{-s^*}^*), \boldsymbol{x}_i^{(p)}\right\rangle \right)$$

$$+ \sum_{q \notin \mathcal{C} \cup \{p\}} \left( \phi'\left(\left\langle \boldsymbol{w}_{s^*}^{(t)}, \boldsymbol{x}_i^{(q)}\right\rangle\right) \left\langle Q_{s^*}(\boldsymbol{w}_{s^*}^*), \boldsymbol{x}_i^{(q)}\right\rangle - \phi'\left(\left\langle \boldsymbol{w}_{-s^*}^{(t)}, \boldsymbol{x}_i^{(q)}\right\rangle\right) \left\langle Q_{-s^*}(\boldsymbol{w}_{-s^*}^*), \boldsymbol{x}_i^{(q)}\right\rangle \right)$$

$$\geq M\beta - o\left(\frac{1}{\text{polylog}(d)}\right)$$

$$\geq \frac{M\beta}{2}.$$

$\square$

By combining Lemma D.6 and Lemma D.7, we can obtain the following result.

**Lemma D.8.** *Suppose the event $E_{\mathrm{init}}$ occurs.*

$$\frac{\eta}{n} \sum_{t=T_{\mathrm{Cutout}}}^{T^*} \sum_{i \in \mathcal{V}_{s^*,k^*}} \mathbb{E}_{\mathcal{C} \sim \mathcal{D}_{\mathcal{C}}} [\ell\left(y_i f_{\mathbf{W}^{(t)}}(\mathbf{X}_{i,\mathcal{C}})\right)] \leq \left\| Q\left(\mathbf{W}^{(T_{\mathrm{Cutout}})}\right) - Q(\mathbf{W}^*) \right\|^2 + 2\eta T^* e^{-\frac{M\beta}{4}},$$

*where $\|\cdot\|$ denotes the Frobenius norm.*

*Proof of Lemma D.8.* Note that for any $T_{\mathrm{Cutout}} \leq t < T^*$,

$$Q\left(\mathbf{W}^{(t+1)}\right) = Q\left(\mathbf{W}^{(t)}\right) - \frac{\eta}{n} \nabla_{\mathbf{W}} \sum_{i \in \mathcal{V}_{s^*,k^*}} \mathbb{E}_{\mathcal{C} \sim \mathcal{D}_{\mathcal{C}}} \left[\ell\left(y_i f_{\mathbf{W}^{(t)}}(\mathbf{X}_{i,\mathcal{C}})\right)\right].$$

Therefore, we have

$$\left\| Q\left(\mathbf{W}^{(t)}\right) - Q\left(\mathbf{W}^*\right) \right\|^2 - \left\| Q\left(\mathbf{W}^{(t+1)}\right) - Q\left(\mathbf{W}^*\right) \right\|^2$$

$$= \frac{2\eta}{n} \left\langle \nabla_{\mathbf{W}} \sum_{i \in \mathcal{V}_{s^*,k^*}} \mathbb{E}_{\mathcal{C} \sim \mathcal{D}_{\mathcal{C}}} \left[\ell\left(y_i f_{\mathbf{W}^{(t)}}(\mathbf{X}_{i,\mathcal{C}})\right)\right], Q\left(\mathbf{W}^{(t)}\right) - Q\left(\mathbf{W}^*\right) \right\rangle$$

$$- \frac{\eta^2}{n^2} \left\| \nabla_{\mathbf{W}} \sum_{i \in \mathcal{V}_{s^*,k^*}} \mathbb{E}_{\mathcal{C} \sim \mathcal{D}_{\mathcal{C}}} \left[\ell\left(y_i f_{\mathbf{W}^{(t)}}(\mathbf{X}_{i,\mathcal{C}})\right)\right] \right\|^2$$

$$= \frac{2\eta}{n} \left\langle \nabla_{\mathbf{W}} \sum_{i \in \mathcal{V}_{s^*,k^*}} \mathbb{E}_{\mathcal{C} \sim \mathcal{D}_{\mathcal{C}}} \left[\ell(y_i f_{\mathbf{W}^{(t)}}(\mathbf{X}_{i,\mathcal{C}}))\right], Q\left(\mathbf{W}^{(t)}\right) \right\rangle$$

$$- \frac{2\eta}{n} \sum_{i \in \mathcal{V}_{s^*,k^*}} \langle \mathbb{E}_{\mathcal{C} \sim \mathcal{D}_{\mathcal{C}}} \left[\ell'(y_i f_{\mathbf{W}^{(t)}}(\mathbf{X}_{i,\mathcal{C}})) \nabla_{\mathbf{W}} y_i f_{\mathbf{W}^{(t)}}(\mathbf{X}_{i,\mathcal{C}})\right], Q\left(\mathbf{W}^*\right) \rangle$$

$$- \frac{\eta^2}{n^2} \left\| \nabla_{\mathbf{W}} \sum_{i \in \mathcal{V}_{s^*,k^*}} \mathbb{E}_{\mathcal{C} \sim \mathcal{D}_{\mathcal{C}}} \left[\ell\left(y_i f_{\mathbf{W}^{(t)}}(\mathbf{X}_{i,\mathcal{C}})\right)\right] \right\|^2$$

$$\geq \frac{2\eta}{n} \left\langle \nabla_{\mathbf{W}} \sum_{i \in \mathcal{V}_{s^*,k^*}} \mathbb{E}_{\mathcal{C} \sim \mathcal{D}_{\mathcal{C}}} \left[\ell(y_i f_{\mathbf{W}^{(t)}}(\mathbf{X}_{i,\mathcal{C}}))\right], Q\left(\mathbf{W}^{(t)}\right) \right\rangle$$

$$- \frac{M\beta\eta}{n} \sum_{i \in \mathcal{V}_{s^*,k^*}} \mathbb{E}_{\mathcal{C} \sim \mathcal{D}_{\mathcal{C}}} \left[\ell'(y_i f_{\mathbf{W}^{(t)}}(\mathbf{X}_{i,\mathcal{C}}))\right] - \frac{\eta^2}{n^2} \left\| \nabla_{\mathbf{W}} \sum_{i \in \mathcal{V}_{s^*,k^*}} \mathbb{E}_{\mathcal{C} \sim \mathcal{D}_{\mathcal{C}}} \left[\ell\left(y_i f_{\mathbf{W}^{(t)}}(\mathbf{X}_{i,\mathcal{C}})\right)\right] \right\|^2,$$

where the last inequality is due to Lemma D.7. By the chain rule, for each $\mathcal{C} \subset [P]$ with $|\mathcal{C}| = C$, we have

$$\left\langle \nabla_{\mathbf{W}} \sum_{i \in \mathcal{V}_{s^*,k^*}} \ell(y_i f_{\mathbf{W}^{(t)}}(\mathbf{X}_{i,\mathcal{C}})), Q\left(\mathbf{W}^{(t)}\right) \right\rangle$$

$$= \sum_{i \in \mathcal{V}_{s^*,k^*}} \left[ \ell'(y_i f_{\mathbf{W}^{(t)}}(\mathbf{X}_{i,\mathcal{C}})) \right.$$

$$\left. \times \sum_{p \notin \mathcal{C}} \left( \phi'\left(\left\langle \mathbf{w}_{s^*}^{(t)}, \mathbf{x}_i^{(p)} \right\rangle\right) \left\langle Q_{s^*}\left(\mathbf{w}_{s^*}^{(t)}\right), \mathbf{x}_i^{(p)} \right\rangle - \phi'\left(\left\langle \mathbf{w}_{-s^*}^{(t)}, \mathbf{x}_i^{(p)} \right\rangle\right) \left\langle Q_{-s^*}\left(\mathbf{w}_{-s^*}^{(t)}\right), \mathbf{x}_i^{(p)} \right\rangle \right) \right].$$

For each $s \in \{\pm 1\}$, $i \in \mathcal{V}_{s^*,k^*}$, and $p \in [P]$,

$$\left| \left\langle \mathbf{w}_s^{(t)}, \mathbf{x}_i^{(p)} \right\rangle - \left\langle Q_s\left(\mathbf{w}_s^{(t)}\right), \mathbf{x}_i^{(p)} \right\rangle \right|$$

$$= \left| \left\langle \mathbf{w}_s^{(t)} - Q_s\left(\mathbf{w}_s^{(t)}\right), \mathbf{x}_i^{(p)} \right\rangle \right|$$

$$\leq \sum_{j\in[n]\setminus\mathcal{V}_{s^*,k^*},q\in[P]\setminus\{p_i^*\}} \left| \left\langle \rho_s^{(t)}(j,q)\frac{\xi_j^{(q)}}{\left\|\xi_j^{(q)}\right\|^2}, \boldsymbol{x}_i^{(p)} \right\rangle \right|$$

$$+ \alpha \sum_{j\in\mathcal{F}_1\setminus\mathcal{V}_{s^*,k^*}} \rho_s^{(t)}(j,\tilde{p}_j)\left\|\xi_j^{(\tilde{p}_j)}\right\|^{-2}\left|\left\langle \boldsymbol{v}_{1,1}, \boldsymbol{x}_i^{(p)}\right\rangle\right|$$

$$+ \alpha \sum_{j\in\mathcal{F}_{-1}\setminus\mathcal{V}_{s^*,k^*}} \rho_s^{(t)}(j,\tilde{p}_j)\left\|\xi_j^{(\tilde{p}_j)}\right\|^{-2}\left|\left\langle \boldsymbol{v}_{-1,1}, \boldsymbol{x}_i^{(p)}\right\rangle\right|$$

$$\leq \widetilde{\mathcal{O}}\left(nP\beta^{-1}\sigma_{\mathrm{d}}\sigma_{\mathrm{b}}^{-1}d^{-\frac{1}{2}}\right) + \widetilde{\mathcal{O}}\left(\alpha^2\beta^{-1}n\sigma_{\mathrm{d}}^{-2}d^{-1}\right)$$

$$= o\left(\frac{1}{\mathrm{polylog}(d)}\right),$$

where the last inequality is due to Lemma D.1 and the event $E_{\mathrm{init}}$. By Lemma F.1,

$$\sum_{p\notin\mathcal{C}}\left(\phi'\left(\left\langle \boldsymbol{w}_{s^*}^{(t)}, \boldsymbol{x}_i^{(p)}\right\rangle\right)\left\langle Q_{s^*}\left(\boldsymbol{w}_{s^*}^{(t)}\right), \boldsymbol{x}_i^{(p)}\right\rangle - \phi'\left(\left\langle \boldsymbol{w}_{-s^*}^{(t)}, \boldsymbol{x}_i^{(p)}\right\rangle\right)\left\langle Q_{-s^*}\left(\boldsymbol{w}_{s^*}^{(t)}\right), \boldsymbol{x}_i^{(p)}\right\rangle\right)$$

$$\leq \sum_{p\notin\mathcal{C}}\left(\phi\left(\left\langle \boldsymbol{w}_{s^*}^{(t)}, \boldsymbol{x}_i^{(p)}\right\rangle\right) - \phi\left(\left\langle \boldsymbol{w}_{-s^*}^{(t)}, \boldsymbol{x}_i^{(p)}\right\rangle\right)\right) + rP + o\left(\frac{1}{\mathrm{polylog}(d)}\right)$$

$$= y_i f_{\boldsymbol{W}^{(t)}}(\boldsymbol{X}_{i,\mathcal{C}}) + o\left(\frac{1}{\mathrm{polylog}(d)}\right),$$

where the last equality is due to $r = o\left(\frac{1}{\mathrm{polylog}(d)}\right)$. Therefore, we have

$$\left\|Q\left(\boldsymbol{W}^{(t)}\right) - Q\left(\boldsymbol{W}^*\right)\right\|^2 - \left\|Q\left(\boldsymbol{W}^{(t+1)}\right) - Q(\boldsymbol{W}^*)\right\|^2$$

$$\geq \frac{2\eta}{n}\sum_{i\in\mathcal{V}_{s^*,k^*}}\mathbb{E}_{\mathcal{C}\sim\mathcal{D}_\mathcal{C}}\left[\ell'\left(y_i f_{\boldsymbol{W}^{(t)}}(\boldsymbol{X}_{i,\mathcal{C}})\right)\left(y_i f_{\boldsymbol{W}^{(t)}}(\boldsymbol{X}_{i,\mathcal{C}}) + o\left(\frac{1}{\mathrm{polylog}(d)}\right) - \frac{M\beta}{2}\right)\right]$$

$$- \frac{\eta^2}{n^2}\left\|\nabla_{\boldsymbol{W}}\sum_{i\in\mathcal{V}_{s^*,k^*}}\mathbb{E}_{\mathcal{C}\sim\mathcal{D}_\mathcal{C}}\left[\ell(y_i f_{\boldsymbol{W}^{(t)}}(\boldsymbol{X}_{i,\mathcal{C}}))\right]\right\|^2$$

$$\geq \frac{2\eta}{n}\sum_{i\in\mathcal{V}_{s^*,k^*}}\mathbb{E}_{\mathcal{C}\sim\mathcal{D}_\mathcal{C}}\left[\ell'(y_i f_{\boldsymbol{W}^{(t)}}(\boldsymbol{X}_{i,\mathcal{C}}))\left(y_i f_{\boldsymbol{W}^{(t)}}(\boldsymbol{X}_{i,\mathcal{C}}) - \frac{M\beta}{4}\right)\right]$$

$$- \frac{\eta^2}{n^2}\left\|\nabla_{\boldsymbol{W}}\sum_{i\in\mathcal{V}_{s^*,k^*}}\mathbb{E}_{\mathcal{C}\sim\mathcal{D}_\mathcal{C}}\left[\ell(y_i f_{\boldsymbol{W}^{(t)}}(\boldsymbol{X}_{i,\mathcal{C}}))\right]\right\|^2.$$

From the convexity of $\ell(\cdot)$,

$$\sum_{i\in\mathcal{V}_{s^*,k^*}}\mathbb{E}_{\mathcal{C}\sim\mathcal{D}_\mathcal{C}}\left[\ell'(y_i f_{\boldsymbol{W}^{(t)}}(\boldsymbol{X}_{i,\mathcal{C}}))\left(y_i f_{\boldsymbol{W}^{(t)}}(\boldsymbol{X}_{i,\mathcal{C}}) - \frac{M\beta}{4}\right)\right]$$

$$\geq \sum_{i\in\mathcal{V}_{s^*,k^*}}\mathbb{E}_{\mathcal{C}\sim\mathcal{D}_\mathcal{C}}\left[\left(\ell(y_i f_{\boldsymbol{W}^{(t)}}(\boldsymbol{X}_{i,\mathcal{C}})) - \ell\left(\frac{M\beta}{4}\right)\right)\right]$$

$$\geq \sum_{i\in\mathcal{V}_{s^*,k^*}}\mathbb{E}_{\mathcal{C}\sim\mathcal{D}_\mathcal{C}}\left[\ell(y_i f_{\boldsymbol{W}^{(t)}}(\boldsymbol{X}_{i,\mathcal{C}}))\right] - ne^{-\frac{M\beta}{4}}.$$

In addition, by Lemma F.3,

$$\frac{\eta^2}{n^2}\left\|\nabla\sum_{i\in\mathcal{V}_{s^*,k^*}}\mathbb{E}_{\mathcal{C}\sim\mathcal{D}_\mathcal{C}}\left[\ell\left(y_i f_{\boldsymbol{W}^{(t)}}(\boldsymbol{X}_{i,\mathcal{C}})\right)\right]\right\|^2$$

$$
\leq \frac{8\eta^2 P^2 \sigma_{\mathrm{d}}^2 d |\mathcal{V}_{s^*,k^*}|}{n^2} \sum_{i \in \mathcal{V}_{s^*,k^*}} \mathbb{E}_{\mathcal{C} \sim \mathcal{D}_\mathcal{C}} [\ell(y_i f_{\boldsymbol{W}^{(t)}}(\boldsymbol{X}_{i,\mathcal{C}}))]
$$

$$
\leq \frac{\eta}{n} \sum_{i \in \mathcal{V}_{s^*,k^*}} \mathbb{E}_{\mathcal{C} \sim \mathcal{D}_\mathcal{C}} [\ell(y_i f_{\boldsymbol{W}^{(t)}}(\boldsymbol{X}_{i,\mathcal{C}}))],
$$

where the last inequality is due to (A8), and we have

$$
\left\| Q\left(\boldsymbol{W}^{(t)}\right) - Q(\boldsymbol{W}^*) \right\|^2 - \left\| Q\left(\boldsymbol{W}^{(t+1)}\right) - Q(\boldsymbol{W}^*) \right\|^2
$$
$$
\geq \frac{\eta}{n} \sum_{i \in \mathcal{V}_{s^*,k^*}} \mathbb{E}_{\mathcal{C} \sim \mathcal{D}_\mathcal{C}} [\ell(y_i f_{\boldsymbol{W}^{(t)}}(\boldsymbol{X}_{i,\mathcal{C}}))] - 2\eta e^{-\frac{M\beta}{4}}.
$$

From telescoping summation, we have

$$
\frac{\eta}{n} \sum_{t=T_{\mathrm{Cutout}}}^{T^*} \sum_{i \in \mathcal{V}_{s^*,k^*}} \mathbb{E}_{\mathcal{C} \sim \mathcal{D}_\mathcal{C}} [\ell\left(y_i f_{\boldsymbol{W}^{(t)}}(\boldsymbol{X}_{i,\mathcal{C}})\right)] \leq \left\| Q\left(\boldsymbol{W}^{(T_{\mathrm{Cutout}})}\right) - Q\left(\boldsymbol{W}^*\right) \right\|^2 + 2\eta T^* e^{-\frac{M\beta}{4}}.
$$

$\square$

Finally, we can prove that the model cannot learn extremely rare features within $T^*$ iterations.

**Lemma D.9.** *Suppose the event $E_{\mathrm{init}}$ occurs. For any $T \in [T_{\mathrm{Cutout}}, T^*]$, we have $\gamma_s^{(T)}(s^*, k^*) = \widetilde{\mathcal{O}}(\alpha^2 \beta^{-2})$ for each $s \in \{\pm 1\}$.*

*Proof of Lemma D.9.* For any $T \in [T_{\mathrm{Cutout}}, T^*]$, we have

$$
\gamma_s^{(T)}(s^*, k^*) = \gamma_s^{(T_{\mathrm{Cutout}})}(s^*, k^*) + \frac{\eta}{n} \sum_{t=T_{\mathrm{Cutout}}}^{T-1} \sum_{i \in \mathcal{V}_{s^*,k^*}} \mathbb{E}_{\mathcal{C} \sim \mathcal{D}_\mathcal{C}} \left[ g_{i,\mathcal{C}}^{(t)} \cdot \mathbb{1}_{p \notin \mathcal{C}} \right] \phi'\left( \left\langle \boldsymbol{w}_s^{(t)}, \boldsymbol{v}_{s^*,k^*} \right\rangle \right)
$$

$$
\leq \gamma_s^{(T_{\mathrm{Cutout}})}(s^*, k^*) + \frac{\eta}{n} \sum_{t=T_{\mathrm{Cutout}}}^{T-1} \sum_{i \in \mathcal{V}_{s^*,k^*}} \mathbb{E}_{\mathcal{C} \sim \mathcal{D}_\mathcal{C}} \left[ g_{i,\mathcal{C}}^{(t)} \right]
$$

$$
\leq \gamma_s^{(T_{\mathrm{Cutout}})}(s^*, k^*) + \frac{\eta}{n} \sum_{t=T_{\mathrm{Cutout}}}^{T-1} \sum_{i \in \mathcal{V}_{s^*,k^*}} \mathbb{E}_{\mathcal{C} \sim \mathcal{D}_\mathcal{C}} \left[ \ell\left(y_i f_{\boldsymbol{W}^{(t)}}(\boldsymbol{X}_{i,\mathcal{C}})\right) \right],
$$

where the first inequality is due to $\phi' \leq 1$ and the second inequality is due to $-\ell' \leq \ell$. From the result of Section D.2.3, $\gamma_s^{(T_{\mathrm{Cutout}})}(s^*, k^*) \leq \alpha^2 \beta^{-1}$ and by Lemma D.8 and Lemma D.6, we have

$$
\frac{\eta}{n} \sum_{t=T_{\mathrm{Cutout}}}^{(T-1)} \sum_{i \in \mathcal{V}_{s^*,k^*}} \mathbb{E}_{\mathcal{C} \sim \mathcal{D}_\mathcal{C}} [\ell\left(y_i f_{\boldsymbol{W}^{(t)}}(\boldsymbol{X}_{i,\mathcal{C}})\right)] \leq \frac{\eta}{n} \sum_{t=T_{\mathrm{Cutout}}}^{(T^*)} \sum_{i \in \mathcal{V}_{s^*,k^*}} \mathbb{E}_{\mathcal{C} \sim \mathcal{D}_\mathcal{C}} [\ell\left(y_i f_{\boldsymbol{W}^{(t)}}(\boldsymbol{X}_{i,\mathcal{C}})\right)]
$$
$$
\leq \left\| Q\left(\boldsymbol{W}^{(T_{\mathrm{Cutout}})}\right) - Q(\boldsymbol{W}^*) \right\|^2 + 2\eta T^* e^{-\frac{M\beta}{2}}
$$
$$
\leq 8M^2 P |\mathcal{V}_{s^*,k^*}| \sigma_{\mathrm{b}}^{-2} d^{-1} + 2\eta T^* e^{-\frac{M\beta}{4}}
$$
$$
= \widetilde{\mathcal{O}}\left(\alpha^2 \beta^{-2}\right).
$$

The last line is due to (A6) and $M = 4\beta^{-1} \log\left(\frac{2\eta\beta^2 T^*}{\alpha^2}\right)$. This finishes the proof. $\square$

**What We Have So Far.** Suppose the event $E_{\mathrm{init}}$ occurs. For any $t \in [T_{\mathrm{Cutout}}, T^*]$, we have

- (Learn common/rare features): $\gamma_s^{(t)}(s, k) + \beta \gamma_{-s}^{(t)}(s, k) = \Omega(1)$ for each $s \in \{\pm 1\}$ and $k \in \mathcal{K}_C \cup \mathcal{K}_R$
- (Overfit augmented data with extremely rare features or no feature): For each $i \in [n], k \in \mathcal{K}_E, \mathcal{C} \subset [P]$ with $|\mathcal{C}| = C$ such that (1) $i \in \mathcal{V}_{y_i,k}$ and $p_i^* \notin \mathcal{C}$ or (2) $i \in [n]$ and $p_i^* \in \mathcal{C}$

$$
\sum_{p \notin \mathcal{C} \cup \{p_i^*\}} \left( \rho_{y_i}^{(t)}(i, p) + \beta \rho_{-y_i}^{(t)}(i, p) \right) = \Omega(1).
$$

- (Cannot learn extreme features): $\gamma_s^{(t)}(s,k), \gamma_{-s}^{(t)}(s,k) = \mathcal{O}\left(\alpha^2\beta^{-2}\right)$ for each $s \in \{\pm 1\}$ and $k \in \mathcal{K}_E$.

- For any $s \in \{\pm 1\}, i \in [n]$, and $p \in [P] \setminus \{p_i^*\}, \rho_s^{(t)}(i,p) = \widetilde{\mathcal{O}}\left(\beta^{-1}\right)$,

### D.2.5 Train and Test Accuracy

In this step, we will prove that the model trained by Cutout has perfect training accuracy on both augmented data and original data but has near-random guesses on test data with extremely rare data.

For any $i \in \mathcal{V}_{s,k}$ with $s \in \{\pm 1\}$, $k \in \mathcal{K}_C \cup \mathcal{K}_\mathcal{R}$ and $\mathcal{C} \subset [P]$ with $|\mathcal{C}| = C$ and $p_i^* \notin \mathcal{C}$,

$$
\begin{aligned}
&y_i f_{\boldsymbol{W}^{(t)}}(\boldsymbol{X}_{i,\mathcal{C}}) \\
&= \sum_{p \notin \mathcal{C}} \left( \phi\left(\left\langle \boldsymbol{w}_s^{(t)}, \boldsymbol{x}_i^{(p)} \right\rangle\right) - \phi\left(\left\langle \boldsymbol{w}_{-s}^{(t)}, \boldsymbol{x}_i^{(p)} \right\rangle\right) \right) \\
&= \gamma_s^{(t)}(s,k) + \beta\gamma_{-s}^{(t)}(s,k) + \sum_{p \notin \mathcal{C} \cup \{p_i^*\}} \left( \rho_s^{(t)}(i,p) + \beta\rho_{-s}^{(t)}(i,p) \right) - 2(P - C) \cdot o\left(\frac{1}{\text{polylog}(d)}\right) \\
&\geq \gamma_s^{(t)}(s,k) + \beta\gamma_{-s}^{(t)}(s,k) - 2(P - C) \cdot o\left(\frac{1}{\text{polylog}(d)}\right) \\
&= \Omega(1) - o\left(\frac{1}{\text{polylog}(d)}\right) \\
&= \Omega(1),
\end{aligned}
$$

for any $t \in [T_{\text{Cutout}}, T^*]$. In addition, for any $i \in [n]$ and $\mathcal{C} \subset [P]$ with $|\mathcal{C}| = C$ that does not correspond to the case above, by Lemma D.5 and Lemma B.4, we have

$$
\begin{aligned}
&y_i f_{\boldsymbol{W}^{(t)}}(\boldsymbol{X}_{i,\mathcal{C}}) \\
&= \sum_{p \notin \mathcal{C}} \left( \phi\left(\left\langle \boldsymbol{w}_{y_i}^{(t)}, \boldsymbol{x}_i^{(p)} \right\rangle\right) - \phi\left(\left\langle \boldsymbol{w}_{-y_i}^{(t)}, \boldsymbol{x}_i^{(p)} \right\rangle\right) \right) \\
&\geq \sum_{p \notin \mathcal{C} \cup \{p_i^*\}} \left( \rho_{y_i}^{(t)}(i,p) + \beta\rho_{-y_i}^{(t)}(i,p) \right) - 2(P - C) \cdot o\left(\frac{1}{\text{polylog}(d)}\right) \\
&= \Omega(1) - o\left(\frac{1}{\text{polylog}(d)}\right) \\
&= \Omega(1),
\end{aligned}
$$

for any $t \in [T_{\text{Cutout}}, T^*]$. We can conclude that Cutout with $t \in [T_{\text{Cutout}}, T^*]$ iterates achieve perfect training accuracy on augmented data.

Next, we will show that Cutout achieves perfect training accuracy on the original data. For any $i \in [n]$, let us choose $\mathcal{C} \subset [P]$ with $|\mathcal{C}| = C$ such that $p_i^* \in \mathcal{C}$. Then, from the result above, we have

$$
\begin{aligned}
y_i f_{\boldsymbol{W}^{(t)}}(\boldsymbol{X}_i) &= y_i f_{\boldsymbol{W}^{(t)}}(\boldsymbol{X}_{i,\mathcal{C}}) + \sum_{p \in \mathcal{C}} \left( \phi\left(\left\langle \boldsymbol{w}_{y_i}^{(t)}, \boldsymbol{x}_i^{(p)} \right\rangle\right) - \phi\left(\left\langle \boldsymbol{w}_{-y_i}^{(t)}, \boldsymbol{x}_i^{(p)} \right\rangle\right) \right) \\
&\geq y_i f_{\boldsymbol{W}^{(t)}}(\boldsymbol{X}_{i,\mathcal{C}}) + \sum_{p \in \mathcal{C} \setminus \{p_i^*\}} \left( \rho_{y_i}^{(t)}(i,p) + \beta\rho_{-y_i}^{(t)}(i,p) \right) - C \cdot o\left(\frac{1}{\text{polylog}(d)}\right) \\
&\geq \Omega(1),
\end{aligned}
$$

for any $t \in [T_{\text{Cutout}}, T^*]$ and we conclude that Cutout with $t \in [T_{\text{Cutout}}, T^*]$ iterates achieve perfect training accuracy on original data.

Lastly, let us move on to the test accuracy part. Let $(\boldsymbol{X}, y) \sim \mathcal{D}$ be a test data with $\boldsymbol{X} = \left(\boldsymbol{x}^{(1)}, \ldots, \boldsymbol{x}^{(P)}\right) \in \mathbb{R}^{d \times P}$ having feature patch $p^*$, dominant noise patch $\tilde{p}$, and feature vector $\boldsymbol{v}_{y,k}$. We have $\boldsymbol{x}^{(p)} \sim N(\boldsymbol{0}, \sigma_b^2 \boldsymbol{\Lambda})$ for each $p \in [P] \setminus \{p^*, \tilde{p}\}$ and $\boldsymbol{x}^{(\tilde{p})} - \alpha\boldsymbol{v}_{s,1} \sim N(\boldsymbol{0}, \sigma_d^2 \boldsymbol{\Lambda})$ for some $s \in \{\pm 1\}$. Therefore, for all $t \in [T_{\text{Cutout}}, T^*]$ and $p \in [P] \setminus \{p^*, \tilde{p}\}$,

$$
\left| \phi\left(\left\langle \boldsymbol{w}_1^{(t)}, \boldsymbol{x}^{(p)} \right\rangle\right) - \phi\left(\left\langle \boldsymbol{w}_{-1}^{(t)}, \boldsymbol{x}^{(p)} \right\rangle\right) \right|
$$

$$\leq \left| \left\langle \boldsymbol{w}_1^{(t)} - \boldsymbol{w}_{-1}^{(t)}, \boldsymbol{x}^{(p)} \right\rangle \right|$$

$$\leq \left| \left\langle \boldsymbol{w}_1^{(0)} - \boldsymbol{w}_{-1}^{(0)}, \boldsymbol{x}^{(p)} \right\rangle \right| + \sum_{i \in [n], q \in [P] \setminus \{p_i^*\}} \left| \rho_1^{(t)}(i,q) - \rho_{-1}^{(t)}(i,q) \right| \frac{\left| \left\langle \xi_i^{(q)}, \boldsymbol{x}^{(p)} \right\rangle \right|}{\left\| \xi_i^{(q)} \right\|^2}$$

$$\leq \widetilde{\mathcal{O}} \left( \sigma_0 \sigma_{\mathrm{b}} d^{\frac{1}{2}} \right) + \widetilde{\mathcal{O}} \left( nP \beta^{-1} \sigma_{\mathrm{d}} \sigma_{\mathrm{b}}^{-1} d^{-\frac{1}{2}} \right)$$

$$= o \left( \frac{\alpha}{\mathrm{polylog}(d)} \right), \tag{28}$$

with probability at least $1 - o \left( \frac{1}{\mathrm{poly}(d)} \right)$ due to Lemma B.2, (A8), (8), and (9).. In addition, for any $s' \in \{\pm 1\}$, we have

$$\left| \left\langle \boldsymbol{w}_{s'}^{(t)}, \boldsymbol{x}^{(\tilde{p})} - \alpha \boldsymbol{v}_{s,1} \right\rangle \right|$$

$$\leq \left| \left\langle \boldsymbol{w}_{s'}^{(0)}, \boldsymbol{x}^{(\tilde{p})} - \alpha \boldsymbol{v}_{s,1} \right\rangle \right| + \sum_{i \in [n], q \in [P] \setminus \{p_i^*\}} \rho_{s'}^{(t)}(i,q) \frac{\left| \left\langle \xi_i^{(q)}, \boldsymbol{x}^{(\tilde{p})} - \alpha \boldsymbol{v}_{s,1} \right\rangle \right|}{\left\| \xi_i^{(q)} \right\|^2}$$

$$= \widetilde{\mathcal{O}} \left( \sigma_0 \sigma_{\mathrm{d}} d^{\frac{1}{2}} \right) + \widetilde{\mathcal{O}} \left( nP \beta^{-1} \sigma_{\mathrm{d}} \sigma_{\mathrm{b}}^{-1} d^{-\frac{1}{2}} \right)$$

$$= o \left( \frac{\alpha}{\mathrm{polylog}(d)} \right), \tag{29}$$

with probability at least $1 - o \left( \frac{1}{\mathrm{poly}(d)} \right)$ due to Lemma B.2, (A8), (8), and (9).

**Case 1:** $k \in \mathcal{K}_C \cup \mathcal{K}_R$

By Lemma B.2, (A7), and (10),

$$\left| \phi \left( \left\langle \boldsymbol{w}_1^{(t)}, \boldsymbol{x}^{(\tilde{p})} \right\rangle \right) - \phi \left( \left\langle \boldsymbol{w}_{-1}^{(t)}, \boldsymbol{x}^{(\tilde{p})} \right\rangle \right) \right|$$

$$\leq \left| \left\langle \boldsymbol{w}_1^{(t)} - \boldsymbol{w}_{-1}^{(t)}, \boldsymbol{x}^{(\tilde{p})} \right\rangle \right|$$

$$\leq \alpha \left| \left\langle \boldsymbol{w}_1^{(t)} - \boldsymbol{w}_{-1}^{(t)}, \boldsymbol{v}_{s,1} \right\rangle \right| + \left| \left\langle \boldsymbol{w}_1^{(t)} - \boldsymbol{w}_{-1}^{(t)}, \boldsymbol{x}^{(p)} - \alpha \boldsymbol{v}_{s,1} \right\rangle \right|$$

$$\leq \alpha \left( \gamma_1^{(t)}(s,1) + \gamma_{-1}^{(t)}(s,1) \right) + \alpha \left| \left\langle \boldsymbol{w}_1^{(0)}, \boldsymbol{v}_{s,1} \right\rangle \right| + \alpha \left| \left\langle \boldsymbol{w}_{-1}^{(0)}, \boldsymbol{v}_{s,1} \right\rangle \right| + o \left( \frac{1}{\mathrm{polylog}(d)} \right)$$

$$\leq \widetilde{\mathcal{O}} \left( \alpha \beta^{-1} \right) + \widetilde{\mathcal{O}} \left( \alpha \sigma_0 \right) + o \left( \frac{1}{\mathrm{polylog}(d)} \right)$$

$$= o \left( \frac{1}{\mathrm{polylog}(d)} \right), \tag{30}$$

with probability at least $1 - o \left( \frac{1}{\mathrm{poly}(d)} \right)$. Suppose (28) and (30) holds. By Lemma B.4, we have

$$y f_{\boldsymbol{W}^{(t)}}(\boldsymbol{X})$$

$$= \left( \phi \left( \left\langle \boldsymbol{w}_y^{(t)}, \boldsymbol{v}_{y,k} \right\rangle \right) - \phi \left( \left\langle \boldsymbol{w}_{-y}^{(t)}, \boldsymbol{v}_{y,k} \right\rangle \right) \right)$$

$$\quad + \sum_{p \in [P] \setminus \{p^*\}} \left( \phi \left( \left\langle \boldsymbol{w}_y^{(t)}, \boldsymbol{x}^{(p)} \right\rangle \right) - \phi \left( \left\langle \boldsymbol{w}_{-y}^{(t)}, \boldsymbol{x}^{(p)} \right\rangle \right) \right)$$

$$= \gamma_y^{(t)}(y,k) + \beta \gamma_{-y}^{(t)}(y,k) - o \left( \frac{1}{\mathrm{polylog}(d)} \right)$$

$$= \Omega(1) - o \left( \frac{1}{\mathrm{polylog}(d)} \right)$$

$$> 0.$$

Therefore, we have

$$\mathbb{P}_{(\boldsymbol{X},y)\sim\mathcal{D}}\left[yf_{\boldsymbol{W}^{(t)}}(\boldsymbol{X}) > 0 \mid \boldsymbol{x}^{(p^*)} = \boldsymbol{v}_{y,k}, k \in \mathcal{K}_C \cup \mathcal{K}_R\right] \geq 1 - o\left(\frac{1}{\text{poly}(d)}\right). \qquad (31)$$

**Case 2:** $k \in \mathcal{K}_E$

By triangular inequality and $\phi' \leq 1$, we have

$$\begin{aligned}
&\phi\left(\left\langle\boldsymbol{w}_s^{(t)}, \boldsymbol{x}^{(\tilde{p})}\right\rangle\right) - \phi\left(\left\langle\boldsymbol{w}_{-s}^{(t)}, \boldsymbol{x}^{(\tilde{p})}\right\rangle\right) \\
&= \phi\left(\left\langle\boldsymbol{w}_s^{(t)}, \alpha\boldsymbol{v}_{s,1}\right\rangle\right) - \phi\left(\left\langle\boldsymbol{w}_{-s}^{(t)}, \alpha\boldsymbol{v}_{s,1}\right\rangle\right) \\
&\quad + \left(\phi\left(\left\langle\boldsymbol{w}_s^{(t)}, \boldsymbol{x}^{(\tilde{p})}\right\rangle\right) - \phi\left(\left\langle\boldsymbol{w}_s^{(t)}, \alpha\boldsymbol{v}_{s,1}\right\rangle\right)\right) - \left(\phi\left(\left\langle\boldsymbol{w}_{-s}^{(t)}, \boldsymbol{x}^{(\tilde{p})}\right\rangle\right) - \phi\left(\left\langle\boldsymbol{w}_{-s}^{(t)}, \alpha\boldsymbol{v}_{s,1}\right\rangle\right)\right) \\
&\geq \phi\left(\left\langle\boldsymbol{w}_s^{(t)}, \alpha\boldsymbol{v}_{s,1}\right\rangle\right) - \phi\left(\left\langle\boldsymbol{w}_{-s}^{(t)}, \alpha\boldsymbol{v}_{s,1}\right\rangle\right) \\
&\quad - \left|\left\langle\boldsymbol{w}_s^{(t)}, \boldsymbol{x}^{(\tilde{p})} - \alpha\boldsymbol{v}_{s,1}\right\rangle\right| - \left|\left\langle\boldsymbol{w}_{-s}^{(t)}, \boldsymbol{x}^{(\tilde{p})} - \alpha\boldsymbol{v}_{s,1}\right\rangle\right|.
\end{aligned}$$

In addition,

$$\begin{aligned}
&\phi\left(\left\langle\boldsymbol{w}_s^{(t)}, \alpha\boldsymbol{v}_{s,1}\right\rangle\right) - \phi\left(\left\langle\boldsymbol{w}_{-s}^{(t)}, \alpha\boldsymbol{v}_{s,1}\right\rangle\right) \\
&= \left(\phi\left(\alpha\gamma_s^{(t)}(s,1)\right) - \phi\left(-\alpha\gamma_{-s}^{(t)}(s,1)\right)\right) \\
&\quad + \left(\phi\left(\left\langle\boldsymbol{w}_s^{(t)}, \alpha\boldsymbol{v}_{s,1}\right\rangle\right) - \phi\left(\alpha\gamma_s^{(t)}(s,1)\right)\right) \\
&\quad - \left(\phi\left(\left\langle\boldsymbol{w}_{-s}^{(t)}, \alpha\boldsymbol{v}_{s,1}\right\rangle\right) - \phi\left(-\alpha\gamma_{-s}^{(t)}(s,1)\right)\right) \\
&\geq \left(\phi\left(\alpha\gamma_s^{(t)}(s,1)\right) - \phi\left(-\alpha\gamma_{-s}^{(t)}(s,1)\right)\right) \\
&\quad - \alpha\left|\left\langle\boldsymbol{w}_s^{(t)}, \boldsymbol{v}_{s,1}\right\rangle - \gamma_s^{(t)}(s,1)\right| - \alpha\left|\left\langle\boldsymbol{w}_{-s}^{(t)}, \boldsymbol{v}_{s,1}\right\rangle + \gamma_{-s}^{(t)}(s,1)\right| \\
&= \alpha\left(\gamma_s^{(t)}(s,1) + \beta\gamma_{-s}^{(t)}(s,1)\right) - \alpha \cdot o\left(\frac{1}{\text{polylog}(d)}\right) \\
&= \Omega(\alpha),
\end{aligned}$$

where the second equality is due to Lemma B.4 and (A8). If (29) holds, we have

$$\phi\left(\left\langle\boldsymbol{w}_s^{(t)}, \boldsymbol{x}^{(\tilde{p})}\right\rangle\right) - \phi\left(\left\langle\boldsymbol{w}_{-s}^{(t)}, \boldsymbol{x}^{(\tilde{p})}\right\rangle\right) = \Omega(\alpha) - o\left(\frac{\alpha}{\text{polylog}(d)}\right) = \Omega(\alpha). \qquad (32)$$

Note that

$$\begin{aligned}
&yf_{\boldsymbol{W}^{(t)}}(\boldsymbol{X}) \\
&= \phi\left(\left\langle\boldsymbol{w}_y^{(t)}, \boldsymbol{v}_{y,k}\right\rangle\right) - \phi\left(\left\langle\boldsymbol{w}_{-y}^{(t)}, \boldsymbol{v}_{y,k}\right\rangle\right) + \phi\left(\left\langle\boldsymbol{w}_y^{(t)}, \boldsymbol{x}^{(\tilde{p})}\right\rangle\right) - \phi\left(\left\langle\boldsymbol{w}_{-y}^{(t)}, \boldsymbol{x}^{(\tilde{p})}\right\rangle\right) \\
&\quad + \sum_{p\in[P]\setminus\{p^*,\tilde{p}\}}\left(\phi\left(\left\langle\boldsymbol{w}_y^{(t)}, \boldsymbol{x}^{(p)}\right\rangle\right) - \phi\left(\left\langle\boldsymbol{w}_{-y}^{(t)}, \boldsymbol{x}^{(p)}\right\rangle\right)\right),
\end{aligned}$$

and

$$\begin{aligned}
&\left|\phi\left(\left\langle\boldsymbol{w}_y^{(t)}, \boldsymbol{v}_{y,k}\right\rangle\right) - \phi\left(\left\langle\boldsymbol{w}_{-y}^{(t)}, \boldsymbol{v}_{y,k}\right\rangle\right)\right| \\
&\quad + \left|\sum_{p\in[P]\setminus\{p^*,\tilde{p}\}}\left(\phi\left(\left\langle\boldsymbol{w}_y^{(t)}, \boldsymbol{x}^{(p)}\right\rangle\right) - \phi\left(\left\langle\boldsymbol{w}_{-y}^{(t)}, \boldsymbol{x}^{(p)}\right\rangle\right)\right)\right| \\
&\leq \left|\left\langle\boldsymbol{w}_y^{(t)} - \boldsymbol{w}_{-y}^{(t)}, \boldsymbol{v}_{y,k}\right\rangle\right| + o\left(\frac{\alpha}{\text{polylog}(d)}\right)
\end{aligned}$$

$$\leq \gamma_1^{(t)}(y,k) + \gamma_{-1}^{(t)}(y,k) + \left| \left\langle \boldsymbol{w}_y^{(0)} - \boldsymbol{w}_{-y}^{(0)}, \boldsymbol{v}_{y,k} \right\rangle \right| + o\left( \frac{\alpha}{\text{polylog}(d)} \right)$$

$$\leq \mathcal{O}(\alpha^2 \beta^{-2}) + \widetilde{\mathcal{O}}(\sigma_0) + o\left( \frac{\alpha}{\text{polylog}(d)} \right)$$

$$= o\left( \frac{\alpha}{\text{polylog}(d)} \right)$$

$$< \phi\left( \left\langle \boldsymbol{w}_s^{(t)}, \boldsymbol{x}^{(\tilde{p})} \right\rangle \right) - \phi\left( \left\langle \boldsymbol{w}_{-s}^{(t)}, \boldsymbol{x}^{(\tilde{p})} \right\rangle \right),$$

where the first inequality is due to (28), the second-to-last line is due to (A8), (8), and (10) , and the last inequality is due to (32). Therefore, we have $y f_{\boldsymbol{W}^{(t)}}(\boldsymbol{X}) > 0$ if $y = s$. Otherwise, $y f_{\boldsymbol{W}^{(t)}}(\boldsymbol{X}) < 0$.

$$\mathbb{P}_{(\boldsymbol{X},y) \sim \mathcal{D}} \left[ y f_{\boldsymbol{W}^{(t)}}(\boldsymbol{X}) > 0 \mid \boldsymbol{x}^{(p^*)} = \boldsymbol{v}_{y,k}, k \in \mathcal{K}_E \right] = \frac{1}{2} \pm o\left( \frac{1}{\text{poly}(d)} \right). \tag{33}$$

Hence, combining (31) and (33) implies

$$\mathbb{P}_{(\boldsymbol{X},y) \sim \mathcal{D}} \left[ y f_{\boldsymbol{W}^{(t)}}(\boldsymbol{X}) > 0 \right] = \sum_{k \in \mathcal{K}_C \cup \mathcal{K}_R} \rho_k + \frac{1}{2} \left( 1 - \sum_{k \in \mathcal{K}_C \cup \mathcal{K}_R} \rho_k \right) \pm o\left( \frac{1}{\text{poly}(d)} \right)$$

$$= 1 - \frac{1}{2} \sum_{k \in \mathcal{K}_E} \rho_k \pm o\left( \frac{1}{\text{poly}(d)} \right).$$

$\square$

# E  Proof for CutMix

## E.1  Proof of Lemma B.3 for CutMix

For each $i, j \in [n]$ and $\mathcal{S} \subset [P]$, let

$$g_{i,j,\mathcal{S}}^{(t)} := -\frac{|\mathcal{S}|}{P} y_i \ell'\big(y_i f_{\boldsymbol{W}^{(t)}}(\boldsymbol{X}_{i,j,\mathcal{S}})\big) - \left(1 - \frac{|\mathcal{S}|}{P}\right) y_j \ell'\big(y_j f_{\boldsymbol{W}^{(t)}}(\boldsymbol{X}_{i,j,\mathcal{S}})\big).$$

For $s \in \{\pm 1\}$ and iterate $t$,

$$\boldsymbol{w}_s^{(t+1)} - \boldsymbol{w}_s^{(t)}$$

$$= -\eta \nabla_{\boldsymbol{w}_s} \mathcal{L}_{\text{CutMix}}\big(\boldsymbol{W}^{(t)}\big)$$

$$= \frac{\eta}{n^2} \sum_{i,j \in [n]} \mathbb{E}_{\mathcal{S} \sim \mathcal{D}_{\mathcal{S}}} \left[ s g_{i,j,\mathcal{S}}^{(t)} \left( \sum_{p \in \mathcal{S}} \phi'\left(\left\langle \boldsymbol{w}_s^{(t)}, \boldsymbol{x}_i^{(p)} \right\rangle\right) \boldsymbol{x}_i^{(p)} + \sum_{p \notin \mathcal{S}} \phi'\left(\left\langle \boldsymbol{w}_s^{(t)}, \boldsymbol{x}_i^{(p)} \right\rangle\right) \boldsymbol{x}_j^{(p)} \right) \right]$$

$$= \frac{s\eta}{n^2} \sum_{s' \in \{\pm 1\}, k \in [K]} \sum_{i \in \mathcal{V}_{s',k}, j \in [n]} \mathbb{E}_{\mathcal{S} \sim \mathcal{D}_{\mathcal{S}}} \left[ g_{i,j,\mathcal{S}}^{(t)} \mathbb{1}_{p_i^* \in \mathcal{S}} + g_{j,i,\mathcal{S}}^{(t)} \mathbb{1}_{p_i^* \notin \mathcal{S}} \right] \phi'\left(\left\langle \boldsymbol{w}_s^{(t)}, \boldsymbol{v}_{s',k} \right\rangle\right) \boldsymbol{v}_{s',k}$$

$$+ \frac{s\eta}{n^2} \sum_{i,j \in [n], p \in [P] \setminus \{p_i^*\}} \mathbb{E}_{\mathcal{S} \sim \mathcal{D}_{\mathcal{S}}} \left[ g_{i,j,\mathcal{S}}^{(t)} \mathbb{1}_{p \in \mathcal{S}} + g_{j,i,\mathcal{S}}^{(t)} \mathbb{1}_{p \notin \mathcal{S}} \right] \phi'\left(\left\langle \boldsymbol{w}_s^{(t)}, \boldsymbol{x}_i^{(p)} \right\rangle\right) \boldsymbol{x}_i^{(p)}.$$

Hence, if we define $\gamma_s^{(t)}(s', k)$'s and $\rho_s^{(t)}(i, p)$'s recursively by using the rule

$$\gamma_s^{(t+1)}(s', k) = \gamma_s^{(t)}(s', k) + \frac{ss'\eta}{n^2} \sum_{i \in \mathcal{V}_{s',k}, j \in [n]} \mathbb{E}_{\mathcal{S} \sim \mathcal{D}_{\mathcal{S}}} \left[ g_{i,j,\mathcal{S}}^{(t)} \mathbb{1}_{p_i^* \in \mathcal{S}} + g_{j,i,\mathcal{S}}^{(t)} \mathbb{1}_{p_i^* \notin \mathcal{S}} \right] \phi'\left(\left\langle \boldsymbol{w}_s^{(t)}, \boldsymbol{v}_{s',k} \right\rangle\right),$$

$$\rho_s^{(t+1)}(i, p) = \rho_s^{(t)}(i, p) + \frac{s y_i \eta}{n^2} \sum_{j \in [n]} \mathbb{E}_{\mathcal{S} \sim \mathcal{D}_{\mathcal{S}}} \left[ g_{i,j,\mathcal{S}}^{(t)} \mathbb{1}_{p \in \mathcal{S}} + g_{j,i,\mathcal{S}}^{(t)} \mathbb{1}_{p \notin \mathcal{S}} \right] \phi'\left(\left\langle \boldsymbol{w}_s^{(t)}, \boldsymbol{x}_i^{(p)} \right\rangle\right) \left\| \xi_i^{(p)} \right\|^2,$$

starting from $\gamma_s^{(0)}(s'k) = \rho_s^{(0)}(i, p) = 0$ for each $s, s' \in \{\pm 1\}, k \in [K], i \in [n]$ and $p \in [P] \setminus \{p_i^*\}$, then we have

$$\boldsymbol{w}_s^{(t)} = \boldsymbol{w}_s^{(0)} + \sum_{k \in [K]} \gamma_s^{(t)}(s, k) \boldsymbol{v}_{s,k} - \sum_{k \in [K]} \gamma_s^{(t)}(-s, k) \boldsymbol{v}_{-s,k}$$

$$+ \sum_{\substack{i \in \mathcal{V}_s \\ p \in [P] \setminus \{\tilde{p}_i\}}} \rho_s^{(t)}(i, p) \frac{\xi_i^{(p)}}{\left\| \xi_i^{(p)} \right\|^2} - \sum_{\substack{i \in \mathcal{V}_{-s} \\ p \in [P] \setminus \{\tilde{p}_i\}}} \rho_s^{(t)}(i, p) \frac{\xi_i^{(p)}}{\left\| \xi_i^{(p)} \right\|^2}$$

$$+ \alpha \left( \sum_{i \in \mathcal{F}_s} s y_i \rho_s^{(t)}(i, \tilde{p}_i) \frac{\boldsymbol{v}_{s,1}}{\left\| \xi_i^{(\tilde{p}_i)} \right\|^2} + \sum_{i \in \mathcal{F}_{-s}} s y_i \rho_s^{(t)}(i, \tilde{p}_i) \frac{\boldsymbol{v}_{-s,1}}{\left\| \xi_i^{(\tilde{p}_i)} \right\|^2} \right),$$

for each $s \in \{\pm 1\}$. $\qquad\qquad\square$

## E.2  Proof of Theorem 3.3

We will prove that the conclusion of Theorem 3.3 holds when the event $E_{\text{init}}$ occurs. The proof of Theorem 3.3 is structured into the following six steps:

1. Introduce a reparametrization of the CutMix loss $\mathcal{L}_{\text{CutMix}}(\boldsymbol{W})$ to a convex function $h(\boldsymbol{Z})$ for ease of analysis (Section E.2.1).

2. Characterize a global minimum of $h(\boldsymbol{Z})$ (Section E.2.2).

3. Evaluate strong convexity constant in the region near the global minimum of $h(\boldsymbol{Z})$ (Section E.2.3).

4. Show that near stationary point of $h(\boldsymbol{Z})$ is close to a global minimum (Section E.2.4).

5. Prove that gradient descent on the CutMix loss $\mathcal{L}_{\text{CutMix}}(\boldsymbol{W})$ achieves a near-stationary point of the reparametrized function $h(\boldsymbol{Z})$ and perfect accuracy on original training data (Section E.2.5).

6. Evaluate the test accuracy of a model in near-stationary point (Section E.2.6).

### E.2.1 Reparametrization of CutMix Loss

It is complicated to characterize the stationary points of CutMix loss $\mathcal{L}_{\mathrm{CutMix}}(\boldsymbol{W})$ due to its non-convexity. We will overcome this problem by introducing reparameterization of the objective function. Let us define

$$z_i^{(p)} := \phi\left(\left\langle \boldsymbol{w}_1, \boldsymbol{x}_i^{(p)} \right\rangle\right) - \phi\left(\left\langle \boldsymbol{w}_{-1}, \boldsymbol{x}_i^{(p)} \right\rangle\right),$$

for $i \in [n], p \in [P]$ and

$$z_{s,k} := \phi(\langle \boldsymbol{w}_1, \boldsymbol{v}_{s,k} \rangle) - \phi(\langle \boldsymbol{w}_{-1}, \boldsymbol{v}_{s,k} \rangle),$$

for each $s \in \{\pm 1\}, k \in [K]$. We can rewrite CutMix loss $\mathcal{L}_{\mathrm{CutMix}}(\boldsymbol{W})$ as a function $h(\boldsymbol{Z})$ of the defined variables $\boldsymbol{Z} := \{z_{s,k}\}_{s \in \{\pm 1\}, k \in [K]} \cup \{z_i^{(p)}\}_{i \in [n], p \in [P] \setminus \{p_i^*\}}$ as follows.

$$
h(\boldsymbol{Z}) := \frac{1}{n^2} \sum_{i,j \in [n]} \mathbb{E}_{\mathcal{S} \sim \mathcal{D}_\mathcal{S}} \left[ \frac{|\mathcal{S}|}{P} \ell\left( y_i \left( \sum_{p \in \mathcal{S}} z_i^{(p)} + \sum_{p \notin \mathcal{S}} z_j^{(p)} \right) \right) \right.
$$
$$
\left. + \left(1 - \frac{|\mathcal{S}|}{P}\right) \ell\left( y_j \left( \sum_{p \in \mathcal{S}} z_i^{(p)} + \sum_{p \notin \mathcal{S}} z_j^{(p)} \right) \right) \right],
$$

where we write $z_i^{(p_i^*)} = z_{s,k}$ if $i \in \mathcal{V}_{s,k}$. For notational simplicity, let us consider $\boldsymbol{Z}$ as vectors in $\mathbb{R}^{2K + n(P-1)}$ with the standard orthonormal basis $\{\boldsymbol{e}_{s,k}\}_{s \in \{\pm 1\}, k \in [K]} \cup \left\{\boldsymbol{e}_i^{(p)}\right\}_{i \in [n], p \in [P] \setminus \{p_i^*\}}$ which means

$$
\boldsymbol{Z} = \{z_{s,k}\}_{s \in \{\pm 1\}, k \in [K]} \cup \left\{z_i^{(p)}\right\}_{i \in [n], p \in [P] \setminus \{p_i^*\}}
$$
$$
= \sum_{s \in \{\pm 1\}, k \in [K]} z_{s,k} \boldsymbol{e}_{s,k} + \sum_{i \in [n], p \in [P] \setminus \{p_i^*\}} z_i^{(p)} \boldsymbol{e}_i^{(p)}.
$$

If there is no confusion, we will use $\boldsymbol{e}_i^{(p_i^*)}$ to represent $\boldsymbol{e}_{s,k}$, for $i \in \mathcal{V}_{s,k}$.

By the chain rule,

$$\nabla_{\boldsymbol{W}} \mathcal{L}_{\mathrm{CutMix}}(\boldsymbol{W}) = \boldsymbol{J}(\boldsymbol{W}) \nabla_{\boldsymbol{Z}} h(\boldsymbol{Z}),$$

where each column of Jacobian matrix $\boldsymbol{J}(\boldsymbol{W}) \in \mathbb{R}^{2d \times (n(P-1) + 2K)}$ is

$$
\nabla_{\boldsymbol{W}} z_{s,k} = \begin{pmatrix} \phi'(\langle \boldsymbol{w}_1, \boldsymbol{v}_{s,k} \rangle) \boldsymbol{v}_{s,k} \\ -\phi'(\langle \boldsymbol{w}_{-1}, \boldsymbol{v}_{s,k} \rangle) \boldsymbol{v}_{s,k} \end{pmatrix} \in \mathbb{R}^{2d}, \nabla_{\boldsymbol{W}} z_i^{(p)} = \begin{pmatrix} \phi'\left(\left\langle \boldsymbol{w}_1, \boldsymbol{x}_i^{(p)} \right\rangle\right) \boldsymbol{x}_i^{(p)} \\ -\phi'\left(\left\langle \boldsymbol{w}_{-1}, \boldsymbol{x}_i^{(p)} \right\rangle\right) \boldsymbol{x}_i^{(p)} \end{pmatrix} \in \mathbb{R}^{2d}.
$$

Let us characterize the smallest singular value $\sigma_{\min}(\boldsymbol{J}(\boldsymbol{W}))$ of the Jacobian matrix $\boldsymbol{J}(\boldsymbol{W})$. For any unit vector $\boldsymbol{c} = \{c_{s,k}\}_{s \in \{\pm 1\}, k \in [K]} \cup \left\{c_i^{(p)}\right\}_{i \in [n], p \in [P] \setminus \{p_i^*\}} \in \mathbb{R}^{2K + n(P-1)}$, we have

$$
\|\boldsymbol{J}(\boldsymbol{W})\boldsymbol{c}\|^2 = \sum_{s \in \{\pm 1\}, k \in [K]} c_{s,k}^2 \|\nabla_{\boldsymbol{W}} z_{s,k}\|^2 + \sum_{i \in [n], p \in [P] \setminus \{p_i^*\}} \left(c_i^{(p)}\right)^2 \left\|\nabla_{\boldsymbol{W}} z_i^{(p)}\right\|^2
$$
$$
+ \sum_{\substack{s_1, s_2 \in \{\pm 1\}, k_1, k_2 \in [K] \\ (s_1, k_1) \neq (s_2, k_2)}} c_{s_1, k_1} c_{s_2, k_2} \langle \nabla_{\boldsymbol{W}} z_{s_1, k_1}, \nabla_{\boldsymbol{W}} z_{s_2, k_2} \rangle
$$
$$
+ 2 \sum_{\substack{s \in \{\pm 1\}, k \in [K] \\ i \in [n], p \in [P] \setminus \{p_i^*\}}} c_{s,k} c_i^{(p)} \left\langle \nabla_{\boldsymbol{W}} z_{s,k}, \nabla_{\boldsymbol{W}} z_i^{(p)} \right\rangle
$$
$$
+ \sum_{\substack{i \in [n], p \in [P] \setminus \{p_i^*\} \\ j \in [n], q \in [P] \setminus \{p_j^*\} \\ (i,p) \neq (j,q)}} c_i^{(p)} c_j^{(q)} \left\langle \nabla_{\boldsymbol{W}} z_i^{(p)}, \nabla_{\boldsymbol{W}} z_j^{(q)} \right\rangle.
$$

For each $s_1, s_2 \in \{\pm 1\}$, $k_1, k_2 \in [K]$ such that $(s_1, k_1) \neq (s_2, k_2)$, and $i \in [n], p \in [P] \setminus \{p_i^*, \tilde{p}_i\}$,

$$\langle \nabla_{\boldsymbol{W}} z_{s_1,k_1}, \nabla_{\boldsymbol{W}} z_{s_2,k_2} \rangle = \left\langle \nabla_{\boldsymbol{W}} z_{s_1,k_1}, \nabla_{\boldsymbol{W}} z_i^{(p)} \right\rangle = 0,$$

and if $k_1 > 1$

$$\left\langle \nabla_{\boldsymbol{W}} z_{s_1,k_1}, \nabla_{\boldsymbol{W}} z_i^{(p)} \right\rangle = \left\langle \nabla_{\boldsymbol{W}} z_{s_1,k_1}, \nabla_{\boldsymbol{W}} z_i^{(\tilde{p}_i)} \right\rangle = 0,$$

since $\langle \boldsymbol{v}_{s_1,k_1}, \boldsymbol{v}_{s_2,k_2} \rangle = \left\langle \boldsymbol{v}_{s_1,k_1}, \xi_i^{(p)} \right\rangle = \left\langle \boldsymbol{v}_{s_1,k_1}, \xi_i^{(\tilde{p}_i)} \right\rangle = 0$. Also, for each $s \in \{\pm 1\}$ and $i \in \mathcal{F}_s$, then

$$2 \left| c_{s,1} c_i^{(\tilde{p}_i)} \left\langle \nabla_{\boldsymbol{W}} z_{s,1}, \nabla_{\boldsymbol{W}} z_i^{(\tilde{p}_i)} \right\rangle \right|$$

$$= 2 \left| c_{s,1} c_i^{(\tilde{p}_i)} \right| \left( \phi'(\langle \boldsymbol{w}_1, \boldsymbol{v}_{s,1} \rangle) \phi'\left( \left\langle \boldsymbol{w}_1, \boldsymbol{x}_i^{(\tilde{p}_i)} \right\rangle \right) + \phi'(\langle \boldsymbol{w}_{-1}, \boldsymbol{v}_{s,1} \rangle) \phi'\left( \left\langle \boldsymbol{w}_{-1}, \boldsymbol{x}_i^{(\tilde{p}_i)} \right\rangle \right) \right) \alpha$$

$$\leq 4 c_{s,1}^2 \left( \phi'(\langle \boldsymbol{w}_1, \boldsymbol{v}_{s,1} \rangle)^2 + \phi'(\langle \boldsymbol{w}_{-1}, \boldsymbol{v}_{s,1} \rangle)^2 \right) \frac{\alpha^2}{\left\| \boldsymbol{x}_i^{(\tilde{p}_i)} \right\|^2}$$

$$+ \frac{1}{4} \left( c_i^{(\tilde{p}_i)} \right)^2 \left( \phi'\left( \left\langle \boldsymbol{w}_1, \boldsymbol{x}_i^{(\tilde{p}_i)} \right\rangle \right)^2 + \phi'\left( \left\langle \boldsymbol{w}_{-1}, \boldsymbol{x}_i^{(\tilde{p}_i)} \right\rangle \right)^2 \right) \left\| \boldsymbol{x}_i^{(\tilde{p}_i)} \right\|^2$$

$$< \frac{1}{2n} c_{s,1}^2 \left( \phi'(\langle \boldsymbol{w}_1, \boldsymbol{v}_{s,1} \rangle)^2 + \phi'(\langle \boldsymbol{w}_{-1}, \boldsymbol{v}_{s,1} \rangle)^2 \right)$$

$$+ \frac{1}{4} \left( c_i^{(\tilde{p}_i)} \right)^2 \left( \phi'\left( \left\langle \boldsymbol{w}_1, \boldsymbol{x}_i^{(\tilde{p}_i)} \right\rangle \right)^2 + \phi'\left( \left\langle \boldsymbol{w}_{-1}, \boldsymbol{x}_i^{(\tilde{p}_i)} \right\rangle \right)^2 \right) \left\| \boldsymbol{x}_i^{(\tilde{p}_i)} \right\|^2,$$

where the last inequality holds since

$$\left\| \boldsymbol{x}_i^{(\tilde{p}_i)} \right\|^2 = \alpha^2 + \left\| \xi_i^{(\tilde{p}_i)} \right\|^2 \geq \frac{1}{2} \sigma_{\mathrm{d}}^2 d = \omega(n\alpha^2),$$

where we apply the fact from the event $E_{\mathrm{init}}$ defined in Lemma B.2 and (A7). Also,

$$\left\langle \nabla_{\boldsymbol{W}} z_{-s,1}, \nabla_{\boldsymbol{W}} z_i^{(\tilde{p}_i)} \right\rangle = 0.$$

Furthermore, for each $i, j \in [n], p \in [P] \setminus \{p_i^*\}, q \in [P] \setminus \{p_j^*\}$ with $(i, p) \neq (j, q)$ satisfies

$$\left| c_i^{(p)} c_j^{(q)} \left\langle \nabla_{\boldsymbol{W}} z_i^{(p)}, \nabla_{\boldsymbol{W}} z_j^{(q)} \right\rangle \right|$$

$$= \left| c_i^{(p)} c_j^{(q)} \right| \left( \phi'\left( \left\langle \boldsymbol{w}_1, \boldsymbol{x}_i^{(p)} \right\rangle \right) \phi'\left( \left\langle \boldsymbol{w}_1, \boldsymbol{x}_j^{(q)} \right\rangle \right) + \phi'\left( \left\langle \boldsymbol{w}_{-1}, \boldsymbol{x}_i^{(p)} \right\rangle \right) \phi'\left( \left\langle \boldsymbol{w}_{-1}, \boldsymbol{x}_j^{(q)} \right\rangle \right) \right) \left| \left\langle \boldsymbol{x}_i^{(p)}, \boldsymbol{x}_j^{(q)} \right\rangle \right|$$

$$\leq \frac{1}{4Pn} \left( c_i^{(p)} \right)^2 \left( \phi'\left( \left\langle \boldsymbol{w}_1, \boldsymbol{x}_i^{(p)} \right\rangle \right)^2 + \phi'\left( \left\langle \boldsymbol{w}_{-1}, \boldsymbol{x}_i^{(p)} \right\rangle \right)^2 \right) \left\| \boldsymbol{x}_i^{(p)} \right\|^2$$

$$+ \frac{1}{4Pn} \left( c_j^{(q)} \right)^2 \left( \phi'\left( \left\langle \boldsymbol{w}_1, \boldsymbol{x}_j^{(q)} \right\rangle \right)^2 + \phi'\left( \left\langle \boldsymbol{w}_{-1}, \boldsymbol{x}_j^{(q)} \right\rangle \right)^2 \right) \left\| \boldsymbol{x}_j^{(q)} \right\|^2$$

$$= \frac{1}{4Pn} \left( \left( c_i^{(p)} \right)^2 \left\| \nabla_{\boldsymbol{W}} z_i^{(p)} \right\|^2 + \left( c_j^{(q)} \right)^2 \left\| \nabla_{\boldsymbol{W}} z_j^{(q)} \right\|^2 \right)$$

where the last inequality is due to AM-GM inequality and

$$\left\| \boldsymbol{x}_i^{(p)} \right\| \cdot \left\| \boldsymbol{x}_j^{(q)} \right\| \geq 2nP \left| \left\langle \boldsymbol{x}_i^{(p)}, \boldsymbol{x}_j^{(q)} \right\rangle \right|,$$

which we show through a case analysis. For the case $p = \tilde{p}_i$ and $q = \tilde{p}_j$, this inequality holds since

$$\left\| \boldsymbol{x}_i^{(p)} \right\| \cdot \left\| \boldsymbol{x}_j^{(q)} \right\| \geq \left\| \xi_i^{(p)} \right\| \cdot \left\| \xi_j^{(q)} \right\| \geq 2nP \left( \left| \left\langle \xi_i^{(p)}, \xi_j^{(q)} \right\rangle \right| + \alpha^2 \right) \geq 2nP \left| \left\langle \boldsymbol{x}_i^{(p)}, \boldsymbol{x}_j^{(q)} \right\rangle \right|,$$

where the second inequality is due to

$$\frac{1}{2} \left\| \xi_i^{(p)} \right\| \cdot \left\| \xi_j^{(q)} \right\| \geq 2nP \left| \left\langle \xi_i^{(p)}, \xi_j^{(q)} \right\rangle \right|, \quad \frac{1}{2} \left\| \xi_i^{(p)} \right\| \cdot \left\| \xi_j^{(q)} \right\| \geq 2nP\alpha^2.$$

which is implied by the fact from the event $E_{\mathrm{init}}$ defined in Lemma B.2, (A1), (A2), and (A7). In the remaining case,

$$\left\| \boldsymbol{x}_i^{(p)} \right\| \cdot \left\| \boldsymbol{x}_j^{(q)} \right\| \geq \left\| \xi_i^{(p)} \right\| \cdot \left\| \xi_j^{(q)} \right\| \geq 2nP \left| \left\langle \xi_i^{(p)}, \xi_j^{(q)} \right\rangle \right| = 2nP \left| \left\langle \boldsymbol{x}_i^{(p)}, \boldsymbol{x}_j^{(q)} \right\rangle \right|,$$

where the second inequality is due to the fact from event $E_{\text{init}}$ defined in Lemma B.2, (A1), and (A2). For $s \in \{\pm 1\}, k \in [K]$ and $i \in [n], p \in [P] \setminus \{p_i^*\}$,

$$\|\nabla_{\boldsymbol{W}} z_{s,k}\|^2 = \phi'(\langle \boldsymbol{w}_1, \boldsymbol{v}_{s,k}\rangle)^2 + \phi'(\langle \boldsymbol{w}_{-1}, \boldsymbol{v}_{s,k}\rangle)^2 \geq 2\beta^2,$$

and

$$\begin{aligned}
\left\|\nabla_{\boldsymbol{W}} z_i^{(p)}\right\|^2 &= \left(\phi'\left(\left\langle \boldsymbol{w}_1, \boldsymbol{x}_i^{(p)}\right\rangle\right)^2 + \phi'\left(\left\langle \boldsymbol{w}_{-1}, \boldsymbol{x}_i^{(p)}\right\rangle\right)^2\right)\left\|\boldsymbol{x}_i^{(p)}\right\|^2 \\
&\geq \beta^2 \sigma_{i,p}^2 d \\
&\geq \beta^2,
\end{aligned}$$

where the last inequality is due to (8). By merging all inequalities together, we have

$$\begin{aligned}
&\|\boldsymbol{J}(\boldsymbol{W})\boldsymbol{c}\|^2 \\
&= \sum_{s \in \{\pm 1\}, k \in [K]} c_{s,k}^2 \|\nabla_{\boldsymbol{W}} z_{s,k}\|^2 + \sum_{\substack{i \in [n], p \in [P] \setminus \{p_i^*\}}} \left(c_i^{(p)}\right)^2 \left\|\nabla_{\boldsymbol{W}} z_i^{(p)}\right\|^2 \\
&\quad + \sum_{s \in \{\pm 1\}, i \in \mathcal{F}_s} c_{s,1} c_i^{(\tilde{p}_i)} \left\langle \nabla_{\boldsymbol{W}} z_{s,1}, \nabla_{\boldsymbol{W}} z_i^{(\tilde{p}_i)}\right\rangle + \sum_{\substack{i \in [n], p \in [P] \setminus \{p_i^*\} \\ j \in [n], q \in [P] \setminus \{p_j^*\} \\ (i,p) \neq (j,q)}} c_i^{(p)} c_j^{(q)} \left\langle \nabla_{\boldsymbol{W}} z_i^{(p)}, \nabla_{\boldsymbol{W}} z_j^{(q)}\right\rangle \\
&\geq \sum_{s \in \{\pm 1\}, k \in [K]} c_{s,k}^2 \|\nabla_{\boldsymbol{W}} z_{s,k}\|^2 + \sum_{\substack{i \in [n], p \in [P] \setminus \{p_i^*\}}} \left(c_i^{(p)}\right)^2 \left\|\nabla_{\boldsymbol{W}} z_i^{(p)}\right\|^2 \\
&\quad - \sum_{s \in \{\pm 1\}, i \in \mathcal{F}_s} \left(\frac{1}{2n} c_{s,1}^2 \|\nabla_{\boldsymbol{W}} z_{s,1}\|^2 + \frac{1}{4}\left(c_i^{(\tilde{p}_i)}\right)^2 \left\|\nabla_{\boldsymbol{W}} z_i^{(\tilde{p}_i)}\right\|^2\right) \\
&\quad - \frac{1}{4Pn} \sum_{\substack{i \in [n], p \in [P] \setminus \{p_i^*\} \\ j \in [n], q \in [P] \setminus \{p_j^*\} \\ (i,p) \neq (j,q)}} \left(\left(c_i^{(p)}\right)^2 \left\|\nabla_{\boldsymbol{W}} z_i^{(p)}\right\|^2 + \left(c_j^{(q)}\right)^2 \left\|\nabla_{\boldsymbol{W}} z_j^{(q)}\right\|^2\right) \\
&> \frac{1}{4} \sum_{s \in \{\pm 1\}, k \in [K]} c_{s,k}^2 \|\nabla_{\boldsymbol{W}} z_{s,k}\|^2 + \frac{1}{4} \sum_{\substack{i \in [n], p \in [P] \setminus \{p_i^*\}}} \left(c_i^{(p)}\right)^2 \left\|\nabla_{\boldsymbol{W}} z_i^{(p)}\right\|^2 \geq \frac{\beta^2}{4},
\end{aligned}$$

and we conclude $\sigma_{\min}(\boldsymbol{J}(\boldsymbol{W})) \geq \frac{\beta}{2}$ for any $\boldsymbol{W}$.

### E.2.2 Characterization of a Global Minimum of CutMix Loss

In this section, we will check that $h(\boldsymbol{Z})$ is strictly convex and it has a global minimum.

For each $i, j \in [n]$ and $\mathcal{S} \subset [P]$ let us define $\boldsymbol{a}_{i,j,\mathcal{S}} \in \mathbb{R}^{2K+n(P-1)}$ as

$$\boldsymbol{a}_{i,j,\mathcal{S}} = \sum_{p \in \mathcal{S}} \boldsymbol{e}_i^{(p)} + \sum_{p \notin \mathcal{S}} \boldsymbol{e}_j^{(p)},$$

and then

$$h(\boldsymbol{Z}) = \frac{1}{n^2} \sum_{i,j \in [n]} \mathbb{E}_{\mathcal{S} \sim \mathcal{D}_{\mathcal{S}}} \left[\frac{|\mathcal{S}|}{P} \ell(y_i \langle \boldsymbol{a}_{i,j,\mathcal{S}}, \boldsymbol{Z}\rangle) + \left(1 - \frac{|\mathcal{S}|}{P}\right) \ell(y_j \langle \boldsymbol{a}_{i,j,\mathcal{S}}, \boldsymbol{Z}\rangle)\right].$$

Since $\ell(\cdot)$ is convex, $h(\boldsymbol{Z})$ is also convex. Note that

$$\nabla h(\boldsymbol{Z}) = \frac{1}{n^2} \sum_{i,j \in [n]} \mathbb{E}_{\mathcal{S} \sim \mathcal{D}_{\mathcal{S}}} \left[\left(\frac{|\mathcal{S}|}{P} y_i \ell'(y_i \langle \boldsymbol{a}_{i,j,\mathcal{S}}, \boldsymbol{Z}\rangle) + \left(1 - \frac{|\mathcal{S}|}{P}\right) y_j \ell'(y_j \langle \boldsymbol{a}_{i,j,\mathcal{S}}, \boldsymbol{Z}\rangle)\right) \boldsymbol{a}_{i,j,\mathcal{S}}\right],$$

and

$$\nabla^2 h(\boldsymbol{Z})$$

$$= \frac{1}{n^2} \sum_{i,j \in [n]} \mathbb{E}_{\mathcal{S} \sim \mathcal{D}_{\mathcal{S}}} \left[ \left( \frac{|\mathcal{S}|}{P} \ell''(y_i \langle \boldsymbol{a}_{i,j,\mathcal{S}}, \boldsymbol{Z} \rangle) + \left( 1 - \frac{|\mathcal{S}|}{P} \right) \ell''(y_j \langle \boldsymbol{a}_{i,j,\mathcal{S}}, \boldsymbol{Z} \rangle) \right) \boldsymbol{a}_{i,j,\mathcal{S}} \boldsymbol{a}_{i,j,\mathcal{S}}^\top \right]$$

$$= \frac{1}{n^2} \sum_{i,j \in [n]} \mathbb{E}_{\mathcal{S} \sim \mathcal{D}_{\mathcal{S}}} \left[ \ell''(\langle \boldsymbol{a}_{i,j,\mathcal{S}}, \boldsymbol{Z} \rangle) \boldsymbol{a}_{i,j,\mathcal{S}} \boldsymbol{a}_{i,j,\mathcal{S}}^\top \right],$$

where the last equality holds since $\ell''(z) = \ell''(-z)$ for any $z \in \mathbb{R}$. From the equation above, it suffices to show that $\{\boldsymbol{a}_{i,j,\mathcal{S}}\}_{i,j \in [n], \mathcal{S} \subset [P]}$ spans $\mathbb{R}^{2K + n(P-1)}$ to show strict convexity of $h(\boldsymbol{Z})$.

We define a function $I : [P] \to [n]$ such that for each $p \in [P]$, $p^*_{I(p)} = p$ with $\boldsymbol{x}^{(p)}_{I(p)} = \boldsymbol{v}_{1,1}$, where the existence is guaranteed by Lemma B.2 (but not necessarily unique). Then for any $i \in [n]$ and $p \in [p]$, we have

$$\boldsymbol{a}_{i,i,\emptyset} + \sum_{q \in [P] \backslash \{p\}} \boldsymbol{a}_{I(q),i,\{q\}} - (P-1) \boldsymbol{a}_{I(p),i,\{p\}}$$

$$= \sum_{p' \in [P]} \boldsymbol{e}^{(p')}_i + \sum_{q \in [P] \backslash \{p\}} \left( \boldsymbol{e}_{1,1} + \sum_{p' \in [P] \backslash \{q\}} \boldsymbol{e}^{(p')}_i \right) - (P-1) \left( \boldsymbol{e}_{1,1} + \sum_{p' \in [P] \backslash \{p\}} \boldsymbol{e}^{(p')}_i \right)$$

$$= \sum_{p' \in [P]} \boldsymbol{e}^{(p')}_i + \left( (P-1) \boldsymbol{e}^{(p)}_i + (P-2) \sum_{p' \in [P] \backslash \{p\}} \boldsymbol{e}^{(p')}_i \right) - (P-1) \sum_{p' \in [P] \backslash \{p\}} \boldsymbol{e}^{(p')}_i$$

$$= P \boldsymbol{e}^{(p)}_i. \tag{34}$$

Hence, $\{\boldsymbol{a}_{i,j,\mathcal{S}}\}_{i,j \in [n], \mathcal{S} \subset [P]}$ spans $\mathbb{R}^{2K + n(P-1)}$ and $h(\boldsymbol{Z})$ is strictly convex. Thus, it can have at most one global minimum. We want to show the existence of the global minimum and characterize it.

$$n^2 \nabla h(\boldsymbol{Z})$$

$$= \sum_{i,j \in [n]} \mathbb{E}_{\mathcal{S} \sim \mathcal{D}_{\mathcal{S}}} \left[ \left( \frac{|\mathcal{S}|}{P} y_i \ell'(y_i \langle \boldsymbol{a}_{i,j,\mathcal{S}}, \boldsymbol{Z} \rangle) + \left( 1 - \frac{|\mathcal{S}|}{P} \right) y_j \ell'(y_j \langle \boldsymbol{a}_{i,j,\mathcal{S}}, \boldsymbol{Z} \rangle) \right) \boldsymbol{a}_{i,j,\mathcal{S}} \right]$$

$$= 2 \sum_{\substack{i,j \in [n] \\ p \in [P]}} \mathbb{E}_{\mathcal{S} \sim \mathcal{D}_{\mathcal{S}}} \left[ \left( \frac{|\mathcal{S}|}{P} y_i \ell'(y_i \langle \boldsymbol{a}_{i,j,\mathcal{S}}, \boldsymbol{Z} \rangle) + \left( 1 - \frac{|\mathcal{S}|}{P} \right) y_j \ell'(y_j \langle \boldsymbol{a}_{i,j,\mathcal{S}}, \boldsymbol{Z} \rangle) \right) \mathbb{1}_{p \in \mathcal{S}} \right] \boldsymbol{e}^{(p)}_i.$$

We can simplify terms as

$$\sum_{j \in [n]} \mathbb{E}_{\mathcal{S} \sim \mathcal{D}_{\mathcal{S}}} \left[ \left( \frac{|\mathcal{S}|}{P} y_i \ell'(y_i \langle \boldsymbol{a}_{i,j,\mathcal{S}}, \boldsymbol{Z} \rangle) + \left( 1 - \frac{|\mathcal{S}|}{P} \right) y_j \ell'(y_j \langle \boldsymbol{a}_{i,j,\mathcal{S}}, \boldsymbol{Z} \rangle) \right) \mathbb{1}_{p \in \mathcal{S}} \right]$$

$$= \sum_{j \in \mathcal{V}_{y_i}} \mathbb{E}_{\mathcal{S} \sim \mathcal{D}_{\mathcal{S}}} \left[ \left( \frac{|\mathcal{S}|}{P} y_i \ell'(y_i \langle \boldsymbol{a}_{i,j,\mathcal{S}}, \boldsymbol{Z} \rangle) + \left( 1 - \frac{|\mathcal{S}|}{P} \right) y_j \ell'(y_j \langle \boldsymbol{a}_{i,j,\mathcal{S}}, \boldsymbol{Z} \rangle) \right) \mathbb{1}_{p \in \mathcal{S}} \right]$$

$$\quad + \sum_{j \in \mathcal{V}_{-y_i}} \mathbb{E}_{\mathcal{S} \sim \mathcal{D}_{\mathcal{S}}} \left[ \left( \frac{|\mathcal{S}|}{P} y_i \ell'(y_i \langle \boldsymbol{a}_{i,j,\mathcal{S}}, \boldsymbol{Z} \rangle) + \left( 1 - \frac{|\mathcal{S}|}{P} \right) y_j \ell'(y_j \langle \boldsymbol{a}_{i,j,\mathcal{S}}, \boldsymbol{Z} \rangle) \right) \mathbb{1}_{p \in \mathcal{S}} \right]$$

$$= y_i \sum_{j \in \mathcal{V}_{y_i}} \mathbb{E}_{\mathcal{S} \sim \mathcal{D}_{\mathcal{S}}} [\ell'(y_i \langle \boldsymbol{a}_{i,j,\mathcal{S}}, \boldsymbol{Z} \rangle) \mathbb{1}_{p \in \mathcal{S}}]$$

$$\quad + y_i \sum_{j \in \mathcal{V}_{-y_i}} \mathbb{E}_{\mathcal{S} \sim \mathcal{D}_{\mathcal{S}}} \left[ \left( \ell'(y_i \langle \boldsymbol{a}_{i,j,\mathcal{S}}, \boldsymbol{Z} \rangle) + \left( 1 - \frac{|\mathcal{S}|}{P} \right) \right) \mathbb{1}_{p \in \mathcal{S}} \right]$$

$$= y_i |\mathcal{V}_{-y_i}| \mathbb{E}_{\mathcal{S} \sim \mathcal{D}_{\mathcal{S}}} \left[ \left( 1 - \frac{|\mathcal{S}|}{P} \right) \mathbb{1}_{p \in \mathcal{S}} \right] + y_i \sum_{j \in [n]} \mathbb{E}_{\mathcal{S} \sim \mathcal{D}_{\mathcal{S}}} [\ell'(y_i \langle \boldsymbol{a}_{i,j,\mathcal{S}}, \boldsymbol{Z} \rangle) \mathbb{1}_{p \in \mathcal{S}}],$$

where the second equality holds since $\ell'(z) + \ell'(-z) = -1$. Also, for any $p \in [P]$,

$$\mathbb{E}_{\mathcal{S} \sim \mathcal{D}_{\mathcal{S}}} \left[ \left( 1 - \frac{|\mathcal{S}|}{P} \right) \mathbb{1}_{p \in \mathcal{S}} \right] = \frac{1}{P} \sum_{q \in [P]} \mathbb{E}_{\mathcal{S} \sim \mathcal{D}_{\mathcal{S}}} \left[ \left( 1 - \frac{|\mathcal{S}|}{P} \right) \mathbb{1}_{q \in \mathcal{S}} \right]$$

$$= \frac{1}{P}\mathbb{E}_{\mathcal{S}\sim\mathcal{D}_\mathcal{S}}\left[\left(1-\frac{|\mathcal{S}|}{P}\right)\sum_{q\in\mathcal{S}}\mathbb{1}_{q\in\mathcal{S}}\right]$$

$$= \frac{1}{P}\mathbb{E}_{\mathcal{S}\sim\mathcal{D}_\mathcal{S}}\left[\left(1-\frac{|\mathcal{S}|}{P}\right)|\mathcal{S}|\right] = \frac{P-1}{6P}.$$

Hence, if

$$\sum_{j\in[n]}\mathbb{E}_{\mathcal{S}\sim\mathcal{D}_\mathcal{S}}[\ell'(y_i\langle a_{i,j,\mathcal{S}},Z\rangle)\mathbb{1}_{p\in\mathcal{S}}] + \frac{P-1}{6P}|\mathcal{V}_{-y_i}| = 0,$$

for all $i\in[n]$ and $p\in[P]$, then we have $\nabla h(Z) = 0$. Let us consider a specific $Z$ parameterized by $z_1, z_{-1}$, of the form $z_i^{(p)} = y_i z_{y_i}$ for all $i\in[n]$ and $p\in[P]$. We will find a stationary point with this specific form and then it should be the unique global minimum in the entire domain. Then for each $i\in[n]$ and $p\in[P]$, we have

$$\sum_{j\in[n]}\mathbb{E}_{\mathcal{S}\sim\mathcal{D}_\mathcal{S}}[\ell'(y_i\langle a_{i,j,\mathcal{S}},Z\rangle)\mathbb{1}_{p\in\mathcal{S}}]$$

$$= \sum_{j\in\mathcal{V}_{y_i}}\mathbb{E}_{\mathcal{S}\sim\mathcal{D}_\mathcal{S}}[\ell'(y_i\langle a_{i,j,\mathcal{S}},Z\rangle)\mathbb{1}_{p\in\mathcal{S}}] + \sum_{j\in\mathcal{V}_{-y_i}}\mathbb{E}_{\mathcal{S}\sim\mathcal{D}_\mathcal{S}}[\ell'(y_i\langle a_{i,j,\mathcal{S}},Z\rangle)\mathbb{1}_{p\in\mathcal{S}}]$$

$$= |\mathcal{V}_{y_i}|\cdot\mathbb{E}_{\mathcal{S}\sim\mathcal{D}_\mathcal{S}}[\ell'(Pz_{y_i})\mathbb{1}_{p\in\mathcal{S}}] + |\mathcal{V}_{-y_i}|\cdot\mathbb{E}_{\mathcal{S}\sim\mathcal{D}_\mathcal{S}}[\ell'(|\mathcal{S}|z_{y_i}-(P-|\mathcal{S}|)z_{-y_i})\mathbb{1}_{p\in\mathcal{S}}]$$

$$= \frac{1}{P}\sum_{q\in[P]}\left(|\mathcal{V}_{y_i}|\cdot\mathbb{E}_{\mathcal{S}\sim\mathcal{D}_\mathcal{S}}[\ell'(Pz_{y_i})\mathbb{1}_{q\in\mathcal{S}}] + |\mathcal{V}_{-y_i}|\cdot\mathbb{E}_{\mathcal{S}\sim\mathcal{D}_\mathcal{S}}[\ell'(|\mathcal{S}|z_{y_i}-(P-|\mathcal{S}|)z_{-y_i})\mathbb{1}_{q\in\mathcal{S}}]\right)$$

$$= \frac{1}{P}\left(|\mathcal{V}_{y_i}|\cdot\mathbb{E}_{\mathcal{S}\sim\mathcal{D}_\mathcal{S}}\left[\ell'(Pz_{y_i})\sum_{q\in\mathcal{S}}\mathbb{1}_{q\in\mathcal{S}}\right]\right.$$

$$\left. +|\mathcal{V}_{-y_i}|\cdot\mathbb{E}_{\mathcal{S}\sim\mathcal{D}_\mathcal{S}}\left[\ell'(|\mathcal{S}|z_{y_i}-(P-|\mathcal{S}|)z_{-y_i})\sum_{q\in\mathcal{S}}\mathbb{1}_{q\in\mathcal{S}}\right]\right)$$

$$= \frac{1}{P}\left(|\mathcal{V}_{y_i}|\cdot\mathbb{E}_{\mathcal{S}\sim\mathcal{D}_\mathcal{S}}[|\mathcal{S}|\ell'(Pz_{y_i})] + |\mathcal{V}_{-y_i}|\cdot\mathbb{E}_{\mathcal{S}\sim\mathcal{D}_\mathcal{S}}[|\mathcal{S}|\ell'(|\mathcal{S}|z_{y_i}-(P-|\mathcal{S}|)z_{-y_i})]\right)$$

$$= \frac{|\mathcal{V}_{y_i}|}{2}\ell'(Pz_{y_i}) + \frac{|\mathcal{V}_{-y_i}|}{P}\mathbb{E}_{\mathcal{S}\sim\mathcal{D}_\mathcal{S}}[|\mathcal{S}|\ell'(|\mathcal{S}|z_{y_i}-(P-|\mathcal{S}|)z_{-y_i})].$$

From Lemma F.4, there exists a unique minimizer $\hat{Z} = \{\hat{z}_{s,k}\}_{s\in\{\pm1\},k\in[K]}\cup\left\{\hat{z}_i^{(p)}\right\}_{i\in[n],p\in[P]\setminus\{p_i^*\}}$ of $h(Z)$ and it satisfies $s\hat{z}_{s,k} = z_s^* = \Theta(1)$ for all $k\in[K]$ and $y_i\hat{z}_i^{(p)} = z_{y_i}^* = \Theta(1)$ for all $i\in[n]$ and $p\in[P]\setminus\{p_i^*\}$ due to (A1).

### E.2.3 Strong Convexity Near Global Minimum

We will show that $h(Z)$ is strongly convex in a set $\mathcal{G}$ containing a global minimum $\hat{Z}$ where $\mathcal{G}$ is defined as follows.

$$\mathcal{G} := \left\{Z\in\mathbb{R}^{2K+n(P-1)} : \|Z-\hat{Z}\|_\infty < \|\hat{Z}\|_\infty\right\},$$

here $\|\cdot\|_\infty$ is $\ell_\infty$ norm. For any $Z\in\mathcal{G}$ and a unit vector $c\in\mathbb{R}^{2K+n(P-1)}$ with $c = \sum_{s\in\{\pm1\},k\in[K]}c_{s,k}e_{s,k} + \sum_{i\in[n],p\in[P]\setminus\{p_i^*\}}c_i^{(p)}e_i^{(p)}$, we have

$$c^\top\nabla^2 h(Z)c = \frac{1}{n^2}\sum_{i,j\in[n]}\mathbb{E}_{\mathcal{S}\sim\mathcal{D}_\mathcal{S}}\left[\ell''(\langle a_{i,j,\mathcal{S}},Z\rangle)\langle a_{i,j,\mathcal{S}},c\rangle^2\right]$$

$$\geq \frac{\ell''(2P\|\hat{Z}\|_\infty)}{n^2}\sum_{i,j\in[n]}\mathbb{E}_{\mathcal{S}\sim\mathcal{D}_\mathcal{S}}[\langle a_{i,j,\mathcal{S}},c\rangle^2].$$

Note that for each $i \in [n], p \in [P]$, from (34), we have

$$c_i^{(p)} = \left\langle \boldsymbol{c}, \boldsymbol{e}_i^{(p)} \right\rangle = \frac{1}{P}\langle \boldsymbol{c}, \boldsymbol{a}_{i,i,\emptyset}\rangle + \frac{1}{P}\sum_{q \in [P]\setminus\{p\}} \langle \boldsymbol{c}, \boldsymbol{a}_{I(q),i,\{q\}}\rangle - \frac{P-1}{P}\langle \boldsymbol{c}, \boldsymbol{a}_{I(p),i,\{p\}}\rangle,$$

where we use the notational convention $c_i^{(p_i^*)} = c_{s,k}$ for $s \in \{\pm 1\}, k \in [K]$ and $i \in \mathcal{V}_{s,k}$. By Cauchy-Schwartz inequality and the fact that $\mathbb{P}_{\mathcal{S}\sim\mathcal{D}_{\mathcal{S}}}[\mathcal{S} = \emptyset], \mathbb{P}_{\mathcal{S}\sim\mathcal{D}_{\mathcal{S}}}[\mathcal{S} = \{q\}] \geq \frac{1}{P(P+1)}$ for all $q \in [P]$,

$$\left(c_i^{(p)}\right)^2$$

$$= \left( \frac{1}{P}\langle \boldsymbol{c}, \boldsymbol{a}_{i,i,\emptyset}\rangle + \frac{1}{P}\sum_{q \in [P]\setminus\{p\}} \langle \boldsymbol{c}, \boldsymbol{a}_{I(q),i,\{q\}}\rangle - \frac{P-1}{P}\langle \boldsymbol{c}, \boldsymbol{a}_{I(p),i,\{p\}}\rangle \right)^2$$

$$\leq \left( \frac{1}{P^2} + \frac{P-1}{P^2} + \left(-\frac{P-1}{P}\right)^2 \right)\left( \langle \boldsymbol{c}, \boldsymbol{a}_{i,i,\emptyset}\rangle^2 + \sum_{q \in [P]\setminus\{p\}} \langle \boldsymbol{c}, \boldsymbol{a}_{I(q),i,\{q\}}\rangle^2 + \langle \boldsymbol{c}, \boldsymbol{a}_{I(p),i,\{p\}}\rangle^2 \right)$$

$$\leq \left( \frac{1}{P^2} + \frac{P-1}{P^2} + \left(-\frac{P-1}{P}\right)^2 \right) P(P+1)\sum_{i,j\in[n]} \mathbb{E}_{\mathcal{S}\sim\mathcal{D}_{\mathcal{S}}}[\langle \boldsymbol{c}, \boldsymbol{a}_{i,j,\mathcal{S}}\rangle^2]$$

$$\leq 2P^2 \sum_{i,j,\in[n]} \mathbb{E}_{\mathcal{S}\sim\mathcal{D}_{\mathcal{S}}} \left[ \langle \boldsymbol{c}, \boldsymbol{a}_{i,j,\mathcal{S}}\rangle^2 \right].$$

Hence, we have

$$\boldsymbol{c}^\top \nabla^2 h(\boldsymbol{Z})\boldsymbol{c} \geq \frac{\ell''(2P\|\hat{Z}\|_\infty)}{(4K+2n(P-1))P^2n^2}(4K+2n(P-1))P^2 \sum_{i,j\in[n]} \mathbb{E}_{\mathcal{S}\sim\mathcal{D}_{\mathcal{S}}} \left[ \langle \boldsymbol{c}, \boldsymbol{a}_{i,j,\mathcal{S}}\rangle^2 \right]$$

$$\geq \frac{\ell''(2P\|\hat{Z}\|_\infty)}{(4K+2n(P-1))P^2n^2}\left( \sum_{s\in\{\pm 1\},k\in[K]} c_{s,k}^2 + \sum_{i\in[n],q\in[P]\setminus\{p_i^*\}} \left(c_i^{(q)}\right)^2 \right)$$

$$= \frac{\ell''(2P\|\hat{Z}\|_\infty)}{(4K+2n(P-1))P^2n^2},$$

and we conclude $h(\boldsymbol{Z})$ is $\mu$-strongly convex in $\mathcal{G}$ where $\mu := \frac{\ell''(2P\|\hat{Z}\|_\infty)}{(4K+2n(P-1))P^2n^2}$. Due to (A1), (A2), and the fact that $\|\hat{Z}\|_\infty = \Theta(1)$, we have $\mu \geq \frac{1}{\text{poly}(d)}$.

### E.2.4 Near Stationary Points are Close to Global Minimum

In this step, we want to show that near stationary points of $h(\boldsymbol{Z})$ are close to a global minimum $\hat{Z}$.

**Lemma E.1.** *Suppose $\boldsymbol{Z} \in \mathbb{R}^{2K+n(P-1)}$ satisfies $\|\nabla h(\boldsymbol{Z})\| < \mu\epsilon$ with some $0 < \epsilon < \frac{\|\hat{z}\|_\infty}{2}$. Then, we have $\left\|\boldsymbol{Z} - \hat{\boldsymbol{Z}}\right\| < \epsilon$.*

*Proof of Lemma E.1.* If $\boldsymbol{Z} = \hat{\boldsymbol{Z}}$, we immediately have our conclusion. We may assume $\boldsymbol{Z} \neq \hat{\boldsymbol{Z}}$.

Let us define a function $g : \mathbb{R} \to \mathbb{R}$ as $g(t) = h\left(\hat{\boldsymbol{Z}} + t(\boldsymbol{Z} - \hat{\boldsymbol{Z}})\right)$. Then $g$ is convex and

$$g'(t) = \left\langle \nabla h\left(\hat{\boldsymbol{Z}} + t(\boldsymbol{Z} - \hat{\boldsymbol{Z}})\right), \boldsymbol{Z} - \hat{\boldsymbol{Z}} \right\rangle,$$

$$g''(t) = \left(\boldsymbol{Z} - \hat{\boldsymbol{Z}}\right)^\top \nabla^2 h\left(\hat{\boldsymbol{Z}} + t(\boldsymbol{Z} - \hat{\boldsymbol{Z}})\right)\left(\boldsymbol{Z} - \hat{\boldsymbol{Z}}\right).$$

Furthermore, for $0 \leq t \leq t_0$ where $t_0 := \frac{\|\hat{z}\|_\infty}{2\|\boldsymbol{Z}-\hat{\boldsymbol{Z}}\|_\infty}$,

$$\hat{\boldsymbol{Z}} + t(\boldsymbol{Z} - \hat{\boldsymbol{Z}}) \in \mathcal{G}, \qquad \therefore g''(t) \geq \mu\left\|\boldsymbol{Z} - \hat{\boldsymbol{Z}}\right\|^2.$$

We can conclude $g$ is $\mu \left\| \boldsymbol{Z} - \hat{\boldsymbol{Z}} \right\|^2$-strongly convex in $[0, t_0]$. From strong convexity in $[0, t_0]$ and convexity in $\mathbb{R}$, we have

$$(g'(t_0) - g'(0))t_0 = g'(t_0)t_0 \geq \mu \left\| \boldsymbol{Z} - \hat{\boldsymbol{Z}} \right\|^2 t_0^2, \quad (g'(1) - g'(t_0))(1 - t_0) \geq 0.$$

If $t_0 < 1$, we have

$$\|\nabla h(\boldsymbol{Z})\| \left\| \boldsymbol{Z} - \hat{\boldsymbol{Z}} \right\| \geq \left\langle \nabla h(\boldsymbol{Z}), \boldsymbol{Z} - \hat{\boldsymbol{Z}} \right\rangle = g'(1) \geq g'(t_0) \geq \mu \left\| \boldsymbol{Z} - \hat{\boldsymbol{Z}} \right\|^2 t_0,$$

and

$$\|\nabla h(\boldsymbol{Z})\| \geq \mu \left\| \boldsymbol{Z} - \hat{\boldsymbol{Z}} \right\| t_0 = \frac{\mu \left\| \boldsymbol{Z} - \hat{\boldsymbol{Z}} \right\| \left\| \hat{\boldsymbol{Z}} \right\|_\infty}{2 \left\| \boldsymbol{Z} - \hat{\boldsymbol{Z}} \right\|_\infty} \geq \frac{\mu \left\| \hat{\boldsymbol{Z}} \right\|_\infty}{2},$$

this is contradictory. Thus, we have $t_0 \geq 1$ and $\boldsymbol{Z} \in \mathcal{G}$. From the strong convexity of $h(\boldsymbol{Z})$ in $\mathcal{G}$, we have

$$\mu \left\| \boldsymbol{Z} - \hat{\boldsymbol{Z}} \right\| \leq \left\| \nabla h(\boldsymbol{Z}) - \nabla h(\hat{\boldsymbol{Z}}) \right\| = \|\nabla h(\boldsymbol{Z})\| < \mu\epsilon,$$

and we have our conclusion $\left\| \boldsymbol{Z} - \hat{\boldsymbol{Z}} \right\| < \epsilon$. $\qquad\square$

### E.2.5 Gradient Descent Achieves a Near Stationary Point

We will show that $\mathcal{L}_{\text{CutMix}}(\boldsymbol{W})$ is a smooth function.

**Lemma E.2.** *Suppose the event $E_{\text{init}}$ occurs. CutMix Loss $\mathcal{L}_{\text{CutMix}}(\boldsymbol{W})$ is $L$-smooth with $L = 9r^{-1}P\sigma_{\text{d}}^2 d$.*

*Proof of Lemma E.2.* Note that

$$\nabla_{\boldsymbol{w}_1} \mathcal{L}_{\text{CutMix}}(\boldsymbol{W})$$
$$= \frac{1}{n^2} \sum_{i,j \in [n]} \mathbb{E}_{\mathcal{S} \sim \mathcal{D}_{\mathcal{S}}} \left[ \left( \frac{|\mathcal{S}|}{P} y_i \ell'(y_i f_{\boldsymbol{W}}(\boldsymbol{X}_{i,j,\mathcal{S}})) + \left(1 - \frac{|\mathcal{S}|}{P}\right) y_j \ell'(y_j f_{\boldsymbol{W}}(\boldsymbol{X}_{i,j,\mathcal{S}})) \right) \right.$$
$$\left. \times \left( \sum_{p \in \mathcal{S}} \phi'\left( \left\langle \boldsymbol{w}_1, \boldsymbol{x}_i^{(p)} \right\rangle \right) \boldsymbol{x}_i^{(p)} + \sum_{p \notin \mathcal{S}} \phi'\left( \left\langle \boldsymbol{w}_1, \boldsymbol{x}_j^{(p)} \right\rangle \right) \boldsymbol{x}_j^{(p)} \right) \right].$$

Let $\widetilde{\boldsymbol{W}} = \{\widetilde{\boldsymbol{w}}_1, \widetilde{\boldsymbol{w}}_{-1}\}$ and $\overline{\boldsymbol{W}} = \{\overline{\boldsymbol{w}}_1, \overline{\boldsymbol{w}}_{-1}\}$ be any parameters of the neural network $f_{\boldsymbol{W}}$. For any $i, j \in [n]$ and $\mathcal{S} \subset [P]$,

$$\left( \frac{|\mathcal{S}|}{P} y_i \ell'(y_i f_{\widetilde{\boldsymbol{W}}}(\boldsymbol{X}_{i,j,\mathcal{S}})) + \left(1 - \frac{|\mathcal{S}|}{P}\right) y_j \ell'(y_j f_{\widetilde{\boldsymbol{W}}}(\boldsymbol{X}_{i,j,\mathcal{S}})) \right)$$
$$\times \left( \sum_{p \in \mathcal{S}} \phi'\left( \left\langle \widetilde{\boldsymbol{w}}_1, \boldsymbol{x}_i^{(p)} \right\rangle \right) \boldsymbol{x}_i^{(p)} + \sum_{p \notin \mathcal{S}} \phi'\left( \left\langle \widetilde{\boldsymbol{w}}_1, \boldsymbol{x}_j^{(p)} \right\rangle \right) \boldsymbol{x}_j^{(p)} \right)$$
$$- \left( \frac{|\mathcal{S}|}{P} y_i \ell'(y_i f_{\overline{\boldsymbol{W}}}(\boldsymbol{X}_{i,j,\mathcal{S}})) + \left(1 - \frac{|\mathcal{S}|}{P}\right) y_j \ell'(y_j f_{\overline{\boldsymbol{W}}}(\boldsymbol{X}_{i,j,\mathcal{S}})) \right)$$
$$\times \left( \sum_{p \in \mathcal{S}} \phi'\left( \left\langle \overline{\boldsymbol{w}}_1, \boldsymbol{x}_i^{(p)} \right\rangle \right) \boldsymbol{x}_i^{(p)} + \sum_{p \notin \mathcal{S}} \phi'\left( \left\langle \overline{\boldsymbol{w}}_1, \boldsymbol{x}_j^{(p)} \right\rangle \right) \boldsymbol{x}_j^{(p)} \right)$$
$$= \left( \frac{|\mathcal{S}|}{P} y_i \ell'(y_i f_{\widetilde{\boldsymbol{W}}}(\boldsymbol{X}_{i,j,\mathcal{S}})) + \left(1 - \frac{|\mathcal{S}|}{P}\right) y_j \ell'(y_j f_{\widetilde{\boldsymbol{W}}}(\boldsymbol{X}_{i,j,\mathcal{S}})) \right)$$
$$\times \left( \sum_{p \in \mathcal{S}} \phi'\left( \left\langle \widetilde{\boldsymbol{w}}_1, \boldsymbol{x}_i^{(p)} \right\rangle \right) \boldsymbol{x}_i^{(p)} + \sum_{p \notin \mathcal{S}} \phi'\left( \left\langle \widetilde{\boldsymbol{w}}_1, \boldsymbol{x}_j^{(p)} \right\rangle \right) \boldsymbol{x}_j^{(p)} \right)$$
$$- \left( \frac{|\mathcal{S}|}{P} y_i \ell'(y_i f_{\widetilde{\boldsymbol{W}}}(\boldsymbol{X}_{i,j,\mathcal{S}})) + \left(1 - \frac{|\mathcal{S}|}{P}\right) y_j \ell'(y_j f_{\widetilde{\boldsymbol{W}}}(\boldsymbol{X}_{i,j,\mathcal{S}})) \right)$$
$$\times \left( \sum_{p \in \mathcal{S}} \phi'\left( \left\langle \overline{\boldsymbol{w}}_1, \boldsymbol{x}_i^{(p)} \right\rangle \right) \boldsymbol{x}_i^{(p)} + \sum_{p \notin \mathcal{S}} \phi'\left( \left\langle \overline{\boldsymbol{w}}_1, \boldsymbol{x}_j^{(p)} \right\rangle \right) \boldsymbol{x}_j^{(p)} \right)$$

$$+ \left( \frac{|\mathcal{S}|}{P} y_i \ell'(y_i f_{\widetilde{\boldsymbol{W}}}(\boldsymbol{X}_{i,j,\mathcal{S}})) + \left( 1 - \frac{|\mathcal{S}|}{P} \right) y_j \ell'(y_j f_{\widetilde{\boldsymbol{W}}}(\boldsymbol{X}_{i,j,\mathcal{S}})) \right)$$

$$\times \left( \sum_{p \in \mathcal{S}} \phi'\left( \left\langle \overline{\boldsymbol{w}}_1, \boldsymbol{x}_i^{(p)} \right\rangle \right) \boldsymbol{x}_i^{(p)} + \sum_{p \notin \mathcal{S}} \phi'\left( \left\langle \overline{\boldsymbol{w}}_1, \boldsymbol{x}_j^{(p)} \right\rangle \right) \boldsymbol{x}_j^{(p)} \right)$$

$$- \left( \frac{|\mathcal{S}|}{P} y_i \ell'(y_i f_{\overline{\boldsymbol{W}}}(\boldsymbol{X}_{i,j,\mathcal{S}})) + \left( 1 - \frac{|\mathcal{S}|}{P} \right) y_j \ell'(y_j f_{\overline{\boldsymbol{W}}}(\boldsymbol{X}_{i,j,\mathcal{S}})) \right)$$

$$\times \left( \sum_{p \in \mathcal{S}} \phi'\left( \left\langle \overline{\boldsymbol{w}}_1, \boldsymbol{x}_i^{(p)} \right\rangle \right) \boldsymbol{x}_i^{(p)} + \sum_{p \notin \mathcal{S}} \phi'\left( \left\langle \overline{\boldsymbol{w}}_1, \boldsymbol{x}_j^{(p)} \right\rangle \right) \boldsymbol{x}_j^{(p)} \right).$$

Since $|\ell'| \leq 1$,

$$\left| \frac{|\mathcal{S}|}{P} y_i \ell'\left( y_i f_{\widetilde{\boldsymbol{W}}}(\boldsymbol{X}_{i,j,\mathcal{S}}) \right) + \left( 1 - \frac{|\mathcal{S}|}{P} \right) y_j \ell'\left( y_j f_{\widetilde{\boldsymbol{W}}}(\boldsymbol{X}_{i,j,\mathcal{S}}) \right) \right| \leq 1,$$

and since $|\phi'| \leq 1$,

$$\left\| \sum_{p \in \mathcal{S}} \phi'\left( \left\langle \overline{\boldsymbol{w}}_1, \boldsymbol{x}_i^{(p)} \right\rangle \right) \boldsymbol{x}_i^{(p)} + \sum_{p \notin \mathcal{S}} \phi'\left( \left\langle \overline{\boldsymbol{w}}_1, \boldsymbol{x}_j^{(p)} \right\rangle \right) \boldsymbol{x}_j^{(p)} \right\| \leq P \max_{i \in [n], p \in [P]} \left\| \boldsymbol{x}_i^{(p)} \right\|.$$

In addition, since $\phi$ is $r^{-1}$-smooth,

$$\left\| \left( \sum_{p \in \mathcal{S}} \phi'\left( \left\langle \widetilde{\boldsymbol{w}}_1, \boldsymbol{x}_i^{(p)} \right\rangle \right) \boldsymbol{x}_i^{(p)} + \sum_{p \notin \mathcal{S}} \phi'\left( \left\langle \widetilde{\boldsymbol{w}}_1, \boldsymbol{x}_j^{(p)} \right\rangle \right) \boldsymbol{x}_j^{(p)} \right) \right.$$

$$\left. - \left( \sum_{p \in \mathcal{S}} \phi'\left( \left\langle \overline{\boldsymbol{w}}_1, \boldsymbol{x}_i^{(p)} \right\rangle \right) \boldsymbol{x}_i^{(p)} + \sum_{p \notin \mathcal{S}} \phi'\left( \left\langle \overline{\boldsymbol{w}}_1, \boldsymbol{x}_j^{(p)} \right\rangle \right) \boldsymbol{x}_j^{(p)} \right) \right\|$$

$$\leq \sum_{p \in \mathcal{S}} \left| \phi'\left( \left\langle \widetilde{\boldsymbol{w}}_1, \boldsymbol{x}_i^{(p)} \right\rangle \right) - \phi'\left( \left\langle \overline{\boldsymbol{w}}_1, \boldsymbol{x}_i^{(p)} \right\rangle \right) \right| \left\| \boldsymbol{x}_i^{(p)} \right\|$$

$$+ \sum_{p \notin \mathcal{S}} \left| \phi'\left( \left\langle \widetilde{\boldsymbol{w}}_1, \boldsymbol{x}_j^{(p)} \right\rangle \right) - \phi'\left( \left\langle \overline{\boldsymbol{w}}_1, \boldsymbol{x}_j^{(p)} \right\rangle \right) \right| \left\| \boldsymbol{x}_j^{(p)} \right\|$$

$$\leq r^{-1} \sum_{p \in \mathcal{S}} \left| \left\langle \widetilde{\boldsymbol{w}}_1 - \overline{\boldsymbol{w}}_1, \boldsymbol{x}_i^{(p)} \right\rangle \right| \left\| \boldsymbol{x}_i^{(p)} \right\| + r^{-1} \sum_{p \notin \mathcal{S}} \left| \left\langle \widetilde{\boldsymbol{w}}_1 - \overline{\boldsymbol{w}}_1, \boldsymbol{x}_j^{(p)} \right\rangle \right| \left\| \boldsymbol{x}_j^{(p)} \right\|$$

$$\leq r^{-1} P \left( \max_{i \in [n], p \in [P]} \left\| \boldsymbol{x}_i^{(p)} \right\| \right)^2 \left\| \widetilde{\boldsymbol{w}}_1 - \overline{\boldsymbol{w}}_1 \right\|,$$

and since $\ell'$ and $\phi$ are 1-Lipschitz, we have

$$\left| \left( \frac{|\mathcal{S}|}{P} y_i \ell'(y_i f_{\widetilde{\boldsymbol{W}}}(\boldsymbol{X}_{i,j,\mathcal{S}})) + \left( 1 - \frac{|\mathcal{S}|}{P} \right) y_j \ell'(y_j f_{\widetilde{\boldsymbol{W}}}(\boldsymbol{X}_{i,j,\mathcal{S}})) \right) \right.$$

$$\left. - \left( \frac{|\mathcal{S}|}{P} y_i \ell'(y_i f_{\overline{\boldsymbol{W}}}(\boldsymbol{X}_{i,j,\mathcal{S}})) + \left( 1 - \frac{|\mathcal{S}|}{P} \right) y_j \ell'(y_j f_{\overline{\boldsymbol{W}}}(\boldsymbol{X}_{i,j,\mathcal{S}})) \right) \right|$$

$$\leq \left| f_{\widetilde{\boldsymbol{W}}}(\boldsymbol{X}_{i,j,\mathcal{S}}) - f_{\overline{\boldsymbol{W}}}(\boldsymbol{X}_{i,j,\mathcal{S}}) \right|$$

$$\leq \sum_{p \in \mathcal{S}} \left( \left| \left\langle \widetilde{\boldsymbol{w}}_1 - \overline{\boldsymbol{w}}_1, \boldsymbol{x}_i^{(p)} \right\rangle \right| + \left| \left\langle \widetilde{\boldsymbol{w}}_{-1} - \overline{\boldsymbol{w}}_{-1}, \boldsymbol{x}_i^{(p)} \right\rangle \right| \right)$$

$$+ \sum_{p \notin \mathcal{S}} \left( \left| \left\langle \widetilde{\boldsymbol{w}}_1 - \overline{\boldsymbol{w}}_1, \boldsymbol{x}_j^{(p)} \right\rangle \right| + \left| \left\langle \widetilde{\boldsymbol{w}}_{-1} - \overline{\boldsymbol{w}}_{-1}, \boldsymbol{x}_j^{(p)} \right\rangle \right| \right)$$

$$\leq P \max_{i \in [n], j \in [P]} \left\| \boldsymbol{x}_i^{(p)} \right\| \left( \left\| \widetilde{\boldsymbol{w}}_1 - \overline{\boldsymbol{w}}_1 \right\| + \left\| \widetilde{\boldsymbol{w}}_{-1} - \overline{\boldsymbol{w}}_{-1} \right\| \right)$$

$$\leq \sqrt{2} P \max_{i \in [n], j \in [P]} \left\| \boldsymbol{x}_i^{(p)} \right\| \left\| \widetilde{\boldsymbol{W}} - \overline{\boldsymbol{W}} \right\|.$$

Therefore,

$$\left\| \nabla_{\boldsymbol{w}_1} \mathcal{L}_{\mathrm{CutMix}}(\widetilde{\boldsymbol{W}}) - \nabla_{\boldsymbol{w}_1} \mathcal{L}_{\mathrm{CutMix}}(\overline{\boldsymbol{W}}) \right\|$$

$$\leq r^{-1} P \left( \max_{i \in [n], p \in [P]} \left\| \boldsymbol{x}_i^{(p)} \right\| \right)^2 \|\widetilde{\boldsymbol{w}}_1 - \overline{\boldsymbol{w}}_1\| + \sqrt{2} P^2 \left( \max_{i \in [n], p \in [P]} \left\| \boldsymbol{x}_i^{(p)} \right\| \right)^2 \left\| \widetilde{\boldsymbol{W}} - \overline{\boldsymbol{W}} \right\|$$

$$\leq 2 r^{-1} P \left( \max_{i \in [n], p \in [P]} \left\| \boldsymbol{x}_i^{(p)} \right\| \right)^2 \left\| \widetilde{\boldsymbol{W}} - \overline{\boldsymbol{W}} \right\|,$$

where the last equality is due to (A1) and (A8). In the same way, we can obtain

$$\left\| \nabla_{\boldsymbol{w}_{-1}} \mathcal{L}_{\mathrm{CutMix}}(\widetilde{\boldsymbol{W}}) - \nabla_{\boldsymbol{w}_{-1}} \mathcal{L}_{\mathrm{CutMix}}(\overline{\boldsymbol{W}}) \right\| \leq 2 r^{-1} P \left( \max_{i \in [n], p \in [P]} \left\| \boldsymbol{x}_i^{(p)} \right\| \right)^2 \left\| \widetilde{\boldsymbol{W}} - \overline{\boldsymbol{W}} \right\|,$$

and

$$\left\| \nabla \mathcal{L}_{\mathrm{CutMix}}(\widetilde{\boldsymbol{W}}) - \nabla \mathcal{L}_{\mathrm{CutMix}}(\overline{\boldsymbol{W}}) \right\| \leq 4 r^{-1} P \left( \max_{i \in [n], p \in [P]} \left\| \boldsymbol{x}_i^{(p)} \right\| \right)^2 \left\| \widetilde{\boldsymbol{W}} - \overline{\boldsymbol{W}} \right\|$$

$$\leq 9 r^{-1} P \sigma_{\mathrm{d}}^2 d \left\| \widetilde{\boldsymbol{W}} - \overline{\boldsymbol{W}} \right\|,$$

where the last inequality holds since $\left\| \xi_i^{(p)} \right\|^2 < \frac{3}{2} \sigma_{\mathrm{d}}^2 d$ and $\alpha^2 \leq \frac{3}{4} \sigma_{\mathrm{d}}^2 d$ due to (A7). Hence, $\mathcal{L}_{\mathrm{CutMix}}(\boldsymbol{W})$ is $L$-smooth with $L := 9 r^{-1} P \sigma_{\mathrm{d}}^2 d$. $\qquad\square$

Since our objective function $\mathcal{L}_{\mathrm{CutMix}}(\boldsymbol{W})$ is $L$-smooth and $\eta \leq \frac{1}{L}$ due to (A8), descent lemma (see Lemma 3.4 in Bubeck et al. (2015)) implies

$$\mathcal{L}_{\mathrm{CutMix}}\left(\boldsymbol{W}^{(t+1)}\right) - \mathcal{L}_{\mathrm{CutMix}}\left(\boldsymbol{W}^{(t)}\right) \leq -\frac{\eta}{2} \left\| \nabla \mathcal{L}_{\mathrm{CutMix}}\left(\boldsymbol{W}^{(t)}\right) \right\|^2,$$

and by telescoping sum, we have

$$\frac{1}{T} \sum_{t=0}^{T-1} \left\| \nabla \mathcal{L}_{\mathrm{CutMix}}\left(\boldsymbol{W}^{(t)}\right) \right\|^2 \leq \frac{2 \mathcal{L}_{\mathrm{CutMix}}\left(\boldsymbol{W}^{(0)}\right)}{\eta T} = \frac{\Theta(1)}{\eta T}, \tag{35}$$

for any $T > 0$.

Choose $\epsilon = \frac{\mu \beta \|\hat{\boldsymbol{Z}}\|_\infty}{\mathrm{polylog}(d)}$. Then from (35), there exists $T_{\mathrm{CutMix}} \leq \frac{\mathrm{poly}(d)}{\eta}$ such that

$$\left\| \nabla \mathcal{L}_{\mathrm{CutMix}}\left(\boldsymbol{W}^{(T_{\mathrm{CutMix}})}\right) \right\| \leq \epsilon.$$

From characterization of $\sigma_{\min}(\boldsymbol{J}(\boldsymbol{W}))$ in Section E.2.1,

$$\epsilon \geq \left\| \nabla \mathcal{L}_{\mathrm{CutMix}}\left(\boldsymbol{W}^{(T_{\mathrm{CutMix}})}\right) \right\| \geq \sigma_{\min}\left( \boldsymbol{J}\left(\boldsymbol{W}^{(T_{\mathrm{CutMix}})}\right) \right) \left\| \nabla h \left(\boldsymbol{Z}^{(T_{\mathrm{CutMix}})}\right) \right\|$$

$$\geq \frac{\beta}{2} \left\| \nabla h \left(\boldsymbol{Z}^{(T_{\mathrm{CutMix}})}\right) \right\|,$$

and thus

$$\left\| \nabla h \left(\boldsymbol{Z}^{(T_{\mathrm{CutMix}})}\right) \right\| \leq 2 \beta^{-1} \epsilon = \mu \cdot \frac{2 \|\hat{\boldsymbol{Z}}\|_\infty}{\mathrm{polylog}(d)}.$$

For sufficiently large $d$, the RHS becomes smaller than $\mu \cdot \frac{\|\hat{\boldsymbol{Z}}\|_\infty}{4}$. Then, by Lemma E.1 we have seen in Section E.2.4,

$$\left\| \boldsymbol{Z}^{(T_{\mathrm{CutMix}})} - \hat{\boldsymbol{Z}} \right\| \leq \frac{\|\hat{\boldsymbol{Z}}\|_\infty}{4},$$

and thus

$$\phi \left( \left\langle \boldsymbol{w}_{y_i}^{(T_{\mathrm{CutMix}})}, \boldsymbol{x}_i^{(p)} \right\rangle \right) - \phi \left( \left\langle \boldsymbol{w}_{-y_i}^{(T_{\mathrm{CutMix}})}, \boldsymbol{x}_i^{(p)} \right\rangle \right) = \Theta(1),$$

for all $i \in [n]$ and $p \in [P]$, and therefore it reaches perfect training accuracy.

### E.2.6 Test Accuracy of Solution Found by Gradient Descent

The final step is showing that $W^{(T_{\text{CutMix}})}$ reaches almost perfect test accuracy.

From the results of Section E.2.5, we have

$$\phi\left(\left\langle w_s^{(T_{\text{CutMix}})}, v_{s,k}\right\rangle\right) - \phi\left(\left\langle w_{-s}^{(T_{\text{CutMix}})}, v_{s,k}\right\rangle\right) = \Theta(1),$$

$$\phi\left(\left\langle w_{y_i}^{(T_{\text{CutMix}})}, \xi_i^{(p)}\right\rangle\right) - \phi\left(\left\langle w_{-y_i}^{(T_{\text{CutMix}})}, \xi_i^{(p)}\right\rangle\right) = \Theta(1),$$

for each $s \in \{\pm 1\}, k \in [K], i \in [n]$ and $p \in [P] \setminus \{p_i^*\}$.

For any $u > v$, by the mean value theorem, we have

$$\beta(u - v) \le \phi(u) - \phi(v) = (u - v)\frac{\phi(u) - \phi(v)}{u - v} \le (u - v).$$

Hence, we have

$$\phi\left(\left\langle w_s^{(T_{\text{CutMix}})}, v_{s,k}\right\rangle\right) - \phi\left(\left\langle w_{-s}^{(T_{\text{CutMix}})}, v_{s,k}\right\rangle\right) \le \left\langle w_s^{(T_{\text{CutMix}})} - w_{-s}^{(T_{\text{CutMix}})}, v_{s,k}\right\rangle,$$

$$\left\langle w_s^{(T_{\text{CutMix}})} - w_{-s}^{(T_{\text{CutMix}})}, v_{s,k}\right\rangle \le \beta^{-1}\left(\phi\left(\left\langle w_s^{(T_{\text{CutMix}})}, v_{s,k}\right\rangle\right) - \phi\left(\left\langle w_{-s}^{(T_{\text{CutMix}})}, v_{s,k}\right\rangle\right)\right),$$

and

$$\Omega(1) \le \left\langle w_s^{(T_{\text{CutMix}})} - w_{-s}^{(T_{\text{CutMix}})}, v_{s,k}\right\rangle \le \mathcal{O}(\beta^{-1}),$$

for each $s \in \{\pm 1\}$ and $k \in [K]$. Similarly, for all $i \in [n]$ and $p \in [P] \setminus \{p_i^*\}$,

$$\phi\left(\left\langle w_{y_i}^{(T_{\text{CutMix}})}, \xi_i^{(p)}\right\rangle\right) - \phi\left(\left\langle w_{-y_i}^{(T_{\text{CutMix}})}, \xi_i^{(p)}\right\rangle\right) \le \left\langle w_{y_i}^{(T_{\text{CutMix}})} - w_{-y_i}^{(T_{\text{CutMix}})}, \xi_i^{(p)}\right\rangle,$$

$$\left\langle w_{y_i}^{(T_{\text{CutMix}})} - w_{-y_i}^{(T_{\text{CutMix}})}, \xi_i^{(p)}\right\rangle \le \beta^{-1}\left(\phi\left(\left\langle w_{y_i}^{(T_{\text{CutMix}})}, \xi_i^{(p)}\right\rangle\right) - \phi\left(\left\langle w_{-y_i}^{(T_{\text{CutMix}})}, \xi_i^{(p)}\right\rangle\right)\right),$$

and

$$\Omega(1) \le \left\langle w_{y_i}^{(T_{\text{CutMix}})} - w_{-y_i}^{(T_{\text{CutMix}})}, \xi_i^{(p)}\right\rangle \le \mathcal{O}(\beta^{-1}).$$

By Lemma B.3,

$$w_1^{(T_{\text{CutMix}})} - w_{-1}^{(T_{\text{CutMix}})}$$
$$= w_1^{(0)} - w_{-1}^{(0)} + \sum_{s \in \{\pm 1\}, k \in [K]} s\gamma(s, k)v_{s,k} + \sum_{i \in [n], p \in [P] \setminus \{p_i^*\}} y_i \rho(i, p)\frac{\xi_i^{(p)}}{\left\|\xi_i^{(p)}\right\|^2},$$

where for each $s \in \{\pm 1\}$,

$$\gamma(s, 1) = \gamma_1^{(T_{\text{CutMix}})}(s, 1) + \gamma_{-1}^{(T_{\text{CutMix}})}(s, 1)$$
$$+ \alpha\sum_{i \in \mathcal{F}_s} y_i\left(\rho_1^{(T_{\text{CutMix}})}(i, \tilde{p}_i) + \rho_{-1}^{(T_{\text{CutMix}})}(i, \tilde{p}_i)\right)\left\|\xi_i^{(\tilde{p}_i)}\right\|^{-2},$$

and

$$\gamma(s, k) = \gamma_1^{(T_{\text{CutMix}})}(s, k) + \gamma_{-1}^{(T_{\text{CutMix}})}(s, k),$$
$$\rho(i, p) = \rho_1^{(T_{\text{CutMix}})}(i, p) + \rho_{-1}^{(T_{\text{CutMix}})}(i, p),$$

for each $s \in \{\pm 1\}, k \in [K] \setminus \{1\}, i \in [n]$ and $p \in [P] \setminus \{p_i^*\}$. If we choose $j \in [n], q \in [P] \setminus \{p_j^*\}$ such that $\rho(j, q) = \max_{i \in [n], p \in [P] \setminus \{p_i^*\}} \rho(i, p)$, then we have

$$\left\langle w_{y_j}^{(T_{\text{CutMix}})} - w_{-y_j}^{(T_{\text{CutMix}})}, \xi_j^{(q)}\right\rangle$$
$$= \left\langle w_{y_j}^{(0)} - w_{-y_j}^{(0)}, \xi_j^{(q)}\right\rangle + \rho(j, q) + y_j\sum_{\substack{i \in [n], p \in [P] \setminus \{p_i^*\} \\ (i,p) \ne (j,q)}} y_i \rho(i, p)\frac{\left\langle \xi_i^{(p)}, \xi_j^{(q)}\right\rangle}{\left\|\xi_i^{(p)}\right\|^2}.$$

From the event $E_{\mathrm{init}}$ defined in Lemma B.2, (A8), and (8),

$$\left|\left\langle \boldsymbol{w}_{y_j}^{(0)} - \boldsymbol{w}_{-y_j}^{(0)}, \xi_j^{(q)} \right\rangle\right| = o\left(\frac{1}{\mathrm{polylog}(d)}\right) \leq \frac{1}{2}\left\langle \boldsymbol{w}_{y_j}^{(T_{\mathrm{CutMix}})} - \boldsymbol{w}_{-y_j}^{(T_{\mathrm{CutMix}})}, \xi_j^{(q)}\right\rangle,$$

where the inequality holds since $\left\langle \boldsymbol{w}_{y_j}^{(T_{\mathrm{CutMix}})} - \boldsymbol{w}_{-y_j}^{(T_{\mathrm{CutMix}})}, \xi_j^{(q)}\right\rangle = \Omega(1)$. In addition, by triangular inequality, we have

$$\left|\sum_{\substack{i\in[n],p\in[P]\setminus\{p_i^*\}\\(i,p)\neq(j,q)}} y_i\rho(i,p)\frac{\left\langle \xi_i^{(p)}, \xi_j^{(q)}\right\rangle}{\left\|\xi_i^{(p)}\right\|^2}\right| \leq \sum_{\substack{i\in[n],p\in[P]\setminus\{p_i^*\}\\(i,p)\neq(j,q)}} \rho(i,p)\frac{\left|\left\langle \xi_i^{(p)}, \xi_j^{(q)}\right\rangle\right|}{\left\|\xi_i^{(p)}\right\|^2}$$

$$\leq \rho(j,q)\widetilde{\mathcal{O}}\left(nP\sigma_{\mathrm{d}}\sigma_{\mathrm{b}}^{-1}d^{-\frac{1}{2}}\right) \leq \frac{\rho(j,q)}{2},$$

where the last inequality is due to (9). Hence,

$$\frac{1}{3}\rho(j,q) \leq \left|\left\langle \boldsymbol{w}_{y_j}^{(T_{\mathrm{CutMix}})} - \boldsymbol{w}_{-y_j}^{(T_{\mathrm{CutMix}})}, \xi_j^{(q)}\right\rangle\right| \leq 3\rho(j,q)$$

and we have $\rho(j,q) = \widetilde{\mathcal{O}}(\beta^{-1})$.

Let $(\boldsymbol{X},y)\sim\mathcal{D}$ be a test data with $\boldsymbol{X} = \left(\boldsymbol{x}^{(1)}, \ldots, \boldsymbol{x}^{(P)}\right) \in \mathbb{R}^{d\times P}$ having feature patch $p^*$, dominant noise patch $\tilde{p}$, and feature vector $\boldsymbol{v}_{y,k}$. We have $\boldsymbol{x}^{(p)} \sim N(\boldsymbol{0}, \sigma_{\mathrm{b}}^2\boldsymbol{\Lambda})$ for each $p \in [P]\setminus\{p^*,\tilde{p}\}$ and $\boldsymbol{x}^{(\tilde{p})} - \alpha\boldsymbol{v}_{s,1} \sim N(\boldsymbol{0}, \sigma_{\mathrm{d}}^2\boldsymbol{\Lambda})$ for some $s \in \{\pm 1\}$. Therefore, for all $p \in [P]\setminus\{p^*,\tilde{p}\}$

$$\left|\phi\left(\left\langle \boldsymbol{w}_1^{(T_{\mathrm{CutMix}})}, \boldsymbol{x}^{(p)}\right\rangle\right) - \phi\left(\left\langle \boldsymbol{w}_{-1}^{(T_{\mathrm{CutMix}})}, \boldsymbol{x}^{(p)}\right\rangle\right)\right|$$

$$\leq \left|\left\langle \boldsymbol{w}_1^{(T_{\mathrm{CutMix}})} - \boldsymbol{w}_{-1}^{(T_{\mathrm{CutMix}})}, \boldsymbol{x}^{(p)}\right\rangle\right|$$

$$= \left|\left\langle \boldsymbol{w}_1^{(0)} - \boldsymbol{w}_{-1}^{(0)}, \boldsymbol{x}^{(p)}\right\rangle\right| + \sum_{i\in[n],q\in[P]\setminus\{p_i^*\}} \rho(i,q)\frac{\left|\left\langle \xi_i^{(q)}, \boldsymbol{x}^{(p)}\right\rangle\right|}{\left\|\xi_i^{(q)}\right\|^2}$$

$$\leq \widetilde{\mathcal{O}}\left(\sigma_0\sigma_{\mathrm{b}}d^{\frac{1}{2}}\right) + \widetilde{\mathcal{O}}\left(nP\beta^{-1}\sigma_{\mathrm{d}}\sigma_{\mathrm{b}}^{-1}d^{-\frac{1}{2}}\right)$$

$$= o\left(\frac{1}{\mathrm{polylog}(d)}\right), \tag{36}$$

with probability at least $1 - o\left(\frac{1}{\mathrm{poly}(d)}\right)$ due to Lemma B.2. In addition,

$$\left|\phi\left(\left\langle \boldsymbol{w}_1^{(T_{\mathrm{CutMix}})}, \boldsymbol{x}^{(\tilde{p})}\right\rangle\right) - \phi\left(\left\langle \boldsymbol{w}_{-1}^{(T_{\mathrm{CutMix}})}, \boldsymbol{x}^{(\tilde{p})}\right\rangle\right)\right|$$

$$\leq \left|\left\langle \boldsymbol{w}_1^{(T_{\mathrm{CutMix}})} - \boldsymbol{w}_{-1}^{(T_{\mathrm{CutMix}})}, \boldsymbol{x}^{(\tilde{p})}\right\rangle\right|$$

$$\leq \alpha\left|\left\langle \boldsymbol{w}_1^{(T_{\mathrm{CutMix}})} - \boldsymbol{w}_{-1}^{(T_{\mathrm{CutMix}})}, \boldsymbol{v}_{s,1}\right\rangle\right| + \left|\left\langle \boldsymbol{w}_1^{(T_{\mathrm{CutMix}})} - \boldsymbol{w}_{-1}^{(T_{\mathrm{CutMix}})}, \boldsymbol{x}^{(\tilde{p})} - \alpha\boldsymbol{v}_{s,1}\right\rangle\right|$$

$$\leq \alpha\beta^{-1}\left|\phi\left(\left\langle \boldsymbol{w}_1^{(T_{\mathrm{CutMix}})}, \boldsymbol{v}_{s,1}\right\rangle\right) - \phi\left(\left\langle \boldsymbol{w}_{-1}^{(T_{\mathrm{CutMix}})}, \boldsymbol{v}_{s,1}\right\rangle\right)\right|$$

$$+ \left|\left\langle \boldsymbol{w}_1^{(0)} - \boldsymbol{w}_{-1}^{(0)}, \boldsymbol{x}^{(\tilde{p})} - \alpha\boldsymbol{v}_{s,1}\right\rangle\right| + \sum_{i\in[n],q\in[P]\setminus\{p_i^*\}} \rho(i,q)\frac{\left|\left\langle \xi_i^{(q)}, \boldsymbol{x}^{(\tilde{p})} - \alpha\boldsymbol{v}_{s,1}\right\rangle\right|}{\left\|\xi_i^{(q)}\right\|^2}$$

$$\leq \widetilde{\mathcal{O}}\left(\alpha\beta^{-1}\right) + \widetilde{\mathcal{O}}\left(\sigma_0\sigma_{\mathrm{d}}d^{\frac{1}{2}}\right) + \widetilde{\mathcal{O}}\left(nP\beta^{-1}\sigma_{\mathrm{d}}\sigma_{\mathrm{b}}^{-1}d^{-\frac{1}{2}}\right)$$

$$= o\left(\frac{1}{\mathrm{polylog}(d)}\right), \tag{37}$$

with probability at least $1 - o\left(\frac{1}{\mathrm{poly}(d)}\right)$, where the last equality is due to (8), (9), (10), and (A8).

Suppose (36) and (37) holds. Then,

$$
\begin{aligned}
& y f_{\boldsymbol{W}^{(T_{\text{CutMix}})}}(\boldsymbol{X}) \\
&= \left( \phi\left( \left\langle \boldsymbol{w}_y^{(T_{\text{CutMix}})}, \boldsymbol{v}_{y,k} \right\rangle \right) - \phi\left( \left\langle \boldsymbol{w}_{-y}^{(T_{\text{CutMix}})}, \boldsymbol{v}_{y,k} \right\rangle \right) \right) \\
& \quad + \sum_{p\in[P]\backslash\{p^*\}} \left( \phi\left( \left\langle \boldsymbol{w}_y^{(T_{\text{CutMix}})}, \boldsymbol{x}^{(p)} \right\rangle \right) - \phi\left( \left\langle \boldsymbol{w}_{-y}^{(T_{\text{CutMix}})}, \boldsymbol{x}^{(p)} \right\rangle \right) \right) \\
&= \Omega(1) - o\left( \frac{1}{\text{polylog}(d)} \right) \\
& > 0.
\end{aligned}
$$

Hence, we have our conclusion. □

# F  Technical Lemmas

In this section, we introduce technical lemmas that are used for proving the main theorems. We present their proofs here for better readability.

The following lemma is used in Section C.2.4 and Section D.2.4:

**Lemma F.1.** *For any $z, \delta \in \mathbb{R}$,*

$$|\phi(z) - (z + \delta)\phi'(z)| \leq r + |\delta|.$$

*Proof of Lemma F.1.*

$$\phi(z) - z\phi'(z) = \begin{cases} z - \frac{1-\beta}{2}r - z = -\frac{1-\beta}{2}r = -\frac{1-\beta}{2}r & \text{if } z \geq r \\ \frac{1-\beta}{2r}z^2 + \beta z - \left(\frac{1-\beta}{r}z + \beta\right)z = \frac{1-\beta}{2r}z^2 & \text{if } 0 \leq z \leq r , \\ \beta z - \beta z = 0 & \text{if } z < 0 \end{cases}$$

and we obtain

$$|\phi(z) - (z + \delta)\phi'(z)| \leq |\phi(z) - z\phi'(z)| + |\delta||\phi'(z) \leq \frac{1-\beta}{2}r + |\delta| \leq r + |\delta|.$$

$\square$

The following lemma is used in Section C.2.4.

**Lemma F.2.** *Suppose $E_{\mathrm{init}}$ occurs. Then, for any model parameter $\boldsymbol{W} = \{\boldsymbol{w}_1, \boldsymbol{w}_{-1}\}$, we have*

$$\left\|\nabla_{\boldsymbol{W}} \sum_{i \in \mathcal{V}_{s,k}} \ell(y_i f_{\boldsymbol{W}}(\boldsymbol{X}_i))\right\|^2 \leq 8P^2 \sigma_{\mathrm{d}}^2 d|\mathcal{V}_{s,k}| \sum_{i \in \mathcal{V}_{s,k}} \ell(y_i f_{\boldsymbol{W}}(\boldsymbol{X}_i)),$$

*for each $s \in \{\pm 1\}$ and $k \in [K]$.*

*Proof of Lemma F.2.* For each $s \in \{\pm 1\}$ and $i \in [n]$, we have

$$\|\nabla_{\boldsymbol{w}_s} f_{\boldsymbol{W}}(\boldsymbol{X}_i)\| = \left\|\sum_{p \in [P]} \phi'\left(\left\langle \boldsymbol{w}_s, \boldsymbol{x}_i^{(p)} \right\rangle\right) \boldsymbol{x}_i^{(p)}\right\| \leq P \max_{p \in [P]} \left\|\boldsymbol{x}_i^{(p)}\right\| \leq 2P\sigma_{\mathrm{d}} d^{\frac{1}{2}},$$

where the inequality is due to the condition from the event $E_{\mathrm{init}}$ defined in Lemma B.2 and (A7)..
Therefore, for each $s \in \{\pm 1\}$, we have

$$\begin{aligned}
\left\|\nabla_{\boldsymbol{w}_s} \sum_{i \in \mathcal{V}_{s,k}} \ell(y_i f_{\boldsymbol{W}}(\boldsymbol{X}_i))\right\|^2 &= \left\|\sum_{i \in \mathcal{V}_{s,k}} \ell'(y_i f_{\boldsymbol{W}}(\boldsymbol{X}_i)) \nabla_{\boldsymbol{w}_s} f_{\boldsymbol{W}}(\boldsymbol{X}_i)\right\|^2 \\
&\leq \left(\sum_{i \in \mathcal{V}_{s,k}} \ell'(y_i f_{\boldsymbol{W}}(\boldsymbol{X}_i)) \|\nabla_{\boldsymbol{w}_s} f_{\boldsymbol{W}}(\boldsymbol{X}_i)\|\right)^2 \\
&\leq 4P^2 \sigma_{\mathrm{d}}^2 d \left(\sum_{i \in \mathcal{V}_{s,k}} \ell'(y_i f_{\boldsymbol{W}}(\boldsymbol{X}_i))\right)^2 \\
&\leq 4P^2 \sigma_{\mathrm{d}}^2 d|\mathcal{V}_{s,k}| \sum_{i \in \mathcal{V}_{s,k}} (\ell'(y_i f_{\boldsymbol{W}}(\boldsymbol{X}_i)))^2 \\
&\leq 4P^2 \sigma_{\mathrm{d}}^2 d|\mathcal{V}_{s,k}| \sum_{i \in \mathcal{V}_{s,k}} \ell(y_i f_{\boldsymbol{W}}(\boldsymbol{X}_i)).
\end{aligned}$$

The first inequality is due to triangular inequality, the third inequality is due to Cauchy-Schwartz inequality and the last inequality is due to $0 \leq -\ell' \leq 1$, which can be used to show $(\ell')^2 \leq -\ell' \leq \ell$.
As a result, we have our conclusion:

$$\left\|\nabla_{\boldsymbol{W}} \sum_{i \in \mathcal{V}_{s,k}} \ell(y_i f_{\boldsymbol{W}}(\boldsymbol{X}_i))\right\|^2 = \left\|\nabla_{\boldsymbol{w}_1} \sum_{i \in \mathcal{V}_{s,k}} \ell(y_i f_{\boldsymbol{W}}(\boldsymbol{X}_i))\right\|^2 + \left\|\nabla_{\boldsymbol{w}_{-1}} \sum_{i \in \mathcal{V}_{s,k}} \ell(y_i f_{\boldsymbol{W}}(\boldsymbol{X}_i))\right\|^2$$

$$\leq 8P^2\sigma_{\mathrm{d}}^2 d|\mathcal{V}_{s,k}| \sum_{i\in\mathcal{V}_{s,k}} \ell(y_i f_{\boldsymbol{W}}(\boldsymbol{X}_i)).$$

$\square$

The following lemma is used in Section D.2.4.

**Lemma F.3.** *Suppose $E_{\mathrm{init}}$ occurs. Then, for any model parameter $\boldsymbol{W} = \{\boldsymbol{w}_1, \boldsymbol{w}_{-1}\}$, we have*

$$\left\|\nabla \sum_{i\in\mathcal{V}_{s,k}} \mathbb{E}_{\mathcal{C}\sim\mathcal{D}_\mathcal{C}}[\ell(y_i f_{\boldsymbol{W}^{(t)}}(\boldsymbol{X}_{i,\mathcal{C}}))]\right\|^2 \leq 8P^2\sigma_{\mathrm{d}}^2 d|\mathcal{V}_{s,k}| \sum_{i\in\mathcal{V}_{s,k}} \mathbb{E}_{\mathcal{C}\sim\mathcal{D}_\mathcal{C}}[\ell(y_i f_{\boldsymbol{W}^{(t)}}(\boldsymbol{X}_{i,\mathcal{C}}))]$$

*for each $s \in \{\pm 1\}$ and $k \in [K]$.*

*Proof of Lemma F.3.* For each $s \in \{\pm 1\}$, $i \in [n]$ and $\mathcal{C} \subset [P]$ with $|\mathcal{C}| = C$, we have

$$\|\nabla_{\boldsymbol{w}_s} f_{\boldsymbol{W}}(\boldsymbol{X}_{i,\mathcal{C}})\| = \left\|\sum_{p\notin\mathcal{C}} \phi'\left(\left\langle \boldsymbol{w}_s, \boldsymbol{x}_i^{(p)}\right\rangle\right) \boldsymbol{x}_i^{(p)}\right\| \leq P \max_{p\in[P]}\left\|\boldsymbol{x}_i^{(p)}\right\| \leq 2P\sigma_{\mathrm{d}} d^{\frac{1}{2}},$$

where the inequality is due to the condition from the event $E_{\mathrm{init}}$ defined in Lemma B.2 and (A7). Therefore, for any $s \in \{\pm 1\}$, we have

$$\left\|\nabla_{\boldsymbol{w}_s} \sum_{i\in\mathcal{V}_{s,k}} \mathbb{E}_{\mathcal{C}\sim\mathcal{D}_\mathcal{C}}[\ell(y_i f_{\boldsymbol{W}}(\boldsymbol{X}_{i,\mathcal{C}}))]\right\|^2$$

$$= \left\|\sum_{i\in\mathcal{V}_{s,k}} \mathbb{E}_{\mathcal{C}\sim\mathcal{D}_\mathcal{C}}[\ell'(y_i f_{\boldsymbol{W}}(\boldsymbol{X}_{i,\mathcal{C}})) \nabla_{\boldsymbol{w}_s} f_{\boldsymbol{W}}(\boldsymbol{X}_{i,\mathcal{C}})]\right\|^2$$

$$\leq \left(\sum_{i\in\mathcal{V}_{s,k}} \mathbb{E}_{\mathcal{C}\sim\mathcal{D}_\mathcal{C}}\left[\ell'(y_i f_{\boldsymbol{W}}(\boldsymbol{X}_{i,\mathcal{C}})) \|\nabla_{\boldsymbol{w}_s} f_{\boldsymbol{W}}(\boldsymbol{X}_{i,\mathcal{C}})\|\right]\right)^2$$

$$\leq 4P^2\sigma_{\mathrm{d}}^2 d\left(\sum_{i\in\mathcal{V}_{s,k}} \mathbb{E}_{\mathcal{C}\sim\mathcal{D}_\mathcal{C}}\left[\ell'(y_i f_{\boldsymbol{W}}(\boldsymbol{X}_{i,\mathcal{C}}))\right]\right)^2$$

$$\leq 4P^2\sigma_{\mathrm{d}}^2 d|\mathcal{V}_{s,k}| \sum_{i\in\mathcal{V}_{s,k}} \mathbb{E}_{\mathcal{C}\sim\mathcal{D}_\mathcal{C}}\left[(\ell'(y_i f_{\boldsymbol{W}}(\boldsymbol{X}_{i,\mathcal{C}})))^2\right]$$

$$\leq 4P^2\sigma_{\mathrm{d}}^2 d|\mathcal{V}_{s,k}| \sum_{i\in\mathcal{V}_{s,k}} \mathbb{E}_{\mathcal{C}\sim\mathcal{D}_\mathcal{C}}[\ell(y_i f_{\boldsymbol{W}}(\boldsymbol{X}_{i,\mathcal{C}}))].$$

The first inequality is due to triangular inequality, the third inequality is due to Cauchy-Schwartz inequality and the last inequality is due to $0 \leq -\ell' \leq 1$, which can be used to show $(\ell')^2 \leq -\ell' \leq \ell$. As a result, we have our conclusion:

$$\left\|\nabla_{\boldsymbol{W}} \sum_{i\in\mathcal{V}_{s,k}} \mathbb{E}_{\mathcal{C}\sim\mathcal{D}_\mathcal{C}}\left[\ell(y_i f_{\boldsymbol{W}}(\boldsymbol{X}_{i,\mathcal{C}}))\right]\right\|^2$$

$$= \left\|\nabla_{\boldsymbol{w}_1} \sum_{i\in\mathcal{V}_{s,k}} \mathbb{E}_{\mathcal{C}\sim\mathcal{D}_\mathcal{C}}\left[\ell(y_i f_{\boldsymbol{W}}(\boldsymbol{X}_{i,\mathcal{C}}))\right]\right\|^2 + \left\|\nabla_{\boldsymbol{w}_{-1}} \sum_{i\in\mathcal{V}_{s,k}} \mathbb{E}_{\mathcal{C}\sim\mathcal{D}_\mathcal{C}}\left[\ell(y_i f_{\boldsymbol{W}}(\boldsymbol{X}_{i,\mathcal{C}}))\right]\right\|^2$$

$$\leq 8P^2\sigma_{\mathrm{d}}^2 d|\mathcal{V}_{s,k}| \sum_{i\in\mathcal{V}_{s,k}} \mathbb{E}_{\mathcal{C}\sim\mathcal{D}_\mathcal{C}}\left[\ell(y_i f_{\boldsymbol{W}}(\boldsymbol{X}_{i,\mathcal{C}}))\right].$$

$\square$

The following lemma guarantees the existence and characterizes the minimum of the CutMix loss in Section E.2.2.

**Lemma F.4.** *Suppose the event $E_{\text{init}}$ occurs. Let $g_1, g_{-1} : \mathbb{R} \times \mathbb{R} \to \mathbb{R}$ be defined as*

$$g_s(z_1, z_{-1}) := \frac{|\mathcal{V}_s|}{|\mathcal{V}_{-s}|}\ell'(Pz_s) + \frac{2}{P}\mathbb{E}_{\mathcal{S}\sim\mathcal{D}_{\mathcal{S}}}\left[|\mathcal{S}|\ell'(|\mathcal{S}|z_s - (P-|\mathcal{S}|)z_{-s})\right] + \frac{P-1}{3P},$$

*for each $s \in \{\pm 1\}$. There exist unique $z_1^*, z_{-1}^* > 0$ such that $g_1(z_1^*, z_{-1}^*) = g_{-1}(z_1^*, z_{-1}^*) = 0$. Furthermore, we have $z_1^*, z_{-1}^* = \Theta(1)$.*

*Proof of Lemma F.4.* For each $z_1 > 0$,

$$g_{-1}(z_1, 0) = \left(\frac{|\mathcal{V}_{-1}|}{|\mathcal{V}_1|} + 1\right)\cdot\left(-\frac{1}{2}\right) + \frac{2}{P}\mathbb{E}_{\mathcal{S}\sim\mathcal{D}_{\mathcal{S}}}[|\mathcal{S}|\ell'(-(P-|\mathcal{S}|)z_1)] + \frac{P-1}{3P}$$

$$< \left(\frac{|\mathcal{V}_{-1}|}{|\mathcal{V}_1|} + 1\right)\cdot\left(-\frac{1}{2}\right) + \frac{P-1}{3P} < 0,$$

since $\ell'(z) \leq -\frac{1}{2}$ for any $z \leq 0$ and we use $\frac{25}{52}n \leq |\mathcal{V}_1|, |\mathcal{V}_{-1}| \leq \frac{27}{52}n$ from the event $E_{\text{init}}$ defined in Lemma B.2. In addition,

$$g_{-1}(z_1, Pz_1 + \log 9)$$

$$= \frac{|\mathcal{V}_{-1}|}{|\mathcal{V}_1|}\ell'(P^2 z_1 + P\log 9) + \frac{2}{P}\mathbb{E}_{\mathcal{S}\sim\mathcal{D}_{\mathcal{S}}}[|\mathcal{S}|\ell'(|\mathcal{S}|Pz_1 + |\mathcal{S}|\log 9 - (P-|\mathcal{S}|)z_1)] + \frac{P-1}{3P}$$

$$\geq \left(\frac{|\mathcal{V}_{-1}|}{|\mathcal{V}_1|} + 1\right)\ell'(\log 9) + \frac{P-1}{3P}$$

$$> 0,$$

where we use $\frac{25}{52}n \leq |\mathcal{V}_1|, |\mathcal{V}_{-1}| \leq \frac{27}{52}n$ from the event $E_{\text{init}}$ defined in Lemma B.2 and (A1) for the last inequality.

Since $z \mapsto g_{-1}(z_1, z)$ is strictly increasing and by intermediate value theorem, there exists $S : (0, \infty) \to (0, \infty)$ such that $z = S(z_1)$ is a unique solution of $g_{-1}(z_1, z) = 0$ and $S(z_1) < Pz_1 + \log 9$. Note that $S$ is strictly increasing since $g_{-1}(z_1, z_{-1})$ is strictly decreasing with respect to $z_1$ and strictly increasing with respect to $z_{-1}$. Also, if $S(z)$ is bounded above, i.e., there exists some $U > 0$ such that $S(z) \leq U$ for any $z > 0$,

$$\lim_{z\to\infty} g_{-1}(z, S(z))$$

$$= \lim_{z\to\infty}\left(\frac{|\mathcal{V}_{-1}|}{|\mathcal{V}_1|}\ell'(PS(z)) + \frac{2}{P}\mathbb{E}_{\mathcal{S}\sim\mathcal{D}_{\mathcal{S}}}\left[|\mathcal{S}|\ell'(|\mathcal{S}|S(z) - (P-|\mathcal{S}|)z)\right] + \frac{P-1}{3P}\right)$$

$$\leq \lim_{z\to\infty}\left(\frac{|\mathcal{V}_{-1}|}{|\mathcal{V}_1|}\ell'(PU) + \frac{2}{P}\mathbb{E}_{\mathcal{S}\sim\mathcal{D}_{\mathcal{S}}}\left[|\mathcal{S}|\ell'(|\mathcal{S}|U - (P-|\mathcal{S}|)z)\right] + \frac{P-1}{3P}\right)$$

$$\leq -\frac{2}{P}\mathbb{E}_{\mathcal{S}\sim\mathcal{D}_{\mathcal{S}}}\left[|\mathcal{S}|\cdot\mathbb{1}_{|\mathcal{S}|\neq P}\right] + \frac{P-1}{3P} = -\frac{P-1}{P+1} + \frac{P-1}{3P}$$

$$< 0,$$

and it is contradictory. Hence, we have $\lim_{z\to\infty} S(z) = \infty$.

Let us choose $\underline{z} > 0$ such that

$$\underline{z} = \frac{1}{P}\log\left(\frac{3P\left(1 + \frac{|\mathcal{V}_1|}{|\mathcal{V}_{-1}|}\right)}{P-1} - 1\right),$$

and thus

$$\ell'(P\underline{z}) = -\frac{P-1}{3P\left(1 + \frac{|\mathcal{V}_1|}{|\mathcal{V}_{-1}|}\right)}.$$

We have

$$g_1(\underline{z}, S(\underline{z})) = \frac{|\mathcal{V}_1|}{|\mathcal{V}_{-1}|}\ell'(P\underline{z}) + \frac{2}{P}\mathbb{E}_{\mathcal{S}\sim\mathcal{D}_{\mathcal{S}}}\left[|\mathcal{S}|\ell'(|\mathcal{S}|\underline{z} - (P-|\mathcal{S}|)S(\underline{z}))\right] + \frac{P-1}{3P}$$

$$\leq \left(\frac{|\mathcal{V}_1|}{|\mathcal{V}_{-1}|} + 1\right)\ell'(P\underline{z}) + \frac{P-1}{3P}$$

$$= 0.$$

Next, we will prove the existence of $z^* > 0$ such that $g_1(z^*, S(z^*)) > 0$. Let us choose $\epsilon > 0$ such that

$$\epsilon^{-1} = \max\left\{\frac{3P(P+1)|\mathcal{V}_{-1}|}{(P-2)(P+2)|\mathcal{V}_1|} + \frac{3(P-1)}{P-2}, \frac{3}{2}\left(1 + \frac{P(P+1)|\mathcal{V}_{-1}|}{(P-1)(P-2)|\mathcal{V}_1|}\right),\right.$$

$$\left.\frac{12P}{P-7}\left(1 + \frac{|\mathcal{V}_{-1}|}{|\mathcal{V}_1|}\right), \frac{12P(P+1)}{(P-2)(P+2)}\left(1 + \frac{|\mathcal{V}_1|}{|\mathcal{V}_{-1}|}\right)\right\}, \tag{38}$$

and note that $\epsilon = \Theta(1)$. Since $\lim_{z\to\infty} S(z) = \infty$, we can choose $z^*$ such that $\ell'\left(\frac{1}{2}\min\{z^*, S(z^*)\}\right) = -\frac{\epsilon}{2}$. Then, for any $t \geq \frac{z^*}{2}$, we have

$$-\epsilon < \ell'(t) < 0 \quad \text{and} \quad -1 < \ell'(-t) < -1 + \epsilon. \tag{39}$$

From the definition of $S$ and (39) with $t = PS(z^*) > \frac{1}{2}\min\{z^*, S(z^*)\}$, we have

$$\mathbb{E}_{\mathcal{S}\sim\mathcal{D}_{\mathcal{S}}}\left[|\mathcal{S}|\ell'(|\mathcal{S}|S(z^*) - (P-|\mathcal{S}|)z^*)\right] = -\frac{P}{2}\left(\frac{|\mathcal{V}_{-1}|}{|\mathcal{V}_1|}\ell'(PS(z^*)) + \frac{P-1}{3P}\right)$$

$$< -\frac{P-1}{6} + \frac{P|\mathcal{V}_{-1}|}{2|\mathcal{V}_1|}\epsilon. \tag{40}$$

If $S(z^*) - (P-1)z^* \geq 0$, then

$$PS(z^*) > (P-1)S(z^*) - z^* > \ldots > 2S(z^*) - (P-2)z^*$$

$$= z^* + S(z^*) + S(z^*) - (P-1)z^*$$

$$\geq z^* + S(z^*) \geq \frac{1}{2}\min\{z^*, S(z^*)\},$$

and we have

$$-\frac{P-1}{6} + \frac{P|\mathcal{V}_{-1}|}{2|\mathcal{V}_1|}\epsilon > \mathbb{E}_{\mathcal{S}\sim\mathcal{D}_{\mathcal{S}}}\left[|\mathcal{S}|\ell'(|\mathcal{S}|S(z^*) - (P-|\mathcal{S}|)z^*)\right]$$

$$= \frac{1}{P+1}\left(\ell'(S(z^*) - (P-1)z^*) + \sum_{m=2}^{P}m\ell'(mS(z^*) + (P-m)z^*)\right)$$

$$\geq \frac{1}{P+1}\left(-\frac{1}{2} - \left(\frac{P(P+1)}{2} - 1\right)\epsilon\right),$$

where the last inequality is due to (39). This is contradictory to (38), especially the first term inside the maximum, and we have $S(z^*) - (P-1)z^* < 0$. In addition, if $(P-1)S(z^*) - z^* \leq 0$, then

$$Pz^* > (P-1)z^* - S(z^*) > \ldots > 2z^* - (P-2)S(z^*)$$

$$= z^* + S(z^*) + z^* - (P-1)S(z^*)$$

$$\geq z^* + S(z^*) \geq \frac{1}{2}\min\{z^*, S(z^*)\},$$

and we have

$$-\frac{P-1}{6} - \frac{P|\mathcal{V}_{-1}|}{2|\mathcal{V}_1|}\epsilon$$

$$< \mathbb{E}_{\mathcal{S}\sim\mathcal{D}_{\mathcal{S}}}\left[|\mathcal{S}|\ell'(|\mathcal{S}|S(z^*) - (P-|\mathcal{S}|)z^*)\right]$$

$$= \frac{1}{P+1}\left(P\ell'(PS(z^*)) + (P-1)\ell'((P-1)S(z^*) - z^*) + \sum_{m=0}^{P-2}m\ell'(mS(z^*) - (P-m)z^*)\right)$$

$$< \frac{1}{P+1}\left(-\frac{(P-1)(P-2)}{2}(1-\epsilon) - \frac{P-1}{2}\right),$$

where the last inequality is due to (39). This is contradictory to (38), especially the second term inside the maximum, and we have $(P-1)S(z^*) - z^* > 0$. Note that we have

$$-\frac{\epsilon}{2} = \ell'\left(\frac{1}{2}\min\{z^*, S(z^*)\}\right) \geq \ell'\left(\frac{z^*}{2P}\right),$$

and since $\epsilon = \Theta(1)$ in (38), we have $z^* \leq 2P\log\left(\frac{2}{\epsilon} - 1\right) = \mathcal{O}(1)$.

Thus, we have $S(z^*) - (P-1)z^* < 0 < (P-1)S(z^*) - z^* < PS(z^*)$. One can consider dividing the interval $[S(z^*) - (P-1)z^*, PS(z^*)]$ into a grid of length $z^* + S(z^*)$. Then, the interval is equally divided into $P-1$ sub-intervals and $0$ belongs to one of them. In other words, there exists $k \in [P-2]$ such that

$$kS(z^*) - (P-k)z^* \leq 0 < (k+1)S(z^*) - (P-k-1)z^*,$$

and note that if $P = 3$, then $k = 1$. The rest of the proof is divided into two cases: $(k+1)S(z^*) - (P-k-1)z^* \geq \frac{1}{2}(z^* + S(z^*))$ or $(k+1)S(z^*) - (P-k-1)z^* < \frac{1}{2}(z^* + S(z^*))$. In both cases, we show that $g_1(z^*, S(z^*)) > 0$.

**Case 1:** $(k+1)S(z^*) - (P-k-1)z^* \geq \frac{1}{2}(z^* + S(z^*))$

From (39), we have

$$-1 < \ell'(-Pz^*) < \cdots < \cdots < \ell'\big((k-1)S(z^*) - (P-k+1)z^*\big) < -1 + \epsilon,$$

and

$$-\epsilon < \ell'\big((k+1)S(z^*) - (P-k-1)z^*\big) < \cdots < \ell'\big(PS(z^*)\big) < 0.$$

Thus, we have

$$\mathbb{E}_{\mathcal{S}\sim\mathcal{D}_\mathcal{S}}\left[|\mathcal{S}|\ell'\big(|\mathcal{S}|S(z^*) - (P-|\mathcal{S}|)z^*\big)|\right]$$
$$> \frac{1}{P+1}\left(k\ell'\big(kS(z^*) - (P-k)z^*\big) - \frac{k(k-1)}{2}\right) - \frac{P}{2}\epsilon.$$

and we obtain $k > \frac{P-1}{2}$ since

$$\frac{k(k+1)}{2}$$
$$= \frac{k(k-1)}{2} + k$$
$$> -\frac{P(P+1)}{2}\epsilon - (P+1)\mathbb{E}_{\mathcal{S}\sim\mathcal{D}_\mathcal{S}}[|\mathcal{S}|\ell'(|\mathcal{S}|S(z^*) - (P-|\mathcal{S}|)z^*)]$$
$$+ k\ell'(kS(z^*) - (P-k)z^*) + k$$
$$> -\frac{P(P+1)}{2}\left(1 + \frac{|\mathcal{V}_{-1}|}{|\mathcal{V}_1|}\right)\epsilon + \frac{(P-1)(P+1)}{6}$$
$$\geq \frac{\frac{P-1}{2}\left(\frac{P-1}{2} + 1\right)}{2},$$

where the second inequality is due to (40) and the fact that $\ell' \geq -1$, and the last inequality is due to (38), especially the third term inside the maximum. Note that since $k \in \mathbb{N}$, $k \geq \frac{P}{2}$.

Note that from (39), we have

$$-1 < \ell'\big(-PS(z^*)\big) < \cdots < \ell'\big((P-k-1)z^* - (k+1)S(z^*)\big) < -1 + \epsilon,$$

and

$$-\epsilon < \ell'\big((P-k+1)z^* - (k-1)z^*\big) < \cdots < \ell'(Pz^*) < 0.$$

Hence, we obtain

$$\mathbb{E}_{\mathcal{S}\sim\mathcal{D}_\mathcal{S}}\left[|\mathcal{S}|\ell'\big(|\mathcal{S}|z^* - (P-|\mathcal{S}|)S(z^*)\big)\right]$$
$$\geq \frac{1}{P+1}\left(-\frac{(P-k-1)(P-k)}{2} - \frac{1}{2}(P-k) - \big((P-k+1) + \cdots + P\big)\epsilon\right)$$

$$\geq -\frac{(P-k)^2}{2(P+1)} - \frac{1}{P+1} \cdot \frac{P(P+1)}{2} \epsilon$$

$$\geq -\frac{P^2}{8(P+1)} - \frac{P}{2}\epsilon,$$

where we use $k \geq \frac{P}{2}$ for the last inequality. Therefore, we have

$$g_1(z^*, S(z^*)) = \frac{|\mathcal{V}_1|}{|\mathcal{V}_{-1}|}\ell'(Pz^*) + \frac{2}{P}\mathbb{E}_{\mathcal{S}\sim\mathcal{D}_\mathcal{S}}[|\mathcal{S}|\ell'(|\mathcal{S}|z^* - (P - |\mathcal{S}|)S(z^*))] + \frac{P-1}{3P}$$

$$\geq -\left(\frac{|\mathcal{V}_1|}{|\mathcal{V}_{-1}|} + 1\right)\epsilon - \frac{P}{4(P+1)} + \frac{P-1}{3P}$$

$$> 0,$$

where the last inequality is due to (38), especially the fourth term inside the maximum.

**Case 2:** $(k+1)S(z^*) - (P - k - 1)z^* < \frac{1}{2}(z^* + S(z^*))$

In this case, we have $kS(z^*) - (P - k)z^* \leq -\frac{1}{2}(z^* + S(z^*))$. From (39), we have

$$-1 < \ell'(-Pz^*) < \cdots < \ell'\big(kS(z^*) - (P - k)z^*\big) < -1 + \epsilon,$$

and

$$-\epsilon < \ell'\big((k+2)S(z^*) - (P - k - 2)z^*\big) < \cdots < \ell'\big(PS(z^*)\big) < 0.$$

Thus, we have

$$\mathbb{E}_{\mathcal{S}\sim\mathcal{D}_\mathcal{S}}[|\mathcal{S}|\ell'(|\mathcal{S}|S(z^*) - (P - |\mathcal{S}|)z^*))|]$$

$$> \frac{1}{P+1}\left((k+1)\ell'\big((k+1)S(z^*) - (P - k - 1)z^*\big) - \frac{k(k+1)}{2}\right) - \frac{P}{2}\epsilon,$$

and we obtain $k > \frac{P-1}{2}$ since

$$\frac{(k+1)^2}{2}$$

$$= \frac{k(k+1)}{2} + \frac{k+1}{2}$$

$$> -\frac{P(P+1)}{2}\epsilon - (P+1)\mathbb{E}_{\mathcal{S}\sim\mathcal{D}_\mathcal{S}}[|\mathcal{S}|\ell'(|\mathcal{S}|S(z^*) - (P - |\mathcal{S}|)z^*)]$$

$$\quad + (k+1)\ell'((k+1)S(z^*) - (P - k - 1)z^*) + \frac{k+1}{2}$$

$$> -\frac{P(P+1)}{2}\left(1 + \frac{|\mathcal{V}_{-1}|}{|\mathcal{V}_1|}\right)\epsilon + \frac{(P-1)(P+1)}{6}$$

$$> \frac{\left(\frac{P-1}{2} + 1\right)^2}{2},$$

where the second inequality is due to (40) and the fact that $\ell'(z) \geq -\frac{1}{2}$ $\forall z \geq 0$, and the last inequality is due to our (38), especially the third term inside the maximum. Note that since $k \in \mathbb{N}$, we have $k \geq \frac{P}{2}$.

Note that from (39), we have

$$-1 < \ell'\big(-PS(z^*)\big) < \cdots < \ell'\big((P - k - 2)z^* - (k + 2)S(z^*)\big) < -1 + \epsilon,$$

and

$$-\epsilon < \ell'\big((P - k)z^* - kz^*\big) < \cdots < \ell'(Pz^*) < 0.$$

Hence, we obtain

$$\mathbb{E}_{\mathcal{S}\sim\mathcal{D}_\mathcal{S}}[|\mathcal{S}|\ell'(|\mathcal{S}|z^* - (P - |\mathcal{S}|)S(z^*)]$$

$$\geq \frac{1}{P+1}\left(-\frac{(P - k - 1)(P - k)}{2} - ((P - k) + \cdots + P)\epsilon\right)$$

$$\geq -\frac{(P-k)(P-k-1)}{2(P+1)} - \frac{1}{P+1} \cdot \frac{P(P+1)}{2}\epsilon$$

$$\geq -\frac{(P-k)^2}{2(P+1)} - \frac{1}{P+1} \cdot \frac{P(P+1)}{2}\epsilon$$

$$\geq -\frac{P^2}{8(P+1)} - \frac{P}{2}\epsilon,$$

where we use $k \geq \frac{P}{2}$ for the last inequality. Therefore, we have

$$g_1(z^*, S(z^*)) = \frac{|\mathcal{V}_1|}{|\mathcal{V}_{-1}|}\ell'(Pz^*) + \frac{2}{P}\mathbb{E}_{\mathcal{S} \sim \mathcal{D}_S}[|\mathcal{S}|\ell'(|\mathcal{S}|z^* - (P - |\mathcal{S}|)S(z^*))] + \frac{P-1}{3P}$$

$$\geq -\left(\frac{|\mathcal{V}_1|}{|\mathcal{V}_{-1}|} + 1\right)\epsilon - \frac{P}{4(P+1)} + \frac{P-1}{3P}$$

$$> 0,$$

where the last inequality is due to (38), especially the fourth term inside the maximum.

In both cases, we have $g_1(z^*, S(z^*)) > 0$. By intermediate value theorem, there exist unique $z_1^*, z_{-1}^* > 0$ such that $g_1(z_1^*, z_{-1}^*) = g_{-1}(z_1^*, z_{-1}^*) = 0$. In addition, $\underline{z} \leq z_1^* \leq z^*$ and we have $z_1 = \Theta(1)$ since $\underline{z} = \Omega(1)$ and $z^* = \mathcal{O}(1)$. By using a similar argument, we can show that $z_{-1}^* = \Theta(1)$, and we have our conclusion. $\qquad \square$

