# OpenReview forum: "Provable Benefit of Cutout and CutMix for Feature Learning"
_NeurIPS.cc/2024/Conference — NeurIPS 2024 spotlight_

### Official Review · Reviewer_ZwXm · 2024-07-09

**Soundness:** 3
**Presentation:** 3
**Contribution:** 3
**Rating:** 6
**Confidence:** 3

**Summary:**

This paper offers a theoretical explanation for the effectiveness of two practically useful algorithms—Cutout and Cutmix—by applying typical feature learning analysis to multi-patched feature and noise data. The authors present negative results for ERM as a comparison and demonstrate positive results for Cutout and Cutmix, including positive margins and near-optimal test error.

**Strengths:**

This paper provides the first theoretical analysis of two practically useful data augmentation algorithms — Cutout and Cutmix.

**Weaknesses:**

This paper applies a smoothed leaky ReLU activation function in the neural network for technical reason, which differs from the model used in practice.

**Questions:**

1. Can the analysis be generalized to non-smooth neural networks? What technical difficulties will you encounter when generalizing the analysis to non-smooth activation functions like ReLU and leaky ReLU?

2. Why is the near convergence result only presented for the Cutmix setting but not for the Cutout setting?

3. Can you provide stronger results about the margin besides its positiveness? What is the order/magnitude of the margin?

4. Why does the result for Cutout hold for any iteration T in an interval, while the result for Cutmix only guarantees the existence of an iteration that satisfies the properties?

**Limitations:**

Same as weaknesses.

---

> ### Author Rebuttal · Authors · 2024-08-07
>
> We would like to express our appreciation to the reviewer for your valuable and constructive comments.  In the following, we address the points raised by the reviewer.
>
> ## **W1 & Q1. The use of smooth activation and generalization to non-smooth activation**
>
> We note that our activation function differs from those typically used in practice. However, smooth activation functions are widely employed in theoretical studies to analyze the generalization performance of two-layer neural networks. Numerous works have explored theory of deep learning under similar settings as discussed in lines 59–71. Therefore, we believe that studying two-layer networks with smooth activations is a valuable approach for bridging the gap between practice and theory.
>
> Our theoretical analysis can be generalized to non-smooth activations, such as ReLU or Leaky ReLU, in the cases of ERM and Cutout. In fact, our proof did not rely on the smoothness of activations in these cases. However, the smoothness of activation plays a crucial role in the analysis of CutMix. The main difference between the analysis of ERM & Cutout and CutMix lies in our approach: for ERM and Cutout, we directly investigate learning dynamics, whereas for CutMix, we characterize the global minimum and demonstrate that CutMix training can achieve a near-stationary point. To show that the CutMix loss achieves a near-stationary point, we use the descent lemma (Lemma 3.4 in [7]), which is only applicable to smooth objective functions. This is why smoothness is necessary for the CutMix analysis.
>
> ## **Q4. Why does the result for Cutmix only guarantee the existence of an iteration that satisfies the properties?**
>
> The use of the descent lemma makes a distinction between the conditions on the iterations for ERM & Cutout and CutMix. The convergence of a smooth function $f(x)$ with optimizing variables $x$ is usually guaranteed by showing
> $\frac{1}{T} \sum_{t=0}^{T-1} \lVert \nabla f(x^{(t)}) \rVert^2 \leq \frac{C}{T},$
> for some constant $C$. Therefore, $\min_{t =0,1, \dots, T-1}\lVert\nabla f(x^{(t)})\rVert^2 \leq \frac{C}{T}$ and it guarantees only the existence of an iteration that achieves a near-stationary point. We would like to emphasize that, even though our theory only guarantees this existence, our numerical validation (Section 5, Figure 1) shows consistent convergence behavior.
>
> ## **Q2. Why is the near convergence result only presented for the Cutmix setting but not for the Cutout setting?**
>
> The difference between ERM & Cutout and CutMix also results in different convergence criteria. We initially adopted training accuracy as the convergence criterion, following [1]. However, as discussed in lines 245–247, evaluating the training accuracy of augmented data is challenging because it uses augmented data with mixed labels. Therefore, we use the loss gradient as the convergence criterion, which can be guaranteed by the descent lemma. We note that we can also prove that ERM and Cutout achieve near convergence using the descent lemma. However, we adopted perfect training accuracy as the convergence criterion because it provides a more intuitive measure of how well the model fits all training data points. Moreover, due to the monotonic nature of ERM and Cutout training, we can show that the guarantees in our theorems hold for any sufficiently large polynomial time.
>
> ## **Q3. Can you provide stronger results about the margin besides its positiveness? What is the order/magnitude of the margin?**
>
> In our theoretical analysis, we proved that the model achieves an $\Omega(1)$ margin for test data with learned features. If you are interested, you can check line 802 for ERM, line 996 for Cutout, and line 1155 for CutMix.
>
> Thanks for your time and consideration.
>
> Best regards,
>
> Authors

---

> > ### Comment · Area_Chair_1igH · 2024-08-11
> >
> > Dear ZwXm,
> >
> > What are your thoughts after reading the rebuttal and other reviews?
> >
> > Best,
> >
> > AC

---

> > ### Comment · Reviewer_ZwXm · 2024-08-13
> >
> > Thank the authors for the response. I will keep my score.

---

> > > ### Author Response · Authors · 2024-08-13
> > >
> > > We appreciate for your response. If you have any additional questions, please feel free to ask.
> > >
> > > Best regards,
> > >
> > > Authors

---

### Official Review · Reviewer_59hr · 2024-07-09

**Soundness:** 3
**Presentation:** 3
**Contribution:** 3
**Rating:** 6
**Confidence:** 3

**Summary:**

This work theoretically investigates why patch-based data augmentation methods for image recognition, namely, CutOut and CutMix, improve performance based on the framework by [Zou+ICML23].
Specifically, they showed that when a CNN is trained with CutOut and CutMix, it can focus on rare and extremely rare features while ignoring noise.
Contrarily, the model trained with the standard ERM memorizes noise, which results in poorer performance.
These findings align with their numerical results.

**Strengths:**

- This work theoretically reveals why patch-based data augmentation, specifically CutOut and CutMix, improves the performance of CNNs.
These methods are so far powerful but ad-hoc heuristic, so research towards their theoretical understanding is important.
- By comparing ERM, Cutout, and CutMix in a unified framework, the authors highlight the superior ability of CutMix that can exploit extremely rare features in data while effectively ignoring noises.
- The theory well aligns with the numerical experiments on synthetic data.

**Weaknesses:**

- The important notions on features, i.e., common features, rare features, and extremely rare features, are not properly defined, making the manuscript difficult to read. I think improving the description in Section 2.1 would resolve this issue.
- The technical contributions against [Zou et al. 23] are not clearly stated in the manuscript.

**Questions:**

- If some features are extremely rare, I think they are less likely to appear in test data. Why learning it improves the performance? In practice, such features are likely to be attributed to systematic mislabeling.
Clearly defining common features, rare features, and extremely rare features may resolve this question.

**Limitations:**

- Although the checklist says the limitations are discussed in Section 2, I could not find the discussion of limitations in the section.

---

> ### Author Rebuttal · Authors · 2024-08-07
>
> We would like to express our appreciation to the reviewer for your valuable and constructive comments.  In the following, we address the points raised by the reviewer.
>
> ## **W1. Confusion on the notions of features**
>
> We have noticed that the notions of common features, rare features, and extremely rare features were not clearly explained in Definition 2.1. These features have significantly different frequencies, where common features appear much more frequently than rare features, and rare features appear more frequently than extremely rare features. While we describe this in the Assumptions (Section 2.4), we acknowledge that additional discussion near Definition 2.1 would be helpful to readers. We plan to include a more detailed explanation of these feature notions in Definition 2.1 in our next revision.
>
> In global response, we provide clarification on the notions of features and more detailed motivation of our data model. We hope this resolves any confusion or misunderstanding. We will address the question raised due to this confusion.
>
> ## **Q1. Why learning extremely rare features improves the performance?**
>
> As we discussed in the global response, extremely rare features in our setting are still frequent enough to appear in a non-negligible fraction of the training set. These features also have a presence in the test distribution and provide valuable information for correct classification. Hence, learning these features contributes to improved performance.
>
> ## **W2. The technical contributions against Zou et. al. 2023 are not clearly stated.**
>
> We have discussed comparison to Zou et.al. 2023 in lines 123-130 and lines 313-318. While we believe this coverage is sufficient, we provide a more detailed comparison here.
>
> - Zou et.al. 2023 only consider two types of features (common features, rare features) while we consider three types of features(common features, rare features, extremely rare features). This is because Zou et.al.2023 investigate two training methods (vanilla, Mixup) while we study three training methods (ERM, Cutout, CutMix).
> - Zou et.al. 2023 consider quadratic activation, whereas we focus on smoothed leaky ReLU activation. While our analysis of ERM shares the same spirit as the analysis of vanilla training in Zou et.al. 2023, detailed proofs differ due to variants in the network architecture.
> - The main difference between the Mixup analysis in Zou et.al. 2023 and our CutMix analysis lies in the approach: Zou et.al. 2023 prove the benefit of Mixup by directly investigating its learning dynamics, while we show the benefit of CutMix by characterizing the global minimum of CutMix loss. Zou et.al. 2023 prove that Mixup can learn rare features in early phase since it is “boosted” by the learning of common features. In contrast, our analysis shows that global minimizers learn all kinds of features indicating that the benefit of CutMix arises as training approaches convergence. This distinction highlights the differences in the underlying mechanisms: Mixup has benefit in the early stage of training, while the benefits of CutMix arise from the later stages of training.
>
> ## **Limitations. Could not find the discussion of limitations in the Section 2**
>
> Our work has limitations related to the neural network architecture, specifically single-neuron leaky ReLU CNNs. We discussed this in lines 137-151, but we will add a Limitation section in the next revision to explicitly address these technical limitations.
>
> Thanks for your time and consideration.
>
> Best regards,
>
> Authors

---

> > ### Comment · Reviewer_59hr · 2024-08-10
> >
> > We thank the authors for the rebuttal.
> > The authors resolved all my concerns.

---

> > > ### Author Response · Authors · 2024-08-11
> > >
> > > Thank you for your response. We are glad to hear that our explanations were helpful and adequately addressed your concerns and questions. If you have any additional questions, please feel free to ask.
> > >
> > > Best regards,
> > >
> > > Authors

---

### Official Review · Reviewer_pSne · 2024-07-14

**Soundness:** 4
**Presentation:** 4
**Contribution:** 2
**Rating:** 5
**Confidence:** 3

**Summary:**

- This paper investigates the effectiveness of patch-level data augmentation techniques, specifically Cutout and CutMix, in training two-layer neural networks.
- The study compares vanilla training, Cutout training, and CutMix training using a feature-noise data model.
- The findings show that Cutout can learn low-frequency features missed by vanilla training, while CutMix can capture even rarer features, resulting in the highest test accuracy.
- The analysis reveals that CutMix enables the network to learn features and noise vectors evenly, offering new insights into patch-level augmentation.

**Strengths:**

- This paper provides a theoretical analysis of how Cutout and CutMix (i.e., patch-level data augmentation) improve feature learning.
- It specifically explains what features each method learns during the training process, offering deep insights beyond empirical results.

**Weaknesses:**

- While this paper provides a theoretical analysis of patch-level data augmentation methods, it only analyzes Cutout and CutMix, leaving other related methods for future work.
- The theoretical analysis of this paper is expected to offer insights into practical applications and performance improvement, but it lacks detailed discussion in this regard.
- It would be beneficial if the paper could provide more meaningful insights to the readers beyond providing a theoretical analysis, which is the most significant shortcoming of this work.
- It is suggested to simplify the content of the “assumptions” summarized in Section 2.4 (with detailed explanations moved to the appendix) and add more experimental content in Section 5, particularly experiments related to CIFAR10.

**Questions:**

- What insights can be gained from the theoretical analysis presented in the paper? Can this help in developing better patch-level data augmentation techniques compared to existing ones?
- The paper specifies that the NN architecture used is a 2-layer CNN, but shouldn’t it be considered a 1-layer CNN?
- Additionally, the weights W exist only for each class individually. Is it not possible to perform a more general analysis with C channels, as done by Shen et al.?

**Limitations:**

There is no specified section for limitations in this paper. The checklist includes a brief explanation, but the corresponding details are not found in the main paper.

---

> ### Author Rebuttal · Authors · 2024-08-07
>
> We would like to express our appreciation to the reviewer for your valuable and constructive comments. In the following, we address the points raised by the reviewer.
>
> ## **W1. It only analyzes Cutout and CutMix**
>
> As the reviewer would also agree, a complete theory is not built in a single day. We chose Cutout and CutMix as a starting point for understanding patch-level augmentation techniques, and even this first step required a considerable amount of efforts. Also, We would like to emphasize that there has been limited theory works even on CutMix, as we discussed in Section 1.2.
>
> ## **W2&3, Q1. More insights beyond theoretical analysis**
>
> We agree that exploring practical implications beyond the theoretical analysis of existing methods like Cutout and CutMix is valuable. We also believe that understanding the underlying mechanisms of these methods can lead to the development of even more effective techniques. With this in mind, we would like to discuss some potential directions for future applications.
>
> One practical insight we can offer is related to the choice of cutting size $C$ in cutout. Our main intuition behind Cutout is that it outperforms by removing dominant noise patch, or “shortcuts,” which do not generalize to unseen data. Real-world image data contains features and noise across several patches. In practical scenarios, a larger cutting size can be effective in eliminating noise but may also remove important features that the model needs to learn. Thus, there is a trade-off in choosing the optimal cutting size. From this intuition, we believe that for images with a larger portion of background(noise), a larger cutting size is likely to be more effective.
>
> In our theoretical analysis, we show that Cutout and CutMix can learn rare features even though still memorizing some noises. We believe that improving underlying mechanism we have demonstrated can lead to even more effective techniques. Below, we list a couple of potential directions that can not only enable effective feature learning, as Cutout and CutMix do, but also improve these methods by preventing the memorization of noise.
>
> - One limitation of Cutout is that it does not always effectively remove dominant noise. As a result, dominant noise can still persist in the augmented data, leading to potential noise memorization. Developing strategies that can more precisely detect and remove these noise components from the image input could enhance the effectiveness of these methods.
> - The main underlying mechanism of CutMix, as we have outlined in Section 4.2, is that it learns information almost uniformly from all patches in the training data, which allows it to capture even rarer features. However, this approach also involves the memorization of noise, which can potentially degrade performance in real-world scenarios. We believe that a more sophisticated cut and pasting strategy—one that considers the positional information of patches rather than using a uniform approach—could improve the model’s ability to learn more from patches containing label-relevant features and reduce the impact of label-irrelevant noise. Already some recent patch-level augmentation method, such as PuzzleMix[5] and Co-mixup[6], use these types of strategy.
>
> ## **W4. Suggestion regarding paper organization**
>
> We appreciate for your suggestion regarding the paper’s organization. We agree that moving some of the experimental results from the appendix to the main text could improve our draft. However, we also believe that fully presenting the assumptions in the main text is important for the completeness of our theoretical results.  We will carefully consider how to balance these aspects in the next revision, keeping the page limits in mind.
>
> ## **Q2. The paper specifies that the NN architecture used is a 2-layer CNN, but shouldn’t it be considered a 1-layer CNN?**
>
> Whether our network is considered a 2-layer CNN or a 1-layer CNN depends on the perspective. Following the existing results in the literature, we regard it as a 2-layer CNN, where the weights in the second layer are fixed at 1 and -1, making only the first layer trainable. Many other works including [1,2,3] also consider neural networks similar to ours as 2-layer CNNs.
>
> ## **Q3. Is it not possible to perform a more general analysis with $C$ channels, as done by Shen et al.?**
>
> We believe that a more general analysis with multiple neurons as done by Shen et al. 2022 is also possible, and we numerically validate this scenario in Appendix A.2, Figure 3.
>
> As we discussed in line 144-151, using ReLU activation requires multiple neurons since neurons can be negatively initialized and remain unchanged throughout the training. In contrast, leaky ReLU activation always has positive slope, ensuring that a single neuron is often sufficient. Therefore, for mathematical simplicity, we focus on the case where the network has a single neuron for each output, and extension to multi-neuron case may require some additional techniques.
>
> ## **Limitation. There is no specified section for limitations in this paper**
>
> Our work has limitations related to the neural network architecture, specifically single-neuron leaky ReLU CNNs. We discussed this in lines 137-151, but we will add a Limitation section in the next revision to explicitly address these technical limitations.
>
> We hope that our response clarifies your concerns. We would appreciate it if you could consider reassessing our submission.
>
> Best regards,
>
> Authors

---

> > ### Comment · Area_Chair_1igH · 2024-08-11
> >
> > Dear pSne,
> >
> > What are your thoughts after reading the rebuttal and other reviews?
> >
> > Best,
> >
> > AC

---

> > ### Comment · Reviewer_pSne · 2024-08-12
> >
> > Thank you for your response. It alleviated my concerns. I'll raise my rating. The provided insights will be very important, along with the theoretical analysis, in the paper.

---

> > > ### Author Response · Authors · 2024-08-13
> > >
> > > Thank you for your feedback and for reconsidering the score. We are glad to hear that our response addressed your concerns. We would also be happy to hear if you have any additional thoughts or suggestions.
> > >
> > > Best regards,
> > >
> > > Authors

---

### Official Review · Reviewer_mYpt · 2024-07-18

**Soundness:** 3
**Presentation:** 2
**Contribution:** 3
**Rating:** 6
**Confidence:** 4

**Summary:**

The paper aims to provide a novel theoretical insight into the training dynamics of data augmentation methods such as Cutout and Cutmix. It also supports theoretically why Cutout and Cutmix perform better than ERM by showing that the ERM training is unable to learn rare and extremely rare features and that ERM can fit perfectly on the training data while being random on the extremely rare data. Thus the key contributions can be summarized as:

**Comparative Analysis**: The paper reveals that Cutout outperforms ERM by facilitating the learning of rarer features, which ERM fails to do (Theorem 3.1 and Theorem 3.2). CutMix is shown to achieve nearly perfect performance by learning all features (Theorem 3.3).

**ERM Limitations**: The authors propose that the negative results for ERM stem from its tendency to classify training samples by memorizing noise vectors instead of learning meaningful features, particularly when features do not appear frequently enough.

**Cutout Benefits**: Cutout mitigates the challenge faced by ERM by removing some of the strong noise patches, thus enabling the learning of rare features to some extent.

**CutMix Mechanism**: A novel technique to analyse the training dynamics of CutMix. This technique views the non-convex loss as a composition of a convex function and reparameterization, allowing characterization of the global minimum of the loss. It shows that CutMix forces the model to activate almost uniformly across every patch of inputs, facilitating the learning of all features.

**Strengths:**

What I like about the paper:
+ It addresses an important aspect of better understanding data-augmentation methods like Cutout and Cutmix.
+ It is theoretically well-motivated.
+ Provides a simple explanation of common features ≫dominant noises≫ rare features ≫background noise≫ extremely rare features for ERM training dynamics.

**Weaknesses:**

- I believe the writing of the current draft version can be further improved. (see questions)
- The modelling assumptions about 2-layer CNN are questionable. (see questions for further discussion)
- The theory needs further experimental evaluation (real-world experiment on CIFAR-10 is uninformative to the story, see questions)

**Questions:**

1. **Definitions**: The definitions of common, rare and extremely rare features are central to the story and contributions of the paper. However, they are not explicitly defined anywhere in the main draft.  I assume there must be some condition on probability $\rho_k$ for a feature to be rare and extremely rare. Defining these in problem formulation will help improve the readability of the paper significantly.

2. **Model Definition**: The current model definition is not 2-layered CNN in my opinion since it does not contain a hidden layer. I can summarize the model definition as $f(X)=\[1, -1\]^T\phi(\mathbf{W}\mathbf{X})1_{P}$. Where $\mathbf{W}=[w_{1},w_{-1}]$ and $\mathbf{X}\in\mathbb{R}^{d\times P}$, this is akin to a single-layer network.

3. **Simulation Evaluation**: About the numerical simulations, the common features are defined as frequency 0.8, rare as 0.15 and extremely rare as 0.05. Is there any reasoning or theoretical justification to do so? What happens when we change the frequencies of these features to skew them further? Is the assumption that extremely rare feature frequency is rare only in the training and at test time there exists another distribution? Otherwise, why is the low performance of ERM on extremely rare features negative? Also, for Fig 1 (leftmost plot) it is not obvious why Cutmix is non-monotonic here. It rises and then plateaus, perhaps authors can change the plot scale to appropriately demonstrate this point.

4. **Real World Evaluation**: I find the real-world evaluation for the theory extremely lacking. It need not be extensive with many datasets but in the current CIFAR-10 experiments, there is no intuition from the perspective of common, rare and extremely rare features. A similar experiment to simulations on CIFAR-10 would support the point of the paper much better.

5. **Visualisation of extremely rare features**: What would the visualisation of extremely rare and rare features in the case of CIFAR-10 look like? If possible can authors visualise the features learnt by ERM trained and CutOut and CutMix? This would show that extremely rare features learned by cutmix are indeed interpretable, thus important for generalisation as compared to random noise in the dataset.

**Limitations:**

The major limitations of the work are in its choice of definitions, namely model definition and definition of extremely rare features.
1. Limitation of model definition: I am sceptical of the current conclusions generalising to networks with more layers since the current model definition is not really even 2 layered.
2. Limitations of extremely rare features definition: Based alone on the frequency of a feature in the dataset, one can not distinguish noise from extremely rare features unless we have access to the data-generating process, which is often not the case in the real world.

---

> ### Author Rebuttal · Authors · 2024-08-07
>
> We express our gratitude for your valuable comments.  In the following, we address the points raised by the reviewer.
>
> ## **W1, Q1. The definitions of features**
>
> Please refer to our global response. We describe these features in Section 2.4, but we acknowledge that discussion near Definition 2.1 would be helpful to readers. We plan to include a more detailed explanation of these notions in Definition 2.1 in our next revision.
>
> ## **W2, Q2, L1. 2-layered CNN**
> Whether our network is 2-layer or 1-layer may depend on the perspective. Following existing works [1,2,3] in the literature, we regard it as a 2-layer CNN, where the weights in the second layer are fixed at 1 and -1, making only the first layer trainable. In addition, we agree that extension to a deeper neural network is not straightforward. However, we believe that our work sheds light on the benefits of patch-level augmentation, making it a valuable contribution. We will leave the extension to deeper networks as future work.
>
> ## **Q3. Numerical simulations**
> Our choice of frequencies is intended to highlight the distinctions between the three methods. Our findings suggest that applying Cutout and CutMix lowers the “threshold” for the frequency of a feature’s occurrence required for learning it. When we reduce the frequency of a feature from 1 to 0, initially all three methods can learn the feature. As the frequency decreases, ERM may fail to learn the feature while Cutout and CutMix continue to succeed. At even lower frequencies, only CutMix can learn the feature whenever it appears in the training data.
>
> ## **Q3, L2. Confusion regarding extremely rare features**
> We believe there may be some confusion regarding extremely rare features. We have provided explanations with examples for our framework in our global response, and we encourage you to refer to it. We address remaining questions here:
>
> >**Is the assumption that .... extremely rare features negative?**
>
> Our training and test data distributions are identical. As detailed in our general response, learning more features generally improves performance. Although data with extremely rare features make up a small portion of the test distribution, learning these features can enhance test accuracy on that data, leading to overall test-time performance improvements. Thus, low performance of ERM on (extremely) rare features is negative for overall performance.
>
> >**Limitation. One can not distinguish noise from extremely rare features**
>
> We think the reviewer may have confused. Extremely rare features are also label-dependent information that can appear nontrivially many times (albeit relatively rarer) across several data points, while similar or identical noise patches hardly reappear across different data points.
>
> ## **Q3. Non-monotonicity of CutMix**
> In Figure 1, the leftmost plot shows that the curve for CutMix initially rises, then slightly decreases before plateauing. This indicates a non-monotone behavior for CutMix, in contrast to the other methods. Let us clarify why: Due to the use of mixed labels, CutMix loss has global minimizers and they evenly learn all kinds of features. In the early stages of learning, the model tends to overshoot in learning common features because of their faster learning speed compared to other features. This initial overshooting leads to a temporary rise and subsequent decrease in the curve. This non-monotone behavior is precisely the reason why we had to devise a novel proof strategy different from ERM and Cutout.
>
> ## **W3, Q4, Q5. Real world evaluation**
>
> We experimented on CIFAR 10 to support our findings and address the reviewer’s concern.
>
> We train ResNet18 using vanilla training without any augmentation, as well as with Cutout and CutMix, following the same experimental details described in Appendix A.1, except using only 10% of the training set. This data-hungry setting is intended to highlight the benefits of Cutout and CutMix. We then evaluated the trained models on the remaining 90% of the CIFAR training dataset. The reason for evaluating on the remaining training dataset is that we plan to analyze the misclassified data using C-score[4], which is publicly available only for the training dataset.
>
> C-score measures the structural regularity of data, with lower values indicating examples that are more difficult to classify correctly. In our framework, data with harder-to-learn features (corresponding to rarer features) would likely have lower C-scores. Since directly extracting and quantitatively evaluating features learned by the models is challenging, we use the C-score as a proxy to evaluate the misclassified data across models trained by ERM, Cutout, and CutMix.
>
> Figure 2 in our attached pdf in our global response illustrates that Cutout tends to misclassify data with lower C-scores compared to ERM, indicating that Cutout learns more hard-to-learn features than vanilla training. Furthermore, the data misclassified by CutMix has even lower C-scores than those misclassified by Cutout, suggesting that CutMix is effective at learning features that are the most challenging to classify. This observation aligns with our theoretical findings, demonstrating that CutMix captures even more difficult features compared to both ERM and Cutout.
>
> Since directly visualizing features learned by a model is challenging, we present data that was misclassified by the model trained with ERM but correctly classified by the model trained with Cutout instead. In Figure 3 in attached file, we show 7 samples per class with the lowest C-scores, which are considered to have rare features. Similarly, we also visualize data misclassified by the model trained with Cutout but correctly classified by the model trained with CutMix to represent data points with extremely rare features. This approach allows us to interpret some (extremely) rare features in CIFAR 10 ,such as frogs with unusual colors.
>
> Thanks for your time and consideration.
>
> Best regards,
>
> Authors

---

> > ### Comment · Area_Chair_1igH · 2024-08-11
> >
> > Dear mYpt,
> >
> > What are your thoughts after reading the rebuttal and other reviews?
> >
> > Best,
> >
> > AC

---

> > ### Comment · Reviewer_mYpt · 2024-08-11
> > **Thanks for the rebuttal**
> >
> > I thank authors for addressing all my concerns. The rebuttal has cleared my original doubts about the work. Therefore, I raise my score.

---

> > > ### Author Response · Authors · 2024-08-12
> > >
> > > Thank you for your response. We are glad to hear that our response was helpful and resolved your concerns. If you have any further questions, please feel free to ask.
> > >
> > > Best regards,
> > >
> > > Authors

---

### Official Review · Reviewer_hATA · 2024-07-29

**Soundness:** 3
**Presentation:** 3
**Contribution:** 3
**Rating:** 7
**Confidence:** 2

**Summary:**

This paper theoretically analyzed the benefits of two data augmentation techniques, Cutout and CutMix, for feature learning of two-layer and two-neuron convolutional neural networks. The authors considered an ideal data distribution in high dimension with one feature patch, one dominant noise patch, and some background noise patches, where the feature patch is generated by three different features in terms of the population: common, rare, and extremely rare features. This paper showed that classical training dynamic without data augmentation achieves perfect training accuracy but performs poorly on rare and extremely rare features, leading to low test accuracy; Cutout training achieves perfect training accuracy and learns rare features better than ERM, but still struggles with extremely rare features; and CutMix training achieves near-perfect performance by learning all features and noise vectors evenly, resulting in the highest test accuracy among the three methods. The paper validated the theoretical findings through extensive numerical experiments and real-world data experiments on CIFAR-10, showing that CutMix achieves the best performance in terms of test accuracy.

**Strengths:**

The paper provided a novel theoretical framework for understanding the benefits of Cutout and CutMix, filling a gap in the literature where empirical success lacked a theoretical explanation. The presentation of the main theorems (Theorem 3.1, 3.2, and 3.3) is rigorous and clearly shows the differences in learning dynamics and performance between ERM, Cutout, and CutMix..

**Weaknesses:**

1. The use of a feature-noise data model to analyze the effectiveness of different training methods is restrictive. This feature noise patch data ideally separates the noise and feature orthogonally and enables us to consider the dynamics of the weights in feature and noise directions separately. However, in general data itself may contain some useful nonlinear features, which may not be characterized by this ideal model. It would be better to provide more motivations for this data assumption and present some real-world datasets exhibiting common, rare, and extremely rare features simultaneously. Besides this analysis only considered binary classification setting.

2. The asymptotic choice of the hyperparameters in Section 2.4 is not natural. While analyzing high-dimensional limits is necessary for the theoretical analysis, the practical implications and sensitivity to these hyperparameters should be better addressed. Including a more detailed discussion or empirical validation of these assumptions would be better. Additionally, the paper could provide more guidance on how to choose the hyperparameters (or the scaling) in training and data augmentation for practitioners.

3. There is limited discussion on the computational costs associated with implementing Cutout and CutMix compared with ERM. Including a theoretical analysis of memory usage, time complexity, or other computational costs would help in understanding the trade-offs involved.

**Questions:**

1. In Section 1.2, it would be better to include more recent work on feature learning theory. For instance, some references are below.

2. In Cutout training, based on your analysis of training, is there an optimal choice of hyperparameter $C$? Can we set $C=0$ in Theorem 3.2 to get Theorem 3.1?

3. In Section 2.4, why do you assume the number of data points $n$ is much smaller than the feature dimension? Is it because of the over-parameterized model? And how about the large learning rate $\eta$ case, for training dynamics? Many previous papers [Yang and Hu. 2020, Abbe, et al. 2022, Ba, et al. 2022] have shown the benefits of large learning rates to obtain feature learning. Moreover, how do you assume $K$, the number of orthogonal feature directions?

4. In (4), a typo in $w^{(t+1)}_s$

5. The proof strategy of CutMix training is to find the global minimizer directly. Does this strategy also work for ERM and Cutout? Can this method generalize to wider neural networks and more general datasets?

6. In Lemma B.3, the statement is not complete. You define the $\gamma_s^{(t)}$ and $\rho_s^{(t)}$ later for different cases, separately.

-------------------------------------------------------------------------------------
Yang and Hu. 2020. Feature learning in infinite-width neural networks

Abbe, et al. 2022. The merged-staircase property: a necessary and nearly sufficient condition for sgd learning of sparse functions on two-layer neural networks

Ba, et al. 2022. High-dimensional asymptotics of feature learning: How one gradient step improves the representation.

Ben Arous, et al. 2021. Online stochastic gradient descent on non-convex losses from high-dimensional inference.

---

> ### Author Rebuttal · Authors · 2024-08-07
>
> We express our gratitude for your time and valuable comments. We also thank you for bringing to our attention the typo in (4) and unclear statement in Lemma B.3 (we intended to claim the “existence” of such $\gamma_s^{(t)}, \rho_s^{(t)}$ for each training method). We will fix/clarify this in our next revision. In the following, we address the points raised by the reviewer.
>
> ## **W1. Motivation for data**
>
> We provide motivations with examples for our framework in our global response. We would appreciate it if you could refer to it.
>
> ## **W2, Q2&3. Choice of hyperparameters**
>
> We assume the number of data points $n$ is much smaller than the dimension $d$ which corresponds to the over-parameterization regime common in modern neural networks. This allows us to apply high-dimensional probability theory effectively, as you mentioned.
>
> The assumptions on hyperparameters, such as the strength of noise and the frequencies of features, are designed to highlight the distinctions between the three training methods by satisfying the inequalities outlined below line 280. The choices for other hyperparameters are made to ensure convergence within our specific setting. These types of assumptions are also considered in several related works [1,2,3].
>
> Also, we do not impose a strict condition on the total number of features. More important thing is the frequency of each feature rather than the total number of features. Any value of $K$ is possible as long as each $\rho_k$ can satisfies given conditions.
>
> Even though most choices of hyperparameters arise from technical elements of our proof, we can provide intuition on choice of cutting size $C$ in cutout. Our intuition behind Cutout is that it outperforms by removing dominant noise, which do not generalize to unseen data. Thus, $C\geq 1$ is necessary to ensure that the analysis of training dynamics differs from that with $C=0$. The proof strategy for Theorem 3.2 heavily relies on the condition $C\geq1$. Hence, setting $C=0$ in Theorem 3.2 does not derive Theorem 3.1 directly.
>
> In practice, image data contains features and noise across several patches. A larger cutting size can be effective in removing noise but may also remove important features that the model needs to learn. Thus, there is a trade-off in choosing the optimal cutting size. From this intuition, we believe that for images with a larger portion of background(noise), a larger cutting size is likely to be more effective.
>
> ## **W3. The computational costs**
>
> Since our work does not focus on proposing novel algorithms, we believe that a discussion on computational and memory costs is not essential and should not be considered a weakness of our work. However, we address this concern here for the reviewer's benefit.
>
> In the practical implementation of Cutout, a squared region is randomly sampled and this same region is cut from all images within a batch. Similarly, in the implementation of CutMix, a squared region is randomly sampled and then used to cut and paste parts of pairs of data batch and its random permutation. Consequently, the computational and memory costs involved in each iteration of both Cutout and CutMix are comparable to those of vanilla training, differing only by a constant factor.
>
> In our theoretical setting, we consider training using full-batch gradient descent for ERM, and gradient descent on the expected loss for Cutout and CutMix. These idealized training methods involve higher computational and memory costs per epoch since they require more augmented data. For example, Cutout requires $\binom{P}{C} n$ data points and CutMix requires $2^{P-1} n^2$ data points. However, we would like to emphasize that this setting is designed for a theoretical understanding; as outlined above, the practical versions are not computationally burdensome.
>
> ## **Q1&3. Recent works on feature learning theory**
>
> Thank you for suggesting recent works on feature learning theory and for raising questions related to comparisons with these literatures. We were not familiar with these works and have noted that they are somewhat orthogonal to our work due to differences in the definition of “features” compared to our approach and other related previous works.
>
> The notion of features in our work refers to label-relevant information contained in the input that is useful for generalization to unseen data. In contrast, the notion of features in the works you suggested seems to be relate to the outputs of the last hidden layer (i.e., the representation of data point learned by the network). This distinction suggests that the concepts of "features" in these studies differ from those in our approach.
>
> We believe that a large learning rate has less impact within our notion of feature learning since it does not affect trend in learning speed of features and noise which is essential as we described in Section 4.1.
>
> ## **Q5. The proof strategy for CutMix**
>
> The proof strategy for CutMix cannot be applied to ERM and Cutout because their training loss lacks a global minimum due to the exponential tail of the logistic loss ($\lim_{z \rightarrow \infty} \ell(z) = 0$). In contrast, CutMix loss has a global minimum due to its use of mixed labels ($\lim_{z \rightarrow \infty } \ell(z)+ \ell(-z) = \infty$).
>
> Since main idea of our strategy is considering the training loss as a composition of reparameterization and convex functions, we believe that this technique could be extended to a broader class of architectures, datasets, and training methods involving mixed labels. However, the exact characterization of the global minimum in different settings would require techniques beyond those we have used.
>
> Thanks for your time and consideration.
>
> Best regards,
>
> Authors

---

> > ### Comment · Area_Chair_1igH · 2024-08-11
> >
> > Dear hATA,
> >
> > What are your thoughts after reading the rebuttal and other reviews?
> >
> > Best,
> >
> > AC

---

> > > ### Comment · Reviewer_hATA · 2024-08-14
> > > **Official Comment from Reviewer hATA**
> > >
> > > I thank the authors for the detailed response and attachment. After reading the rebuttals and other reviews, I would like to increase the score.

---

> > > > ### Author Response · Authors · 2024-08-14
> > > >
> > > > We appreciate you for your response. We are glad to hear that our responses were helpful and adequately addressed your concerns and questions. We would also be happy to hear if you have any additional thoughts or suggestions.
> > > >
> > > > Best regards,
> > > >
> > > > Authors

---

### Author Rebuttal · Authors · 2024-08-07

Dear reviewers,

We express our gratitude for your time and valuable comments.

Before addressing concerns/questions raised by individual reviewers, we would like to re-emphasize the main intuition behind our theoretical framework and findings.

## **Motivation for feature and noise in our data distribution**

We would like to provide motivation for our data distribution and clarify each component of the distribution. The figure in the attached file illustrates the core ideas behind our feature noise data model, which we believe will help reviewers better understand our approach.

The main characteristic of image data is that the input contains both information relevant to the image labels (which we refer to as "features," e.g., a cat’s face) and information irrelevant to the labels (which we refer to as "noise," e.g., the background). The key difference between these two components is that features can appear in other data points, while noise typically does not appear in other (unseen) data since it is independent of the label. This distinction motivates our approach, where features are sampled from a set of fixed vectors and noise is sampled from a Gaussian distribution. Both features and noise can be used to correctly classify training data, however, only features are useful for correctly classifying unseen test data. Thus, learning features are important for better generalization.

## **Clarification on difference between common, rare, and extremely rare features**

Next, we would like to clarify the notion of common, rare, and extremely rare features. Different features appear in data with different frequencies. For example, in a cat and dog classification task, the number of occurrence of cat’s face and cat’s tail in the dataset might differ significantly, yet both are relevant for the label "cat". The reason we separate features into these three categories is to highlight the distinctions between the three training methods we analyze. Additionally, we emphasize that “extremely rare” features are still likely to appear in a nontrivial fraction of the training data with high probability, as outlined in the second bullet point of Lemma B.2, given the assumptions on hyperparameters in Assumption B.1. We hope this clarification addresses any concerns or misunderstandings expressed by some reviewers.

We plan to include these discussions in the next revised version of our paper.

Additionally, we provide references that will be addressed in individual rebuttals below.

We hope our response helps to resolve any concerns and confusion.

Best regards,

Authors

Reference

[1] Ruoqi Shen, Sébastien Bubeck, and Suriya Gunasekar. Data augmentation as feature manipulation. In ICML 2022

[2] Zeyuan Allen-Zhu and Yuanzhi Li. Towards understanding ensemble, knowledge distillation and self-distillation in deep learning. arXiv 2020

[3] Difan Zou, Yuan Cao, Yuanzhi Li, and Quanquan Gu. The benefits of mixup for feature learning. In ICML 2023

[4] Zihenrg Jiang, Chiyuan Zhang Kunal Talwar, and Michael C. Mozer. Characterizing structural regularities of labeled data in overparameterized models. In ICML 2021

[5] Jang-Hyun Kim, Wonho Choo, and Hyun Oh Song. Puzzle mix: Exploiting saliency and local statistics for optimal mixup. In ICML 2020

[6] Jang-Hyun Kim, Wonho Choo, Hosan Jeong, and Hyun Oh Song. Co-mixup: Saliency guided joint mixup with supermodular diversity. In ICLR 2021

[7] Sébastien Bubeck et al. Convex optimization: Algorithms and complexity. Foundations and Trends®358 in Machine Learning 2015

---

### Decision · Program_Chairs · 2024-09-25

**Decision:**

Accept (spotlight)

**Comment:**

The paper investigates the theoretical benefits of Cutout and CutMix for two-layer neural networks.

The reviewers generally appreciate the theoretical insights of the widely used data augmentation methods. They unanimously voiced that the remaining concerns are adequately addressed by the rebuttal.

I recommend accepting this paper as a spotlight, acknowledging the following values:
- Novel theoretical framework for connecting feature learning and data augmentation
- Inspiring potential future extensions for theoretical analysis or novel augmentation techniques
- Empirical support for the theoretical findings